# Theoretical Insights Into Multiclass Classification:
# A High-dimensional Asymptotic View

**Christos Thrampoulidis**
UC, Santa Barbara
cthrampo@ucsb.edu

**Samet Oymak**
UC, Riverside
oymak@ece.ucr.edu

**Mahdi Soltanolkotabi**
University of Southern California
soltanol@usc.edu

## Abstract

Contemporary machine learning applications often involve classification tasks with many classes. Despite their extensive use, a precise understanding of the statistical properties and behavior of classification algorithms is still missing, especially in modern regimes where the number of classes is rather large. In this paper, we take a step in this direction by providing the first asymptotically precise analysis of linear multiclass classification. Our theoretical analysis allows us to precisely characterize how the test error varies over different training algorithms, data distributions, problem dimensions as well as number of classes, inter/intra class correlations and class priors. Specifically, our analysis reveals that the classification accuracy is highly distribution-dependent with different algorithms achieving optimal performance for different data distributions and/or training/features sizes. Unlike linear regression/binary classification, the test error in multiclass classification relies on intricate functions of the trained model (e.g., correlation between some of the trained weights) whose asymptotic behavior is difficult to characterize. This challenge is already present in simple classifiers, such as those minimizing a square loss. Our novel theoretical techniques allow us to overcome some of these challenges. The insights gained may pave the way for a precise understanding of other classification algorithms beyond those studied in this paper.

## 1   Introduction

Multiclass classification is fundamental to a large number of real-world machine learning applications that demand the ability to automatically distinguish between thousands of different classes. Applications include essentially any problem with categorical outputs spanning natural language processing [SVL14], where a seq2seq decoder has to choose the correct word token, reinforcement learning [JGP16, MXSS20], where the agent has to choose the correct action, to recommendation systems, where the model should recommend the correct movie out of many other options. For instance, YouTube's recommendation system is modeled as an extreme multiclass problem with more than a million classes where each video corresponds to a viable class [CAS16].

The growing list of applications motivate an in-depth exploration of multiclass classification algorithms. Despite their extensive use however, a precise understanding of the statistical properties and behavior of classification algorithms is still missing with many open questions: *What is the total and per class test accuracy? How does this quantity depend on various problem parameters such as data distributions, problem dimensions, etc.? What is the highest test accuracy achievable by any algorithm? What is the best algorithm for each scenario? Which algorithm achieves the highest accuracy on rare or minority classes? How does the answer to the above question change in modern regimes where the number of classes is large?*

Asymptotic analysis in modern high-dimensional regimes where the number of training data and feature sizes grow in tandem with each other provides a promising setting for precisely quantifying the accuracy of classification algorithms as a function of problem variables and resolving the questions above. However, despite the rich literature on precise high-dimensional estimation and more recently

binary classification, multiclass classification is an under-explored venue possibly due to the difficulty of capturing the intricate dependencies between the classes even for relatively simple linear classifiers.

**Contributions.** We initiate a precise asymptotic study of linear multiclass classification in the modern high-dimensional regime, where the sizes of the training data and of the feature vectors grow large at a proportional rate. A key promise of such a precise analysis is that it allows us to accurately compare between different classification algorithms and data models. Compared to linear regression/binary classification, we identify the following crucial challenge: *the test accuracy in multiclass classification relies on intricate cross-correlations between the trained weights of the classifier*. This has two consequences that drive our analysis. First, in order to obtain sharp asymptotics on the test error of any classifier, it is a prerequisite to precisely quantify the asymptotics of these cross-correlations. Second, the test error does not depend on the correlations in closed-form expressions. Thus, to compare between different classifiers, we need efficient numerical and analytic means to evaluate the test error in terms of the correlation matrices. Interestingly, we show that these challenges are already present in simple classifiers, such as minimizing the square loss, and in stylized distributional settings, such as Gaussian features. Our contributions are as follows:

• We study two different data models: a Gaussian Mixtures Model (GMM) and a Multinomial Logit Model (MLM) with Gaussian features. For each one of them, we provide a precise characterization of total and class-wise test accuracy for three different training algorithms: (i) a least-squares (LS) based classifier, (ii) a weighted least-squares (WLS) based classifier, and (iii) a simple per class averaging (Avg) estimator. For the least-squares based classifiers, we develop a new technique to overcome the technical challenge of characterizing the limiting behavior of the weights' cross-correlations. For the per class averaging classifier, we show that it is Bayes optimal for a GMM with equal priors.

• We discuss efficient means of evaluating the test accuracy as a function of the weights' cross-correlations. This, together with the derived asymptotic formulae for the latter, lead to the first precise high-dimensional characterization of how the total/class-wise accuracy varies for different algorithms, data distributions, problem dimensions as well as number of classes, the inter/intra class correlations and class priors. For special problem geometries, we derive precise conditions on the data distribution and on the relative size of the training set over which each of the two studied algorithms dominates.

• We present and discuss numerical simulations that corroborate our theoretical findings. For instance, with an eye towards making classification algorithms more fair/equitable, we use our precise characterization of the class-wise accuracy to demonstrate how different algorithms behave in the presence of rare/minority classes. We also empirically compare the algorithms studied in this paper to other popular losses such as cross-entropy minimization. This allows us to better understand the performance of various algorithms in modern regimes of large number of classes.

**Related Work.** There is a classical body of algorithmic work on multiclass classification, e.g., [CS01, LLW04, WW98, BB99, DB94] and several empirical studies of their comparative performance [RK04, Für02, ASS00, PM05]. A more recent extension of this line of work investigates the effect of the loss function in deep neural networks [HYS16, GCOZ17, KS18, BEH20]. Algorithms for extreme multiclass problems with huge number of classes has also been studied in several [CAL13, YHR+16, DCO20, RCY+19, KMS15] works. On the theory front, numerous works have investigated consistency [Zha04, LLW04, TB07, PSG13, PS16] and finite-sample behavior [KP+02, Gue02, ASS00, LLY+18, CKMY16, LDBK15, Mau16, LDZK19] of multiclass classification algorithms. Our work differs from this literature in that we are interested in *precise* characterizations of the test accuracy rather than order-wise bounds. Here we focus on linear classifiers, but we consider the modern high-dimensional regime in which both the sample size and the features' dimension are large.

Specifically, our theoretical approach to linear multiclass classification fits in the rapidly growing literature on *sharp* high-dimensional asymptotics of convex optimization-based estimators [Don06, Sto09, OH10, CRPW12, ALMT13, DMM11, BM12, ALMT13, Sto13, OTH13, TOH15, Kar13, EK18, DM16, ORS17, TXH18, TAH18, MM18, WWM19, CM19, HL19, BKRS19, ASH19, JSH20]. Most of this line of work studies linear models and regression problems. More recently there has been a surge of interest in sharp analysis of a variety of methods tailored to binary classification models [TAH15, Hua17, CS18, SC19, MLC19b, MLC19a, KA19, SAH19, TPT20b, DKT19, MRSY19, LS20, MKLZ20, Lol20, TPT20a]. Nevertheless, none of these prior works have yet considered multiclass classification settings. Our paper unveils the salient features of the multiclass setting and shows that corresponding results from the binary setting do not directly apply here. We emphasize that this is the case even for seemingly simple one-vs-all (OVA) classifiers, such as

minimizing the square-loss, that involve training a single binary classifier per class [RK04]. The key technical tool behind our sharp analysis is the convex Gaussian min-max Theorem (CGMT) [TOH15, Sto13]. However, a "naive" application of the CGMT on the original optimization of the classifier does not allow us to compute all the necessary correleations between the classfier's weights to precisely capture the total/class-wise errors. Instead, our key idea is to formulate an artificial optimization problem, which captures the missing correlations and at the same time conveniently allows us to leverage the CGMT.

**Notation.** We use $[k]$ to denote $\{1, \ldots, k\}$. We use boldface lowercase letters $\boldsymbol{x}, \boldsymbol{y}, \boldsymbol{\mu}, \ldots$ to denote vectors and boldface uppercase letters $\boldsymbol{X}, \boldsymbol{Y}, \boldsymbol{M}, \ldots$ for matrices. We write $\boldsymbol{e}_\ell$ for the $\ell$-th standard basis vector in $\mathbb{R}^k$. We also write $\boldsymbol{I}_k, \boldsymbol{0}_{k \times k}$ and $\boldsymbol{1}_k$ for the $k \times k$ identity and all-zeros matrices and the $k \times 1$ all-ones vectors. For a vector $\boldsymbol{c} \in \mathbb{R}^k$ we write $\arg \max \boldsymbol{c}$ to denote the index of its largest entry, i.e., $\arg \max \boldsymbol{c} = \arg \max_{j \in [k]} \boldsymbol{c}_i$. The superscript $\dagger$ denotes pseudoinverse. We use $Q(x)$ for the tail of a standard Gaussian (Q-function). Finally, we reserve variables $G_0, G_1, \ldots, G_k \overset{iid}{\sim} \mathcal{N}(0, 1)$ to denote i.i.d. standard Gaussians.

## 2 Problem formulation

We focus on multiclass classification problems with $k$ classes. Specifically, we assume the training data consists of $n$ feature/label pairs $\{(\boldsymbol{x}_i, Y_i)\}_{i=1}^n$ with $\boldsymbol{x}_i \in \mathbb{R}^d$ representing the features and $Y_i \in \{1, 2, \ldots, k\}$ the associated labels representing one of $k$ classes. It will be convenient to also model the labels as one-hot encoded vectors $\boldsymbol{y}_i \in \mathbb{R}^k$ representing one of $k$ classes with one-hot encoding, i.e., $\boldsymbol{y}_i = \boldsymbol{e}_{Y_i}$. Therefore, when convenient we shall use $\{(\boldsymbol{x}_i, \boldsymbol{y}_i)\}_{i=1}^n$ to represent the training data. Throughout, we shall use $\boldsymbol{X} = \begin{bmatrix} \boldsymbol{x}_1 & \boldsymbol{x}_2 & \ldots & \boldsymbol{x}_n \end{bmatrix} \in \mathbb{R}^{d \times n}$, and $\boldsymbol{Y} = \begin{bmatrix} \boldsymbol{y}_1 & \boldsymbol{y}_2 & \ldots & \boldsymbol{y}_n \end{bmatrix} \in \mathbb{R}^{k \times n}$, to denote the matrix of features and their labels aggregated into a matrix, respectively. We shall also use $\boldsymbol{Y}_\ell \in \mathbb{R}^n$ to denote the $\ell$-th row of $\boldsymbol{Y}$. In our analysis we focus on training linear classifiers. Specifically, we use $\boldsymbol{W} = \begin{bmatrix} \boldsymbol{w}_1 & \boldsymbol{w}_2 & \cdots & \boldsymbol{w}_k \end{bmatrix}^T \in \mathbb{R}^{k \times d}$ and $\boldsymbol{b} \in \mathbb{R}^k$ to denote the weights and biases of this linear model, respectively. The overall input-output relationship of the classifier in this case is a function that maps an input vector $\boldsymbol{x} \in \mathbb{R}^d$ into an output of size $k$ via $\boldsymbol{x} \mapsto \boldsymbol{W}\boldsymbol{x} + \boldsymbol{b} \in \mathbb{R}^k$, where a training algorithm is used to train the corresponding weights $\boldsymbol{W} \in \mathbb{R}^{k \times d}$ and biases $\boldsymbol{b} \in \mathbb{R}^k$. Next we detail the data models and training algorithms that are formally studied in this paper. We end this section by discussing how the test error can be calculated for the different data models.

### 2.1 Data Models

In our theoretical analysis we assume the training data $\{(\boldsymbol{x}_i, Y_i)\}_{i=1}^n$ (alternatively $\{(\boldsymbol{x}_i, \boldsymbol{y}_i)\}_{i=1}^n$) are generated i.i.d. according to $(\boldsymbol{x}, Y)/(\boldsymbol{x}, \boldsymbol{y})$. We consider two models for the distribution of $(\boldsymbol{x}, \boldsymbol{y})$ which we detail next. In both models we shall use mean/regressor vectors $\{\boldsymbol{\mu}_\ell\}_{\ell=1}^k \in \mathbb{R}^d$ and aggregate them into columns of a matrix of the form $\boldsymbol{M} := \begin{bmatrix} \boldsymbol{\mu}_1 & \boldsymbol{\mu}_2 & \ldots & \boldsymbol{\mu}_k \end{bmatrix} \in \mathbb{R}^{d \times k}$. In the first model, these vectors represent the mean of the features conditioned on the class, i.e., $\boldsymbol{\mu}_\ell = \mathbb{E}\begin{bmatrix} \boldsymbol{x}|Y = \ell \end{bmatrix}$, whereas in the second model these vectors can be viewed as regressor coefficients. We shall refer to $\{\boldsymbol{\mu}_\ell\}_{\ell=1}^k/\boldsymbol{M}$ as "mean" vectors/matrix in both models. We denote the Grammian matrix of means as $\boldsymbol{\Sigma}_{\boldsymbol{\mu}, \boldsymbol{\mu}} = \boldsymbol{M}^T\boldsymbol{M}$. Furthermore, we shall use $\mu_\ell := \|\boldsymbol{\mu}_\ell\|_{\ell_2}$ to denote the norm of the mean vector $\boldsymbol{\mu}_\ell$.

**Gaussian Mixture Model (GMM).** In this model each example $(\boldsymbol{x}, Y)$ belongs to class $\ell \in [k]$ with probability $\pi_\ell$, i.e., $\mathbb{P}\{Y = \ell\} = \pi_\ell$. We let $\boldsymbol{\pi} = \begin{bmatrix} \pi_1 & \pi_2 & \ldots & \pi_k \end{bmatrix}^T \in \mathbb{R}^k$ denote the vector of priors which of course obeys $\boldsymbol{\pi} \geq \boldsymbol{0}$ and $\boldsymbol{1}^T\boldsymbol{\pi} = 1$. Also, we model the class conditional density of an example in class $\ell$ with an isotropic Gaussian centered at a mean vector $\boldsymbol{\mu}_\ell$. In particular, we say that a data point $(\boldsymbol{x}, Y)$ (or its one-hot encoded representation $(\boldsymbol{x}, \boldsymbol{y})$) follows the GMM model when

$$\mathbb{P}\{Y = \ell\} = \pi_\ell \qquad \text{and} \qquad \boldsymbol{x} = \boldsymbol{\mu}_Y + \boldsymbol{z}, \ \boldsymbol{z} \sim \mathcal{N}(\boldsymbol{0}, \sigma^2 \boldsymbol{I}_d). \tag{2.1}$$

We note that for a training set summarized by the feature and label matrices $\boldsymbol{X}$ and $\boldsymbol{Y}$ with columns generated i.i.d. according to the above distribution we have: $\boldsymbol{X} = \boldsymbol{M}\boldsymbol{Y} + \boldsymbol{Z}$ where $\boldsymbol{Z} \in \mathbb{R}^{d \times n}$ is a Gaussian noise matrix with i.i.d. $\mathcal{N}(0, \sigma^2)$ entries.

**Multinomial Logit Model (MLM).** In this model we assume that feature vectors $\boldsymbol{x}$ are distributed i.i.d. $\mathcal{N}(\boldsymbol{0}, \boldsymbol{I}_d)$ and that the conditional density of the class labels is given by the soft-max function. Concretely, we say that a data point $(\boldsymbol{x}, Y)$ (or its one-hot encoded representation $(\boldsymbol{x}, \boldsymbol{y})$) follows the multinomial logit model when

$$\boldsymbol{x} \sim \mathcal{N}(\boldsymbol{0}, \boldsymbol{I}_d) \qquad \text{and} \qquad \mathbb{P}\{Y = \ell \mid \boldsymbol{x}\} = e^{\langle \boldsymbol{\mu}_\ell, \boldsymbol{x} \rangle} \Big/ \sum_{j \in [k]} e^{\langle \boldsymbol{\mu}_j, \boldsymbol{x} \rangle}. \tag{2.2}$$

## 2.2 Classification algorithms

As mentioned earlier, in this paper we focus on training linear classifiers of the form $x \mapsto Wx + b$ with $W \in \mathbb{R}^{k \times d}$ denoting the weights and $b \in \mathbb{R}^k$ the offset values.

**Least-squares (LS).** In this approach we train a linear classifier $x \mapsto Wx + b$ via a least-squares fit to the training data: $(\widehat{W}, \widehat{b}) := \arg\min_{W,b} \frac{1}{2n} \sum_{i=1}^{n} \|Wx_i + b - y_i\|_{\ell_2}^2 = \frac{1}{2n} \|WX + b\mathbf{1}_n^T - Y\|_F^2$.

**Class averaging (Avg).** This approach uses the following weight and offset values $\widehat{W} := \frac{1}{n}YX^T$ and $\widehat{b} := \frac{1}{n}Y\mathbf{1}$. Let $n_\ell$ be the number of training data from class $\ell$ then, equivalently, $\widehat{w}_\ell = \frac{n_\ell}{n}\left(\frac{1}{n_\ell}\sum_{i: Y_i=\ell}^{n} x_i\right)$ and $\widehat{b}_\ell = \frac{n_\ell}{n}$. Therefore, this classifier picks weights according to the empirical mean of features of each class multiplied by the relative frequency of that class and the offset value as the fraction of data points from that class. We note that this algorithm has the same classification performance as the outcome of the ridge-regularized least-squares with infinite regularization.

**Weighted Least-squares (WLS).** This is a variation of the Least-squares approach where we fit a weighted least squares loss of the form $(\widehat{W}, \widehat{b}) := \arg\min_{W,b} \frac{1}{2n}\left\|\left(WX + b\mathbf{1}_n^T - Y\right)D\right\|_F^2$. Here, $D \in \mathbb{R}^{n \times n}$ is a diagonal matrix with the ith diagonal entry equal to $D_{ii} = \omega_\ell$ when the i-th data point is from class $\ell$ (i.e. $Y_i = \ell$) and $\omega_\ell \geq 0$, $\ell \in [k]$ denote the weights. Aggregating the weights into a vector of the form $\omega = \begin{bmatrix} \omega_1 & \omega_2 & \dots & \omega_k \end{bmatrix}^T \in \mathbb{R}^k$ we can rewrite $D$ in the form $D = \mathrm{diag}\left(Y^T\omega\right)$. In this approach the loss associated to data points to class $\ell$ is weighted by a factor $\omega_\ell^2$. For instance, if the class priors are known, a natural choice might be $\omega_\ell = 1/\sqrt{\pi_\ell}$. Such a weighted approach allows the classification algorithm to focus on rare/minority classes which are not well represented in the training data.

**Cross-entropy (CE).** In this approach the best weight/offset values are determined by fitting a cross entropy loss $(\widehat{W}, \widehat{b}) := \arg\min_{W,b} \frac{1}{n}\sum_{i=1}^{n} \log\left(\frac{\sum_{\ell=1}^{k} e^{\langle \widehat{w}_\ell, x_i \rangle + b_\ell}}{e^{\langle \widehat{w}_{Y_i}, x_i \rangle + b_{Y_i}}}\right)$. Theoretical analysis for CE is substantially more involved and we defer it to future work. Nevertheless, we compare with this classifier in our numerical simulations.

## 2.3 Class-wise and total test classification error

Let $\widehat{W}, \widehat{b}$ denote the parameters of a trained classifier. Now consider a fresh data sample $(x, Y)$ generated according to the same distribution as the training data. Once, we have learned the parameters $\widehat{W}, \widehat{b}$ of the classifier, the class $\widehat{Y}$ predicted by the classifier is made by a winner takes it all strategy, as follows, $\widehat{Y} = \arg\max_{j \in [k]} \langle \widehat{w}_j, x \rangle + \widehat{b}_j$. Therefore, the classification error condition on the the true label being $c$, which we shall refer to as the *class-wise test error*, is equal to

$$\mathbb{P}_{e|c} := \mathbb{P}\left\{\widehat{Y} \neq Y \mid Y = c\right\} = \mathbb{P}\left\{\langle \widehat{w}_c, x \rangle + \widehat{b}_c \leq \max_{j \neq c} \langle \widehat{w}_j, x \rangle + \widehat{b}_j\right\}. \tag{2.3}$$

Correspondingly, the *total classification error* is given by

$$\mathbb{P}_e := \mathbb{P}\left\{\widehat{Y} \neq Y\right\} = \mathbb{P}\left\{\arg\max_{j \in [k]}\{\langle \widehat{w}_j, x \rangle + \widehat{b}_j\} \neq Y\right\} = \mathbb{P}\left\{\langle \widehat{w}_Y, x \rangle + \widehat{b}_Y \leq \max_{j \neq Y} \langle \widehat{w}_j, x \rangle + \widehat{b}_j\right\}. \tag{2.4}$$

For both the GMM and MLM, the classification error depends on the vector of intercepts $\widehat{b} \in \mathbb{R}^k$ and the following key "correlation" matrices: $\Sigma_{w,w} := \widehat{W}\widehat{W}^T$ and $\Sigma_{w,\mu} := \widehat{W}M$.

**GMM.** In model (2.1), the test error probability is explicitly given by

$$\mathbb{P}_e = \mathbb{P}\left\{\arg\max \left(\sigma g + \widehat{b} + \Sigma_{w,\mu}e_Y\right) \neq Y\right\}, \quad \text{where } g \sim \mathcal{N}\left(0, \Sigma_{w,w}\right), \tag{2.5}$$

and $Y$ is independent of $g$ with probability mass function $\mathbb{P}\{Y = \ell\} = \pi_\ell, \ \ell \in [k]$.

**MLM.** In model (2.2), the test error probability is explicitly given by

$$\mathbb{P}_e = \mathbb{P}\left\{\arg\max \left(g + \widehat{b}\right) \neq Y(h)\right\}, \quad \text{where } \begin{bmatrix} g \\ h \end{bmatrix} \sim \mathcal{N}\left(0, \begin{bmatrix} \Sigma_{w,w} & \Sigma_{w,\mu} \\ \Sigma_{w,\mu}^T & \Sigma_{\mu,\mu} \end{bmatrix}\right), \tag{2.6}$$

and $\mathbb{P}\{Y(h) = \ell\} = e^{h_\ell}/\sum_{j \in [k]} e^{h_j}, \ \ell \in [k]$.

**Calculating the class-wise/total misclassifcation errors.** The identities (2.5) and (2.6) (see Section D.1 for a proof) as well as similar ones for the class-wise test error demonstrate that the total/class-wise errors only depend on the correlation matrices $\boldsymbol{\Sigma}_{\boldsymbol{w},\boldsymbol{w}}$ and $\boldsymbol{\Sigma}_{\boldsymbol{w},\boldsymbol{\mu}}$, the offset values $\widehat{\boldsymbol{b}}$ and the the class conditional means. For instance, as we show in the supplementary for GMM the class-wise errors are given by

$$\mathbb{P}_{e|c} = 1 - \mathbb{P}\left\{\boldsymbol{S}_c^{1/2}\,\boldsymbol{z} \geq \boldsymbol{t}_c\right\}, \tag{2.7}$$

where $\boldsymbol{z}$ is a Gaussian random vector distributed as $\mathcal{N}(\boldsymbol{0}, \sigma^2\boldsymbol{I}_{k-1})$, $\boldsymbol{S}_c \in \mathbb{R}^{(k-1)\times(k-1)}$ is a symmetric matrix such that its $i, j$ element is given by $[\boldsymbol{S}_c]_{ij} := \langle \widehat{\boldsymbol{w}}_c - \widehat{\boldsymbol{w}}_j, \widehat{\boldsymbol{w}}_c - \widehat{\boldsymbol{w}}_i \rangle$ and $\boldsymbol{t}_c \in \mathbb{R}^{k-1}$ a vector with entries $[\boldsymbol{t}_c]_i := \langle \widehat{\boldsymbol{w}}_i - \widehat{\boldsymbol{w}}_c, \boldsymbol{\mu}_c \rangle + (\widehat{\boldsymbol{b}}_i - \widehat{\boldsymbol{b}}_c)$. Similarly, based on (2.7) the total classification error in GMM is equal to $\mathbb{P}_e = \sum_{\ell=1}^k \pi_\ell \mathbb{P}_{e|c} = 1 - \sum_{\ell=1}^k \pi_\ell \mathbb{P}\left\{\boldsymbol{S}_c^{1/2}\,\boldsymbol{z} \geq \boldsymbol{t}_c\right\}$. As also detailed in the supplementary, the class-wise/total test errors for MLM similarly depends on quantities of the form $\mathbb{P}\{\boldsymbol{A}\boldsymbol{z} \geq \boldsymbol{t}\}$ with $\boldsymbol{z}$ a standard Gaussian random vector, $\boldsymbol{A}$ and $\boldsymbol{t}$ depending only on correlation matrices, conditional means and classifier offset-values; see Section D.3. There are a variety of algorithmic approaches to calculate $\mathbb{P}\{\boldsymbol{A}\boldsymbol{z} \geq \boldsymbol{t}\}$ once $\boldsymbol{A}$ and $\boldsymbol{t}$ are known based on Monte Carlo methods. Analytic bounds on this quantity have also been studied in the literature, e.g., [HH03, SL80]; see more details in Section D.

## 2.4 High-dimensional regime

This paper derives sharp asymptotic formulae for the class-wise and total classification error of averaging and (weighted) LS algorithms for GMM and MLM. We defer all our proofs to the appendix. All our results hold in the following high-dimensional regime with finite $k$.

**Assumption 1** *We focus on a double asymptotic regime where $n, d \to \infty$ at a fixed ratio $\gamma = d/n > 0$.*

For the (weighted) least-squares classifier, we focus here in the overdetermined regime $\gamma < 1$. However, our approach is also directly applicable to regularized (or min-norm) LS/WLS in the overparameterized regime $\gamma > 1$.

For a sequence of random variables $\mathcal{X}_{n,d}$ that converges in probability to some constant $c$ in the limit above, we simply write $\mathcal{X}_{n,d} \xrightarrow{P} c$. For a random vector/matrix $\boldsymbol{v}_{n,d}/\boldsymbol{V}_{n,d}$ and a deterministic vector/matrix $\boldsymbol{c}/\boldsymbol{C}$, the expressions $\boldsymbol{v}_{n,d} \xrightarrow{P} \boldsymbol{c}$ and $\boldsymbol{V}_{n,d} \xrightarrow{P} \boldsymbol{C}$ are to be understood entry-wise.

# 3 Results for Gaussian Mixture Model

In this section we discuss the asymptotics of the intercepts/correlation matrices for the averaging and the LS classifiers for the GMM. The derived formulas can be directly plugged in (2.5) and (2.7) to obtain asymptotics for the total and class-wise test error, respectively. We end this section by also characterizing the Bayes optimal estimator in this model when priors are balanced $\pi_\ell = 1/k, \ell \in [k]$. Additional results on the performance of Weighted LS are deferred to the appendix.

## 3.1 Class averaging classifier

**Proposition 3.1** *Consider data generated according to GMM in an asymptotic regime with any $\gamma > 0$. For the averaging estimator discussed in Section 2.2, the following high-dimensional limits hold*

$$\widehat{\boldsymbol{b}} \xrightarrow{P} \boldsymbol{\pi}, \quad \boldsymbol{\Sigma}_{\boldsymbol{w}\boldsymbol{\mu}} \xrightarrow{P} diag(\boldsymbol{\pi}) \cdot \boldsymbol{\Sigma}_{\boldsymbol{\mu},\boldsymbol{\mu}}, \tag{3.1a}$$

$$\boldsymbol{\Sigma}_{\boldsymbol{w},\boldsymbol{w}} \xrightarrow{P} \gamma\sigma^2 \cdot diag(\boldsymbol{\pi}) + diag(\boldsymbol{\pi}) \cdot \boldsymbol{\Sigma}_{\boldsymbol{\mu},\boldsymbol{\mu}} \cdot diag(\boldsymbol{\pi}). \tag{3.1b}$$

The above result allows us to precisely characterize the behavior of the averaging estimator in the high-dimensional regime. Let us consider a few special cases.

**Two classes.** Consider the special case with two classes with class priors $\pi_1 = 1 - \pi_2 =: \pi$. In this case we can compute the class-wise misclassification probabilities $\mathbb{P}_{e|1}$ and $\mathbb{P}_{e|2}$ explicitly. Specifically using (3.1), we have $\boldsymbol{S}_1 = \|\pi\boldsymbol{\mu}_1 - (1-\pi)\boldsymbol{\mu}_2\|_{\ell_2}^2 + \gamma\sigma^2$ and $t_1 = (1-2\pi) + (1-\pi)\langle\boldsymbol{\mu}_1,\boldsymbol{\mu}_2\rangle - \pi\|\boldsymbol{\mu}_1\|_{\ell_2}^2$.

Substituting the latter two in (2.7) we arrive at $\mathbb{P}_{e|1} \xrightarrow{P} Q\left(\frac{\pi\|\boldsymbol{\mu}_1\|_{\ell_2}^2 - (1-\pi)\langle\boldsymbol{\mu}_1,\boldsymbol{\mu}_2\rangle + 2\pi - 1}{\sqrt{\|\pi\boldsymbol{\mu}_1 - (1-\pi)\boldsymbol{\mu}_2\|_{\ell_2}^2 + \gamma\sigma^2}}\right)$. In the case of equal priors $\pi = \pi_1 = \pi_2 = 1/2$, antipodal and equal energy of the means, i.e., $\boldsymbol{\mu}_1 = -\boldsymbol{\mu}_2$ and $\mu := \|\boldsymbol{\mu}_1\|_{\ell_2} = \|\boldsymbol{\mu}_2\|_{\ell_2}$, we can use the above to conclude that $\mathbb{P}_{e|1} = \mathbb{P}_{e|2} = \frac{1}{2}\mathbb{P}_e = \frac{1}{2}Q\left(\sqrt{\frac{\mu^2}{\mu^2 + \gamma\sigma^2}}\right)$. This formula recovers the result of [MKLZ20] for this special case. Also, as mentioned in [MKLZ20], the formula matches the Bayes optimal error computed in [LM19] for Gaussian mean vectors. This shows that the class averaging method is Bayes optimal in this very simple setting. In Section 3.3, we generalize this result to multiple classes: we show that the average estimator is (asymptotically) Bayes optimal for balanced classes and equal-energy Gaussian means for any $k \geq 2$.

**Orthogonal means, equal priors and equal energy.** Next we focus on a special case with orthogonal means $\langle\boldsymbol{\mu}_i, \boldsymbol{\mu}_j\rangle = 0$, $i \neq j \in [k]$ of equal energy $\mu^2 := \|\boldsymbol{\mu}_i\|_{\ell_2}^2$ and of equal priors $\pi_i = \pi = 1/k$ for $i \in [k]$. In this case, the class-wise miss-classification error converges to $\mathbb{P}_{e|c} \xrightarrow{P} 1 - \mathbb{P}\{\boldsymbol{S}_c^{1/2}\boldsymbol{z} > \boldsymbol{t}\}$, where $\boldsymbol{S}_c = \pi(\pi\mu^2 + \gamma\sigma^2)(\boldsymbol{I}_{k-1} + \boldsymbol{1}_{k-1}\boldsymbol{1}_{k-1}^T)$ and $\boldsymbol{t} = -\pi\mu^2\boldsymbol{1}_{k-1}$. Defining $u_{\text{Avg}} := \frac{\mu^2}{\sigma}\sqrt{\frac{1}{\mu^2 + k\gamma\sigma^2}}$, after some algebraic manipulations the total classification error of the averaging estimator in this case is given by $\mathbb{P}_{e|c} = \mathbb{P}_{e,\text{Avg}} \xrightarrow{P} \mathbb{P}\left\{G_0 + \max_{j \in [k-1]} G_j \geq u_{\text{Avg}}\right\}$, where $G_0, \ldots, G_{k-1} \overset{iid}{\sim} \mathcal{N}(0,1)$.

## 3.2 Least-squares classifier

This section focuses on characterizing the intercepts and correlation matrices for the least-squares classifier. To present our results, we assume that the Grammian matrix has eigenvalue decomposition

$$\boldsymbol{\Sigma}_{\boldsymbol{\mu},\boldsymbol{\mu}} = \boldsymbol{M}^T\boldsymbol{M} = \boldsymbol{V}\boldsymbol{\Sigma}^2\boldsymbol{V}^T, \qquad \boldsymbol{\Sigma} > \boldsymbol{0}_{r\times r}, \ \boldsymbol{V} \in \mathbb{R}^{k\times r}, \ r \leq k. \tag{3.2}$$

with $\boldsymbol{\Sigma}$ a diagonal positive-definite matrix and $\boldsymbol{V}$ an orthonormal matrix obeying $\boldsymbol{V}^T\boldsymbol{V} = \boldsymbol{I}_r$.

**Theorem 3.2** *Consider data generated according to GMM in an asymptotic regime with $\gamma < 1$. In addition to (3.2), define the following two positive (semi)-definite matrices: $\boldsymbol{P} := diag(\boldsymbol{\pi}) - \boldsymbol{\pi}\boldsymbol{\pi}^T \succeq \boldsymbol{0}_{k\times k}$ and $\boldsymbol{\Delta} := \sigma^2\boldsymbol{I}_r + \boldsymbol{\Sigma}\boldsymbol{V}^T\boldsymbol{P}\boldsymbol{V}\boldsymbol{\Sigma} > \boldsymbol{0}_{r\times r}$. Then, for the least-squares linear classifier $(\widehat{\boldsymbol{W}}, \widehat{\boldsymbol{b}})$ the following limits are true asymptotically*

$$\widehat{\boldsymbol{b}} \xrightarrow{P} \boldsymbol{\pi} - \boldsymbol{P}\boldsymbol{V}\boldsymbol{\Sigma}\boldsymbol{\Delta}^{-1}\boldsymbol{\Sigma}\boldsymbol{V}^T\boldsymbol{\pi}, \quad \boldsymbol{\Sigma}_{\boldsymbol{w},\boldsymbol{\mu}} \xrightarrow{P} \boldsymbol{P}\boldsymbol{V}\boldsymbol{\Sigma}\boldsymbol{\Delta}^{-1}\boldsymbol{\Sigma}\boldsymbol{V}^T, \tag{3.3a}$$

$$\boldsymbol{\Sigma}_{\boldsymbol{w},\boldsymbol{w}} \xrightarrow{P} \frac{\gamma}{(1-\gamma)\sigma^2}\boldsymbol{P} + \boldsymbol{P}\boldsymbol{V}\boldsymbol{\Sigma}\boldsymbol{\Delta}^{-1}\left(\boldsymbol{\Delta}^{-1} - \frac{\gamma}{(1-\gamma)\sigma^2}\boldsymbol{I}_r\right)\boldsymbol{\Sigma}\boldsymbol{V}^T\boldsymbol{P}. \tag{3.3b}$$

The above result allows us to precisely characterize the behavior of the least-squares classifier in the high-dimensional regime. In Section H.2, we specialize (3.3) to the case of orthogonal means. Compared to the weight vectors $\widehat{\boldsymbol{w}}_i, i \in [k]$ of the class averaging classifier that are also (asymptotically) orthogonal when means are orthogonal, this is *not* the case for LS. We show next that these spurious correlations only hurt the classification error when classes are balanced.

**Proposition 3.3** *Consider the case of orthogonal, equal energy-means $\boldsymbol{\Sigma}_{\boldsymbol{\mu},\boldsymbol{\mu}} = \mu\boldsymbol{I}_k$, balanced priors $\pi_i = 1/k$, $i \in [k]$ and $\gamma < 1$. Setting $u_{\text{LS}} := \frac{\mu^2}{\sigma}\sqrt{\frac{1-\gamma}{\mu^2 + k\gamma\sigma^2}}$, it holds that $\mathbb{P}_{e,\text{LS}} \xrightarrow{P} \mathbb{P}\{G_0 + \max_{j \in [k-1]} G_j \geq u_{\text{LS}}\}$. Specifically, since $u_{\text{LS}} = u_{\text{Avg}}\sqrt{1-\gamma} < u_{\text{Avg}}$, the averaging estimator strictly outperforms LS for all $0 < \gamma < 1$ and $k \geq 2$ in this setting.*

## 3.3 Bayes estimator for the balanced Gaussian Mixture Model

To check how far the above algorithms are from the lowest misclassification error achievable by any algorithm in this section, we consider a Bayesian setting with Gaussian mean vectors and we derive the Bayes-optimal risk for the case of equal priors. Recall that the Bayes estimator $\hat{Y} = \arg\max_{\ell \in [k]} \mathbb{P}\{Y = \ell \mid \boldsymbol{X}, \boldsymbol{Y}, \boldsymbol{x}\}$ minimizes the risk $\mathbb{P}_e = \mathbb{P}\{\hat{Y} \neq Y\} = \mathbb{E}_{\boldsymbol{X},\boldsymbol{Y},\boldsymbol{x},Y}[\mathbb{1}[\hat{Y} \neq Y]]$.

**Proposition 3.4** *Consider $\boldsymbol{\mu}_i \overset{iid}{\sim} \mathcal{N}(\boldsymbol{0}, \frac{\mu^2}{d}\text{I}_d)$ and $\pi_i = 1/k$ for all $i \in [k]$. Set $u_{\text{Bayes}} := \frac{\mu^2}{\sigma}\frac{1}{\sqrt{\mu^2 + k\gamma\sigma^2}}$. Then, the Bayes risk converges to $\mathbb{P}\left\{G_0 + \max_{\ell \in [k-1]} G_\ell \geq u_{\text{Bayes}}\right\}$.*

Under Gaussian prior, the means are asymptotically orthogonal and equal-energy. As shown earlier, in this setting, $\mathbb{P}_{e,\mathrm{Avg}} \xrightarrow{P} \mathbb{P}\left\{ G_0 + \max_{\ell \in [k-1]} G_\ell \geq u_{\mathrm{Avg}} \right\}$. But, $u_{\mathrm{Avg}} = u_{\mathrm{Bayes}}$. Thus, the averaging method is (asymptotically) Bayes optimal for equal-norm, orthogonal means and balanced classes. An analogous result was' derived in [LM19, MKLZ20], but only for binary classification.

# 4    Results for Multinomial Logit Model

In this section we discuss the asymptotics of the intercepts/correlation matrices for MLM. We present results for arbitrary mean-vectors as well as special cases where the means are mutually orthogonal. Recall the eigenvalue decomposition of the Grammian $\boldsymbol{\Sigma}_{\boldsymbol{\mu},\boldsymbol{\mu}} = \boldsymbol{V}\boldsymbol{\Sigma}^2\boldsymbol{V}^T$ in (3.2). In order to state our results, it is convenient to introduce the following probability vectors in $\mathbb{R}^k$ and $\mathbb{R}^{k^2}$:

$$\boldsymbol{\pi} := \mathbb{E}\left[ \frac{e^{\boldsymbol{V}\boldsymbol{\Sigma}\boldsymbol{g}}}{\mathbf{1}_k^T e^{\boldsymbol{V}\boldsymbol{\Sigma}\boldsymbol{g}}} \right] \in \mathbb{R}^k \ \ \text{and} \ \ \boldsymbol{\Pi} := \mathbb{E}\left[ \frac{\left(e^{\boldsymbol{V}\boldsymbol{\Sigma}\boldsymbol{g}}\right)\left(e^{\boldsymbol{V}\boldsymbol{\Sigma}\boldsymbol{g}}\right)^T}{\left(\mathbf{1}_k^T e^{\boldsymbol{V}\boldsymbol{\Sigma}\boldsymbol{g}}\right)^2} \right] \in \mathbb{R}^{k \times k}, \ \ \text{where } \boldsymbol{g} \sim \mathcal{N}(\mathbf{0}, \boldsymbol{I}_r). \quad (4.1)$$

Note that $\boldsymbol{\pi}$ and $\boldsymbol{\Pi}$ are the first and second moments of the soft-max mapping of $\boldsymbol{V}\boldsymbol{\Sigma}\boldsymbol{g} \sim \mathcal{N}(\mathbf{0}, \boldsymbol{\Sigma}_{\boldsymbol{\mu},\boldsymbol{\mu}})$. In fact, for the MLM in (2.2) it holds that $\mathbb{P}\{Y = \ell\} = \mathbb{E}[\mathbb{P}\{Y = \ell \,|\, \boldsymbol{x}\}] = \mathbb{E}[\frac{e^{\boldsymbol{e}_\ell^T \boldsymbol{V}\boldsymbol{\Sigma}\boldsymbol{g}}}{\mathbf{1}_k^T e^{\boldsymbol{V}\boldsymbol{\Sigma}\boldsymbol{g}}}] = \boldsymbol{\pi}_\ell, \ \ell \in [k]$ since $\boldsymbol{M}^T\boldsymbol{x}$ is distributed as $\boldsymbol{V}\boldsymbol{\Sigma}\boldsymbol{g}$. Thus, $\boldsymbol{\pi}$ is the vector of class priors (which explains the slight abuse of notation here in relation to our notation for the class priors of the GMM).

## 4.1    Class averaging classifier

**Proposition 4.1** *Consider data generated according to MLM in an asymptotic regime with any $\gamma > 0$. For the averaging classifier, the following high-dimensional limits hold*

$$\widehat{\boldsymbol{b}} \xrightarrow{P} \boldsymbol{\pi}\,, \quad \boldsymbol{\Sigma}_{\boldsymbol{w},\boldsymbol{\mu}} \xrightarrow{P} (diag(\boldsymbol{\pi}) - \boldsymbol{\Pi}) \cdot \boldsymbol{\Sigma}_{\boldsymbol{\mu},\boldsymbol{\mu}}\,, \quad (4.2a)$$

$$\boldsymbol{\Sigma}_{\boldsymbol{w},\boldsymbol{w}} \xrightarrow{P} \gamma \cdot diag(\boldsymbol{\pi}) + (diag(\boldsymbol{\pi}) - \boldsymbol{\Pi}) \, \boldsymbol{\Sigma}_{\boldsymbol{\mu},\boldsymbol{\mu}} \, (diag(\boldsymbol{\pi}) - \boldsymbol{\Pi})\,. \quad (4.2b)$$

Using Gaussian decomposition in (2.6) and checking from (4.2) that $\boldsymbol{\Sigma}_{\boldsymbol{w},\boldsymbol{w}} - \boldsymbol{\Sigma}_{\boldsymbol{w},\boldsymbol{\mu}}\boldsymbol{\Sigma}_{\boldsymbol{\mu},\boldsymbol{\mu}}^\dagger\boldsymbol{\Sigma}_{\boldsymbol{w},\boldsymbol{\mu}}^T \xrightarrow{P} \gamma \cdot \mathrm{diag}(\boldsymbol{\pi})$ the test error obtains the following explicit form:

$$\mathbb{P}_{e,\mathrm{Avg}} \xrightarrow{P} \mathbb{P}\left\{ \arg\max\left\{ \sqrt{\gamma} \cdot \mathrm{diag}(\sqrt{\boldsymbol{\pi}}) \cdot \widetilde{\boldsymbol{g}} + (\mathrm{diag}(\boldsymbol{\pi}) - \boldsymbol{\Pi}) \cdot \boldsymbol{V}\boldsymbol{\Sigma} \cdot \boldsymbol{g} + \boldsymbol{\pi} \right\} \neq Y(\boldsymbol{g}) \right\}, \quad (4.3)$$

where $\widetilde{\boldsymbol{g}} \sim \mathcal{N}(\mathbf{0}, \boldsymbol{I}_k)$, $\boldsymbol{g} \sim \mathcal{N}(\mathbf{0}, \boldsymbol{I}_r)$ and $\mathbb{P}\{Y(\boldsymbol{g}) = c\} = e^{\boldsymbol{e}_c^T \boldsymbol{V}\boldsymbol{\Sigma}\boldsymbol{g}} \big/ \sum_{j \in [k]} e^{\boldsymbol{e}_j^T \boldsymbol{V}\boldsymbol{\Sigma}\boldsymbol{g}}, c \in [k]$.

## 4.2    Least-squares classifier

This section focuses on characterizing the intercepts and correlation matrices for the least-squares classifier. We also use the result to characterize conditions under which LS outperforms averaging.

**Theorem 4.2** *Consider data generated according to MLM in an asymptotic regime with $0 < \gamma < 1$. Recall the notation in (4.1). For the LS classifier, the following high-dimensional limits hold.*

$$\widehat{\boldsymbol{b}} \xrightarrow{P} \boldsymbol{\pi}\,, \quad \boldsymbol{\Sigma}_{\boldsymbol{w},\boldsymbol{\mu}} \xrightarrow{P} (diag(\boldsymbol{\pi}) - \boldsymbol{\Pi}) \cdot \boldsymbol{\Sigma}_{\boldsymbol{\mu},\boldsymbol{\mu}}\,, \quad (4.4a)$$

$$\boldsymbol{\Sigma}_{\boldsymbol{w},\boldsymbol{w}} \xrightarrow{P} \frac{\gamma}{1-\gamma} \cdot \left( diag(\boldsymbol{\pi}) - \boldsymbol{\pi}\boldsymbol{\pi}^T \right) + \frac{1-2\gamma}{1-\gamma} \cdot (diag(\boldsymbol{\pi}) - \boldsymbol{\Pi}) \cdot \boldsymbol{\Sigma}_{\boldsymbol{\mu},\boldsymbol{\mu}} \cdot (diag(\boldsymbol{\pi}) - \boldsymbol{\Pi})\,. \quad (4.4b)$$

It is interesting to observe that (4.4a) is identical to (4.2a). However, the cross-correlations in $\boldsymbol{\Sigma}_{\boldsymbol{w},\boldsymbol{w}}$ differ. We prove below that this leads to an improved performance of the LS classifier for large sample sizes. First, Theorem 4.2 can be used to check that

$$\boldsymbol{\Sigma}_{\boldsymbol{w},\boldsymbol{w}} - \boldsymbol{\Sigma}_{\boldsymbol{w},\boldsymbol{\mu}}\boldsymbol{\Sigma}_{\boldsymbol{\mu},\boldsymbol{\mu}}^\dagger\boldsymbol{\Sigma}_{\boldsymbol{w},\boldsymbol{\mu}}^T \xrightarrow{P} \frac{\gamma}{1-\gamma}\left( \mathrm{diag}(\boldsymbol{\pi}) - \boldsymbol{\pi}\boldsymbol{\pi}^T - (\mathrm{diag}(\boldsymbol{\pi}) - \boldsymbol{\Pi})\,\boldsymbol{\Sigma}_{\boldsymbol{\mu},\boldsymbol{\mu}}\,(\mathrm{diag}(\boldsymbol{\pi}) - \boldsymbol{\Pi}) \right).$$

Thus, the only change in the test-error formula compared to (4.3) is the term $\gamma \cdot \mathrm{diag}(\boldsymbol{\pi})$ substituted by the matrix above.

**Proposition 4.3** *Assume orthogonal, equal-energy means* $\boldsymbol{\Sigma}_{\boldsymbol{\mu},\boldsymbol{\mu}} = \mu^2 \boldsymbol{I}_k$, $k \geq 2$. *Let* $\gamma_\star = \frac{\mu^2 k}{(k-1)^2} \Big( 1 - k \, \mathbb{E}\Big[ \frac{e^{2\mu G_1}}{(\sum_{\ell \in [k]} e^{\mu G_\ell})^2} \Big] \Big)^2 \in (0,1)$. *Then, with probability 1 as* $n \to \infty$, $\mathbb{P}_{e,\mathrm{LS}} < \mathbb{P}_{e,\mathrm{Avg}} \iff \gamma < \gamma_\star$.

## 5 Numerical Results

This section validates our theory via numerical experiments and provides further insights on multiclass classification. See also Section A for more extensive experiments. We study the class-wise/total test misclassification error in both GMM and MLM for different sample sizes, number of classes and class priors. In line with Section 2.2 we consider four algorithms: (i) Averaging (Avg), (ii) LS, (iii) Weighted LS (WLS) with the $i$th class weighted by $\omega_\ell^2 = 1/\pi_\ell$, (iv) Cross-Entropy (CE).

Figures 1 and 2 focus on GMM with $k = 9$ classes, $d = 300$ and $\|\boldsymbol{\mu}_i\|_{\ell_2}^2 = 15$. To model different class prior probabilities, we use the distribution $\pi_1 = \pi_2 = \pi_3 = 0.5, \pi_4 = 0.5, \pi_5 = 0.5, \pi_6 = 0.25, \pi_7 = 0.25, \pi_8 = 0.25, \pi_9 = 1/21$. We consider three scenarios: (a) orthogonal means, equal prior ($\pi_i = 1/9$); (b) orthogonal means, different prior; (c) correlated means with pairwise correlation coefficient equal to 0.5 (i.e., $\langle \boldsymbol{\mu}_i, \boldsymbol{\mu}_j \rangle / (\|\boldsymbol{\mu}_i\|_{\ell_2} \|\boldsymbol{\mu}_j\|_{\ell_2}) = 0.5$ for $i \neq j$) and different priors as discussed above. Figure 1 shows the test miss-classification errors as a function of $\gamma := d/n$. In all scenarios our theoretical predictions are a near perfect match to the empirical performance. In scenario (a), class-wise averaging achieves the lowest error as predicted by Proposition 3.4. However, in scenario (b) where the means have different norms the averaging method has higher misclassification error compared with CE, LS and WLS for large sample sizes (small $\gamma$). We note that both LS and WLS achieve lower errors compared with CE as the sample size grows. Scenario (c) is similar to (b). However, due to class correlations, the errors are uniformly higher. Figure 2 shows the corresponding class-wise miss-classification errors for the smallest $\gamma$ in Figure 1 ($\gamma = 0.117$). In scenario (a), errors are equal which is expected given the equal class priors. In scenarios (b) and (c) however, due to different priors, large classes 7,8,9 achieve best accuracy. The performance difference is most visible for the averaging approach. LS mitigates this issue to some extent, while WLS creates the flattest class-wise errors suggesting that it can reduce the miss-classification error on small/minority classes.

Figure 3 focuses on orthogonal classes with varying number of classes $k$ where $\|\boldsymbol{\mu}_i\|_{\ell_2}^2 = 15$ and $d \in \{50, 100, 200\}$ with $kd/n = k\gamma$ fixed at $k\gamma = 20/11$. It plots the ratio of the empirical error probability and our theoretical prediction as $k$ grows until $k = d$. Two observations are worth mentioning here. (1) The accuracy of our predictions noticeably improves as the problem dimension $d, n$ grow as expected given the asymptotic nature of our analysis. Interestingly, the convergence appears to be noticeably faster (as a function of $d$) for the LS rather than the Averaging classifier. (2) Our theoretical results formally require that $k$ is fixed while $d$ (and $n$) grow large. Yet, the presented experimental results suggest that they might also hold for large $k$ under the shown scaling. This is a fascinating research question that we believe is worth investigating further.

Figure 4 provides experiments on MLM with $k = 9$ orthogonal classes. Unlike GMM, CE achieves the best performance in MLM. In Figure 4 (a), classes have same norms $\|\mu_i\|_{\ell_2} = 10$, while in Figure 4 (b) we have quadrupled the norms of classes 7,8,9 and doubled the norms of classes 4,5,6. This disparity between the norms seems to help improve the CE accuracy, but hurt LS/averaging accuracy for small $\gamma$. Finally, Figure 4 (c) shows the class-wise probability of error associated with (b) for $\gamma = 0.117$ and demonstrates that LS outperforms averaging.

## 6 Future Directions

This work aims at initiating a precise asymptotic study of multiclass classifiers that provides a promising setting for resolving a rich set of open questions regarding the (comparative) performance of classification algorithms as a function of the involved problem variables. As mentioned, even understanding the statistical performance of one-vs-all multiclass classifiers does not follow directly from the existing literature on binary classifiers. Extending the results of this paper to the one-vs-all logistic and SVM classifiers would allow for a principled comparison among these different choices. A possibly more challenging, albeit mathematically intriguing and practically relevant task, is characterizing the asymptotics of more complicated (non-separable) losses, such as the cross-entropy loss. For this, even characterizing the asymptotic behavior of the correlations $\boldsymbol{\Sigma}_{\boldsymbol{w},\boldsymbol{\mu}}$

requires new ideas. The previously mentioned study of "extreme multiclass classification" in which the number of classes $k$ is very large is another fascinating direction.

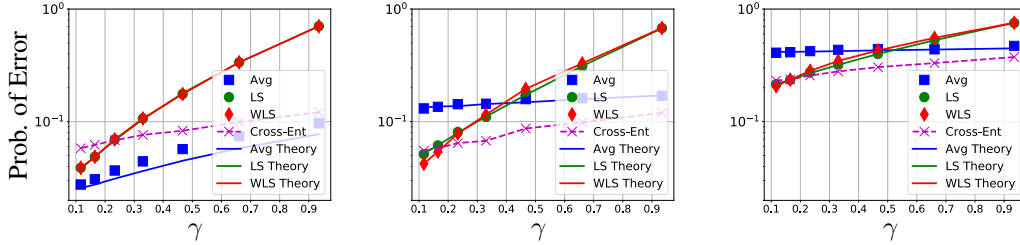

Figure 1: GMM with $k = 9, d = 300$. (a) orthogonal, equal prior, (b) orthogonal, different prior, (c) correlated, different prior.

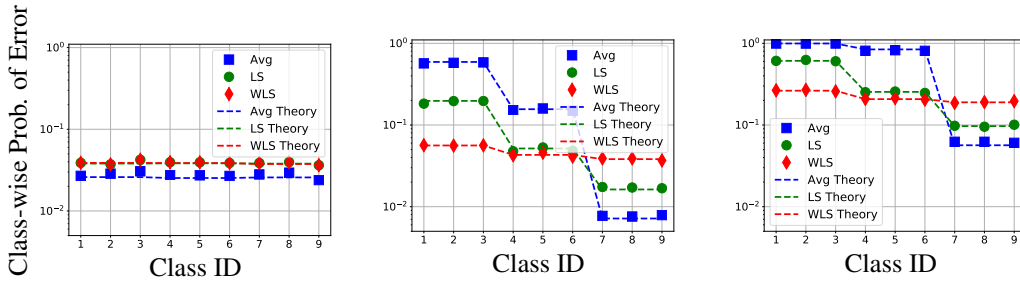

Figure 2: Class-wise probability of errors corresponding to Figure 1 with $\gamma = 0.117$.

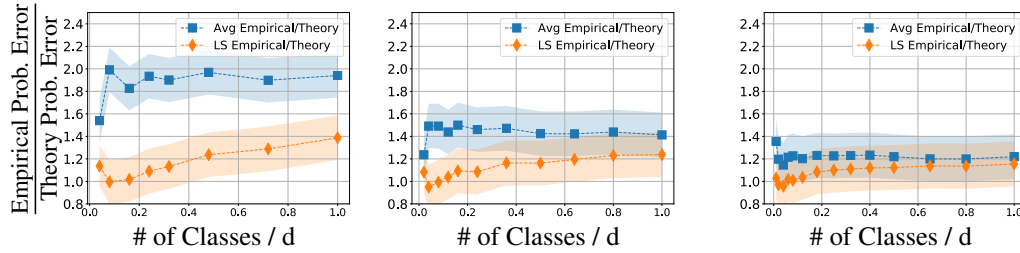

Figure 3: GMM, $K/d$ is varied from 0 to 1 while keeping $K\gamma$ constant for (a) $d = 50$, (b) $d = 100$, (c) $d = 200$.

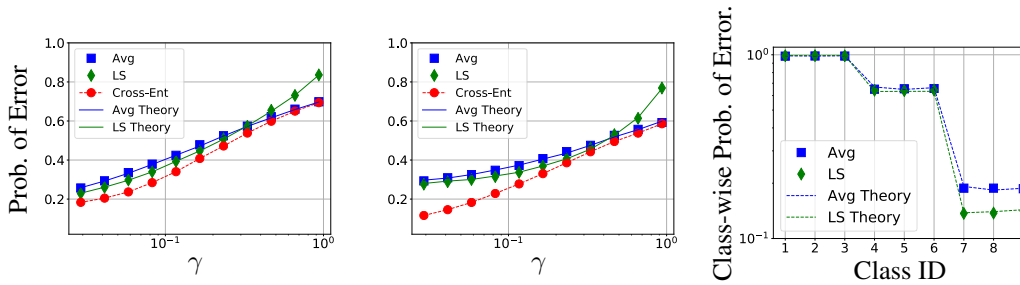

Figure 4: MLM with orthogonal means for (a) equal norms and (b) different norms. (c) Class-wise probability errors for (b).

## Acknowledgments

C. Thrampoulidis was partially supported by the NSF under Grant Numbers CCF-2009030 and HDR-1934641. S. Oymak is partially supported by the NSF award CNS-1932254. M. Soltanolkotabi is supported by the Packard Fellowship in Science and Engineering, a Sloan Research Fellowship in Mathematics, an NSF-CAREER under award #1846369, the Air Force Office of Scientific Research Young Investigator Program (AFOSR-YIP) under award #FA9550 − 18 − 1 − 0078, DARPA Learning with Less Labels (LwLL) and FastNICS programs, and NSF-CIF awards #1813877 and #2008443.

## Broader Impact

In this paper we develop a precise and asymptotically exact understanding of the statistical behavior of a variety of classification algorithms. In particular we precisely, characterize how the total and class-wise accuracy varies under different training algorithms, data distributions, problem dimensions, inter/intra class correlations and class priors. Despite being theoretical/foundational in nature it has potential for broader practical impact. In particular, our precise characterization of class-wise accuracy allows us to understand how different training algorithms impact accuracy of machine learning algorithms on rare/minority classes. Such a precise understanding may help guide the development of more fair/equitable algorithms. On the flip side, such insights may potentially also be used nefariously enabling the marginalization of rare/minority classes by developing algorithms that reduce their class-wise accuracy.

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
