[Supplementary Material]

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

{\boldsymbol{W}}, \widehat{\boldsymbol{b}}) := \arg\min_{\boldsymbol{W}, \boldsymbol{b}} \frac{1}{2n} \sum_{i=1}^{n} \left\| \boldsymbol{W}\boldsymbol{x}_i + \boldsymbol{b} - \boldsymbol{y}_i \right\|_{\ell_2}^2 = \frac{1}{2n} \left\| \boldsymbol{W}\boldsymbol{X} + \boldsymbol{b}\boldsymbol{1}_n^T - \boldsymbol{Y} \right\|_F^2.$$

**Class averaging (Avg).** This approach uses the following weight and offset values

$$\widehat{\boldsymbol{W}} := \frac{1}{n} \boldsymbol{Y}\boldsymbol{X}^T \quad \text{and} \quad \widehat{\boldsymbol{b}} := \frac{1}{n} \boldsymbol{Y}\boldsymbol{1}.$$

Let $n_\ell$ be the number of training data from class $\ell$ then, equivalently, $\widehat{\boldsymbol{w}}_\ell = \frac{n_\ell}{n} \left( \frac{1}{n_\ell} \sum_{i:\, Y_i = \ell}^{n} \boldsymbol{x}_i \right)$ and $\widehat{\boldsymbol{b}}_\ell = \frac{n_\ell}{n}$. Therefore, this classifier picks weights according to the empirical mean of features of each class multiplied by the relative frequency of that class and the offset value as the fraction of data points from that class. We note that this algorithm has the same classification performance as the outcome of the ridge-regularized least-squares with infinite regularization.

**Weighted Least-squares (WLS).** This is a variation of the Least-squares approach where we fit a weighted least squares loss of the form

$$(\widehat{\boldsymbol{W}}, \widehat{\boldsymbol{b}}) := \arg\min_{\boldsymbol{W}, \boldsymbol{b}} \frac{1}{2n} \left\| \left( \boldsymbol{W}\boldsymbol{X} + \boldsymbol{b}\boldsymbol{1}_n^T - \boldsymbol{Y} \right) \boldsymbol{D} \right\|_F^2.$$

Here, $\boldsymbol{D} \in \mathbb{R}^{n \times n}$ is a diagonal matrix with the ith diagonal entry equal to $D_{ii} = \omega_\ell$ when the i-th data point is from class $\ell$ (i.e. $Y_i = \ell$) and $\omega_\ell \geq 0$, $\ell \in [k]$ denote the weights. Aggregating the weights into a vector of the form $\boldsymbol{\omega} = \begin{bmatrix} \omega_1 & \omega_2 & \dots & \omega_k \end{bmatrix}^T \in \mathbb{R}^k$ we can rewrite $\boldsymbol{D}$ in the form

$$\boldsymbol{D} = \text{diag}\left( \boldsymbol{Y}^T \boldsymbol{\omega} \right).$$

In this approach the loss associated to data points to class $\ell$ is weighted by a factor $\omega_\ell^2$. For instance, if the class priors are known, a natural choice might be $\omega_\ell = 1/\sqrt{\pi_\ell}$. Such a weighted approach allows the classification algorithm to focus on rare/minority classes which are not well represented in the training data.

**Cross-entropy (CE).** In this approach the best weight/offset values are determined by fitting a cross entropy loss $(\widehat{\boldsymbol{W}}, \widehat{\boldsymbol{b}}) := \arg\min_{\boldsymbol{W}, \boldsymbol{b}} \frac{1}{n} \sum_{i=1}^{n} \log\left( \frac{\sum_{\ell=1}^{k} e^{\langle \widehat{\boldsymbol{w}}_\ell, \boldsymbol{x}_i \rangle + b_\ell}}{e^{\langle \widehat{\boldsymbol{w}}_{Y_i}, \boldsymbol{x}_i \rangle + b_{Y_i}}} \right)$. Theoretical analysis for CE is substantially more involved and we defer it to future work. Nevertheless, we compare with this classifier in our numerical simulations.

## 2.3 Class-wise and total test classification error

Let $\widehat{\boldsymbol{W}}, \widehat{\boldsymbol{b}}$ denote the parameters of a trained classifier. Now consider a fresh data sample $(\boldsymbol{x}, Y)$ generated according to the same distribution as the training data. Once, we have learned the parameters $\widehat{\boldsymbol{W}}, \widehat{\boldsymbol{b}}$ of the classifier, the class $\widehat{Y}$ predicted by the classifier is made by a winner takes it all strategy, as follows, $\widehat{Y} = \arg\max_{j \in [k]} \langle \widehat{\boldsymbol{w}}_j, \boldsymbol{x} \rangle + \widehat{\boldsymbol{b}}_j$. Therefore, the classification error condition on the the true label being $c$, which we shall refer to as the *class-wise test error*, is equal to

$$\mathbb{P}_{e|c} := \mathbb{P}\left\{ \widehat{Y} \neq Y \big| Y = c \right\} = \mathbb{P}\left\{ \langle \widehat{\boldsymbol{w}}_c, \boldsymbol{x} \rangle + \widehat{\boldsymbol{b}}_c \leq \max_{j \neq c} \langle \widehat{\boldsymbol{w}}_j, \boldsymbol{x} \rangle + \widehat{\boldsymbol{b}}_j \right\}. \tag{2.3}$$

Correspondingly, the *total classification error* is given by

$$\mathbb{P}_e := \mathbb{P}\left\{\widehat{Y} \neq Y\right\} = \mathbb{P}\left\{\arg\max_{j\in[k]}\{\langle\widehat{\boldsymbol{w}}_j,\boldsymbol{x}\rangle + \widehat{\boldsymbol{b}}_j\} \neq Y\right\} = \mathbb{P}\left\{\langle\widehat{\boldsymbol{w}}_Y,\boldsymbol{x}\rangle + \widehat{\boldsymbol{b}}_Y \leq \max_{j\neq Y}\langle\widehat{\boldsymbol{w}}_j,\boldsymbol{x}\rangle + \widehat{\boldsymbol{b}}_j\right\}. \quad (2.4)$$

For both the GMM and MLM, the classification error depends on the vector of intercepts $\widehat{\boldsymbol{b}} \in \mathbb{R}^k$ and the following key "correlation" matrices:

$$\boldsymbol{\Sigma}_{\boldsymbol{w},\boldsymbol{w}} := \widehat{\boldsymbol{W}}\widehat{\boldsymbol{W}}^T \quad \text{and} \quad \boldsymbol{\Sigma}_{\boldsymbol{w},\boldsymbol{\mu}} := \widehat{\boldsymbol{W}}\boldsymbol{M}.$$

**GMM.** In model (2.1), the test error probability is explicitly given by

$$\mathbb{P}_e = \mathbb{P}\left\{\arg\max\;(\sigma\,\boldsymbol{g} + \widehat{\boldsymbol{b}} + \boldsymbol{\Sigma}_{\boldsymbol{w},\boldsymbol{\mu}}\boldsymbol{e}_Y) \neq Y\right\}, \quad \text{where } \boldsymbol{g} \sim \mathcal{N}\left(\boldsymbol{0},\boldsymbol{\Sigma}_{\boldsymbol{w},\boldsymbol{w}}\right), \quad (2.5)$$

and $Y$ is independent of $\boldsymbol{g}$ with probability mass function $\mathbb{P}\{Y = \ell\} = \pi_\ell, \;\; \ell \in [k]$.

**MLM.** In model (2.2), the test error probability is explicitly given by

$$\mathbb{P}_e = \mathbb{P}\left\{\arg\max\;(\boldsymbol{g} + \widehat{\boldsymbol{b}}) \neq Y(\boldsymbol{h})\right\}, \quad \text{where } \begin{bmatrix}\boldsymbol{g}\\\boldsymbol{h}\end{bmatrix} \sim \mathcal{N}\left(\boldsymbol{0}, \begin{bmatrix}\boldsymbol{\Sigma}_{\boldsymbol{w},\boldsymbol{w}} & \boldsymbol{\Sigma}_{\boldsymbol{w},\boldsymbol{\mu}}\\ \boldsymbol{\Sigma}_{\boldsymbol{w},\boldsymbol{\mu}}^T & \boldsymbol{\Sigma}_{\boldsymbol{\mu},\boldsymbol{\mu}}\end{bmatrix}\right), \quad (2.6)$$

and $\mathbb{P}\{Y(\boldsymbol{h}) = \ell\} = e^{\boldsymbol{h}_\ell}/\sum_{j\in[k]}e^{\boldsymbol{

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

**Proposition 3.4** *Consider* $\mu_i \overset{iid}{\sim} \mathcal{N}(0, \frac{\mu^2}{d}\mathrm{I}_d)$ *and* $\pi_i = 1/k$ *for all* $i \in [k]$. *Set* $u_{\mathrm{Bayes}} := \frac{\mu^2}{\sigma} \frac{1}{\sqrt{\mu^2+k\gamma\sigma^2}}$. *Then, the Bayes risk converges to* $\mathbb{P}\left\{ G_0 + \max_{\ell\in[k-1]} G_\ell \geq u_{\mathrm{Bayes}} \right\}$.

Under Gaussian prior, the means are asymptotically orthogonal and equal-energy. As shown earlier, in this setting, $\mathbb{P}_{e,\mathrm{Avg}} \xrightarrow{P} \mathbb{P}\left\{ G_0 + \max_{\ell\in[k-1]} G_\ell \geq u_{\mathrm{Avg}} \right\}$. But, $u_{\mathrm{Avg}} = u_{\mathrm{Bayes}}$. Thus, the averaging method is (asymptotically) Bayes optimal for equal-norm, orthogonal means and balanced classes. An analogous result was derived in [LM19, MKLZ20], but only for binary classification.

## 4 Results for Multinomial Logit Model

In this section we discuss the asymptotics of the intercepts/correlation matrices for MLM. We present results for arbitrary mean-vectors as well as special cases where the means are mutually orthogonal. Recall the eigenvalue decomposition of the Grammian $\Sigma_{\mu,\mu} = V\Sigma^2 V^T$ in (3.2). In order to state our results, it is convenient to introduce the following probability vectors in $\mathbb{R}^k$ and $\mathbb{R}^{k^2}$:

$$\pi := \mathbb{E}\left[\frac{e^{V\Sigma g}}{\mathbf{1}_k^T e^{V\Sigma g}}\right] \in \mathbb{R}^k \;\; \text{and} \;\; \Pi := \mathbb{E}\left[\frac{\left(e^{V\Sigma g}\right)\left(e^{V\Sigma g}\right)^T}{\left(\mathbf{1}_k^T e^{V\Sigma g}\right)^2}\right] \in \mathbb{R}^{k\times k}, \;\; \text{where } g \sim \mathcal{N}(0, I_r). \quad (4.1)$$

Note that $\pi$ and $\Pi$ are the first and second moments of the soft-max mapping of $V\Sigma g \sim \mathcal{N}(0, \Sigma_{\mu,\mu})$. In fact, for the MLM in (2.2) it holds that $\mathbb{P}\{Y = \ell\} = \mathbb{E}[\mathbb{P}\{Y = \ell \mid x\}] = \mathbb{E}\left[\frac{e^{e_\ell^T V\Sigma g}}{\mathbf{1}_k^T e^{V\Sigma g}}\right] = \pi_\ell$, $\ell \in [k]$ since $M^T x$ is distributed as $V\Sigma g$. Thus, $\pi$ is the vector of class priors (which explains the slight abuse of notation here in relation to our notation for the class priors of the GMM).

### 4.1 Class averaging classifier

**Proposition 4.1** *Consider data generated according to MLM in an asymptotic regime with any* $\gamma > 0$. *For the averaging classifier, the following high-dimensional limits hold*

$$\widehat{b} \xrightarrow{P} \pi, \quad \Sigma_{w,\mu} \xrightarrow{P} (diag(\pi) - \Pi) \cdot \Sigma_{\mu,\mu}, \quad (4.2a)$$

$$\Sigma_{w,w} \xrightarrow{P} \gamma \cdot diag(\pi) + (diag(\pi) - \Pi) \cdot \Sigma_{\mu,\mu} (diag(\pi) - \

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

## Footnotes

[1] Note the slight abuse of notation compared to the definitions in (K.30) and (K.30). This "renaming" should not be confusing as the constant $\gamma\cdot\eta$ (that is different between the two definitions) cancels when computing $\begin{bmatrix}\boldsymbol{a}\\\boldsymbol{b}_\ell\end{bmatrix}=\boldsymbol{A}^{-1}\boldsymbol{c}_\ell$ (see (K.12)).

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

# Contents

Figure 5: The threshold $\gamma_\star$ of Proposition 4.3 as a function of the number of classes $k$ and the means' energy $\mu$. LS provably outperforms class-averaging for $\gamma < \gamma_\star$.

Figure 6: Class-wise probabilities of error for MLM with (a) orthogonal means and (b) correlated means.

Figure 7: Precise theoretical predictions compared to a theoretical upper bound (UB) obtained by Union bound that does *not* require knowledge of the off-diagonal entries of $\Sigma_{w,w}$. GMM with orthogonal means and (a) equal priors, different strengths; (b) different priors, different strengths; (c) different priors, equal strengths. See text for details.

## A    Additional Numerical Results

In this section, we provide further numerical experiments.

First, in Figure 5 we investigate the question: *When does least-squares provably outperform averaging?* Our Proposition 4.3 provides a fundamental transition point in sample complexity above which least-squares is provably better than averaging under MLM. In Figure 5, we visualize $\gamma_\star$ as a function

of different number of classes as well as different levels of mean energy. Least-squares outperform averaging in the region below the lines displayed in Figure 5. Our key message is that least-squares work better when the sample complexity is higher and the problem is less noisy. As the number of classes $k$ increase, the problem becomes more difficult/noisy and we require a larger sample complexity to ensure classifier achieves a similar amount of accuracy as small $k$. Following this intuition, as $k$ increases, $\gamma_\star$ shifts smaller due to larger sample requirement. Similarly energy $\mu$ directly controls the noise level of the problem, i.e., larger $\mu$ results in a larger signal-to-noise ratio. Thus, as we increase $\mu$, $\gamma_\star$ increases as well because same test accuracy can be achieved with smaller sample size.

Second, Figure 6 provides further experiments on the class-wise probabilities of the MLM model with $k = 9$ classes for $\gamma = 0.1$. Classes 1,2,3 have norms $\|\boldsymbol{\mu}_i\|_{\ell_2} = 15$, while we have quadrupled the norms of classes 7,8,9 and doubled the norms of classes 4,5,6. In scenario (a) the means are orthogonal and in scenario (b) the means are highly correlated. The WLS shown corresponds to the following choice of weights: $\omega_i^2 = 1/\boldsymbol{\pi}_\ell, \ell \in [k]$, where $\boldsymbol{\pi}_\ell$ is the $\ell$th entry of the vector $\boldsymbol{\pi}$ in (4.1). The theoretical predictions for the class-wise error probabilities are computed using formula (D.7). As was the case for the GMM in Figure 2, we see that WLS creates the flattest class-wise errors.

Finally, in Figure 7 we investigate the following question: *To what extent pairwise class correlations are necessary for performance prediction?* Specifically, we consider a GMM setup with $k = 5$ classes and orthogonal means under three scenarios: (a) $\pi_1 = \pi_2 = \pi_3 = \pi_4 = \pi_5$, $4\|\boldsymbol{\mu}_1\|_{\ell_2} = 4\|\boldsymbol{\mu}_2\|_{\ell_2} = 2\|\boldsymbol{\mu}_3\|_{\ell_2} = 2\|\boldsymbol{\mu}_4\|_{\ell_2} = \|\boldsymbol{\mu}_5\|_{\ell_2} = 4\sqrt{3}$; (b) $4\pi_1 = 4\pi_2 = 2\pi_3 = 2\pi_4 = \pi_5$, $4\|\boldsymbol{\mu}_1\|_{\ell_2} = 4\|\boldsymbol{\mu}_2\|_{\ell_2} = 2\|\boldsymbol{\mu}_3\|_{\ell_2} = 2\|\boldsymbol{\mu}_4\|_{\ell_2} = \|\boldsymbol{\mu}_5\|_{\ell_2} = 4\sqrt{3}$; (c) $4\pi_1 = 4\pi_2 = 2\pi_3 = 2\pi_4 = \pi_5$, $\|\boldsymbol{\mu}_1\|_{\ell_2} = \|\boldsymbol{\mu}_2\|_{\ell_2} = \|\boldsymbol{\mu}_3\|_{\ell_2} = \|\boldsymbol{\mu}_4\|_{\ell_2} = \|\boldsymbol{\mu}_5\|_{\ell_2} = \sqrt{3}$. The solid lines are exact performance predictions based on our theory for averaging and least-squares estimators. The dashed lines are the theoretical upper bounds, which do *not* require the knowledge of cross-correlations between the classes (i.e., off-diagonal entries of $\boldsymbol{\Sigma}_{w,w}$ are unknown). These bounds are calculated by applying a union bound to the class-wise probabilities $\mathbb{P}_{e|c}$, $c \in [k]$ in (D.1) and further appropriately bounding the off-diagonal entries of $\boldsymbol{\Sigma}_{w,w}$ in terms of the self-correlations of the classes, i.e., only the diagonal entries of $\boldsymbol{\Sigma}_{w,w}$. Please see Section D.4.3 for details. Overall, the bounds shown only depend on $\widehat{b}, \boldsymbol{\Sigma}_{w,\mu}$ and $\text{diag}(\boldsymbol{\Sigma}_{w,w})$, which can all be obtained by studying the properties of isolated least-squares on individual classes without understanding their pairwise relations. While this suggests a simpler method to calculate theoretical bounds, there is a visible gap between such upper bounds and exact bounds and this gap is particularly more visible in the third scenario (c), where the bound becomes vacuous for LS. The gap remains visible in scenarios (a) and (b). At least, in these two cases comparing the bounds for averaging and LS to each other reveals the transition in performance gain between the two estimators. However, the cross-point of the curves does not coincide with the true one. This empirical study emphasizes the fact that pairwise correlations are indeed critical for exact asymptotic analysis and naive approaches cannot reproduce, in general, the results of our sharp analysis.

# B    Additional Results on Weighted Least-squares classifiers

## B.1    WLS for GMM

We now focus on characterizing the intercepts/correlation matrices for the WLS classifier.

**Theorem B.1** *Consider data generated according to GMM and $\gamma < 1$. Consider a weighted LS classifier with weights $\boldsymbol{D} = diag(\omega_1, \ldots, \omega_k)$ and let $\eta$ be the unique solution to $\sum_{\ell=1}^{k} \frac{\pi_\ell \omega_\ell^2}{\omega_\ell^2 + \eta} = \gamma$. Also define $\boldsymbol{P} := diag(\widetilde{\boldsymbol{\pi}}) - \widetilde{\boldsymbol{\pi}}\widetilde{\boldsymbol{\pi}}^T \geq \mathbf{0}_{k \times k}$ and $\boldsymbol{\Delta} := \sigma^2 \boldsymbol{I}_r + \boldsymbol{\Sigma} \boldsymbol{V}^T \boldsymbol{P} \boldsymbol{V} \boldsymbol{\Sigma} > \mathbf{0}_{r \times r}$ with the entries of $\widetilde{\boldsymbol{\pi}}$ given by $\widetilde{\pi}_\ell = \frac{1}{\gamma} \frac{\pi_\ell \omega_\ell^2}{\omega_\ell^2 + \eta}$. Then, for the WLS linear classifier $(\widehat{\boldsymbol{W}}, \widehat{\boldsymbol{b}})$ the following asymptotic limits hold*

$$\widehat{\boldsymbol{b}} \xrightarrow{P} \widetilde{\boldsymbol{\pi}} - \boldsymbol{P}\boldsymbol{V}\boldsymbol{\Sigma}\boldsymbol{\Delta}^{-1}\boldsymbol{\Sigma}\boldsymbol{V}^T\widetilde{\boldsymbol{\pi}}, \quad \boldsymbol{\Sigma}_{w,\mu} \xrightarrow{P} \boldsymbol{P}\boldsymbol{V}\boldsymbol{\Sigma}\boldsymbol{\Delta}^{-1}\boldsymbol{\Sigma}\boldsymbol{V}^T, \tag{B.1a}$$

$$\boldsymbol{\Sigma}_{w,w} \xrightarrow{P} \frac{\zeta}{\sigma^2}\boldsymbol{P} + \boldsymbol{P}\boldsymbol{V}\boldsymbol{\Sigma}\boldsymbol{\Delta}^{-1}\Big(\boldsymbol{\Delta}^{-1} - \frac{\zeta}{\sigma^2}\boldsymbol{I}_r\Big)\boldsymbol{\Sigma}\boldsymbol{V}^T\boldsymbol{P} + \frac{\eta\zeta}{\sigma^2}\boldsymbol{Q}. \tag{B.1b}$$

*Here, $\zeta := \gamma\big/\Big(\eta \sum_{\ell=1}^{k} \frac{\pi_\ell \omega_\ell^2}{(\omega_\ell^2 + \eta)^2}\Big)$ and $\boldsymbol{Q} \in \mathbb{R}^{k \times k}$ is a known matrix depending on various problem parameters. Its precise value is given in (J.18).*

Surprisingly, the effect of the weights is essentially equivalent to adjusting the class priors from $\boldsymbol{\pi}$ to $\widetilde{\boldsymbol{\pi}}$ defined in the theorem (modulo the extra additive term in the cross correlation matrix $\boldsymbol{\Sigma}_{\boldsymbol{w},\boldsymbol{w}}$). This shows that weighted LS has similar performance to an un-weighted LS applied to a model with different class priors $\widetilde{\boldsymbol{\pi}}$. This characterization allows us to precisely understand how different weighting schemes can alter test accuracy for rare/minority classes.

## B.2 WLS for MLM

Theorem B.2 predicts the asymptotic performance of *weighted* least-squares for data generated according to MLM.

**Theorem B.2** *Consider data generated according to MLM and $\gamma < 1$. Consider a weighted LS classifier with weights $\boldsymbol{D} = diag(\omega_1, \ldots, \omega_k)$ and let $\eta$ be the unique solution to $\sum_{\ell=1}^{k} \frac{\pi_\ell \omega_\ell^2}{\omega_\ell^2 + \eta} = \gamma$. Also define vector $\boldsymbol{\nu} \in \mathbb{R}^k$ with entries given by $\nu_\ell = \frac{1}{\gamma} \frac{\omega_\ell^2}{\omega_\ell^2 + \eta}$ and matrix*

$$\boldsymbol{\Delta} = \mathbb{E}\left[ \left( \boldsymbol{\nu}^T \boldsymbol{v} \right) \boldsymbol{g} \boldsymbol{g}^T \right] - \boldsymbol{\Sigma} \boldsymbol{V}^T \left( diag(\boldsymbol{\pi}) - \boldsymbol{\Pi} \right) \boldsymbol{\nu} \boldsymbol{\nu}^T \left( diag(\boldsymbol{\pi}) - \boldsymbol{\Pi} \right) \boldsymbol{V} \boldsymbol{\Sigma} > \boldsymbol{0}_{r \times r}, \tag{B.2}$$

*where $\boldsymbol{v} \in \mathbb{R}^k$ is a random vector with entries $V_\ell = e^{\boldsymbol{e}_\ell^T \boldsymbol{V} \boldsymbol{\Sigma} \boldsymbol{g}} \big/ \sum_{\ell' \in [k]} e^{\boldsymbol{e}_{\ell'} \boldsymbol{V} \boldsymbol{\Sigma} \boldsymbol{g}}$ for $\boldsymbol{g} \sim \mathcal{N}(\boldsymbol{0}, \boldsymbol{I}_r)$. Then, for the WLS linear classifier $\left( \widehat{\boldsymbol{W}}, \widehat{\boldsymbol{b}} \right)$ the following asymptotic limits hold*

$$\widehat{\boldsymbol{b}} \xrightarrow{P} diag(\boldsymbol{\nu})\boldsymbol{\pi} - diag(\boldsymbol{\nu}) \left( \boldsymbol{I}_k - \boldsymbol{\pi} \boldsymbol{\nu}^T \right) \left( diag(\boldsymbol{\pi}) - \boldsymbol{\Pi} \right) \boldsymbol{V} \boldsymbol{\Sigma} \boldsymbol{\Delta}^{-1} \boldsymbol{\Sigma} \boldsymbol{V}^T \left( diag(\boldsymbol{\pi}) - \boldsymbol{\Pi} \right) \boldsymbol{\nu}. \tag{B.3a}$$

$$\boldsymbol{\Sigma}_{\boldsymbol{w},\boldsymbol{\mu}} \xrightarrow{P} diag(\boldsymbol{\nu}) \left( \boldsymbol{I}_k - \boldsymbol{\pi} \boldsymbol{\nu}^T \right) \left( diag(\boldsymbol{\pi}) - \boldsymbol{\Pi} \right) \boldsymbol{V} \boldsymbol{\Sigma} \boldsymbol{\Delta}^{-1} \boldsymbol{\Sigma} \boldsymbol{V}^T. \tag{B.3b}$$

*The corresponding formula for the asymptotic limit of the cross-correlation matrix $\boldsymbol{\Sigma}_{\boldsymbol{w},\boldsymbol{w}}$ is given in (K.38) in Section K.*

Of course, the theorem above includes Theorem 4.2 as a special case. Indeed, we show how setting $\omega_\ell = 1$, $\ell \in [k]$ recovers the solution for (un-weighted) LS. First, solving for $\eta$ simply gives $\eta = \frac{1}{\gamma} - 1$. Thus, $\boldsymbol{\nu} = \boldsymbol{1}_k$. Also, observe in (4.1) that $(diag(\boldsymbol{\pi}) - \boldsymbol{\Pi}) \boldsymbol{1}_k = \boldsymbol{0}$ and $\boldsymbol{1}^T \boldsymbol{v} = 1$. Thus, (B.2) reduces to $\boldsymbol{\Delta} = \mathbb{E}[\boldsymbol{g} \boldsymbol{g}^T] = \boldsymbol{I}_r$. With these, it can be readily checked that (B.3a) and (B.3b) simplify to the expressions in (4.4a).

The term $\mathbb{E}\left[ \left( \boldsymbol{\nu}^T \boldsymbol{v} \right) \boldsymbol{g} \boldsymbol{g}^T \right]$ in (B.2) can be computed using Monte Carlo sampling. It is also possible to slightly simplify the calcuations involved using Gaussian integration by parts as shown in Lemma C.3. As mentioned, the formula that predicts $\boldsymbol{\Sigma}_{\boldsymbol{w},\boldsymbol{w}}$ is given in (K.38). While somewhat more complicated than formulae (B.3), the expression that we provide is also explicit. Numerical simulations shown in Figure 6 in Section A validate the accuracy of the theoretical predictions of the theorem.

# C  Preliminaries

In this section we gather a few preliminary results that will be used later on in our proofs.

## C.1  Slepian's inequality

**Lemma C.1 (Slepian's inequality [LT91])** *Let $\boldsymbol{g} \sim \mathcal{N}(\boldsymbol{0}, \boldsymbol{S})$ and $\widetilde{\boldsymbol{g}} \sim \mathcal{N}(\boldsymbol{0}, \boldsymbol{R})$ such that for all $i, j \in [k]$:*

$$\boldsymbol{S}_{ii} = \boldsymbol{R}_{ii}, \qquad \text{and} \qquad \boldsymbol{S}_{ij} \geq \boldsymbol{R}_{ij}.$$

*Then, for any $\boldsymbol{t} \in \mathbb{R}^k$ it holds that*

$$\mathbb{P}\left\{ \bigcup_{j \in [k]} \{\boldsymbol{g}_j \geq \boldsymbol{t}_j\} \right\} \leq \mathbb{P}\left\{ \bigcup_{j \in [k]} \{\widetilde{\boldsymbol{g}}_j \geq \boldsymbol{t}_j\} \right\}.$$

*Equivalently, letting $\boldsymbol{z} \sim \mathcal{N}(\boldsymbol{0}, \boldsymbol{I}_k)$,*

$$1 - \mathbb{P}\left\{ \boldsymbol{S}^{1/2} \boldsymbol{z} \leq \boldsymbol{t} \right\} \leq 1 - \mathbb{P}\left\{ \boldsymbol{R}^{1/2} \boldsymbol{z} \leq \boldsymbol{t} \right\}.$$

## C.2 Gaussian integration by parts

We say that a function $F : \mathbb{R}^m \to \mathbb{R}$ is of moderate growth if for each $c > 0$,

$$\lim_{\|x\|_{\ell_2} \to \infty} F(x) \exp\left(-c \|x\|_{\ell_2}^2\right) = 0.$$

The following result is a direct application of Gaussian integration by parts; for instance, see [FR13, Prop. 8.29].

**Lemma C.2 (Gaussian integration by parts (GIP) )** *Let $g \sim \mathcal{N}(0, I_r)$ and function $f : \mathbb{R}^r \to \mathbb{R}$ such that $f$ and all its first and second order partial derivatives are of moderate growth. Then, the following statements are true:*

*(i)* $\mathbb{E}[f(g)g] = \mathbb{E}[\nabla f(g)]$.

*(ii)* $\mathbb{E}[f(g)gg^T] = \mathbb{E}[f(g)] I_r + \mathbb{E}[\nabla^2 f(g)]$.

The following is a corollary of Lemma C.2 applied to the soft-max function.

**Lemma C.3 (GIP for the Softmax)** *Let $g \sim \mathcal{N}(0_r, I_r)$ and random vector $v = [V_1, V_2, \dots, V_k]^T$ with entries:*

$$v = \frac{e^{V\Sigma g}}{1_k^T e^{V\Sigma g}}, \qquad V_\ell = \frac{e^{e_\ell^T V\Sigma g}}{\sum_{j \in [k]} e^{e_j^T V\Sigma g}}, \quad \ell \in [k]. \tag{C.1}$$

*Further recall the notation of $\pi$ and $\Pi$ in (4.1). The following statements are true:*

*(i)* $\mathbb{E}[v] = \pi$.

*(ii)* *For all $i \in [r], \ell \in [k]$,*

$$\mathbb{E}[g_i V_\ell] = (e_i^T \Sigma V^T e_\ell) \pi_\ell - e_i^T \Sigma V^T \sum_{j \in [k]} e_j \Pi_{ij},$$

*and in matrix form:*

$$\mathbb{E}[vg^T] = (diag(\pi) - \Pi) V\Sigma.$$

*(iii)* *For all $\ell \in [k]$ let $s_\ell : \mathbb{R}^k \to \mathbb{R}$ denote the soft-max function: $s_\ell(x) = \frac{e^{x_\ell}}{\sum_{i \in [k]} e^{x_i}}$. Then,*

$$\mathbb{E}[V_\ell gg^T] = \pi_\ell I_r + \Sigma^T V^T \mathbb{E}[\nabla^2 s_\ell(V\Sigma g)] V\Sigma.$$

## C.3 Block matrix inversion

**Lemma C.4 (Block matrix inversion)** *Let $T = \begin{bmatrix} A & b \\ b^T & \delta \end{bmatrix}$ be an invertible block matrix. Then*

$$T^{-1}\begin{bmatrix} \mathbf{f} \\ \epsilon \end{bmatrix} = \begin{bmatrix} \Delta^{-1}\left(\mathbf{f} - \frac{\epsilon}{\delta} b\right) \\ \frac{\epsilon}{\delta} - \frac{1}{\delta} b^T \Delta^{-1}\left(\mathbf{f} - \frac{\epsilon}{\delta} b\right) \end{bmatrix} \tag{C.2}$$

*where $\Delta = A - \frac{1}{\delta} bb^T > 0$ is the Schur complement.*

# D Calculating and bounding the missclassification error

## D.1 Proof of (2.5) and (2.6)

**GMM.** Starting from (2.4) and using the fact that $x_i = \mu_Y + z = Me_Y + z$, $z \sim \mathcal{N}(0, \sigma^2 I_k)$, we have that

$$\mathbb{P}_e = \mathbb{P}\left\{ \arg\max_{j \in [k]}\{\langle \widehat{w}_j, Me_Y \rangle + \langle \widehat{w}_j, z \rangle + \widehat{b}_j\} \neq Y \right\},$$

or, in matrix-form:

$$\mathbb{P}_e = \mathbb{P}\left\{ \arg\max\{\widehat{\boldsymbol{W}}\boldsymbol{M}\boldsymbol{e}_Y + \widehat{\boldsymbol{W}}\boldsymbol{z} + \widehat{\boldsymbol{b}}\} \neq Y \right\}.$$

Recall that $\boldsymbol{\Sigma}_{\boldsymbol{w},\boldsymbol{\mu}} := \widehat{\boldsymbol{W}}\boldsymbol{M}$ and note that $\widehat{\boldsymbol{W}}\boldsymbol{z}$ is a zero-mean Gaussian vector with covariance matrix $\sigma^2 \widehat{\boldsymbol{W}}\widehat{\boldsymbol{W}}^T = \sigma^2 \boldsymbol{\Sigma}_{\boldsymbol{w},\boldsymbol{w}}$ in order to conclude with the desired formula in (2.5).

**MLM.** Recall from (2.2) that $\boldsymbol{x} \sim \mathcal{N}(\boldsymbol{0}, \boldsymbol{I}_d)$ and $Y$ is distributed such that $\mathbb{P}\{Y = \ell \mid \boldsymbol{x}\} = e^{\langle \boldsymbol{\mu}_\ell, \boldsymbol{x} \rangle} / \sum_{j \in [k]} e^{\langle \boldsymbol{\mu}_j, \boldsymbol{x} \rangle}$. Let $\boldsymbol{g} = \widehat{\boldsymbol{W}}\boldsymbol{x}$ and $\boldsymbol{h} = \boldsymbol{M}^T \boldsymbol{x}$. In this notation, (2.4) becomes

$$\mathbb{P}_e = \mathbb{P}\left\{ \arg\max\,(\boldsymbol{g} + \widehat{\boldsymbol{b}}) \neq Y \right\},$$

with $\mathbb{P}\{Y = \ell \mid \boldsymbol{x}\} = e^{\boldsymbol{h}_\ell} / \sum_{j \in [k]} e^{\boldsymbol{h}_j}$. To complete the proof of (2.6), it is easy to check that $\begin{bmatrix} \boldsymbol{g} \\ \boldsymbol{h} \end{bmatrix}$ defined above is jointly Gaussian with zero-mean and covariance matrix $\begin{bmatrix} \boldsymbol{\Sigma}_{\boldsymbol{w},\boldsymbol{w}} & \boldsymbol{\Sigma}_{\boldsymbol{w},\boldsymbol{\mu}} \\ \boldsymbol{\Sigma}_{\boldsymbol{w},\boldsymbol{\mu}}^T & \boldsymbol{\Sigma}_{\boldsymbol{\mu},\boldsymbol{\mu}} \end{bmatrix}$.

## D.2 Class-wise and total miss-classification error for GMM

The class-wise miss-classification error for GMM is given by

$$\mathbb{P}_{e|c} = \mathbb{P}\left( \exists j \neq c \,:\, \langle \widehat{\boldsymbol{w}}_c - \widehat{\boldsymbol{w}}_j, \boldsymbol{z} \rangle \leq \langle \widehat{\boldsymbol{w}}_j - \widehat{\boldsymbol{w}}_c, \boldsymbol{\mu}_c \rangle + (\widehat{\boldsymbol{b}}_j - \widehat{\boldsymbol{b}}_c) \right) \tag{D.1}$$

$$= 1 - \mathbb{P}\left( \forall j \neq c \,:\, \langle \widehat{\boldsymbol{w}}_c - \widehat{\boldsymbol{w}}_j, \boldsymbol{z} \rangle \geq \langle \widehat{\boldsymbol{w}}_j - \widehat{\boldsymbol{w}}_c, \boldsymbol{\mu}_c \rangle + (\widehat{\boldsymbol{b}}_j - \widehat{\boldsymbol{b}}_c) \right), \tag{D.2}$$

where we used that $\boldsymbol{x} = \boldsymbol{\mu}_c + \boldsymbol{z}$. Let $\boldsymbol{S}_c \in \mathbb{R}^{(k-1) \times (k-1)}$ be a symmetric matrix and $\boldsymbol{t}_c \in \mathbb{R}^{k-1}$ a vector with entries:

$$[\boldsymbol{t}_c]_j := \langle \widehat{\boldsymbol{w}}_j - \widehat{\boldsymbol{w}}_c, \boldsymbol{\mu}_c \rangle + (\widehat{\boldsymbol{b}}_j - \widehat{\boldsymbol{b}}_c) \quad j \neq c \in [k] \tag{D.3a}$$

$$[\boldsymbol{S}_c]_{ij} := \langle \widehat{\boldsymbol{w}}_c - \widehat{\boldsymbol{w}}_j, \widehat{\boldsymbol{w}}_c - \widehat{\boldsymbol{w}}_i \rangle, \quad i, j \neq c \in [k]. \tag{D.3b}$$

Then, we can rewrite (D.2) as

$$\mathbb{P}_{e|c} := 1 - \mathbb{P}\left\{ \boldsymbol{S}_c^{1/2}\boldsymbol{z} \geq \boldsymbol{t}_c \right\}, \tag{D.4}$$

where the inequality in the rightmost expression applies entry-wise.

Further, by using the law of total probability we have

$$\mathbb{P}_e = \sum_{c=1}^k \pi_c \mathbb{P}_{e|c} = \sum_{c=1}^k \pi_c \left( 1 - \mathbb{P}\left\{ \boldsymbol{S}_c^{1/2}\boldsymbol{z} \geq \boldsymbol{t}_c \right\} \right).$$

## D.3 Class-wise and total miss-classification error for MLM

In this section, we derive an explicit formula for the class-wise error for MLM. Recall (2.6):

$$\mathbb{P}_e = \mathbb{P}\left\{ \arg\max\,(\boldsymbol{g} + \widehat{\boldsymbol{b}}) \neq Y(\boldsymbol{h}) \right\}, \quad \text{where} \quad \begin{bmatrix} \boldsymbol{g} \\ \boldsymbol{h} \end{bmatrix} \sim \mathcal{N}\left( \boldsymbol{0}, \begin{bmatrix} \boldsymbol{\Sigma}_{\boldsymbol{w},\boldsymbol{w}} & \boldsymbol{\Sigma}_{\boldsymbol{w},\boldsymbol{\mu}} \\ \boldsymbol{\Sigma}_{\boldsymbol{w},\boldsymbol{\mu}}^T & \boldsymbol{\Sigma}_{\boldsymbol{\mu},\boldsymbol{\mu}} \end{bmatrix} \right),$$

and $\mathbb{P}\{Y(\boldsymbol{h}) = \ell\} = e^{\boldsymbol{h}_\ell} / \sum_{j \in [k]} e^{\boldsymbol{h}_j}, \quad \ell \in [k]$. Using Gaussian decomposition we can write $\boldsymbol{g} = \widetilde{\boldsymbol{g}} + \boldsymbol{\Sigma}_{\boldsymbol{w},\boldsymbol{\mu}}\boldsymbol{\Sigma}_{\boldsymbol{\mu},\boldsymbol{\mu}}^\dagger \boldsymbol{h}$ where $\widetilde{\boldsymbol{g}} \sim \mathcal{N}(\boldsymbol{0}_k, \boldsymbol{\Sigma}_{\boldsymbol{w},\boldsymbol{w}} - \boldsymbol{\Sigma}_{\boldsymbol{w},\boldsymbol{\mu}}\boldsymbol{\Sigma}_{\boldsymbol{\mu},\boldsymbol{\mu}}^\dagger \boldsymbol{\Sigma}_{\boldsymbol{w},\boldsymbol{\mu}})$. Using this, we have

$$\mathbb{P}_e = \mathbb{P}\left\{ \arg\max\,(\widetilde{\boldsymbol{g}} + \boldsymbol{\Sigma}_{\boldsymbol{w},\boldsymbol{\mu}}\boldsymbol{\Sigma}_{\boldsymbol{\mu},\boldsymbol{\mu}}^\dagger \boldsymbol{h} + \widehat{\boldsymbol{b}}) \neq Y(\boldsymbol{h}) \right\}$$

$$= \mathbb{P}\left\{ \arg\max\,(\widetilde{\boldsymbol{g}} + \boldsymbol{\Sigma}_{\boldsymbol{w},\boldsymbol{\mu}}\boldsymbol{\Sigma}_{\boldsymbol{\mu},\boldsymbol{\mu}}^\dagger \boldsymbol{h} + \widehat{\boldsymbol{b}}) \neq Y(\boldsymbol{h}) \right\}$$

$$= \sum_{c \in [k]} \mathbb{E}_{\boldsymbol{h},\widetilde{\boldsymbol{g}}}\left[ \mathbb{P}\left\{ Y(\boldsymbol{h}) = c \right\} \cdot \mathbb{1}\left\{ \arg\max\,(\widetilde{\boldsymbol{g}} + \boldsymbol{\Sigma}_{\boldsymbol{w},\boldsymbol{\mu}}\boldsymbol{\Sigma}_{\boldsymbol{\mu},\boldsymbol{\mu}}^\dagger \boldsymbol{h} + \widehat{\boldsymbol{b}}) \neq c \right\} \right]$$

$$= \sum_{c \in [k]} \mathbb{E}_{\boldsymbol{h},\widetilde{\boldsymbol{g}}}\left[ \frac{e^{\boldsymbol{h}_c}}{\sum_{\ell \in [k]} e^{\boldsymbol{h}_\ell}} \left( 1 - \prod_{j \neq c} \mathbb{1}\left\{ \widetilde{\boldsymbol{g}}_c + [\boldsymbol{\Sigma}_{\boldsymbol{w},\boldsymbol{\mu}}\boldsymbol{\Sigma}_{\boldsymbol{\mu},\boldsymbol{\mu}}^\dagger \boldsymbol{h}]_c + \widehat{\boldsymbol{b}}_c \geq \widetilde{\boldsymbol{g}}_j + [\boldsymbol{\Sigma}_{\boldsymbol{w},\boldsymbol{\mu}}\boldsymbol{\Sigma}_{\boldsymbol{\mu},\boldsymbol{\mu}}^\dagger \boldsymbol{h}]_j + \widehat{\boldsymbol{b}}_j \right\} \right) \right]$$

$$= \sum_{c \in [k]} \mathbb{E}_{\boldsymbol{h}}\left[ \frac{e^{\boldsymbol{h}_c}}{\sum_{\ell \in [k]} e^{\boldsymbol{h}_\ell}} \left( 1 - \mathbb{P}_{\boldsymbol{z} \sim \mathcal{N}(\boldsymbol{0}, \boldsymbol{I}_{k-1})}\left\{ \boldsymbol{S}_c^{1/2}\boldsymbol{z} \geq \boldsymbol{t}_c(\boldsymbol{h}) \right\} \right) \right], \tag{D.5}$$

where in the last line

$$[\boldsymbol{t}_c(\boldsymbol{h})]_j = \widehat{\boldsymbol{b}}_j - \widehat{\boldsymbol{b}}_c + [\boldsymbol{\Sigma}_{\boldsymbol{w},\boldsymbol{\mu}}\boldsymbol{\Sigma}_{\boldsymbol{\mu},\boldsymbol{\mu}}^\dagger\boldsymbol{h}]_j - [\boldsymbol{\Sigma}_{\boldsymbol{w},\boldsymbol{\mu}}\boldsymbol{\Sigma}_{\boldsymbol{\mu},\boldsymbol{\mu}}^\dagger\boldsymbol{h}]_c, \quad j \neq c \in [k] \tag{D.6a}$$

$$[\boldsymbol{S}_c]_{i,j} = (\boldsymbol{e}_c - \boldsymbol{e}_j)^T \left(\boldsymbol{\Sigma}_{\boldsymbol{w},\boldsymbol{w}} - \boldsymbol{\Sigma}_{\boldsymbol{w},\boldsymbol{\mu}}\boldsymbol{\Sigma}_{\boldsymbol{\mu},\boldsymbol{\mu}}^\dagger\boldsymbol{\Sigma}_{\boldsymbol{w},\boldsymbol{\mu}}\right)(\boldsymbol{e}_c - \boldsymbol{e}_i), \quad i,j \neq c \in [k]. \tag{D.6b}$$

Further recalling the decomposition $\mathbb{P}_e = \sum_c \mathbb{P}\{\widehat{Y} \neq Y | Y = c\} \mathbb{P}\{Y = c\}$ and noting that

$$\mathbb{P}\{Y = c\} = \mathbb{E}_{\boldsymbol{x}}\left[\mathbb{P}\{Y = c|\boldsymbol{x}\}\right] = \mathbb{E}_{\boldsymbol{x}}\left[\frac{1}{\left(1 + \sum_{j \neq c} e^{(\boldsymbol{\mu}_j - \boldsymbol{\mu}_c)^T\boldsymbol{x}}\right)}\right] = \boldsymbol{\pi}_c$$

we can see from (D.5) that the class-wise error probabilities can be calculated as follows:

$$\mathbb{P}_{e|c} = \mathbb{P}\{\widehat{Y} \neq Y | Y = c\} = \frac{1}{\boldsymbol{\pi}_c}\,\mathbb{E}_{\boldsymbol{h}\sim\mathcal{N}(\boldsymbol{0}_k,\boldsymbol{\Sigma}_{\boldsymbol{\mu},\boldsymbol{\mu}})}\left[\frac{e^{\boldsymbol{h}_c}}{\sum_{\ell\in[k]} e^{\boldsymbol{h}_\ell}}\left(1 - \mathbb{P}_{\boldsymbol{z}\sim\mathcal{N}(\boldsymbol{0},\boldsymbol{I}_{k-1})}\left\{\boldsymbol{S}_c^{1/2}\boldsymbol{z} \geq \boldsymbol{t}_c(\boldsymbol{h})\right\}\right)\right],$$
$$\tag{D.7}$$

where $\boldsymbol{\pi}_c$ is the $c^{\text{th}}$ entry of the vector $\boldsymbol{\pi}$ in (4.1) and $\boldsymbol{S}_c, \boldsymbol{t}_c(\boldsymbol{h})$ are defined in (D.6).

## D.4 Evaluating and bounding tail probabilities of multivariate Gaussians

In Sections D.3 and D.2, we expressed the class-wise probability of missclassification error for both GMM and MLM in the following convenient form for $\boldsymbol{z} \sim \mathcal{N}(\boldsymbol{0}, \boldsymbol{I}_{k-1})$,

$$1 - \mathbb{P}\{\boldsymbol{A}^{1/2}\boldsymbol{z} \leq \boldsymbol{t}\} = \mathbb{P}\left\{\bigcup_{i\in[k-1]} \{\widetilde{\boldsymbol{a}}_i^T\boldsymbol{z} \geq \boldsymbol{t}_i\}\right\}. \tag{D.8}$$

Here, $\boldsymbol{A} \geq \boldsymbol{0} \in \mathbb{R}^{(k-1)\times(k-1)}, \boldsymbol{t} \in \mathbb{R}^{k-1}$ are appropriate coefficient matrices (see (D.4) and (D.5)) and $\widetilde{\boldsymbol{a}}_i$ denotes the $i$th row of the matrix $\boldsymbol{A}^{1/2}$. For example, (D.8) maps to (D.4) for $\boldsymbol{A} \leftarrow \boldsymbol{S}_c$ and $\boldsymbol{t} \leftarrow (-\boldsymbol{t}_c)$.

The formulation above is convenient both in our theoretical analysis, as well as, in simulations. In the rest of this section, we briefly discuss some relevant tools that allow to further simplify or bound expressions in the form of (D.8).

### D.4.1 A special case: Rank-one update of Identity

First, we discuss the case where the coefficient matrix $\boldsymbol{A}$ and vector $\boldsymbol{t}$ in (D.8) take the special form $\boldsymbol{A} \propto \boldsymbol{I} + \boldsymbol{1}\boldsymbol{1}^T$ and $\boldsymbol{t} \propto \boldsymbol{1}$. This special case appears in some of the stylized symmetric problem settings studied in this paper, such as classification problems with orthogonal and equally-balanced means.

**Lemma D.1** *Let $\boldsymbol{A} = \boldsymbol{I}_k + \boldsymbol{1}_k\boldsymbol{1}_k^T$ and $\boldsymbol{g} \sim \mathcal{N}(\boldsymbol{0}, \boldsymbol{A})$. Then, for any $t \in \mathbb{R}$,*

$$1 - \mathbb{P}\{\boldsymbol{g} \leq t\boldsymbol{1}_k\} = \mathbb{P}\{G_0 + \max_{i\in[k]} G_i \geq t\}, \quad G_0, G_1, \ldots, G_k \overset{iid}{\sim} \mathcal{N}(0,1). \tag{D.9}$$

**Proof** For each $i \in [n]$, we can decompose $\boldsymbol{g}_i = G_0 + G_i$, where $G_0, G_1, \ldots, G_k$ are iid standard normals. Indeed, it can be readily checked from this that $\mathbb{E}[\boldsymbol{g}_i^2] = 2$ and $\mathbb{E}[\boldsymbol{g}_i\boldsymbol{g}_j] = \mathbb{E}[G_0^2] = 1$, $i \neq j$, which is consistent with $\boldsymbol{g} \sim \mathcal{N}(\boldsymbol{0}, \boldsymbol{A})$. Thus, we can write

$$1 - \mathbb{P}\{\boldsymbol{g} \leq t\boldsymbol{1}_k\} = \mathbb{P}\{\max_{i\in[k]} \boldsymbol{g}_i \geq t\} = \mathbb{P}\{G_0 + \max_{i\in[k]} G_i \geq t\},$$

which completes the proof. ∎

### D.4.2 Slepian's bound

When the matrix $\boldsymbol{A}$ does not have the special structure assumed by Lemma D.1, it is not possible in general to provide simple expressions as the one in (D.9). Yet, it might be possible to obtain

upper bounds of the same simple form. Such simple bounds can be useful for theoretical interpretations of otherwise complicated formulae, or can provide efficient means for quick (but, non-tight) implementations.

In this section, we discuss Slepian's inequality (see C.1) as a useful tool in this direction. Assume that $a = \min_{i,j \in [k]} \boldsymbol{A}_{ij} \geq 0$. To begin, note that $\boldsymbol{A} \geq (\text{diag}(\boldsymbol{A}) - a\boldsymbol{I}) + a\boldsymbol{1}\boldsymbol{1}^T$, where the inequality holds element-wise and equality is true for the diagonal elements. Then, one can apply Slepian's Lemma C.1 to upper bound the conditional probability of error in (D.8) with the following simple bound:

$$1 - \mathbb{P}\{\boldsymbol{A}^{1/2}\boldsymbol{g} \leq \boldsymbol{t}\} \leq 1 - \mathbb{P}\left\{ \left( (\text{diag}(\boldsymbol{A}) - a\boldsymbol{I}) + a\boldsymbol{1}\boldsymbol{1}^T \right)^{1/2} \boldsymbol{g} \leq \boldsymbol{t} \right\}$$

$$\leq \mathbb{P}\left\{ \bigcup_{j \in [k]} \left\{ G_0 + G_j \sqrt{[\boldsymbol{A}]_{jj}/a - 1} \geq [\boldsymbol{t}]_j/a \right\} \right\}, \qquad G_0, G_1, \ldots, G_k \overset{iid}{\sim} \mathcal{N}(0,1).$$

In the second line above, we used the Gaussian decomposition of Lemma D.1.

### D.4.3 Simple bounds for GMM

**Union bound.** Of course, it is also possible to apply (a simpler) union bound to upper bound the tail probability in (D.8). Here, we show explicitly the result of applying union bound to the class-wise error probabilities of the GMM. Specifically, consider (D.1). An application of the union bound leads to the following:

$$\mathbb{P}_{e|c} = \mathbb{P}\left( \exists j \neq c \,:\, \langle \widehat{\boldsymbol{w}}_c - \widehat{\boldsymbol{w}}_j, \boldsymbol{z} \rangle \leq \langle \widehat{\boldsymbol{w}}_j - \widehat{\boldsymbol{w}}_c, \boldsymbol{\mu}_c \rangle + (\widehat{\boldsymbol{b}}_j - \widehat{\boldsymbol{b}}_c) \right) \tag{D.10}$$

$$\leq \sum_{j \neq c} \mathbb{P}_{G \sim \mathcal{N}(0,1)}\left\{ \|\widehat{\boldsymbol{w}}_j - \widehat{\boldsymbol{w}}_c\|_{\ell_2} G \leq \langle \widehat{\boldsymbol{w}}_j - \widehat{\boldsymbol{w}}_c, \boldsymbol{\mu}_c \rangle + (\widehat{\boldsymbol{b}}_j - \widehat{\boldsymbol{b}}_c) \right\}$$

$$= \sum_{j \neq c} Q\left( \langle \frac{\widehat{\boldsymbol{w}}_c - \widehat{\boldsymbol{w}}_j}{\|\widehat{\boldsymbol{w}}_c - \widehat{\boldsymbol{w}}_j\|_{\ell_2}}, \boldsymbol{\mu}_c \rangle + \frac{\widehat{\boldsymbol{b}}_c - \widehat{\boldsymbol{b}}_j}{\|\widehat{\boldsymbol{w}}_c - \widehat{\boldsymbol{w}}_j\|_{\ell_2}} \right)$$

$$= \sum_{j \neq c} Q\left( \frac{-[\boldsymbol{t}_c]_j}{\sqrt{[\boldsymbol{S}_c]_{jj}}} \right) \tag{D.11}$$

$$\leq (k-1) \cdot Q(d_{\min}),$$

where in (D.11) $\boldsymbol{S}_c, \boldsymbol{t}_c$ are defined in (D.3) and in the last line we denote $d_{\min} := \min_{j \neq c}\left\{ -[\boldsymbol{t}_c]_j \big/ \sqrt{[\boldsymbol{S}_c]_{j,j}} \right\}$.

**Union bound without knowledge of cross-correlations** $\langle \boldsymbol{w}_i, \boldsymbol{w}_j \rangle$, $i \neq j$. It is worth noting that the upper bound in (D.11) requires knowledge of the cross-correlations $\langle \widehat{\boldsymbol{w}}_j, \widehat{\boldsymbol{w}}_c \rangle$, $j \neq c$, i.e., of the off-diagonal entries of $\boldsymbol{\Sigma}_{\boldsymbol{w},\boldsymbol{w}}$. Thankfully, our analysis allows predicting these values. For comparison, we ask wether it is possible to further upper bound the class-wise error probability if only the diagonal entries of $\boldsymbol{\Sigma}_{\boldsymbol{w},\boldsymbol{w}}$ (i.e., the norms $\|\widehat{\boldsymbol{w}}_j\|_{\ell_2}, j \in [k]$) were known. A simple answer to this questions is as follows. Observe that $\sqrt{[\boldsymbol{S}_c]_{jj}} = \|\widehat{\boldsymbol{w}}_c - \widehat{\boldsymbol{w}}_j\|_{\ell_2} \leq \|\widehat{\boldsymbol{w}}_c\|_{\ell_2} + \|\widehat{\boldsymbol{w}}_j\|_{\ell_2}$. Thus, if $[\boldsymbol{t}_c]_j < 0$ then the $j$th term in (D.11) is further upper bounded by $Q\left( \frac{-[\boldsymbol{t}_c]_j}{\|\widehat{\boldsymbol{w}}_c\|_{\ell_2} + \|\widehat{\boldsymbol{w}}_j\|_{\ell_2}} \right)$:

$$\mathbb{P}_{e|c} \leq \begin{cases} \sum_{j \neq c} Q\left( \frac{-[\boldsymbol{t}_c]_j}{\|\widehat{\boldsymbol{w}}_c\|_{\ell_2} + \|\widehat{\boldsymbol{w}}_j\|_{\ell_2}} \right) & \text{if } \boldsymbol{t}_c \leq \boldsymbol{0}, \\ 1 & \text{otherwise.} \end{cases}$$

Unfortunately, this bound becomes non-trivial for the class-wise probability of error only if

$$[\boldsymbol{t}_c]_j < 0 \Leftrightarrow \langle \widehat{\boldsymbol{w}}_c, \boldsymbol{\mu}_c \rangle + \widehat{\boldsymbol{b}}_c \geq \langle \widehat{\boldsymbol{w}}_j, \boldsymbol{\mu}_c \rangle + \widehat{\boldsymbol{b}}_j \text{ for all } j \neq c.$$

Intuitively, this assumes a regime wherethe weight vector $\widehat{\boldsymbol{w}}_c$ corresponding to class $c$ aligns better with the corresponding mean vector $\boldsymbol{\mu}_c$ than the rest of the weight vectors $\widehat{\boldsymbol{w}}_j$, $j \neq c$. This emphasizes the important role of the cross-correlation matrix $\boldsymbol{\Sigma}_{\boldsymbol{w},\boldsymbol{w}}$ (including the off-diagonals) for accurate performance prediction. For an illustration, we have implemented this bound and have compared it to our sharp predictions in Figure 7.

**Oracle lower bound.** For completeness, we briefly discuss an oracle lower bound for the class-wise probability of error in GMM. Specifically, assume that the means $\boldsymbol{\mu}_i$, $i \in [n]$ are known. Then

the problem of classifying a new sample $\boldsymbol{x}$ is a $k$-ary hypothesis testing problem with Gaussian conditionals. Denote $\mathbb{P}_{\text{genie,Bayes}}$ the Bayes error of this hypothesis testing problem. Clearly $\mathbb{P}_{\text{genie,Bayes}}$ is a lower bound on the error of any classifier that is trained on data. The paper [WBW$^+$16] further lower bounds $\mathbb{P}_{\text{genie,Bayes}}$ in terms of the Bayesian probability of errors between every two classes as follows:

$$\mathbb{P}_e \geq \mathbb{P}_{\text{genie,Bayes}} \geq \frac{2}{k} \sum_{i \neq j} \pi_i \, \mathbb{P}_{\text{genie,ij}} \tag{D.12}$$

$$= \frac{2}{k} \sum_{i \neq j} \frac{\pi_i}{\pi_i + \pi_j} \left\{ \pi_i \cdot Q\left( \frac{\|\boldsymbol{\mu}_i - \boldsymbol{\mu}_j\|_{\ell_2}}{2} + \frac{\log(\pi_i/\pi_j)}{\|\boldsymbol{\mu}_i - \boldsymbol{\mu}_j\|_{\ell_2}} \right) + \pi_j \cdot Q\left( \frac{\|\boldsymbol{\mu}_i - \boldsymbol{\mu}_j\|_{\ell_2}}{2} - \frac{\log(\pi_i/\pi_j)}{\|\boldsymbol{\mu}_i - \boldsymbol{\mu}_j\|_{\ell_2}} \right) \right\}.$$

where $\mathbb{P}_{\text{genie,ij}}$ is the Bayesian error between classes $i$ and $j$ with priors $\frac{\pi_i}{\pi_i+\pi_j}$ and $\frac{\pi_j}{\pi_i+\pi_j}$. For the last equality we have used the well-known formula for the Bayesian probability of binary Gaussian hypothesis testing. In the case of equal-priors the genie lower bound above simplifies to

$$\mathbb{P}_e \geq \frac{2}{k} \sum_{i \neq j} \pi_i \cdot Q\left( \frac{\|\boldsymbol{\mu}_i - \boldsymbol{\mu}_j\|_{\ell_2}}{2} \right). \tag{D.13}$$

Note that, in contrast to (D.13), our analysis allows for precise evaluations of the missclassification error $\mathbb{P}_e$.

# E  The Class-averaging estimator

## E.1  Proofs for GMM

### E.1.1  GMM: Proof of Proposition 3.1

The first statement (3.1a) follows directly from the fact that $\frac{1}{n}\mathbf{1}^T \boldsymbol{Y}_i = \frac{n_i}{n} \xrightarrow{P} \pi_i$. For the next two statements note that

$$\widehat{\boldsymbol{w}}_i = \frac{1}{n} \boldsymbol{M} \boldsymbol{Y} \boldsymbol{Y}_i + \frac{1}{n} \boldsymbol{Z} \boldsymbol{Y}_i = \frac{1}{n} \sum_{j=1}^k \boldsymbol{\mu}_j (\boldsymbol{Y}_j^T \boldsymbol{Y}_i) + \frac{1}{n} \boldsymbol{Z} \boldsymbol{Y}_i = \frac{\|\boldsymbol{Y}_i\|_{\ell_2}^2}{n} \boldsymbol{\mu}_i + \frac{1}{n} \boldsymbol{Z} \boldsymbol{Y}_i, \tag{E.1}$$

where in the last line we used orthogonality of the rows $\boldsymbol{Y}_j$ of the matrix $\boldsymbol{Y}$:

$$\langle \boldsymbol{Y}_i, \boldsymbol{Y}_j \rangle = 0, \forall i \neq j \in [k]. \tag{E.2}$$

To conclude simply use the facts that for all $i \in [k]$:

  (i)  $\frac{\|\boldsymbol{Y}_i\|_{\ell_2}^2}{n} = \frac{n_i}{n} \xrightarrow{P} \pi_i$.

  (ii)  $\boldsymbol{Z} \boldsymbol{Y}_i \sim \sigma \|\boldsymbol{Y}_i\|_{\ell_2} \boldsymbol{g}_i$ with $\boldsymbol{g}_i \overset{iid}{\sim} \mathcal{N}(\mathbf{0}, \boldsymbol{I}_d)$ because of (E.2).

  (iii)  $\frac{\|\boldsymbol{g}_i\|_{\ell_2}}{\sqrt{n}} \xrightarrow{P} \sqrt{\gamma}$ and $\frac{1}{\sqrt{n}} \langle \boldsymbol{g}_i, \boldsymbol{\mu}_j \rangle \xrightarrow{P} 0$.

## E.2  Proofs for MLM

### E.2.1  Proof of Proposition 4.1

Let us define $\boldsymbol{g} \sim \mathcal{N}(\mathbf{0}, \boldsymbol{I}_r)$ and random vector $\boldsymbol{v} = [V_1, V_2, \ldots, V_k]^T$ with entries:

$$\boldsymbol{v} = \frac{e^{\boldsymbol{V}\boldsymbol{\Sigma}\boldsymbol{g}}}{\mathbf{1}_k^T e^{\boldsymbol{V}\boldsymbol{\Sigma}\boldsymbol{g}}}, \qquad V_i = \frac{e^{\boldsymbol{e}_i^T \boldsymbol{V}\boldsymbol{\Sigma}\boldsymbol{g}}}{\sum_{j\in[k]} e^{\boldsymbol{e}_j^T \boldsymbol{V}\boldsymbol{\Sigma}\boldsymbol{g}}}, \quad i \in [k]. \tag{E.3}$$

We will prove the following three statements:

$$\widehat{\boldsymbol{b}} \xrightarrow{P} \mathbb{E}[\boldsymbol{v}] \tag{E.4a}$$

$$\boldsymbol{\Sigma}_{\boldsymbol{w},\boldsymbol{\mu}} \xrightarrow{P} \mathbb{E}[\boldsymbol{v}\boldsymbol{g}^T] \boldsymbol{\Sigma} \boldsymbol{V}^T \tag{E.4b}$$

$$\boldsymbol{\Sigma}_{\boldsymbol{w},\boldsymbol{w}} \xrightarrow{P} \gamma \cdot \text{diag}(\mathbb{E}[\boldsymbol{v}]) + \mathbb{E}[\boldsymbol{v}\boldsymbol{g}^T] \cdot \mathbb{E}[\boldsymbol{g}\boldsymbol{v}^T]. \tag{E.4c}$$

These lead to (4.2) using Lemma C.3. Therefore, in what follows, we prove (E.4)

For the intercepts $\widehat{\boldsymbol{b}}_\ell$, $\ell \in [k]$ it holds that

$$\widehat{\boldsymbol{b}}_\ell = \frac{1}{n}\mathbf{1}^T \boldsymbol{Y}_\ell \xrightarrow{P} \mathbb{P}\{Y = \ell\} = \mathbb{E}\left[\frac{e^{\boldsymbol{h}_\ell}}{\sum_{j\in[k]} e^{\boldsymbol{h}_j}}\right],$$

where $\boldsymbol{h} \sim \mathcal{N}(\mathbf{0}_k, \boldsymbol{\Sigma}_{\boldsymbol{\mu\mu}})$. To deduce the first statement in (E.4a), note that $\boldsymbol{h} \overset{(\mathrm{D})}{=} \boldsymbol{V}\boldsymbol{\Sigma}\boldsymbol{g}$.

Continuing with the vectors $\widehat{\boldsymbol{w}}_\ell$, $\ell \in [k]$, recall that $\boldsymbol{w}_\ell = \frac{1}{n}\boldsymbol{X}\boldsymbol{Y}_\ell = \frac{1}{n}\sum_{i_\ell\in[n]}\boldsymbol{x}_{i_\ell}[\boldsymbol{Y}_\ell]_{i_\ell}$. Consider the singular decomposition

$$\boldsymbol{M} = \boldsymbol{U}\boldsymbol{\Sigma}\boldsymbol{V}^T = \begin{bmatrix} \boldsymbol{u}_1 & \boldsymbol{u}_2 & \dots & \boldsymbol{u}_r \end{bmatrix} \operatorname{diag}(\sigma_1, \sigma_2, \dots, \sigma_r) \begin{bmatrix} \boldsymbol{v}_1^T \\ \boldsymbol{v}_2^T \\ \dots \\ \boldsymbol{v}_r^T \end{bmatrix},$$

with $\boldsymbol{U} \in \mathbb{R}^{d\times r}$, $\boldsymbol{\Sigma} \in \mathbb{R}^{r\times r}$, and $\boldsymbol{V} \in \mathbb{R}^{k\times r}$ where $r = \operatorname{rank}(\boldsymbol{M}) \leq k$. Decompose $\boldsymbol{X} \in \mathbb{R}^{d\times n}$ as $\boldsymbol{X} = \boldsymbol{U}\boldsymbol{U}^T\boldsymbol{X} + \boldsymbol{P}^\perp\boldsymbol{X}$ with $\boldsymbol{P}^\perp = \boldsymbol{I}_d - \boldsymbol{U}\boldsymbol{U}^T$. With this notation we compute

$$\langle \boldsymbol{w}_\ell, \boldsymbol{\mu}_c\rangle = \frac{1}{n}\sum_{i\in[n]}\boldsymbol{x}_i^T\boldsymbol{\mu}_c[\boldsymbol{Y}_\ell]_i = \frac{1}{n}\sum_{j=1}^r\sum_{i\in[n]}(\boldsymbol{x}_i^T\boldsymbol{u}_j)\cdot(\boldsymbol{\mu}_c^T\boldsymbol{u}_j)[\boldsymbol{Y}_\ell]_i + \frac{1}{n}\sum_{i\in[n]}\cdot(\boldsymbol{\mu}_c^T\boldsymbol{P}^\perp\boldsymbol{x}_i)[\boldsymbol{Y}_\ell]_i$$

$$\xrightarrow{P} \sum_{j=1}^r(\boldsymbol{e}_c^T\boldsymbol{V}\boldsymbol{\Sigma}\boldsymbol{e}_j)\mathbb{E}\left[\boldsymbol{g}_j\frac{\left(e^{\boldsymbol{e}_\ell^T\boldsymbol{V}\boldsymbol{\Sigma}\boldsymbol{g}}\right)}{\sum_{\ell'\in[k]}e^{\boldsymbol{e}_{\ell'}^T\boldsymbol{V}\boldsymbol{\Sigma}\boldsymbol{g}}}\right]$$

$$= \sum_{j=1}^r\mathbb{E}[V_\ell\boldsymbol{g}_j]\left(\boldsymbol{e}_j^T\boldsymbol{\Sigma}\boldsymbol{V}^T\boldsymbol{e}_c\right). \tag{E.5}$$

Here, we have recognized that for every $i \in [n]: \boldsymbol{U}^T\boldsymbol{x}_i \sim \boldsymbol{g}$, and also, conditioned on $\boldsymbol{x}_i$: $[\boldsymbol{Y}_\ell]_i \sim \operatorname{Bern}\left(e^{\boldsymbol{\mu}_\ell^T\boldsymbol{x}_i}/\sum_{\ell'}e^{\boldsymbol{\mu}_{\ell'}^T\boldsymbol{x}_i}\right)$ and $\boldsymbol{\mu}_\ell^T\boldsymbol{x}_i = \boldsymbol{e}_\ell^T\boldsymbol{V}\boldsymbol{\Sigma}\boldsymbol{U}^T\boldsymbol{x}_i \sim \boldsymbol{e}_\ell^T\boldsymbol{V}\boldsymbol{\Sigma}\boldsymbol{g}$, $\ell \in [k]$. This shows the second statement in (E.4b) when expressed in matrix form.

We proceed similarly with the proof of the last statement in (E.4c) as follows:

$$\langle \boldsymbol{w}_\ell, \boldsymbol{w}_c\rangle = \frac{1}{n^2}\sum_{i_\ell\in[n],i_c\in[n]}\boldsymbol{x}_{i_\ell}^T\boldsymbol{x}_{i_c}[\boldsymbol{Y}_\ell]_{i_\ell}[\boldsymbol{Y}_c]_{i_c}$$

$$= \frac{1}{n^2}\sum_{j=1}^r\sum_{i_\ell\in[n],i_c\in[n]}(\boldsymbol{x}_{i_\ell}^T\boldsymbol{u}_j)(\boldsymbol{x}_{i_c}^T\boldsymbol{u}_j)[\boldsymbol{Y}_\ell]_{i_\ell}[\boldsymbol{Y}_c]_{i_c} + \frac{1}{n^2}\sum_{i_\ell\in[n],i_c\in[n]}(\boldsymbol{P}^\perp\boldsymbol{x}_{i_\ell})^T(\boldsymbol{P}^\perp\boldsymbol{x}_{i_c})[\boldsymbol{Y}_\ell]_{i_\ell}[\boldsymbol{Y}_c]_{i_c}$$

For $i_\ell = i_c = i \in [n]$ note that

$$\frac{1}{n^2}\sum_{i\in[n]}\sum_{j=1}^r(\boldsymbol{x}_i^T\boldsymbol{u}_j)(\boldsymbol{x}_i^T\boldsymbol{u}_j)[\boldsymbol{Y}_\ell]_i[\boldsymbol{Y}_c]_i \xrightarrow{P} 0,$$

while, for $i_\ell \neq i_c$,

$$\frac{1}{n^2}\sum_{i_\ell\neq i_c\in[n]}\sum_{j=1}^r(\boldsymbol{x}_{i_\ell}^T\boldsymbol{u}_j)[\boldsymbol{Y}_\ell]_{i_\ell}(\boldsymbol{x}_{i_c}^T\boldsymbol{u}_j)[\boldsymbol{Y}_c]_{i_c} \xrightarrow{P} \sum_{j=1}^r\mathbb{E}\left[\boldsymbol{g}_j\cdot\frac{\left(e^{\boldsymbol{e}_\ell^T\boldsymbol{V}\boldsymbol{\Sigma}\boldsymbol{g}}\right)}{\sum_{\ell'\in[k]}e^{\boldsymbol{e}_{\ell'}^T\boldsymbol{V}\boldsymbol{\Sigma}\boldsymbol{g}}}\right]\mathbb{E}\left[\boldsymbol{g}_j\cdot\frac{\left(e^{\boldsymbol{e}_c^T\boldsymbol{V}\boldsymbol{\Sigma}\boldsymbol{g}}\right)}{\sum_{\ell'\in[k]}e^{\boldsymbol{e}_{\ell'}^T\boldsymbol{V}\boldsymbol{\Sigma}\boldsymbol{g}}}\right]$$

$$= \sum_{j=1}^r\mathbb{E}[V_\ell\boldsymbol{g}_j]\mathbb{E}[\boldsymbol{g}_jV_c] = \boldsymbol{e}_\ell^T\boldsymbol{E}[\boldsymbol{v}\boldsymbol{g}^T]\cdot\boldsymbol{E}[\boldsymbol{g}\boldsymbol{v}^T]\boldsymbol{e}_c^T$$

Furthermore,

$$\frac{1}{n}\sum_{i\in[n]}\|\boldsymbol{P}^\perp\boldsymbol{x}_i\|_{\ell_2}^2[\boldsymbol{Y}_\ell]_i[\boldsymbol{Y}_c]_i^2 \xrightarrow{P} \gamma\cdot\mathbb{1}_{\ell,c}\cdot\mathbb{E}\left[\left(\frac{e^{\boldsymbol{e}_\ell^T\boldsymbol{V}\boldsymbol{\Sigma}\boldsymbol{g}}}{\sum_{\ell'\in[k]}e^{\boldsymbol{e}_{\ell'}^T\boldsymbol{V}\boldsymbol{\Sigma}\boldsymbol{g}}}\right)\right] = \gamma\cdot\boldsymbol{e}_\ell^T\operatorname{diag}(\mathbb{E}[\boldsymbol{v}])\boldsymbol{e}_c.$$

Combining the last two displays results in (E.4b), as desired.

### E.2.2 Orthogonal means

Here, we specialize the general result of Proposition 4.1 to the special case of orthogonal means: $\langle \boldsymbol{\mu}_i, \boldsymbol{\mu}_j \rangle = 0, \forall i \neq j$. Recall the notation $\mu_i = \|\boldsymbol{\mu}_i\|_{\ell_2}, i \in [k]$. Then, in this case the parameters in (4.1) are simply given by the following

$$\boldsymbol{\pi}_i := \mathbb{E}\Big[\frac{e^{\mu_i G_i}}{\sum_{\ell \in [k]} e^{\mu_\ell G_\ell}}\Big], i \in [k] \quad \text{and} \quad \boldsymbol{\Pi}_{ij} := \mathbb{E}\Big[\frac{e^{\mu_i G_i} e^{\mu_j G_j}}{\big(\sum_{\ell \in [k]} e^{\mu_\ell G_\ell}\big)^2}\Big], \; i, j \in [k]. \tag{E.6}$$

Specifically, (4.3) can be equivalently expressed as

$$\mathbb{P}_{e,\text{Avg}} \xrightarrow{P} \mathbb{P}\big(\arg\max_{\ell \in [k]} \{\gamma \cdot \text{diag}(\boldsymbol{\pi}) \cdot \widetilde{\boldsymbol{g}} + (\text{diag}(\boldsymbol{\pi}) - \boldsymbol{\Pi}) \cdot \boldsymbol{\Sigma}\boldsymbol{g}\} \neq Y(\boldsymbol{g})\big) \tag{E.7}$$

$$= \mathbb{P}\big(\bigcup_{j \neq Y}\{\gamma \cdot \boldsymbol{\pi}_\ell \cdot \widetilde{\boldsymbol{g}}_\ell \geq \gamma \cdot \boldsymbol{\pi}_Y \cdot \widetilde{\boldsymbol{g}}_Y + (\boldsymbol{e}_Y - \boldsymbol{e}_\ell)^T (\text{diag}(\boldsymbol{\pi}) - \boldsymbol{\Pi})\boldsymbol{\Sigma}\boldsymbol{g} + (\boldsymbol{\pi}_Y - \boldsymbol{\pi}_\ell)\}\big), \tag{E.8}$$

where $\boldsymbol{g}, \widetilde{\boldsymbol{g}} \overset{iid}{\sim} \mathcal{N}(\mathbf{0}, \boldsymbol{I}_k)$, $\mathbb{P}(Y(\boldsymbol{g}) = c) = \frac{e^{\mu_c \boldsymbol{g}_c}}{\sum_{\ell \in [k]} e^{\mu_\ell \boldsymbol{g}_\ell}}$ and $\boldsymbol{\Sigma} = \text{diag}(\mu_1, \ldots, \mu_k)$.

## F   On the Bayes risk of GMM: Proof of Proposition 3.4

Without loss of generality in this proof we assume $\sigma = 1$. The general result follows by simply replacing $(\mu, \sigma)$ with $(\frac{\mu}{\sigma}, 1)$ and using the proof for $\sigma = 1$. Recall that the feature vectors $\boldsymbol{x}_1, \ldots, \boldsymbol{x}_n$ of the training data set are given by:

$$\boldsymbol{x}_i = \boldsymbol{M}\boldsymbol{y}_i + \boldsymbol{z}_i, \quad i \in [n],$$

where the matrix of means $\boldsymbol{M} \in \mathbb{R}^{d \times k}$ has iid Gaussian entries with variance $\mu^2/d$, $\boldsymbol{z}_i \overset{iid}{\sim} \mathcal{N}(\mathbf{0}, \boldsymbol{I}_d)$ and $\boldsymbol{y}_i \overset{iid}{\sim} \text{Unif}(\boldsymbol{e}_1, \ldots, \boldsymbol{e}_k)$ with $\boldsymbol{e}_j$ denoting the $j^{th}$ canonical vector in $\mathbb{R}^k$. By definition here the Bayes estimator is the maximum-likelihood (ML) estimator. By applying the law of total probability and by successive application of the Bayes rule we have the following chain of reformulations of the ML:

$$\hat{\boldsymbol{y}}_{n+1} = \arg\max_{\boldsymbol{e}_j, \, j \in [k]} P\{\boldsymbol{y} = \boldsymbol{e}_j \mid \boldsymbol{X}, \boldsymbol{Y}, \boldsymbol{x}_{n+1}\}$$

$$= \arg\max_{\boldsymbol{e}_j, \, j \in [k]} \int P\{\boldsymbol{y} = \boldsymbol{e}_j \mid \boldsymbol{M}, \boldsymbol{X}, \boldsymbol{Y}, \boldsymbol{x}_{n+1}\} P\{\boldsymbol{M} \mid \boldsymbol{X}, \boldsymbol{Y}, \boldsymbol{x}_{n+1}\} \, d\boldsymbol{M}$$

$$= \arg\max_{\boldsymbol{e}_j, \, j \in [k]} \int \frac{P\{\boldsymbol{x}_{n+1} \mid \boldsymbol{y} = \boldsymbol{e}_j, \boldsymbol{M}, \boldsymbol{X}, \boldsymbol{Y}\} \cdot P\{\boldsymbol{y} = \boldsymbol{e}_j \mid \boldsymbol{M}, \boldsymbol{X}, \boldsymbol{Y}\}}{P\{\boldsymbol{x}_{n+1} \mid \boldsymbol{M}, \boldsymbol{X}, \boldsymbol{Y}\}} P\{\boldsymbol{M} \mid \boldsymbol{X}, \boldsymbol{Y}, \boldsymbol{x}_{n+1}\} \, d\boldsymbol{M}$$

$$= \arg\max_{\boldsymbol{e}_j, \, j \in [k]} \int P\{\boldsymbol{x}_{n+1} \mid \boldsymbol{y} = \boldsymbol{e}_j, \boldsymbol{M}\} \frac{P\{\boldsymbol{M} \mid \boldsymbol{X}, \boldsymbol{Y}, \boldsymbol{x}_{n+1}\}}{P\{\boldsymbol{x}_{n+1} \mid \boldsymbol{M}, \boldsymbol{X}, \boldsymbol{Y}\}} \, d\boldsymbol{M} \tag{F.1}$$

$$= \arg\max_{\boldsymbol{e}_j, \, j \in [k]} \int P\{\boldsymbol{x}_{n+1} \mid \boldsymbol{y} = \boldsymbol{e}_j, \boldsymbol{M}\} \frac{P\{\boldsymbol{M} \mid \boldsymbol{X}, \boldsymbol{Y}\}}{P\{\boldsymbol{x}_{n+1} \mid \boldsymbol{X}, \boldsymbol{Y}\}} \, d\boldsymbol{M}$$

$$= \arg\max_{\boldsymbol{e}_j, \, j \in [k]} \int P\{\boldsymbol{x}_{n+1} \mid \boldsymbol{y} = \boldsymbol{e}_j, \boldsymbol{M}\} P\{\boldsymbol{M} \mid \boldsymbol{X}, \boldsymbol{Y}\} \, d\boldsymbol{M} \tag{F.2}$$

$$= \arg\max_{\boldsymbol{e}_j, \, j \in [k]} \int P\{\boldsymbol{x}_{n+1} \mid \boldsymbol{y} = \boldsymbol{e}_j, \boldsymbol{M}\} P\{\boldsymbol{X} \mid \boldsymbol{M}, \boldsymbol{Y}\} P\{\boldsymbol{M}\} \, d\boldsymbol{M} \tag{F.3}$$

To arrive in (F.1) we used that $P(\boldsymbol{y} = \boldsymbol{e}_j \mid \boldsymbol{M}, \boldsymbol{X}, \boldsymbol{Y}) = \pi, \; \forall j \in [k]$ and $P(\boldsymbol{x}_{n+1} \mid \boldsymbol{y} = \boldsymbol{e}_j, \boldsymbol{M}, \boldsymbol{X}, \boldsymbol{Y}) = P(\boldsymbol{x}_{n+1} \mid \boldsymbol{y} = \boldsymbol{e}_j, \boldsymbol{M})$. Also, (F.2) follows by recognizing that $P(\boldsymbol{x}_{n+1} \mid \boldsymbol{X}, \boldsymbol{Y}) > 0$ is independent of the variable of integration $\boldsymbol{M}$ and of the optimization variable $j$. For the same reasons, in (F.3) we have ignored the normalizing term $P(\boldsymbol{X} | \boldsymbol{Y})$.

Recalling that $\boldsymbol{z}_{n+1} \sim \mathcal{N}(\mathbf{0}, \boldsymbol{I}_d)$, we have that $P(\boldsymbol{x}_{n+1} \mid \boldsymbol{y} = \boldsymbol{e}_j, \boldsymbol{M}) \propto \exp\big(-\|\boldsymbol{x}_{n+1} - \boldsymbol{\mu}_j\|_{\ell_2}^2/2\big)$ where $\propto$ hides constant positive terms. Moreover, the posterior probability of the mean matrix given

the training data is given by

$$P\left(\boldsymbol{X} \mid \boldsymbol{M}, \boldsymbol{Y}\right) \cdot P\left(\boldsymbol{M}\right) \propto \exp\left(-\frac{\|\boldsymbol{X} - \boldsymbol{M}\boldsymbol{Y}\|_{\ell_2}^2}{2}\right) \cdot \exp\left(-\frac{\|\boldsymbol{M}\|_{\ell_2}^2}{2(\mu^2/d)}\right)$$

$$\propto \prod_{c=1}^{k}\left\{\exp\left(-\frac{\|\boldsymbol{\mu}_c\|_{\ell_2}^2}{2(\mu^2/d)}\right) \cdot \prod_{i\in\mathcal{C}_c}\exp\left(-\frac{\|\boldsymbol{x}_i - \boldsymbol{\mu}_c\|_{\ell_2}^2}{2}\right)\right\}, \qquad \text{(F.4)}$$

where we denote by $\mathcal{C}_c$ the collection of training samples that belong to class $c \in [k]$, i.e. $\mathcal{C}_c = \{i \in [n] \mid \boldsymbol{y}_i = \boldsymbol{e}_c\}$.

With these the objective function of the ML rule in (F.3) becomes:

$$\hat{\boldsymbol{y}}_{n+1} = \arg\max_{j\in[k]} \mathcal{I}(j, \mathcal{C}_j \cap \{n+1\}) \cdot \prod_{\substack{c=1 \\ c\neq j}}^{k} \mathcal{I}(c, \mathcal{C}_c), \qquad \text{(F.5)}$$

where for $\ell \in [k]$ and a subset $\mathcal{A} \subset [n+1]$ we denote

$$\mathcal{I}(\ell, \mathcal{A}) := \int \mathrm{d}\boldsymbol{\mu}_\ell \exp\left(-\frac{\|\boldsymbol{\mu}_\ell\|_{\ell_2}^2}{2(\mu^2/d)}\right) \cdot \exp\left(-\sum_{i\in\mathcal{A}} \frac{\|\boldsymbol{x}_i - \boldsymbol{\mu}_\ell\|_{\ell_2}^2}{2}\right).$$

By completing the squares and invoking a gaussian integral it can be shown that

$$\mathcal{I}(\ell, \mathcal{A}) := \sqrt{\frac{(d/\mu^2 + |\mathcal{A}|)}{(2\varpi)^d}} \exp\left(-\frac{\left(1 - \frac{1}{d/\mu^2+|\mathcal{A}|}\right)}{2}\sum_{i\in\mathcal{A}}\|\boldsymbol{x}_i\|_{\ell_2}^2 + \frac{1}{2\left(d/\mu^2 + |\mathcal{A}|\right)}\sum_{i\in\mathcal{A}}\langle\boldsymbol{x}_i, \sum_{\substack{j\in\mathcal{A}\\j\neq i}}\boldsymbol{x}_j\rangle\right)$$

$$:= \sqrt{\frac{(d/\mu^2 + |\mathcal{A}|)}{(2\varpi)^d}} \exp\left(-\frac{1}{2\left(\frac{d}{\mu^2} + |\mathcal{A}|\right)}\left(\left(\frac{d}{\mu^2} + |\mathcal{A}| - 1\right)\sum_{i\in\mathcal{A}}\|\boldsymbol{x}_i\|_{\ell_2}^2 - \sum_{i\in\mathcal{A}}\langle\boldsymbol{x}_i, \sum_{\substack{j\in\mathcal{A}\\j\neq i}}\boldsymbol{x}_j\rangle\right)\right)$$

$$:= \sqrt{\frac{(d/\mu^2 + |\mathcal{A}|)}{(2\varpi)^d}} \exp\left(-\frac{1}{2\left(\frac{d/n}{\mu^2} + |\mathcal{A}|/n\right)}\left(\left(\frac{d/n}{\mu^2} + \frac{|\mathcal{A}|}{n} - \frac{1}{n}\right)\sum_{i\in\mathcal{A}}\|\boldsymbol{x}_i\|_{\ell_2}^2 - \frac{1}{n}\sum_{i\in\mathcal{A}}\langle\boldsymbol{x}_i, \sum_{\substack{j\in\mathcal{A}\\j\neq i}}\boldsymbol{x}_j\rangle\right)\right).$$

Using this in (F.5) we have that

$$\hat{\boldsymbol{y}}_{n+1} = \arg\max_{j\in[k]} \mathcal{I}(j) \cdot \exp\left(-\frac{1}{2\left(\frac{d/n}{\mu^2} + \frac{n_j+1}{n}\right)}\left(\left(\frac{d/n}{\mu^2} + \frac{n_j}{n}\right)\|\boldsymbol{x}_{n+1}\|_{\ell_2}^2 - \frac{2}{n}\langle\boldsymbol{x}_{n+1}, \sum_{\ell\in\mathcal{C}_j}\boldsymbol{x}_\ell\rangle\right)\right),$$

$$\text{(F.6)}$$

where $\xi(n_c) := \frac{d/n}{\mu^2} + \frac{n_c}{n},\ c \in [k]$ and

$$\mathcal{I}(j) := \left\{\prod_{\substack{c=1\\c\neq j}}^{k} e^{-\frac{1}{2\xi(n_c)}\left(\left(\xi(n_c) - \frac{1}{n}\right)\sum_{i\in\mathcal{C}_c}\|\boldsymbol{x}_i\|_{\ell_2}^2 - \frac{1}{n}\sum_{i\in\mathcal{C}_c}\langle\boldsymbol{x}_i, \Sigma_{\substack{\ell\in\mathcal{C}_c\\\ell\neq i}}\boldsymbol{x}_\ell\rangle\right)}\right\} \cdot e^{-\frac{1}{2\left(\xi(n_j)+\frac{1}{n}\right)}\left(\xi(n_j)\sum_{i\in\mathcal{C}_j}\|\boldsymbol{x}_i\|_{\ell_2}^2 - \frac{1}{n}\sum_{i\in\mathcal{C}_j}\langle\boldsymbol{x}_i, \Sigma_{\substack{\ell\in\mathcal{C}_j\\\ell\neq i}}\boldsymbol{x}_\ell\rangle\right)}.$$

We conclude that

$$\hat{\boldsymbol{y}}_{n+1} = \arg\max_{j\in[k]} \log\left(\mathcal{I}(j)\right) - \frac{1}{2\left(\frac{d/n}{\mu^2} + \frac{n_j+1}{n}\right)}\left\{\left(\frac{d/n}{\mu^2} + \frac{n_j}{n}\right)\|\boldsymbol{x}_{n+1}\|_{\ell_2}^2 - \frac{2}{n}\langle\boldsymbol{x}_{n+1}, \sum_{\ell\in\mathcal{C}_j}\boldsymbol{x}_\ell\rangle\right\}$$

$$= \arg\max_{j\in[k]} \log\left(\mathcal{I}(j)\right) + \frac{1}{2\left(\frac{d/n}{\mu^2} + \frac{n_j+1}{n}\right)}\left\{\frac{2}{n}\langle\boldsymbol{x}_{n+1}, \sum_{\ell\in\mathcal{C}_j}\boldsymbol{x}_\ell\rangle\right\}. \qquad \text{(F.7)}$$

Next, we evaluate the objective in (F.7) in the asymptotic limit $n, d \to \infty, n/d = \gamma$. First, since $n_c/n \xrightarrow{P} \pi$, note that $\mathcal{I}(j) - \mathcal{I}(\ell) \xrightarrow{P} 0$ for all $\ell, j \in [k]$. Moreover, note that

$$\frac{1}{n}\langle \boldsymbol{x}_{n+1}, \sum_{\ell \in \mathcal{C}_j} \boldsymbol{x}_\ell \rangle = \frac{1}{n}\langle \boldsymbol{M}\boldsymbol{y}_{n+1} + \boldsymbol{z}_{n+1}, n_j \boldsymbol{\mu}_j + \sum_{\ell \in \mathcal{C}_j} \boldsymbol{z}_\ell \rangle$$

$$= \frac{n_j}{n}\langle \boldsymbol{M}\boldsymbol{y}_{n+1}, \boldsymbol{\mu}_j \rangle + \frac{n_j}{n}\langle \boldsymbol{z}_{n+1}, \boldsymbol{\mu}_j \rangle + \frac{1}{n}\sum_{\ell \in \mathcal{C}_j}\langle \boldsymbol{M}\boldsymbol{y}_{n+1}, \boldsymbol{z}_\ell \rangle + \frac{1}{n}\sum_{\ell \in \mathcal{C}_j}\langle \boldsymbol{z}_{n+1}, \boldsymbol{z}_\ell \rangle$$

(F.8)

For each one of the four terms in (F.8), we have the following by the CLT:

$$\frac{n_j}{n}\langle \boldsymbol{M}\boldsymbol{y}_{n+1}, \boldsymbol{\mu}_\ell \rangle \xrightarrow{P} \pi \mu^2 \langle \boldsymbol{y}_{n+1}, \boldsymbol{e}_j \rangle$$

$$\frac{n_j}{n}\langle \boldsymbol{z}_{n+1}, \boldsymbol{\mu}_j \rangle \xrightarrow{(D)} \mathcal{N}(0, \pi^2 r^2)$$

$$\frac{1}{n}\sum_{\ell \in \mathcal{C}_j}\langle \boldsymbol{M}\boldsymbol{y}_{n+1}, \boldsymbol{z}_\ell \rangle \xrightarrow{P} 0$$

$$\frac{1}{n}\sum_{\ell \in \mathcal{C}_j}\langle \boldsymbol{z}_{n+1}, \boldsymbol{z}_\ell \rangle \xrightarrow{(D)} \mathcal{N}(0, \pi\gamma),$$

where in the last line we used the fact that $\frac{1}{\sqrt{n_j}}\sum_{\ell \in \mathcal{C}_j} \frac{\langle \boldsymbol{z}_{n+1}, \boldsymbol{z}_\ell \rangle}{\sqrt{n}} \xrightarrow{(D)} \mathcal{N}(0, \gamma)$.

Therefore, in the asymptotic limit, the Bayes estimator is the solution to:

$$\hat{\boldsymbol{y}}_{n+1} = \arg\max_{\boldsymbol{e}_j, j \in [k]} \pi \mu^2 \langle \boldsymbol{y}_{n+1}, \boldsymbol{e}_j \rangle + \sqrt{\pi(\pi\mu^2 + \gamma)} G_j, \quad G_1, \ldots, G_k \overset{iid}{\sim} \mathcal{N}(0, 1). \quad \text{(F.9)}$$

As such, the probability of error is

$$\mathbb{P}_e = \mathbb{P}\{\hat{\boldsymbol{y}}_{n+1} \neq \boldsymbol{y}_{n+1}\} = \mathbb{P}\left\{\pi\mu^2 + \sqrt{\pi(\pi\mu^2 + \gamma)} G_0 \le \max_{\ell \in [k-1]} \sqrt{\pi(\pi\mu^2 + \gamma)} G_\ell\right\}$$

$$= \mathbb{P}\left\{G_0 + \max_{\ell \in [k-1]} G_\ell \ge \mu^2 \sqrt{\frac{\pi}{\pi\mu^2 + \gamma}}\right\}. \quad \text{(F.10)}$$

# G   Proof outline for least-squares: key ideas and challenges

In this section, we provide a proof sketch for the analysis of the multiclass least-squares (LS) classifier.

Specifically, we discuss our approach towards specifying the high-dimensional limits of the key quantities needed to evaluate the classification error: $\boldsymbol{b}, \boldsymbol{\Sigma}_{\boldsymbol{w},\boldsymbol{\mu}}$, and, $\boldsymbol{\Sigma}_{\boldsymbol{w},\boldsymbol{w}}$. For simplicity, we focus here on the performance of the LS classifier GMM. We note that our proofs for the MLM and the Weighted Least-Squares (WLS) classifiers follow the same general strategy, but in some parts require more involved and intricate analysis and derivations. Our proof follows the following general steps; see the appendix for complete details and derivations.

**Step I: Decomposing the loss across classes.** Recall from Section 2.2 that the multiclass LS classifier produces a linear classifier $\boldsymbol{x} \mapsto \boldsymbol{W}\boldsymbol{x} + \boldsymbol{b}$ via a least-squares fit to the training data:

$$(\widehat{\boldsymbol{W}}, \widehat{\boldsymbol{b}}) := \frac{1}{2n}\left\|\boldsymbol{W}\boldsymbol{X} + \boldsymbol{b}\mathbf{1}_n^T - \boldsymbol{Y}\right\|_F^2. \quad \text{(G.1)}$$

Notice that the objective function above is separable. That is,

$$\frac{1}{2n}\left\|\boldsymbol{W}\boldsymbol{X} + \boldsymbol{b}\mathbf{1}_n^T - \boldsymbol{Y}\right\|_F^2 = \frac{1}{2n}\sum_{\ell=1}^k \left\|\boldsymbol{X}^T \boldsymbol{w}_\ell + b_\ell \mathbf{1}_n - \boldsymbol{Y}_\ell\right\|_{\ell_2}^2.$$

Hence, for each $\ell \in [k]$,

$$(\widehat{\boldsymbol{w}}_\ell, \widehat{b}_\ell) = \arg\min_{\boldsymbol{w}_\ell, b_\ell} \frac{1}{2n}\left\|\boldsymbol{X}^T \boldsymbol{w}_\ell + b_\ell \mathbf{1}_n - \boldsymbol{Y}_\ell\right\|_{\ell_2}^2. \quad \text{(G.2)}$$

This decomposition is convenient for analysis as it is easier to compute the statistical properties of the simple single-output LS in (G.2) compared to the multi-output objective in (G.1). Indeed, as we show, in Step III, this simplification will eventually allow us to compute the high-dimensional behavior of the following key quantities for all $\ell \in [k]$: (i) the intercept $\widehat{b}_\ell$, (ii) the mean-correlations $\langle \widehat{w}_\ell, \mu_c \rangle$, $c \in [k]$, (iii) the norm $\|\widehat{w}_\ell\|_{\ell_2}$.

**Step II: Reduction to an Auxiliary Optimization (AO) problem via CGMT.** To calculate the high-dimensional statistical behavior of (G.2) we use the Convex Gaussian min-max Theorem (CGMT) [Sto13, TOH15] framework. We provide a brief introduction of the CGMT machinery in Section G.1. Roughly stated, this framework allows us to replace a *Primary Optimization* (PO) problem of the form (G.2) with an *Auxiliary Optimization* (AO) problem that is simpler to analyze, but is predictive of the behavior of the latter. For instance, for the PO in (G.2) in the GMM, after some algebraic manipulations, the AO problem takes the form

$$\frac{1}{2} \left( \min_{w_\ell, b_\ell} \; \frac{1}{\sqrt{n}} \left\| \sigma \|w_\ell\|_{\ell_2} \, g + Y^T M^T w_\ell + b_\ell 1_n - Y_\ell \right\|_{\ell_2} + \frac{1}{\sqrt{n}} \sigma h^T w_\ell \right)_+^2, \qquad (G.3)$$

where $(x)_+ := \max(0, x)$ and $g \in \mathbb{R}^n$ and $h \in \mathbb{R}^d$ are two independent Gaussian random vectors distributed as $\mathcal{N}(0, I_n)$ and $\mathcal{N}(0, I_d)$.

**Step III: Simplification of the AO and computing $\Sigma_{w,\mu}$ and $b$.** In this step we carry out a series of intricate calculations to further simplify (G.3) and characterize its various asymptotic properties. At a high-level, we follow the principled machinery introduced in [TOH15, TAH18], organizing our analysis in three intermediate steps: (a) Scalarization; (b) Convergence analysis; and (c) Deterministic analysis. We note that each one of these intermediate steps for the multiclass setting is more involved than in previously considered regression and binary classification settings. The detailed derivations are deferred to the Appendix H.1. At the end of this analysis step, we have computed the high-dimensional behavior of the intercepts $\widehat{b}_\ell$, $\ell \in [k]$, the mean-correlations $\langle \widehat{w}_\ell, \mu_c \rangle$, $\ell, c \in [k]$, the norms $\|\widehat{w}_\ell\|_{\ell_2}$, $\ell \in [k]$ and the LS training loss $\|X^T \widehat{w}_\ell + \widehat{b}_\ell 1_n - Y_\ell\|_{\ell_2}$. In particular, for GMM these calculations allow us to conclude the following limits for all $\ell \in [k]$:

$$\widehat{b}_\ell \xrightarrow{P} \pi_\ell \left(1 - (e_\ell - \pi)^T V \Sigma \Delta^{-1} \Sigma V^T \pi \right), \qquad M^T \widehat{w}_\ell \xrightarrow{P} \pi_\ell V \Sigma \Delta^{-1} \Sigma V^T (e_\ell - \pi),$$
$$(G.4)$$

and

$$\|w_\ell\|_{\ell_2}^2 \xrightarrow{P} \frac{\gamma}{(1-\gamma)\sigma^2} \pi_\ell (1 - \pi_\ell) + \pi_\ell^2 (e_\ell - \pi)^T V \Sigma \Delta^{-1} \left( \Delta^{-1} - \frac{\gamma}{(1-\gamma)\sigma^2} I_r \right) \Sigma V^T (e_\ell - \pi).$$
$$(G.5)$$

where $\Delta := \sigma^2 I_r + \Sigma V^T P V \Sigma > 0_{r \times r}$ and $P := \operatorname{diag}(\pi) - \pi \pi^T$.

Expressing (G.4) in matrix form leads to (3.3a) in Theorem 3.2. Thus, it remains to prove (3.3b), i.e., to determine the high-dimensional limit of $\Sigma_{w,w}$. Note that (G.5) already determines the diagonal entries of $\Sigma_{w,w}$. However, thus far, our analysis treats the optimization of each classifier $\widehat{w}_\ell$, $\ell \in [k]$ independently and provides no information for the cross-correlation $\langle \widehat{w}_\ell, \widehat{w}_c \rangle, \ell \neq c \in [k]$

**Step IV: Computing $\Sigma_{w,w}$ and capturing cross-correlations.** The final and most involved part of our analysis is characterizing the asymptotic behavior of $\Sigma_{w,w}$. To see why this is particularly challenging note that the reduction from (G.1) to (G.2) "breaks" the dependence of all $\widehat{w}_1, \widehat{w}_2, \ldots, \widehat{w}_k$ on the *same* feature matrix $X$. Capturing this dependence is crucial in determining the "cross-correlations" $\langle \widehat{w}_\ell, \widehat{w}_c \rangle$, $\ell \neq c$. As noted in Section 2.3 the matrix $\Sigma_{w,w}$ is needed to calculate the class-wise and total miss-classification errors. Unfortunately, the CGMT is *not* directly applicable to the multi-output LS optimization in (G.1). Our idea to circumvent this challenge builds on the following simple observation: the vector $\widehat{w}_{\ell,c} = \widehat{w}_\ell + \widehat{w}_c$ is itself the solution to another simple single-output LS problem.

**Lemma G.1** *For $\ell \neq c \in [k]$, let $\widehat{w}_\ell$, $\widehat{w}_c$ be the $\ell$ and $c$-th row of $\widehat{W}$ which is the solution to the multi-output least-squares minimization* (G.1). *Denote $\widehat{w}_{\ell,c} := \widehat{w}_\ell + \widehat{w}_c$. Then, $\widehat{w}_{\ell,c}$ is a minimizer in*

*the following single-output least-squares problem:*

$$\widehat{\boldsymbol{w}}_{\ell,c} = \arg\min_{\boldsymbol{w},b} \frac{1}{2n}\left\| \boldsymbol{Y}_\ell + \boldsymbol{Y}_c - \boldsymbol{X}^T\boldsymbol{w} - b\mathbf{1}_n \right\|_{\ell_2}^2.$$

Thanks to Lemma G.1, we can use the CGMT to characterize the limiting behavior of $\|\widehat{\boldsymbol{w}}_\ell + \widehat{\boldsymbol{w}}_c\|_{\ell_2}$. These calculations are similar to (but, in certain cases, such as for weighted least-squares, more involved than) those in Steps II and III above. Now note that an asymptotic characterization of $\|\widehat{\boldsymbol{w}}_\ell + \widehat{\boldsymbol{w}}_c\|_{\ell_2}$ immediately yields the asymptotic characterization of $\langle\widehat{\boldsymbol{w}}_\ell, \widehat{\boldsymbol{w}}_c\rangle$ as

$$\langle\widehat{\boldsymbol{w}}_\ell, \widehat{\boldsymbol{w}}_c\rangle = \frac{\|\widehat{\boldsymbol{w}}_\ell + \widehat{\boldsymbol{w}}_c\|_{\ell_2}^2 - \|\widehat{\boldsymbol{w}}_\ell\|_{\ell_2}^2 - \|\widehat{\boldsymbol{w}}_c\|_{\ell_2}^2}{2}, \tag{G.6}$$

and $\|\widehat{\boldsymbol{w}}_\ell\|_{\ell_2}$, $\|\widehat{\boldsymbol{w}}_c\|_{\ell_2}$ are already computed in Step IV (cf. (G.5)). For the GMM, the analysis in this step allow us to calculate the asymptotic behavior of $\boldsymbol{\Sigma}_{w,w}$ as promised in (3.3b) in Theorem 3.2:

$$\boldsymbol{\Sigma}_{\boldsymbol{w},\boldsymbol{w}} \xrightarrow{P} \frac{\gamma}{(1-\gamma)\sigma^2}\boldsymbol{P} + \boldsymbol{P}\boldsymbol{V}\boldsymbol{\Sigma}\boldsymbol{\Delta}^{-1}\Big(\boldsymbol{\Delta}^{-1} - \frac{\gamma}{(1-\gamma)\sigma^2}\boldsymbol{I}_r\Big)\boldsymbol{\Sigma}\boldsymbol{V}^T\boldsymbol{P}.$$

## G.1  Background on the CGMT

The CGMT is an extension of Gordon's Gaussian min-max inequality (GMT) [Gor88]. In the context of high-dimensional inference problems, Gordon's inequality was first successfully used in the study oh sharp phase-transitions in noiseless Compressed Sensing [Sto09, CRPW12, ALMT13, Sto09]. More recently, [Sto13] (see also [ALMT13, Sec. 10.3]) discovered that Gordon's inequality is essentially tight for certain convex problems. A concrete and general formulation of this idea was given by [TOH15] and was called the CGMT.

In order to summarize the essential ideas, consider the following two Gaussian processes:

$$X_{\boldsymbol{w},\boldsymbol{u}} := \boldsymbol{u}^T\boldsymbol{G}\boldsymbol{w} + \psi(\boldsymbol{w},\boldsymbol{u}), \tag{G.7a}$$

$$Y_{\boldsymbol{w},\boldsymbol{u}} := \|\boldsymbol{w}\|_{\ell_2}\boldsymbol{g}^T\boldsymbol{u} + \|\boldsymbol{u}\|_{\ell_2}\boldsymbol{h}^T\boldsymbol{w} + \psi(\boldsymbol{w},\boldsymbol{u}), \tag{G.7b}$$

where: $\boldsymbol{G} \in \mathbb{R}^{n\times d}$, $\boldsymbol{g} \in \mathbb{R}^n$, $\boldsymbol{h} \in \mathbb{R}^d$, they all have entries iid Gaussian; the sets $\mathcal{S}_{\boldsymbol{w}} \subset \mathbb{R}^d$ and $\mathcal{S}_{\boldsymbol{u}} \subset \mathbb{R}^n$ are compact; and, $\psi : \mathbb{R}^d \times \mathbb{R}^n \to \mathbb{R}$. For these two processes, define the following (random) min-max optimization programs, which are refered to as the *primary optimization* (PO) problem and the *auxiliary optimization* AO:

$$\Phi(\boldsymbol{G}) = \min_{\boldsymbol{w}\in\mathcal{S}_{\boldsymbol{w}}} \max_{\boldsymbol{u}\in\mathcal{S}_{\boldsymbol{u}}} X_{\boldsymbol{w},\boldsymbol{u}}, \tag{G.8a}$$

$$\phi(\boldsymbol{g},\boldsymbol{h}) = \min_{\boldsymbol{w}\in\mathcal{S}_{\boldsymbol{w}}} \max_{\boldsymbol{u}\in\mathcal{S}_{\boldsymbol{u}}} Y_{\boldsymbol{w},\boldsymbol{u}}. \tag{G.8b}$$

If the sets $\mathcal{S}_{\boldsymbol{w}}$ and $\mathcal{S}_{\boldsymbol{u}}$ are convex and *bounded*, and $\psi$ is continuous *convex-concave* on $\mathcal{S}_{\boldsymbol{w}} \times \mathcal{S}_{\boldsymbol{u}}$, then, for any $\nu \in \mathbb{R}$ and $t > 0$, it holds [TOH15, Thm. 3]:

$$\mathbb{P}\left(|\Phi(\boldsymbol{G}) - \nu| > t\right) \leq 2\,\mathbb{P}\left(|\phi(\boldsymbol{g},\boldsymbol{h}) - \nu| > t\right). \tag{G.9}$$

In words, concentration of the optimal cost of the AO problem around $q^*$ implies concentration of the optimal cost of the corresponding PO problem around the same value $q^*$. Asymptotically, if we can show that $\phi(\boldsymbol{g},\boldsymbol{h}) \xrightarrow{P} q^*$, then we can conclude that $\Phi(\boldsymbol{G}) \xrightarrow{P} q^*$. Moreover, starting from (G.9) and under appropriate strict convexity conditions, the CGMT shows that concentration of the optimal solution of the AO problem implies concentration of the optimal solution of the PO around the same value. For example, if minimizers of (G.8b) satisfy $\|\boldsymbol{w}_\phi(\boldsymbol{g},\boldsymbol{h})\|_{\ell_2} \xrightarrow{P} \alpha^*$ for some $\alpha^* > 0$, then, the same holds true for the minimizers of (G.8a): $\|\boldsymbol{w}_\Phi(\boldsymbol{G})\|_{\ell_2} \xrightarrow{P} \alpha^*$. Thus, one can analyze the AO to infer corresponding properties of the PO, the premise being of course that the former is simpler to handle than the latter.

In [TAH18], the authors introduce a principled machinery that allows to (a) express a quite general family of convex inference optimization problems in the form of the PO and (b) properly analyze the corresponding AO. In particular, the analysis of the AO is performed in three intermediate steps. First, the (random) optimization over vector variables is simplified to an easier optimization over only few scalar variables, termed the "scalarized AO". After the scalarization step, it is possible to establish (uniform) convergence of the scalarized AO to a deterministic min-max optimization problem over only a few scalar variables. The convergence step is followed by the analysis of the latter deterministic problem, which leads to the desired asymptotic characterizations.

# H  Least-squares for GMM

## H.1  Proof of Theorem 3.2

### H.1.1  Computing $\Sigma_{w,\mu}$

The LS classifier solves:

$$
\min_{\boldsymbol{W}\in\mathbb{R}^{k\times d},\, \boldsymbol{b}\in\mathbb{R}^k} \quad \frac{1}{2n}\left\|\boldsymbol{W}\boldsymbol{X}+\boldsymbol{b}\mathbf{1}_n^T-\boldsymbol{Y}\right\|_F^2 = \sum_{\ell=1}^{k}\min_{\boldsymbol{w}_\ell,b_\ell}\ \frac{1}{2n}\left\|\boldsymbol{X}^T\boldsymbol{w}_\ell+b_\ell\mathbf{1}_n-\boldsymbol{Y}_\ell\right\|_{\ell_2}^2
$$

$$
=\sum_{\ell=1}^{k}\min_{\boldsymbol{w}_\ell,b_\ell}\ \frac{1}{2n}\left\|\boldsymbol{Y}^T\boldsymbol{M}^T\boldsymbol{w}_\ell+\boldsymbol{Z}^T\boldsymbol{w}_\ell+b_\ell\mathbf{1}_n-\boldsymbol{Y}_\ell\right\|_{\ell_2}^2 .
$$

Define

$$
\mathcal{L}_{PO}\left(\boldsymbol{w}_\ell,b_\ell\right):=\frac{1}{2n}\left\|\boldsymbol{Y}^T\boldsymbol{M}^T\boldsymbol{w}_\ell+\boldsymbol{Z}^T\boldsymbol{w}_\ell+b_\ell\mathbf{1}_n-\boldsymbol{Y}_\ell\right\|_{\ell_2}^2 . \tag{H.1}
$$

**Identifying the AO.** To continue further note that by duality we have

$$
\min_{\boldsymbol{w}_\ell,b_\ell}\ \mathcal{L}_{PO}\left(\boldsymbol{w}_\ell,b_\ell\right)=\min_{\boldsymbol{w}_\ell,b_\ell}\ \max_{\boldsymbol{s}}\ \frac{1}{n}\left(\boldsymbol{s}^T\boldsymbol{Y}^T\boldsymbol{M}^T\boldsymbol{w}_\ell+\boldsymbol{s}^T\boldsymbol{Z}^T\boldsymbol{w}_\ell+b_\ell\boldsymbol{s}^T\mathbf{1}_n-\boldsymbol{s}^T\boldsymbol{Y}_\ell-\frac{\|\boldsymbol{s}\|_{\ell_2}^2}{2}\right).
$$

Note that the above is jointly convex in $(\boldsymbol{w}_\ell,b_\ell)$ and concave in $\boldsymbol{s}$ and the Gaussian matrix $\boldsymbol{Z}$ is independent of everything else. Thus, the objective is in the form of (G.7a) and so we consider the corresponding Auxiliary Optimization (AO) problem:

$$
\min_{\boldsymbol{w}_\ell,b_\ell}\ \max_{\boldsymbol{s}}\ \frac{1}{n}\left(\boldsymbol{s}^T\boldsymbol{Y}^T\boldsymbol{M}^T\boldsymbol{w}_\ell+\sigma\|\boldsymbol{w}_\ell\|_{\ell_2}\boldsymbol{g}^T\boldsymbol{s}+\sigma\|\boldsymbol{s}\|_{\ell_2}\boldsymbol{h}^T\boldsymbol{w}_\ell+b_\ell\boldsymbol{s}^T\mathbf{1}_n-\boldsymbol{s}^T\boldsymbol{Y}_\ell-\frac{\|\boldsymbol{s}\|_{\ell_2}^2}{2}\right),
$$

where $\boldsymbol{g}\in\mathbb{R}^n$ and $\boldsymbol{h}\in\mathbb{R}^d$ are independent Gaussian random vectors with i.i.d. $\mathcal{N}(0,1)$ entries. Maximizing over the direction of $\boldsymbol{s}$ and setting its norm $\beta=\|\boldsymbol{s}\|_{\ell_2}$ we arrive at

$$
\min_{\boldsymbol{w}_\ell,b_\ell}\ \max_{\beta\geq0}\ \frac{1}{n}\left(\beta\left\|\sigma\|\boldsymbol{w}_\ell\|_{\ell_2}\boldsymbol{g}+\boldsymbol{Y}^T\boldsymbol{M}^T\boldsymbol{w}_\ell+b_\ell\mathbf{1}_n-\boldsymbol{Y}_\ell\right\|_{\ell_2}+\beta\sigma\boldsymbol{h}^T\boldsymbol{w}_\ell-\frac{\beta^2}{2}\right)
$$

$$
=\min_{\boldsymbol{w}_\ell,b_\ell}\ \frac{1}{2n}\left(\left\|\sigma\|\boldsymbol{w}_\ell\|_{\ell_2}\boldsymbol{g}+\boldsymbol{Y}^T\boldsymbol{M}^T\boldsymbol{w}_\ell+b_\ell\mathbf{1}_n-\boldsymbol{Y}_\ell\right\|_{\ell_2}+\sigma\boldsymbol{h}^T\boldsymbol{w}_\ell\right)_+^2
$$

$$
=\frac{1}{2}\left(\min_{\boldsymbol{w}_\ell,b_\ell}\ \frac{1}{\sqrt{n}}\left\|\sigma\|\boldsymbol{w}_\ell\|_{\ell_2}\boldsymbol{g}+\boldsymbol{Y}^T\boldsymbol{M}^T\boldsymbol{w}_\ell+b_\ell\mathbf{1}_n-\boldsymbol{Y}_\ell\right\|_{\ell_2}+\frac{1}{\sqrt{n}}\sigma\boldsymbol{h}^T\boldsymbol{w}_\ell\right)_+^2
$$

**Scalarization of the AO.** For convenience, define

$$
\bar{\phi}_{AO,\ell}:=\min_{\boldsymbol{w}_\ell,b_\ell}\ \frac{1}{\sqrt{n}}\left\|\sigma\|\boldsymbol{w}_\ell\|_{\ell_2}\boldsymbol{g}+\boldsymbol{Y}^T\boldsymbol{M}^T\boldsymbol{w}_\ell+b_\ell\mathbf{1}_n-\boldsymbol{Y}_\ell\right\|_{\ell_2}+\frac{\sigma}{\sqrt{n}}\boldsymbol{h}^T\boldsymbol{w}_\ell. \tag{H.2}
$$

To continue, consider the singular value decomposition

$$
\boldsymbol{M}=\boldsymbol{U}\boldsymbol{\Sigma}\boldsymbol{V}^T=\begin{bmatrix}\boldsymbol{u}_1 & \boldsymbol{u}_2 & \ldots & \boldsymbol{u}_r\end{bmatrix}\operatorname{diag}(\sigma_1,\sigma_2,\ldots,\sigma_r)\begin{bmatrix}\boldsymbol{v}_1^T\\ \boldsymbol{v}_2^T\\ \ldots\\ \boldsymbol{v}_r^T\end{bmatrix}, \tag{H.3}
$$

with $\boldsymbol{U}\in\mathbb{R}^{d\times r}$, $\boldsymbol{\Sigma}\in\mathbb{R}^{r\times r}$, and $\boldsymbol{V}\in\mathbb{R}^{k\times r}$ where $r=\operatorname{rank}(\boldsymbol{M})\leq k$. We further decompose $\boldsymbol{w}_\ell$ in its projections on the orthogonal columns $\boldsymbol{u}_1,\ldots,\boldsymbol{u}_r$ of $\boldsymbol{U}$:

$$
\boldsymbol{w}_\ell=\sum_{i=1}^{r}\alpha_i\boldsymbol{u}_i+\alpha_0\boldsymbol{w}_\ell^\perp,
$$

where $\left\| \boldsymbol{w}_\ell^\perp \right\|_{\ell_2} = 1$ and $\boldsymbol{U}^T \boldsymbol{w}_\ell^\perp = \boldsymbol{0}$, $\alpha_0 \geq 0$ and we denote

$$\alpha_i := \boldsymbol{u}_i^T \boldsymbol{w}_\ell, i \in [r]. \tag{H.4}$$

We also define $\boldsymbol{\alpha} = \begin{bmatrix} \alpha_1 & \alpha_2 & \dots & \alpha_k \end{bmatrix}^T$. In this notation, we have

$$\bar{\phi}_{AO,\ell}(\boldsymbol{g}, \boldsymbol{h}) := \min_{\alpha_0 \geq 0,\, \boldsymbol{\alpha} \in \mathbb{R}^r,\, b_\ell} \frac{1}{\sqrt{n}} \left\| \sigma \sqrt{\alpha_0^2 + \|\boldsymbol{\alpha}\|_{\ell_2}^2}\, \boldsymbol{g} + \boldsymbol{Y}^T \boldsymbol{V} \boldsymbol{\Sigma} \boldsymbol{\alpha} + b_\ell \boldsymbol{1}_n - \boldsymbol{Y}_\ell \right\|_{\ell_2}$$

$$+ \sum_{i=1}^{r} \alpha_i \sigma \frac{\boldsymbol{h}^T \boldsymbol{u}_i}{\sqrt{n}} + \frac{\alpha_0 \sigma}{\sqrt{n}} \min_{\boldsymbol{w}_\ell^\perp} \left( \boldsymbol{h}^T \boldsymbol{w}_\ell^\perp \right)$$

$$= \min_{\alpha_0 \geq 0,\, \boldsymbol{\alpha} \in \mathbb{R}^r,\, b_\ell} \frac{1}{\sqrt{n}} \left\| \sigma \sqrt{\alpha_0^2 + \|\boldsymbol{\alpha}\|_{\ell_2}^2}\, \boldsymbol{g} + \boldsymbol{Y}^T \boldsymbol{V} \boldsymbol{\Sigma} \boldsymbol{\alpha} + b_\ell \boldsymbol{1}_n - \boldsymbol{Y}_\ell \right\|_{\ell_2}$$

$$+ \sum_{i=1}^{r} \alpha_i \sigma \frac{\boldsymbol{h}^T \boldsymbol{u}_i}{\sqrt{n}} - \alpha_0 \sigma \frac{\|\boldsymbol{h}^\perp\|_{\ell_2}}{\sqrt{n}}, \tag{H.5}$$

where in the second line we denote $\boldsymbol{h}^\perp$ the projection of $\boldsymbol{h}$ onto the complement subspace of the span of $\boldsymbol{u}_1, \dots, \boldsymbol{u}_r$ and we recalled that $\|\boldsymbol{w}_\ell^\perp\|_{\ell_2} = 1$ and $\langle \boldsymbol{w}_\ell^\perp, \boldsymbol{u}_i \rangle = 0, i \in [r]$.

**Convergence of the AO.** First, note that

$$\frac{1}{n} \left\| \boldsymbol{Y}^T \boldsymbol{V} \boldsymbol{\Sigma} \boldsymbol{\alpha} + b_\ell \boldsymbol{1}_n - \boldsymbol{Y}_\ell \right\|_{\ell_2}^2 = \frac{1}{n} \left\| \boldsymbol{Y}^T \left( \boldsymbol{V} \boldsymbol{\Sigma} \boldsymbol{\alpha} - \boldsymbol{e}_\ell \right) + b_\ell \boldsymbol{1}_n \right\|_{\ell_2}^2$$

$$= \frac{1}{n} \left\| \boldsymbol{Y}^T \left( \boldsymbol{V} \boldsymbol{\Sigma} \boldsymbol{\alpha} - \boldsymbol{e}_\ell \right) \right\|_{\ell_2}^2 + b_\ell^2 + \frac{2}{n} b_\ell \boldsymbol{1}_n^T \boldsymbol{Y}^T \left( \boldsymbol{V} \boldsymbol{\Sigma} \boldsymbol{\alpha} - \boldsymbol{e}_\ell \right)$$

$$= \operatorname{trace} \left( \left( \boldsymbol{V} \boldsymbol{\Sigma} \boldsymbol{\alpha} - \boldsymbol{e}_\ell \right)^T \operatorname{diag} \left( \frac{n_1}{n}, \frac{n_2}{n}, \dots, \frac{n_k}{n} \right) \left( \boldsymbol{V} \boldsymbol{\Sigma} \boldsymbol{\alpha} - \boldsymbol{e}_\ell \right) \right)$$

$$+ b_\ell^2 + 2 b_\ell \begin{bmatrix} \frac{n_1}{n} & \frac{n_2}{n} & \dots & \frac{n_k}{n} \end{bmatrix} \left( \boldsymbol{V} \boldsymbol{\Sigma} \boldsymbol{\alpha} - \boldsymbol{e}_\ell \right)$$

Thus

$$\frac{1}{n} \left\| \boldsymbol{Y}^T \boldsymbol{V} \boldsymbol{\Sigma} \boldsymbol{\alpha} + b_\ell \boldsymbol{1}_n - \boldsymbol{Y}_\ell \right\|_{\ell_2}^2 \xrightarrow{P} \operatorname{trace} \left( \left( \boldsymbol{V} \boldsymbol{\Sigma} \boldsymbol{\alpha} - \boldsymbol{e}_\ell \right)^T \operatorname{diag} \left( \boldsymbol{\pi} \right) \left( \boldsymbol{V} \boldsymbol{\Sigma} \boldsymbol{\alpha} - \boldsymbol{e}_\ell \right) \right)$$

$$+ b_\ell^2 + 2 b_\ell \boldsymbol{\pi}^T \left( \boldsymbol{V} \boldsymbol{\Sigma} \boldsymbol{\alpha} - \boldsymbol{e}_\ell \right)$$

$$= \boldsymbol{\alpha}^T \left( \boldsymbol{\Sigma} \boldsymbol{V}^T \operatorname{diag}(\boldsymbol{\pi}) \boldsymbol{V} \boldsymbol{\Sigma} \right) \boldsymbol{\alpha} - 2 \pi_\ell \boldsymbol{\alpha}^T \boldsymbol{\Sigma} \boldsymbol{V}^T \boldsymbol{e}_\ell + 2 b_\ell \boldsymbol{\alpha}^T \boldsymbol{\Sigma} \boldsymbol{V}^T \boldsymbol{\pi}$$

$$+ b_\ell^2 - 2 b_\ell \pi_\ell + \pi_\ell.$$

At this point, observe that we have reduced the AO to an optimization problem over only $r + 2$ scalar variables. Using the law of large numbers, the fact that $\|\boldsymbol{h}^\perp\|_{\ell_2}$ concentrates around $\sqrt{d - r}$ and $(d - r)/n \xrightarrow{P} \gamma$, as well as the limit calculation above, it is not hard to see that for fixed $\alpha_0, b_\ell$ and $\boldsymbol{\alpha} = [\alpha_1, \dots, \alpha_r]^T \in \mathbb{R}^r$, the objective function in (H.5) converges to the following:

$$\mathcal{D}_\ell(\alpha_0, \boldsymbol{\alpha}, b_\ell)$$

$$:= \sqrt{\alpha_0^2 \sigma^2 + \boldsymbol{\alpha}^T \left( \sigma^2 \boldsymbol{I}_r + \boldsymbol{\Sigma} \boldsymbol{V}^T \operatorname{diag}(\boldsymbol{\pi}) \boldsymbol{V} \boldsymbol{\Sigma} \right) \boldsymbol{\alpha} - 2 \boldsymbol{\alpha}^T \left( \pi_\ell \boldsymbol{\Sigma} \boldsymbol{V}^T \boldsymbol{e}_\ell - b_\ell \boldsymbol{\Sigma} \boldsymbol{V}^T \boldsymbol{\pi} \right) + b_\ell^2 - 2 b_\ell \pi_\ell + \pi_\ell}$$

$$- \alpha_0 \sigma \sqrt{\gamma}, \tag{H.6}$$

We will show in the next paragraph that the argument inside the square-root in (H.6) is a convex quadratic over $(\alpha_0, \boldsymbol{\alpha}, b_\ell)$ (see (H.19)). Thus, the function $\mathcal{D}_\ell(\alpha_0, \boldsymbol{\alpha}, b_\ell)$ is jointly convex. Using uniform convergence of convex functions over compact sets [AG82, Cor.. II.1], we arrive at

$$\bar{\phi}_{AO,\ell}(\boldsymbol{g}, \boldsymbol{h}) \xrightarrow{P} \min_{\alpha_0 \geq 0, \boldsymbol{\alpha}, b_\ell} \mathcal{D}_\ell(\alpha_0, \boldsymbol{\alpha}, b_\ell). \tag{H.7}$$

**Deterministic Analysis.** Here, we analyze the deterministic scalar minimization on the RHS of (H.7). Define

$$\boldsymbol{A} := \begin{bmatrix} \sigma^2 \boldsymbol{I}_r + \boldsymbol{\Sigma} \boldsymbol{V}^T \operatorname{diag}(\boldsymbol{\pi}) \boldsymbol{V} \boldsymbol{\Sigma} & \boldsymbol{\Sigma} \boldsymbol{V}^T \boldsymbol{\pi} \\ \boldsymbol{\pi}^T \boldsymbol{V} \boldsymbol{\Sigma} & 1 \end{bmatrix} \quad \text{and} \quad \boldsymbol{c}_\ell = \begin{bmatrix} \boldsymbol{\Sigma} \boldsymbol{V}^T \boldsymbol{e}_\ell \\ 1 \end{bmatrix}, \tag{H.8}$$

and observe that we can write

$$\mathcal{D}_\ell(\alpha_0, \boldsymbol{\alpha}, b_\ell) = \sqrt{\alpha_0^2 \sigma^2 + \pi_\ell + \begin{bmatrix} \boldsymbol{\alpha}^T & b_\ell \end{bmatrix} \boldsymbol{A} \begin{bmatrix} \boldsymbol{\alpha} \\ b_\ell \end{bmatrix} - 2\pi_\ell \boldsymbol{c}_\ell^T \begin{bmatrix} \boldsymbol{\alpha} \\ b_\ell \end{bmatrix}} - \alpha_0 \sigma \sqrt{\gamma}. \tag{H.9}$$

First, note that the matrix $\boldsymbol{A}$ is positive definite. This can be checked by computing the Schur complement of $\boldsymbol{A}$:

$$\boldsymbol{\Delta} := \sigma^2 \boldsymbol{I}_r + \boldsymbol{\Sigma} \boldsymbol{V}^T \boldsymbol{P} \boldsymbol{V} \boldsymbol{\Sigma} := \sigma^2 \boldsymbol{I}_r + \boldsymbol{\Sigma} \boldsymbol{V}^T \left( \mathrm{diag}(\boldsymbol{\pi}) - \boldsymbol{\pi} \boldsymbol{\pi}^T \right) \boldsymbol{V} \boldsymbol{\Sigma} > \boldsymbol{0}_{r \times r}. \tag{H.10}$$

Positive definiteness above holds because $\boldsymbol{P} := \left( \mathrm{diag}(\boldsymbol{\pi}) - \boldsymbol{\pi} \boldsymbol{\pi}^T \right) \succeq \boldsymbol{0}_{k \times k}$. Thus the term under the square-root in (H.9) is a strictly convex quadratic. Thus, $\mathcal{D}_\ell$ is jointly convex in its arguments.

To simplify the RHS of (H.7) we proceed by minimizing $\mathcal{D}_\ell(\alpha_0, \boldsymbol{\alpha}, b_\ell)$ over $(\boldsymbol{\alpha}, b_\ell)$ which from Lemma (C.3) is equal to

$$\begin{bmatrix} \widehat{\boldsymbol{\alpha}} \\ \widehat{b_\ell} \end{bmatrix} = \pi_\ell \boldsymbol{A}^{-1} \boldsymbol{c}_\ell = \pi_\ell \begin{bmatrix} \boldsymbol{I} & \boldsymbol{0} \\ -\boldsymbol{\pi}^T \boldsymbol{V} \boldsymbol{\Sigma} & 1 \end{bmatrix} \begin{bmatrix} \boldsymbol{\Delta}^{-1} & \boldsymbol{0} \\ \boldsymbol{0}^T & 1 \end{bmatrix} \begin{bmatrix} \boldsymbol{I} & -\boldsymbol{\Sigma} \boldsymbol{V}^T \boldsymbol{\pi} \\ \boldsymbol{0}^T & 1 \end{bmatrix} \begin{bmatrix} \boldsymbol{\Sigma} \boldsymbol{V}^T \boldsymbol{e}_\ell \\ 1 \end{bmatrix}$$

$$= \pi_\ell \begin{bmatrix} \boldsymbol{I} & \boldsymbol{0} \\ -\boldsymbol{\pi}^T \boldsymbol{V} \boldsymbol{\Sigma} & 1 \end{bmatrix} \begin{bmatrix} \boldsymbol{\Delta}^{-1} & \boldsymbol{0} \\ \boldsymbol{0}^T & 1 \end{bmatrix} \begin{bmatrix} -\boldsymbol{\Sigma} \boldsymbol{V}^T (\boldsymbol{\pi} - \boldsymbol{e}_\ell) \\ 1 \end{bmatrix}$$

$$= \pi_\ell \begin{bmatrix} -\boldsymbol{\Delta}^{-1} \boldsymbol{\Sigma} \boldsymbol{V}^T (\boldsymbol{\pi} - \boldsymbol{e}_\ell) \\ 1 + \boldsymbol{\pi}^T \boldsymbol{V} \boldsymbol{\Sigma} \boldsymbol{\Delta}^{-1} \boldsymbol{\Sigma} \boldsymbol{V}^T (\boldsymbol{\pi} - \boldsymbol{e}_\ell) \end{bmatrix}. \tag{H.11}$$

Thus, the minimum value attained is

$$-\pi_\ell^2 \begin{bmatrix} -(\boldsymbol{\pi} - \boldsymbol{e}_\ell)^T \boldsymbol{V} \boldsymbol{\Sigma} & 1 \end{bmatrix} \begin{bmatrix} \boldsymbol{\Delta}^{-1} & \boldsymbol{0} \\ \boldsymbol{0}^T & 1 \end{bmatrix} \begin{bmatrix} -\boldsymbol{\Sigma} \boldsymbol{V}^T (\boldsymbol{\pi} - \boldsymbol{e}_\ell) \\ 1 \end{bmatrix} = -\pi_\ell^2 \left( 1 + (\boldsymbol{\pi} - \boldsymbol{e}_\ell)^T \boldsymbol{V} \boldsymbol{\Sigma} \boldsymbol{\Delta}^{-1} \boldsymbol{\Sigma} \boldsymbol{V}^T (\boldsymbol{\pi} - \boldsymbol{e}_\ell) \right).$$

Using the above, (H.7) reduces to

$$\bar{\phi}_{AO,\ell}(\boldsymbol{g}, \boldsymbol{h}) \xrightarrow{P} \min_{\alpha_0 \geq 0} \sqrt{\alpha_0^2 \sigma^2 + \pi_\ell - \pi_\ell^2 \left( 1 + (\boldsymbol{\pi} - \boldsymbol{e}_\ell)^T \boldsymbol{V} \boldsymbol{\Sigma} \boldsymbol{\Delta}^{-1} \boldsymbol{\Sigma} \boldsymbol{V}^T (\boldsymbol{\pi} - \boldsymbol{e}_\ell) \right)} - \alpha_0 \sigma \sqrt{\gamma}. \tag{H.12}$$

Setting the derivative with respect to $\alpha_0$ to zero we arrive at

$$\frac{\alpha_0 \sigma^2}{\sqrt{\alpha_0^2 \sigma^2 + \pi_\ell - \pi_\ell^2 \left( 1 + (\boldsymbol{\pi} - \boldsymbol{e}_\ell)^T \boldsymbol{V} \boldsymbol{\Sigma} \boldsymbol{\Delta}^{-1} \boldsymbol{\Sigma} \boldsymbol{V}^T (\boldsymbol{\pi} - \boldsymbol{e}_\ell) \right)}} = \sigma \sqrt{\gamma}.$$

Thus,

$$\widehat{\alpha}_0 = \frac{1}{\sigma} \sqrt{\frac{\gamma}{1 - \gamma}} \sqrt{\pi_\ell (1 - \pi_\ell) - \pi_\ell^2 (\boldsymbol{\pi} - \boldsymbol{e}_\ell)^T \boldsymbol{V} \boldsymbol{\Sigma} \boldsymbol{\Delta}^{-1} \boldsymbol{\Sigma} \boldsymbol{V}^T (\boldsymbol{\pi} - \boldsymbol{e}_\ell)}. \tag{H.13}$$

Plugging the latter into (H.12) we arrive at

$$\bar{\phi}_{AO,\ell}(\boldsymbol{g}, \boldsymbol{h}) \xrightarrow{P} \sqrt{1 - \gamma} \sqrt{\pi_\ell (1 - \pi_\ell) - \pi_\ell^2 (\boldsymbol{\pi} - \boldsymbol{e}_\ell)^T \boldsymbol{V} \boldsymbol{\Sigma} \boldsymbol{\Delta}^{-1} \boldsymbol{\Sigma} \boldsymbol{V}^T (\boldsymbol{\pi} - \boldsymbol{e}_\ell)}.$$

**Asymptotic predictions.** First, from (H.11) the bias term converges as follows:

$$\widehat{b}_\ell \xrightarrow{P} \pi_\ell \left( 1 + \boldsymbol{\pi}^T \boldsymbol{V} \boldsymbol{\Sigma} \boldsymbol{\Delta}^{-1} \boldsymbol{\Sigma} \boldsymbol{V}^T (\boldsymbol{\pi} - \boldsymbol{e}_\ell) \right).$$

Thus,

$$\widehat{\boldsymbol{b}} \xrightarrow{P} \mathrm{diag}(\boldsymbol{\pi}) \left( \boldsymbol{1}_k + \left( \boldsymbol{\pi} \boldsymbol{1}_k^T - \boldsymbol{I}_k \right) \boldsymbol{V} \boldsymbol{\Sigma} \boldsymbol{\Delta}^{-1} \boldsymbol{\Sigma} \boldsymbol{V}^T \boldsymbol{\pi} \right).$$

Recall from (H.4) that $\boldsymbol{\alpha} = \boldsymbol{U}^T \boldsymbol{w}_\ell$. Thus, the correlations $\langle \boldsymbol{\mu}_i, \boldsymbol{w}_\ell \rangle$, $i \in [k]$ converge as follows:

$$\boldsymbol{M}^T \boldsymbol{w}_\ell = \boldsymbol{V} \boldsymbol{\Sigma} \boldsymbol{U}^T \boldsymbol{w}_\ell \xrightarrow{P} \boldsymbol{V} \boldsymbol{\Sigma} \widehat{\boldsymbol{\alpha}} = -\pi_\ell \boldsymbol{V} \boldsymbol{\Sigma} \boldsymbol{\Delta}^{-1} \boldsymbol{\Sigma} \boldsymbol{V}^T (\boldsymbol{\pi} - \boldsymbol{e}_\ell). \tag{H.14}$$

Here, convergence applies element-wise to the entries of the involved random vectors. Moreover, from the analysis above we can predict the limit of the norm $\|\boldsymbol{w}_\ell\|_{\ell_2}$. For this, note that $\|\boldsymbol{w}_\ell\|_{\ell_2}^2 = \widehat{\alpha}_0^2 + \widehat{\boldsymbol{\alpha}}^T \widehat{\boldsymbol{\alpha}}$. Thus,

$$\|\boldsymbol{w}_\ell\|_{\ell_2}^2 \xrightarrow{P} \frac{\gamma}{(1 - \gamma)\sigma^2} \pi_\ell (1 - \pi_\ell) + \pi_\ell^2 (\boldsymbol{\pi} - \boldsymbol{e}_\ell)^T \boldsymbol{V} \boldsymbol{\Sigma} \boldsymbol{\Delta}^{-1} \left( \boldsymbol{\Delta}^{-1} - \frac{\gamma}{(1 - \gamma)\sigma^2} \boldsymbol{I}_r \right) \boldsymbol{\Sigma} \boldsymbol{V}^T (\boldsymbol{\pi} - \boldsymbol{e}_\ell). \tag{H.15}$$

### H.1.2 Computing $\Sigma_{w,w}$

In the previous section we used the CGMT to predict the bias $\widehat{b}_\ell$, the correlations $\langle \boldsymbol{\mu}_i, \widehat{\boldsymbol{w}}_\ell \rangle$, $i[k]$ and the norm $\|\widehat{\boldsymbol{w}}_\ell\|_{\ell_2}$ for all $\ell \in [k]$ members of the multi-output classifier. Here, we show how to compute the limits of the cross-correlations $\langle \widehat{\boldsymbol{w}}_\ell, \widehat{\boldsymbol{w}}_j \rangle, \ell \neq j \in [k]$.

**Lemma H.1** *For $\ell \neq j \in [k]$, let $\widehat{\boldsymbol{w}}_\ell \, \widehat{\boldsymbol{w}}_j$ be solutions to the least-squares minimization*

$$(\widehat{\boldsymbol{w}}_\ell, \widehat{\boldsymbol{w}}_j, \widehat{b}_\ell, \widehat{b}_j) = \arg \min_{\boldsymbol{w}_\ell, \boldsymbol{w}_j, b_\ell, b_j} \left\{ \frac{1}{2n} \left\| \boldsymbol{Y}_\ell - \boldsymbol{X}^T \boldsymbol{w}_\ell - b_\ell \mathbf{1}_n \right\|_{\ell_2}^2 + \frac{1}{2n} \left\| \boldsymbol{Y}_j - \boldsymbol{X}^T \boldsymbol{w}_j - b_j \mathbf{1}_n \right\|_{\ell_2}^2 \right\}.$$

*Denote $\widehat{\boldsymbol{w}}_{\ell,j} := \widehat{\boldsymbol{w}}_\ell + \widehat{\boldsymbol{w}}_j$ and $\widehat{b}_{\ell,j} := \widehat{b}_\ell + \widehat{b}_j$. Then, $(\widehat{\boldsymbol{w}}_{\ell,j}, \widehat{b}_{\ell,j})$ is a minimizer in the following least-squares problem:*

$$(\widehat{\boldsymbol{w}}_{\ell,j}, \widehat{b}_{\ell,j}) = \arg \min_{\boldsymbol{w}, b} \frac{1}{2n} \left\| \boldsymbol{Y}_\ell + \boldsymbol{Y}_j - \boldsymbol{X}^T \boldsymbol{w} - b \mathbf{1}_n \right\|_{\ell_2}^2. \tag{H.16}$$

**Proof** Clearly the minimization in (H.16) is convex. Thus, it suffices to prove that $\widehat{\boldsymbol{w}}_\ell + \widehat{\boldsymbol{w}}_j$ satisfies the KKT conditions. First, by optimality of $\widehat{\boldsymbol{w}}_\ell$, we have that

$$\boldsymbol{X} \left( \boldsymbol{Y}_\ell - \boldsymbol{X}^T \widehat{\boldsymbol{w}}_\ell - \widehat{b}_\ell \mathbf{1}_n \right) = 0$$

Similarly, for $\widehat{\boldsymbol{w}}_j$:

$$\boldsymbol{X} \left( \boldsymbol{Y}_j - \boldsymbol{X}^T \widehat{\boldsymbol{w}}_j - \widehat{b}_j \mathbf{1}_n \right) = 0.$$

Adding the equations on the above displays we find that

$$\boldsymbol{X} \left( \boldsymbol{Y}_\ell + \boldsymbol{Y}_j - \boldsymbol{X}^T (\widehat{\boldsymbol{w}}_j + \widehat{\boldsymbol{w}}_\ell) - (\widehat{b}_j + \widehat{b}_\ell) \mathbf{1}_n \right) = 0.$$

Recognize that this coincides with the optimality condition for (H.16). Thus, the proof is complete. ∎

Thanks to Lemma H.1, we can use the CGMT to characterize the limiting behavior of $\|\widehat{\boldsymbol{w}}_\ell + \widehat{\boldsymbol{w}}_j\|_{\ell_2}$. Observe that this immediately gives the limit of $\langle \widehat{\boldsymbol{w}}_\ell, \widehat{\boldsymbol{w}}_j \rangle$ since

$$\langle \widehat{\boldsymbol{w}}_\ell, \widehat{\boldsymbol{w}}_j \rangle = \frac{\|\widehat{\boldsymbol{w}}_\ell + \widehat{\boldsymbol{w}}_j\|_{\ell_2}^2 - \|\widehat{\boldsymbol{w}}_\ell\|_{\ell_2}^2 - \|\widehat{\boldsymbol{w}}_j\|_{\ell_2}^2}{2}. \tag{H.17}$$

The analysis of (H.16) is very similar to that of (H.1); thus, most details are omitted. Similar to (H.5) we can relate (H.16) with the following AO problem:

$$\bar{\phi}_{AO,\ell,j}(\boldsymbol{g}, \boldsymbol{h}) := \min_{\beta_0 \geq 0, \boldsymbol{\beta} \in \mathbb{R}^r, b_{\ell,j}} \frac{1}{\sqrt{n}} \left\| \sigma \sqrt{\beta_0^2 + \|\boldsymbol{\beta}\|_{\ell_2}^2} \, \boldsymbol{g} + \boldsymbol{Y}^T \boldsymbol{V} \boldsymbol{\Sigma} \boldsymbol{\beta} + b_{\ell,j} \mathbf{1}_n - \boldsymbol{Y}_\ell - \boldsymbol{Y}_j \right\|_{\ell_2}$$

$$+ \sigma \sum_{i=1}^r \beta_i \frac{\boldsymbol{h}^T \boldsymbol{u}_i}{\sqrt{n}} - \sigma \beta_0 \frac{\|\boldsymbol{h}^\perp\|_{\ell_2}}{\sqrt{n}}, \tag{H.18}$$

where we have decomposed

$$\boldsymbol{w}_{\ell,j} = \sum_{i=1}^r \beta_i \boldsymbol{u}_i + \beta_0 \boldsymbol{w}_{\ell,j}^\perp,$$

with $\|\boldsymbol{w}_{\ell,j}^\perp\|_{\ell_2} = 1$ and $\boldsymbol{U}^T \boldsymbol{w}_{\ell,j}^\perp = \mathbf{0}_r$.

Using a calculation similar to the one leading to (H.9) we can show that (H.18) converges point-wise in $\beta_0, \boldsymbol{\beta} = [\beta_1, \ldots, \beta_r], b_{\ell,j}$ to the following:

$$\mathcal{D}_\ell(\beta_0, \boldsymbol{\beta}, b_{\ell,j}) = \sqrt{\beta_0^2 \sigma^2 + \pi_\ell + \pi_j + \begin{bmatrix} \boldsymbol{\beta}^T & b_{\ell,j} \end{bmatrix} \boldsymbol{A} \begin{bmatrix} \boldsymbol{\beta} \\ b_{\ell,j} \end{bmatrix} - 2 \boldsymbol{d}_{\ell,j}^T \begin{bmatrix} \boldsymbol{\beta} \\ b_{\ell,j} \end{bmatrix}} - \beta_0 \sigma \sqrt{\gamma}, \tag{H.19}$$

where $\boldsymbol{A}$ is as in (H.8) and we have further defined

$$\boldsymbol{d}_{\ell,j} := \begin{bmatrix} \pi_\ell \boldsymbol{\Sigma} \boldsymbol{V}^T \boldsymbol{e}_\ell + \pi_j \boldsymbol{\Sigma} \boldsymbol{V}^T \boldsymbol{e}_j \\ \pi_\ell + \pi_j \end{bmatrix}.$$

Thus, similar to (H.11) we can compute the minimizer of the deterministic objective in (H.19):

$$
\begin{bmatrix} \widehat{\boldsymbol{\beta}} \\ \widehat{b}_{\ell,j} \end{bmatrix} = \begin{bmatrix} -\boldsymbol{\Delta}^{-1}\boldsymbol{\Sigma}\boldsymbol{V}^T\left(\pi_\ell(\boldsymbol{\pi}-\boldsymbol{e}_\ell)+\pi_j(\boldsymbol{\pi}-\boldsymbol{e}_j)\right) \\ \pi_\ell + \pi_j + \boldsymbol{\pi}^T\boldsymbol{V}\boldsymbol{\Sigma}\boldsymbol{\Delta}^{-1}\boldsymbol{\Sigma}\boldsymbol{V}^T\left(\pi_\ell(\boldsymbol{\pi}-\boldsymbol{e}_\ell)+\pi_j(\boldsymbol{\pi}-\boldsymbol{e}_j)\right) \end{bmatrix},
\tag{H.20}
$$

and

$$
\widehat{\beta}_0 = \frac{1}{\sigma}\sqrt{\frac{\gamma}{1-\gamma}}\sqrt{\pi_\ell+\pi_j-(\pi_\ell+\pi_j)^2-\left(\pi_\ell(\boldsymbol{\pi}-\boldsymbol{e}_\ell)+\pi_j(\boldsymbol{\pi}-\boldsymbol{e}_j)\right)^T\boldsymbol{V}\boldsymbol{\Sigma}\boldsymbol{\Delta}^{-1}\boldsymbol{\Sigma}\boldsymbol{V}^T\left(\pi_\ell(\boldsymbol{\pi}-\boldsymbol{e}_\ell)+\pi_j(\boldsymbol{\pi}-\boldsymbol{e}_j)\right)},
\tag{H.21}
$$

where recall that $\boldsymbol{\Delta}$ is as in (H.10).

From the CGMT, we have that $\|\widehat{\boldsymbol{w}}_\ell + \widehat{\boldsymbol{w}}_j\|_{\ell_2}^2 \xrightarrow{P} \widehat{\beta}_0^2 + \|\boldsymbol{\beta}\|_{\ell_2}^2$. Combining this with the calculations above, we conclude that

$$
\|\widehat{\boldsymbol{w}}_\ell + \widehat{\boldsymbol{w}}_j\|_{\ell_2}^2 \xrightarrow{P} \frac{\gamma}{(1-\gamma)\sigma^2}\left(\pi_\ell+\pi_j\right)\left(1-\pi_\ell-\pi_j\right)
$$

$$
+ \left(\pi_\ell(\boldsymbol{\pi}-\boldsymbol{e}_\ell)+\pi_j(\boldsymbol{\pi}-\boldsymbol{e}_j)\right)^T\boldsymbol{V}\boldsymbol{\Sigma}\boldsymbol{\Delta}^{-1}\left(\boldsymbol{\Delta}^{-1}-\frac{\gamma}{(1-\gamma)\sigma^2}\boldsymbol{I}_r\right)\boldsymbol{\Sigma}\boldsymbol{V}^T\left(\pi_\ell(\boldsymbol{\pi}-\boldsymbol{e}_\ell)+\pi_j(\boldsymbol{\pi}-\boldsymbol{e}_j)\right)
\tag{H.22}
$$

Finally, using (H.22) and (H.15) in (H.17) it follows that

$$
\langle \boldsymbol{w}_\ell, \boldsymbol{w}_j \rangle \xrightarrow{P} \pi_\ell\pi_j\left(-\frac{\gamma}{(1-\gamma)\sigma^2}+(\boldsymbol{\pi}-\boldsymbol{e}_\ell)^T\boldsymbol{V}\boldsymbol{\Sigma}\boldsymbol{\Delta}^{-1}\left(\boldsymbol{\Delta}^{-1}-\frac{\gamma}{(1-\gamma)\sigma^2}\boldsymbol{I}_r\right)\boldsymbol{\Sigma}\boldsymbol{V}^T(\boldsymbol{\pi}-\boldsymbol{e}_j)\right).
\tag{H.23}
$$

## H.2 Orthogonal means

Here, we specialize the asymptotic predictions of Theorem 3.2 to the case of orthogonal means $\langle \boldsymbol{\mu}_i, \boldsymbol{\mu}_j \rangle = 0$, $i \neq j$.

**Corollary H.2 (Orthogonal means)** *Consider the case of orthogonal means, i.e. $\langle \boldsymbol{\mu}_i, \boldsymbol{\mu}_j \rangle = 0$, $\forall i \neq j$ and $\gamma < 1$ with Euclidean norms given by $\mu_i = \|\boldsymbol{\mu}_i\|_{\ell_2}$. Define the following parameters for $i \in [k]$:*

$$
\rho_i := \pi_i\sigma^2\big/\left(\sigma^2+\pi_i\mu_i^2\right) \quad and \quad \beta_i = \rho_i\sigma^2\big/\left(\sigma^2-\sum_{i=1}^k \pi_i\rho_i\mu_i^2\right).
$$

*Then, the following asymptotic limits hold for the least-squares classifier, for all $i,j \in [k]$:*

$$
\widehat{b}_i \xrightarrow{P} \beta_i, \qquad \langle \widehat{\boldsymbol{w}}_i, \boldsymbol{\mu}_j \rangle \xrightarrow{P} \frac{1}{\sigma}(\mathbb{1}_{ij}-\beta_i)\rho_j\mu_j,
\tag{H.24a}
$$

$$
\langle \widehat{\boldsymbol{w}}_i, \widehat{\boldsymbol{w}}_j \rangle \xrightarrow{P} \frac{1}{\sigma^4}\beta_i\beta_j\sum_{\ell=1}^k \rho_\ell^2\mu_\ell^2 - \frac{1}{\sigma^4}\beta_i\rho_j^2\mu_j^2 - \frac{1}{\sigma^4}\beta_j\rho_i^2\mu_i^2 - \frac{\gamma\beta_i\rho_j}{(1-\gamma)\sigma^2} + \frac{\mathbb{1}_{ij}}{\sigma^2}\left(\frac{\gamma}{(1-\gamma)}\rho_i+\frac{1}{\sigma^2}\rho_i^2\mu_i^2\right)
\tag{H.24b}
$$

*Furthermore, if the means have equal norms $\mu := \mu_i$ and the classes are balanced: $\pi_i = 1/k$, $i \in [k]$, then, setting $u_{\mathrm{LS}} := \frac{\mu^2}{\sigma}\sqrt{\frac{1-\gamma}{\mu^2+k\gamma\sigma^2}}$, it holds that*

$$
\mathbb{P}_e = \mathbb{P}\left\{G_0 + \max_{j\in[k-1]} G_j \geq u_{\mathrm{LS}}\right\}, \quad G_0, G_1, \ldots, G_{k-1} \overset{iid}{\sim} \mathcal{N}(0,1).
\tag{H.25}
$$

**Proof** This is a direct corollary of Theorem 3.2. Indeed, (H.24) can be derived from (3.3) after substituting $\boldsymbol{V} = \boldsymbol{I}_k$, $\boldsymbol{\Sigma} = \mathrm{diag}(\mu_1, \mu_2, \ldots, \mu_k)$ and some algebra steps that we omit for brevity.

Instead, we outline below how to conclude (H.25) from (H.24). Assume that $\mu_i = \mu$, $\forall i \in [k]$ and $\pi_i = \pi = 1/k$, $\forall i \in [k]$. Recall from (2.7) that $\mathbb{P}\left(\text{error} \mid \boldsymbol{y} = \boldsymbol{e}_c\right) = 1 - \mathbb{P}\left(\boldsymbol{S}_c^{1/2}\boldsymbol{z} > \boldsymbol{t}\right)$, and using (H.24) it can be checked that

$$
\boldsymbol{S}_c = \frac{\pi}{1+\pi\mu^2}\left(\frac{\pi\mu^2}{1+\pi\mu^2}+\frac{\gamma}{1-\gamma}\right)(\boldsymbol{I}_k+\mathbf{1}_k\mathbf{1}_k^T) \quad \text{and} \quad \boldsymbol{t} = -\frac{\pi\mu^2}{1+\pi\mu^2}\mathbf{1}.
$$

Thus, setting

$$u_{\mathrm{LS}} := \mu^2 \sqrt{\frac{\pi}{\pi\mu^2 + \left(\frac{\gamma}{1-\gamma}\right)\left(1 + \pi\mu^2\right)}} = \mu^2 \sqrt{\frac{1-\gamma}{\mu^2 + \gamma/\pi}}, \tag{H.26}$$

and applying Lemma (D.1), the probability of error is given by the advertised expression. ∎

# I  Least-squares for MLM

## I.1  Proof of Theorem 4.2

### I.1.1  Computing $\Sigma_{w,\mu}$

Assume that $\boldsymbol{X}, \boldsymbol{Y}$ are generated from the MLM.

Fix any $\ell \in [k]$. The classifier parameters $\widehat{\boldsymbol{w}}_\ell, \widehat{\boldsymbol{b}}_\ell$ minimize the following objective function $\mathcal{L}_{PO}\left(\boldsymbol{w}_\ell, b_\ell\right) := \frac{1}{2n}\left\|\boldsymbol{X}^T\boldsymbol{w}_\ell + b_\ell\boldsymbol{1}_n - \boldsymbol{Y}_\ell\right\|_{\ell_2}^2$.

**Identifying the AO.** To continue further note that by duality we have

$$\min_{\boldsymbol{w}_\ell, b_\ell}\ \mathcal{L}_{PO}\left(\boldsymbol{w}_\ell, b_\ell\right) = \min_{\boldsymbol{w}_\ell, b_\ell}\ \max_{\boldsymbol{s}}\ \frac{1}{n}\left(\boldsymbol{s}^T\boldsymbol{X}^T\boldsymbol{w}_\ell + b_\ell\boldsymbol{s}^T\boldsymbol{1}_n - \boldsymbol{s}^T\boldsymbol{Y}_\ell - \frac{\|\boldsymbol{s}\|_{\ell_2}^2}{2}\right), \tag{I.1}$$

and the optimization is jointly convex in $\left(\boldsymbol{w}_\ell, b_\ell\right)$ and concave in $\boldsymbol{s}$. Here, note that $\boldsymbol{Y}_\ell$ depends on the Gaussian matrix $\boldsymbol{X}$. Thus, before applying the CGMT, we need to break this dependence as follows. Consider the singular value decomposition

$$\boldsymbol{M} = \boldsymbol{U}\boldsymbol{\Sigma}\boldsymbol{V}^T = \begin{bmatrix} \boldsymbol{u}_1 & \boldsymbol{u}_2 & \dots & \boldsymbol{u}_r \end{bmatrix} \mathrm{diag}(\sigma_1, \sigma_2, \dots, \sigma_r) \begin{bmatrix} \boldsymbol{v}_1^T \\ \boldsymbol{v}_2^T \\ \dots \\ \boldsymbol{v}_r^T \end{bmatrix}, \tag{I.2}$$

with $\boldsymbol{U} \in \mathbb{R}^{d \times r}$, $\boldsymbol{\Sigma} \in \mathbb{R}^{r \times r}$, and $\boldsymbol{V} \in \mathbb{R}^{k \times r}$ where $r = \mathrm{rank}(\boldsymbol{M}) \le k$. For every $i \in [n]$, we decompose $\boldsymbol{x}_i$ in its projection on the subspace spanned orthogonal columns $\boldsymbol{u}_1, \dots, \boldsymbol{u}_r$ as follows:

$$\boldsymbol{x}_i = \boldsymbol{U}\boldsymbol{U}^T\boldsymbol{X}_i + \boldsymbol{P}^\perp\boldsymbol{X}_i = \boldsymbol{U}\tilde{\boldsymbol{g}}_i + \boldsymbol{P}^\perp\boldsymbol{x}_i,$$

where $\boldsymbol{P}^\perp = \boldsymbol{I}_r - \boldsymbol{U}\boldsymbol{U}^T$, and we denote

$$\widetilde{\boldsymbol{G}} := \begin{bmatrix} \tilde{\boldsymbol{g}}_1 & \tilde{\boldsymbol{g}}_2 & \dots & \tilde{\boldsymbol{g}}_n \end{bmatrix}, \quad \tilde{\boldsymbol{g}}_i := \boldsymbol{U}^T\boldsymbol{x}_i \in \mathbb{R}^r, i \in [n]. \tag{I.3}$$

Recalling that $\boldsymbol{x}_i \sim \mathcal{N}(\boldsymbol{0}, \boldsymbol{I}_d)$ note that

$$\tilde{\boldsymbol{g}}_i \sim \mathcal{N}(\boldsymbol{0}, \boldsymbol{I}_r) \quad \text{and} \quad \tilde{\boldsymbol{g}}_i \perp \boldsymbol{P}^\perp\boldsymbol{x}_i. \tag{I.4}$$

Further recall that for all $i \in [n]$, conditioned on $\boldsymbol{x}_i$

$$[\boldsymbol{Y}_\ell]_i \sim \mathrm{Bern}\left(\frac{e^{\boldsymbol{\mu}_\ell^T\boldsymbol{x}_i}}{\sum_{\ell' \in [k]} e^{\boldsymbol{\mu}_{\ell'}^T\boldsymbol{x}_i}}\right) \sim \mathrm{Bern}\left(\frac{e^{\boldsymbol{e}_\ell^T\boldsymbol{V}\boldsymbol{\Sigma}\tilde{\boldsymbol{g}}_i}}{\sum_{\ell' \in [k]} e^{\boldsymbol{e}_{\ell'}^T\boldsymbol{V}\boldsymbol{\Sigma}\tilde{\boldsymbol{g}}_i}}\right), \tag{I.5}$$

where we used (I.3) and the SVD decomposition of $\boldsymbol{M}$. In this notation, we can rewrite the PO as follows:

$$\min_{\boldsymbol{w}_\ell, b_\ell}\ \max_{\boldsymbol{s}}\ \frac{1}{n}\left(\boldsymbol{s}^T\boldsymbol{X}^T\boldsymbol{P}^\perp\boldsymbol{w}_\ell + \boldsymbol{s}^T\widetilde{\boldsymbol{G}}^T\boldsymbol{U}^T\boldsymbol{w}_\ell + b_\ell\boldsymbol{s}^T\boldsymbol{1}_n - \boldsymbol{s}^T\boldsymbol{Y}_\ell - \frac{\|\boldsymbol{s}\|_{\ell_2}^2}{2}\right)$$

From (I.4) and (I.5) notice that $\boldsymbol{Y}_\ell$ depends only on $\widetilde{\boldsymbol{G}}$ and $\widetilde{\boldsymbol{G}}$ is independent of $\boldsymbol{X}^T\boldsymbol{P}^\perp$. Therefore, the corresponding Auxiliary Optimization (AO) problem becomes

$$\min_{\boldsymbol{w}_\ell, b_\ell}\ \max_{\boldsymbol{s}}\ \frac{1}{n}\left(\|\boldsymbol{P}^\perp\boldsymbol{w}_\ell\|_{\ell_2}\boldsymbol{g}^T\boldsymbol{s} + \|\boldsymbol{s}\|_{\ell_2}\boldsymbol{h}^T\boldsymbol{P}^\perp\boldsymbol{w}_\ell + \boldsymbol{s}^T\widetilde{\boldsymbol{G}}^T\boldsymbol{U}^T\boldsymbol{w}_\ell + b_\ell\boldsymbol{s}^T\boldsymbol{1}_n - \boldsymbol{s}^T\boldsymbol{Y}_\ell - \frac{\|\boldsymbol{s}\|_{\ell_2}^2}{2}\right),$$
$$\tag{I.6}$$

where $\boldsymbol{g} \in \mathbb{R}^n$ and $\boldsymbol{h} \in \mathbb{R}^d$ are iid Gaussian vectors independent of everything else.

**Scalarization of the AO.** Maximizing over the direction of $\boldsymbol{s}$ and denoting its norm $\beta = \|\boldsymbol{s}\|_{\ell_2} \geq 0$ we arrive at

$$\min_{\boldsymbol{w}_\ell, b_\ell} \max_{\beta \geq 0} \frac{1}{n} \left( \beta \left\| \|\boldsymbol{P}^\perp \boldsymbol{w}_\ell\|_{\ell_2} \boldsymbol{g} + \boldsymbol{G}^T \boldsymbol{U}^T \boldsymbol{w}_\ell + b_\ell \mathbf{1}_n - \boldsymbol{Y}_\ell \right\|_{\ell_2} + \beta \boldsymbol{h}^T \boldsymbol{P}^\perp \boldsymbol{w}_\ell - \frac{\beta^2}{2} \right)$$

$$= \min_{\boldsymbol{w}_\ell, b_\ell} \frac{1}{2n} \left( \left\| \|\boldsymbol{P}^\perp \boldsymbol{w}_\ell\|_{\ell_2} \boldsymbol{g} + \widetilde{\boldsymbol{G}}^T \boldsymbol{U}^T \boldsymbol{w}_\ell + b_\ell \mathbf{1}_n - \boldsymbol{Y}_\ell \right\|_{\ell_2} + \boldsymbol{h}^T \boldsymbol{P}^\perp \boldsymbol{w}_\ell \right)_+^2$$

$$= \frac{1}{2} \left( \min_{\boldsymbol{w}_\ell, b_\ell} \frac{1}{\sqrt{n}} \left\| \|\boldsymbol{P}^\perp \boldsymbol{w}_\ell\|_{\ell_2} \boldsymbol{g} + \widetilde{\boldsymbol{G}}^T \boldsymbol{U}^T \boldsymbol{w}_\ell + b_\ell \mathbf{1}_n - \boldsymbol{Y}_\ell \right\|_{\ell_2} + \frac{1}{\sqrt{n}} \boldsymbol{h}^T \boldsymbol{P}^\perp \boldsymbol{w}_\ell \right)_+^2 \tag{I.7}$$

In the remaining, we focus in the inner minimization above. Let us denote

$$\boldsymbol{a} := \boldsymbol{U}^T \boldsymbol{w}_\ell \quad \text{and} \quad \alpha_0 = \|\boldsymbol{P}^\perp \boldsymbol{w}_\ell\|_{\ell_2}.$$

Notice that $\boldsymbol{a} \perp \boldsymbol{P}^\perp \boldsymbol{w}_\ell$ and thus the orthogonal decomposition $\boldsymbol{w}_\ell = \boldsymbol{U}\boldsymbol{a} + \boldsymbol{P}^\perp \boldsymbol{w}_\ell$. With this observation, we can optimize over the direction of $\boldsymbol{P}^T \boldsymbol{w}_\ell$ in (I.7) by aligning it with $-\boldsymbol{P}^T \boldsymbol{h}$. With this, the minimization in (I.7) reduces to the following

$$\min_{\boldsymbol{a}, \alpha_0 \geq 0, b_\ell} \frac{1}{\sqrt{n}} \left\| \alpha_0 \boldsymbol{g} + \widetilde{\boldsymbol{G}}^T \boldsymbol{a} + b_\ell \mathbf{1}_n - \boldsymbol{Y}_\ell \right\|_{\ell_2} - \alpha_0 \frac{1}{\sqrt{n}} \|\boldsymbol{P}^\perp \boldsymbol{h}\|_{\ell_2}. \tag{I.8}$$

**Convergence of the AO.** First, we argue on point-wise convergence of the objective function in (I.8). Fix $\boldsymbol{a}, \alpha_0$ and $b_\ell$. From the WLLN, $\frac{1}{\sqrt{n}} \|\boldsymbol{P}^\perp \boldsymbol{h}\|_{\ell_2} \xrightarrow{P} \sqrt{\gamma}$ and

$$\frac{1}{n} \left\| \alpha_0 \boldsymbol{g} + \widetilde{\boldsymbol{G}}^T \boldsymbol{a} + b_\ell \mathbf{1}_n - \boldsymbol{Y}_\ell \right\|_{\ell_2}^2 = \frac{1}{n} \sum_{i=1}^n \left( \alpha_0 g_i + \boldsymbol{a}^T \tilde{\boldsymbol{g}}_i + b_\ell - [\boldsymbol{Y}_\ell]_i \right)^2 \xrightarrow{P} \mathbb{E}\left[ \left( \alpha_0 G_0 + \boldsymbol{a}^T \boldsymbol{g} + b_\ell - Y_\ell \right)^2 \right], \tag{I.9}$$

where the expectation is over $\boldsymbol{g} \sim \mathcal{N}(\mathbf{0}_r, \boldsymbol{I}_r)$ (with some abuse of notation) and

$$Y_\ell \sim \text{Bern}(V_\ell) \quad \text{and} \quad V_\ell = \frac{e^{\boldsymbol{e}_\ell^T \boldsymbol{V} \boldsymbol{\Sigma} \boldsymbol{g}}}{\sum_{\ell'=1}^k e^{\boldsymbol{e}_{\ell'} \boldsymbol{V} \boldsymbol{\Sigma} \boldsymbol{g}}}. \tag{I.10}$$

Therefore, point-wise on $\boldsymbol{a}, \alpha_0$ and $b_\ell$, the objective of the AO converges to

$$\mathcal{D}_\ell(\alpha_0, \boldsymbol{\alpha}, b_\ell) := \sqrt{\mathbb{E}\left[ \left( \alpha_0 G_0 + \boldsymbol{a}^T \boldsymbol{g} + b_\ell - Y_\ell \right)^2 \right]} - \alpha_0 \sqrt{\gamma}. \tag{I.11}$$

Next, with an argument based on convexity and compactness similar to that in "Convergence analysis of the AO" in Section H it can be argued that the convergence above is uniform. Thus,

$$(\text{I.8}) \xrightarrow{P} \min_{\alpha_0 \geq 0, \boldsymbol{\alpha}, b_\ell} \mathcal{D}_\ell(\alpha_0, \boldsymbol{\alpha}, b_\ell). \tag{I.12}$$

**Deterministic analysis of the AO.** Here, we solve the deterministic minimization problem in (I.12). Optimization over $b_\ell$ is straightforward. By setting

$$b_\ell = \mathbb{E}[Y_\ell] = \mathbb{E}[V_\ell],$$

we now have to optimize

$$\min_{\alpha_0 \geq 0, \boldsymbol{\alpha}} \sqrt{\alpha_0^2 + \mathbb{E}\left[ (\boldsymbol{a}^T \boldsymbol{g} - Y_\ell)^2 \right] - (\mathbb{E}[V_\ell])^2} - \alpha_0 \sqrt{\gamma}. \tag{I.13}$$

By direct differentiation and first-order optimality, we compute the optimal values as follows:

$$\widetilde{a}_j = \mathbb{E}[\boldsymbol{g}_j Y_\ell] = \mathbb{E}[\boldsymbol{g}_j V_\ell], \ j \in [r], \tag{I.14}$$

$$\widetilde{\alpha}_0^2 = \frac{\gamma}{1-\gamma} \left( \text{Var}[Y_\ell] - \sum_{j=1}^r (\mathbb{E}[\boldsymbol{g}_j V_\ell])^2 \right) = \frac{\gamma}{1-\gamma} \left( \mathbb{E}[V_\ell] - (\mathbb{E}[V_\ell])^2 - \sum_{j=1}^r (\mathbb{E}[\boldsymbol{g}_j V_\ell])^2 \right). \tag{I.15}$$

**Asymptotic Predictions.** From the analysis above, we conclude with the following limits about the solution $\widehat{\boldsymbol{b}}_\ell, \widehat{\boldsymbol{w}}_\ell$ of the PO:

$$\widehat{\boldsymbol{b}}_\ell \xrightarrow{P} \mathbb{E}[V_\ell] \tag{I.16a}$$

$$\langle \boldsymbol{\mu}_c, \widehat{\boldsymbol{w}}_\ell \rangle \xrightarrow{P} \boldsymbol{e}_c^T \boldsymbol{V} \boldsymbol{\Sigma} \mathbb{E}[\boldsymbol{g} V_\ell], \quad c \in [k] \tag{I.16b}$$

$$\|\widehat{\boldsymbol{w}}_\ell\|_{\ell_2}^2 \xrightarrow{P} \sum_{j=1}^r (\mathbb{E}[\boldsymbol{g}_j V_\ell])^2 + \frac{\gamma}{1-\gamma} \left( \mathbb{E}[V_\ell] - (\mathbb{E}[V_\ell])^2 - \sum_{j=1}^r (\mathbb{E}[\boldsymbol{g}_j V_\ell])^2 \right) \tag{I.16c}$$

$$= \frac{\gamma}{1-\gamma} \left( \mathbb{E}[V_\ell] - (\mathbb{E}[V_\ell])^2 \right) + \frac{1-2\gamma}{1-\gamma} \sum_{j=1}^r (\mathbb{E}[\boldsymbol{g}_j V_\ell])^2. \tag{I.16d}$$

Recall the notation in (4.1). Note that $\mathbb{E}[V_\ell] = \boldsymbol{\pi}_\ell$. Moreover, using Gaussian integration by parts Lemma C.3, it can be shown that $\mathbb{E}[V_\ell \boldsymbol{g}] = \boldsymbol{\Sigma} \boldsymbol{V}^T (\mathrm{diag}(\boldsymbol{\pi}) - \boldsymbol{\Pi}) \boldsymbol{e}_\ell$. Using these and writing and in matrix form, we arrive at (4.4a).

### I.1.2  Computing $\Sigma_{w,w}$

Here, we prove (4.4b). Specifically, we compute the correlations $\langle \widehat{\boldsymbol{w}}_\ell, \widehat{\boldsymbol{w}}_c \rangle$, $\ell \ne c \in [k]$ by following the strategy of Section H.1.2. Specifically, in view of Lemma H.1 we need to study the following PO:

$$\min_{\boldsymbol{w},b} \ \max_{\boldsymbol{s}} \ \frac{1}{n} \left( \boldsymbol{s}^T \boldsymbol{X}^T \boldsymbol{w} + b \boldsymbol{s}^T \boldsymbol{1}_n - \boldsymbol{s}^T (\boldsymbol{Y}_\ell + \boldsymbol{Y}_c) - \frac{\|\boldsymbol{s}\|_{\ell_2}^2}{2} \right) \tag{I.17}$$

which is minimized by $\widehat{\boldsymbol{w}}_\ell + \widehat{\boldsymbol{w}}_c$. Thus the analysis will lead us to an asymptotic formula for $\|\widehat{\boldsymbol{w}}_\ell + \widehat{\boldsymbol{w}}_c\|_{\ell_2}$. This when combined with the formulae for $\|\widehat{\boldsymbol{w}}_\ell\|_{\ell_2}$ and $\|\widehat{\boldsymbol{w}}_c\|_{\ell_2}$ in (I.16d) will give the desired.

The analysis of (I.17) is almost identical to the analysis of (I.1) in the previous section. Specifically, without repeating all the details for brevity, it can be shown that the AO of (I.17) converges to the following (cf. (I.11)):

$$\mathcal{D}_\ell(\alpha_0, \boldsymbol{\alpha}, b_\ell) := \sqrt{\mathbb{E}\left[ \left( \alpha_0 G_0 + \boldsymbol{a}^T \boldsymbol{g} + b_\ell - Y_{\ell,c} \right)^2 \right]} - \alpha_0 \sqrt{\gamma}, \tag{I.18}$$

where as before $G_0 \sim \mathcal{N}(0,1), \boldsymbol{g} \sim \mathcal{N}(\boldsymbol{0}_r, \boldsymbol{I}_r)$, only now (I.10) is modified to:

$$Y_{\ell,c} \sim \mathrm{Bern}(V_c + V_\ell) \quad \text{and as before:} \quad V_\ell = \frac{e^{\boldsymbol{e}_\ell^T \boldsymbol{V} \boldsymbol{\Sigma} \boldsymbol{g}}}{\sum_{\ell'=1}^k e^{\boldsymbol{e}_{\ell'}^T \boldsymbol{V} \boldsymbol{\Sigma} \boldsymbol{g}}}. \tag{I.19}$$

With these, it can be shown that

$$\|\widehat{\boldsymbol{w}}_\ell + \widehat{\boldsymbol{w}}_c\|_{\ell_2}^2 \xrightarrow{P} \sum_{j=1}^r (\mathbb{E}[\boldsymbol{g}_j (V_c + V_\ell)])^2 + \frac{\gamma}{1-\gamma} \left( \mathbb{E}[V_c + V_\ell] - (\mathbb{E}[V_c + V_\ell])^2 - \sum_{j=1}^r (\mathbb{E}[\boldsymbol{g}_j (V_c + V_\ell)])^2 \right)$$

Combining this with (I.16d), we conclude that for $\ell \ne c \in [k]$:

$$\langle \widehat{\boldsymbol{w}}_\ell, \widehat{\boldsymbol{w}}_c \rangle \xrightarrow{P} \frac{1-2\gamma}{1-\gamma} \sum_{j=1}^r \mathbb{E}[\boldsymbol{g}_j V_c] \mathbb{E}[\boldsymbol{g}_j V_\ell] - \frac{\gamma}{1-\gamma} \mathbb{E}[V_c] \mathbb{E}[V_\ell]. \tag{I.20}$$

This shows (4.4b) after applying Gaussian integration by parts and expressing it in matrix form; see Lemma C.3.

### I.2  Orthogonal means and equal-energy

Here, we use Theorem 4.2 to prove that, in contrast to the GMM, in the MLM under orthogonal and equal-energy means: LS outperforms the averaging classifier for large enough sample sizes. Assuming orthogonal means of equal energy $\mu$:

$$\boldsymbol{\pi} = \pi_1 \boldsymbol{1}_k = (1/k) \boldsymbol{1}_k, \tag{I.21}$$

$$\boldsymbol{\Pi} = (\boldsymbol{\Pi}_{11} - \boldsymbol{\Pi}_{12}) \boldsymbol{I}_k + \boldsymbol{\Pi}_{12} \boldsymbol{1}_k \boldsymbol{1}_k^T \quad \text{with} \quad \boldsymbol{\Pi}_{12} = \frac{1 - k^2 \boldsymbol{\Pi}_{11}^2}{k(k-1)} \quad \text{and} \quad \boldsymbol{\Pi}_{11} = \mathbb{E}\left[ \frac{e^{2\mu G_1}}{\left( \sum_{\ell \in [k]} e^{\mu G_\ell} \right)^2} \right].$$

Then,

$$\mathbf{\Sigma_{w,w}} - \mathbf{\Sigma_{w,\mu}}\mathbf{\Sigma_{\mu,\mu}^{-1}}\mathbf{\Sigma_{w,\mu}^T} \xrightarrow{P} \frac{\gamma}{1-\gamma} \cdot (p\mathbf{I}_k - q\mathbf{1}_k\mathbf{1}_k) , \tag{I.22}$$

where we defined

$$p := \boldsymbol{\pi}_1 - \mu^2(\boldsymbol{\pi}_1 - \mathbf{\Pi}_{11} + \mathbf{\Pi}_{12})^2 \text{ and } q := (\boldsymbol{\pi}_1^2 + \mathbf{\Pi}_{12}^2\mu^2 k - 2\mu^2\mathbf{\Pi}_{12}(\boldsymbol{\pi}_1 - \mathbf{\Pi}_{11} + \mathbf{\Pi}_{12})). \tag{I.23}$$

Thus, similar to (4.3) and with the same notation,

$$\mathbb{P}_{e,\mathrm{LS}} \xrightarrow{P} \mathbb{P}\left\{ \arg\max_{\ell \in [k]} \left\{ \sqrt{\frac{\gamma}{1-\gamma}} \cdot \left(p\mathbf{I}_k - q\mathbf{1}_k\mathbf{1}_k^T\right)^{1/2} \cdot \widetilde{\boldsymbol{g}} + \mu \cdot (\mathrm{diag}(\boldsymbol{\pi}) - \mathbf{\Pi})\,\boldsymbol{g} \right\} \neq Y(\boldsymbol{g}) \right\}. \tag{I.24}$$

In (I.24) (as well as in (4.3)), note that the matrices multiplying $\widetilde{\boldsymbol{g}}$ and $\boldsymbol{g}$ have all the form of a rank one update of a (scaled) identity matrix. It turns out that we can exploit this structure to simplify the formulae for the test error even further. Importantly, this lets us directly compare $\mathbb{P}_{e,\mathrm{LS}}$ and $P_{e,\mathrm{Avg}}$ of the two classifiers. These are detailed in Section I.3.

## I.3  Proof of Proposition 4.3

In (E.7) and (I.24), we showed the following limits for orthogonal means of equal-energy $\mu > 0$:

$$\mathbb{P}_{e,\mathrm{Avg}} \xrightarrow{P} \mathbb{P}\left( \arg\max_{\ell \in [k]} \left\{ \gamma \cdot \boldsymbol{\pi}_1 \cdot \mathbf{I}_k \cdot \widetilde{\boldsymbol{g}} + \mu\left((\boldsymbol{\pi}_1 - \mathbf{\Pi}_{11}) \cdot \mathbf{I}_k + \mathbf{\Pi}_{12}\mathbf{1}_k\mathbf{1}_k^T\right) \cdot \boldsymbol{g} \right\} \neq Y(\boldsymbol{g}) \right)$$

$$\mathbb{P}_{e,\mathrm{LS}} \xrightarrow{P} \mathbb{P}\left( \arg\max_{\ell \in [k]} \left\{ \sqrt{\frac{\gamma}{1-\gamma}} \cdot \left(p\mathbf{I}_k - q\mathbf{1}_k\mathbf{1}_k^T\right)^{1/2} \cdot \widetilde{\boldsymbol{g}} + \mu\left((\boldsymbol{\pi}_1 - \mathbf{\Pi}_{11}) \cdot \mathbf{I}_k + \mathbf{\Pi}_{12}\mathbf{1}_k\mathbf{1}_k^T\right) \cdot \boldsymbol{g} \right\} \neq Y(\boldsymbol{g}) \right),$$

where $\mathbb{P}(Y(\boldsymbol{g}) = \ell) = \frac{e^{\mu g_\ell}}{\sum_{j \in [k]} e^{\mu g_j}}$ and we have further used (I.21) and the notation in (I.23).

We compare the expression on the RHS in the above display by applying Lemma I.1 below with the following substitutions

$$\boldsymbol{g} \leftarrow \widetilde{\boldsymbol{g}}, \quad \boldsymbol{h} \leftarrow \boldsymbol{g}, \quad c(\boldsymbol{h}) \leftarrow Y(\boldsymbol{g})$$

$$p_2 \leftarrow \frac{\gamma}{1-\gamma}\left(\boldsymbol{\pi}_1 - \mu^2(\boldsymbol{\pi}_1 - \mathbf{\Pi}_{11} + \mathbf{\Pi}_{12})^2\right), \quad q_2 \leftarrow \frac{\gamma}{1-\gamma}(\boldsymbol{\pi}_1^2 + \mathbf{\Pi}_{12}^2\mu^2 k - 2\mu^2\mathbf{\Pi}_{12}(\boldsymbol{\pi}_1 - \mathbf{\Pi}_{11} + \mathbf{\Pi}_{12})),$$

$$p_1 \leftarrow \gamma\boldsymbol{\pi}_1, \quad q_1 \leftarrow 0.$$

This shows that with probability 1, $\mathbb{P}_{e,\mathrm{LS}} < \mathbb{P}_{e,\mathrm{Avg}}$ if and only if $p_2 < p_1 \Leftrightarrow \gamma < \gamma_\star = \mu^2\left(\boldsymbol{\pi}_1 - \mathbf{\Pi}_{11} + \mathbf{\Pi}_{12}\right)^2/\boldsymbol{\pi}_1$. To retrieve (??), recall that $\boldsymbol{\pi}_1 = 1/k$ and $k\mathbf{\Pi}_{11} + (k^2 - k)\mathbf{\Pi}_{12} = 1$. The only thing left to prove is that $\gamma_\star < 1$. To see this note that $p_2 > 0$ from positive semi-definiteness of the Schur matrix in (I.23). It takes simple algebra to conclude that $p_2 > 0 \implies \gamma_\star < 1$.

**Lemma I.1** *Let $k \geq 2$, $\boldsymbol{g} \sim \mathcal{N}(\mathbf{0}, \mathbf{I}_k)$ $\boldsymbol{h} \sim \mathcal{N}(\mathbf{0}, \mathbf{I}_k)$, and discrete random variable $c(\boldsymbol{h})$ such that $\mathbb{P}(c(\boldsymbol{h}) = \ell) = e^{h_\ell}/\sum_{j \in [k]} e^{h_j}$. Consider the function $F : \mathbb{R}_{>0} \times \mathbb{R} \to [0, 1]$ defined as follows*

$$F(p, q) = \mathbb{P}\left( \arg\max\left\{ \left(p\mathbf{I}_k - q\mathbf{1}_k\mathbf{1}_k^T\right)^{1/2}\boldsymbol{g} + \left(\alpha\mathbf{I}_k - \beta\mathbf{1}_k\mathbf{1}_k^T\right)^{1/2}\boldsymbol{h} \right\} \neq c(\boldsymbol{h}) \right),$$

*such that $p\mathbf{I}_k - q\mathbf{1}_k\mathbf{1}_k^T > 0$ and fixed $\alpha\mathbf{I}_k - \beta\mathbf{1}_k\mathbf{1}_k^T > 0$. Then, the following statements are true.*

1. $F(p, q) = \mathbb{P}\left( \arg\max\left\{ \sqrt{p} \cdot \boldsymbol{g} + \sqrt{\alpha}\,\boldsymbol{h} \right\} \neq c(\boldsymbol{h}) \right)$.

2. *For $0 < p_2 < p_1$ and any $q_1 < \frac{p_1}{k}, q_2 < \frac{p_2}{k}$, it holds that $F(p_2, q_2) < F(p_1, q_1)$.*

**Proof** Fix any $p > 0, q \leq \frac{p}{k}$. Denote $\boldsymbol{T} := \left(p\mathbf{I}_k - q\mathbf{1}_k\mathbf{1}_k^T\right)^{1/2}$ and $\boldsymbol{S} := \left(\alpha\mathbf{I}_k - \beta\mathbf{1}_k\mathbf{1}_k^T\right)^{1/2}$ for convenience. It can be checked that $\boldsymbol{T} := \left(\sqrt{p}\mathbf{I}_k + \frac{\sqrt{p-qk}-\sqrt{p}}{k}\mathbf{1}_k\mathbf{1}_k^T\right)$ and $\boldsymbol{S} := \left(\sqrt{\alpha}\mathbf{I}_k + \frac{\sqrt{\alpha-\beta k}-\sqrt{\alpha}}{k}\mathbf{1}_k\mathbf{1}_k^T\right)$. From these, it follows directly that

$$F(p, q) = \mathbb{P}\left( \arg\max\left\{ \sqrt{p} \cdot \boldsymbol{g} + \sqrt{\alpha}\,\boldsymbol{h} \right\} \neq c(\boldsymbol{h}) \right).$$

This shows the first statement.

Next, we show the second statement. Using the distribution of $c(\boldsymbol{h})$ and symmetry we have the following chain of equalities:

$$
\begin{aligned}
1 - F(p,q) &= \mathbb{P}\left\{\arg\max_{j\in[k]}\left\{\sqrt{p}\cdot\boldsymbol{g} + \sqrt{\alpha}\,\boldsymbol{h}\right\} = c(\boldsymbol{h})\right\} \\
&= k\cdot\mathbb{E}\left[\frac{e^{h_k}}{\sum_{j\in[k]}e^{h_j}}\cdot\mathbb{1}\left\{\arg\max\left\{\sqrt{p}\cdot\boldsymbol{g} + \sqrt{\alpha}\,\boldsymbol{h}\right\} = k\right\}\right] \\
&= k\cdot\mathbb{E}\left[\frac{e^{h_k}}{\sum_{j\in[k]}e^{h_j}}\cdot\prod_{j\in[k-1]}\mathbb{1}\left\{\sqrt{p}\cdot\boldsymbol{g}_j + \sqrt{\alpha}\,\boldsymbol{h}_j < \sqrt{p}\cdot\boldsymbol{g}_k + \sqrt{\alpha}\,\boldsymbol{h}_k\right\}\right] \\
&= k\cdot\mathbb{E}\left[\frac{e^{h_k}}{\sum_{j\in[k]}e^{h_j}}\cdot\prod_{j\in[k-1]}\mathbb{1}\left\{\boldsymbol{g}_j < \boldsymbol{g}_k + \frac{\sqrt{\alpha}\,\boldsymbol{h}_k - \sqrt{\alpha}\,\boldsymbol{h}_j}{\sqrt{p}}\right\}\right] \\
&= k\cdot\mathbb{E}\left[\frac{e^{h_k}}{\sum_{j\in[k]}e^{h_j}}\cdot\prod_{j\in[k-1]}Q\left(\boldsymbol{g}_k + \frac{\sqrt{\alpha}\,\boldsymbol{h}_j - \sqrt{\alpha}\,\boldsymbol{h}_k}{\sqrt{p}}\right)\right] =: k\cdot G(\sqrt{p}),\qquad\text{(I.25)}
\end{aligned}
$$

where in the last line we used the rotational symmetry of the Gaussian distribution:

$$
\mathbb{P}\left\{\boldsymbol{g}_j < \boldsymbol{g}_k + \frac{\sqrt{\alpha}\,\boldsymbol{h}_k - \sqrt{\alpha}\,\boldsymbol{h}_j}{\sqrt{p}}\,\Big|\,\boldsymbol{h}_1,\dots,\boldsymbol{h}_k\right\} = \mathbb{P}\left\{\boldsymbol{g}_j > \boldsymbol{g}_k + \frac{\sqrt{\alpha}\,\boldsymbol{h}_j - \sqrt{\alpha}\,\boldsymbol{h}_k}{\sqrt{p}}\,\Big|\,\boldsymbol{h}_1,\dots,\boldsymbol{h}_k\right\},
$$

and the fact that $\boldsymbol{g}_1,\dots,\boldsymbol{g}_{k-1}$ are independent.

Next, we will show that the function $\mathcal{G}(\cdot)$ defined above is strictly decreasing in $(0,\infty)$. Towards this goal, using $Q'(x) = -\frac{1}{\sqrt{2\pi}}e^{-x^2/2} = -\phi(x)$ and using the shorthand

$$
H_{kj} = \boldsymbol{h}_j - \boldsymbol{h}_k,\; j\in[k],
$$

we may compute the derivative of $\mathcal{G}$ at any $s > 0$ as follows:

$$
\begin{aligned}
\frac{\mathrm{d}\mathcal{G}(s)}{\mathrm{d}s} &= \sum_{i\in[k-1]}\mathbb{E}\left[\frac{e^{h_k}}{\sum_{j\in[k]}e^{h_j}}\cdot\phi\left(\boldsymbol{g}_k + \frac{\sqrt{\alpha}\,H_{ki}}{s}\right)\cdot\frac{\sqrt{\alpha}\,H_{ki}}{s^2}\cdot\prod_{j\neq i\in[k-1]}Q\left(\boldsymbol{g}_k + \frac{\sqrt{\alpha}\,H_{kj}}{s}\right)\right] \\
&= \sum_{i\in[k-1]}\mathbb{E}\left[\frac{1}{\sum_{j\in[k]}e^{H_{kj}}}\cdot\phi\left(\boldsymbol{g}_k + \frac{\sqrt{\alpha}\,H_{ki}}{s}\right)\cdot\frac{\sqrt{\alpha}\,H_{ki}}{s^2}\cdot\prod_{j\neq i\in[k-1]}Q\left(\boldsymbol{g}_k + \frac{\sqrt{\alpha}\,H_{kj}}{s}\right)\right] \\
&= \sum_{i\in[k-1]}\frac{\sqrt{\alpha}}{s^2}\mathbb{E}\left[H_{ki}\cdot\mathcal{A}_i\left(\boldsymbol{g}_k,\{H_{kj}\}_{j\in[k-1]}\right)\right],\qquad\text{(I.26)}
\end{aligned}
$$

where in the last line we have defined

$$
\mathcal{A}_i\left(\boldsymbol{g}_k,\{H_{kj}\}_{j\in[k-1]}\right) := \frac{1}{1 + \sum_{j\in[k-1]}e^{H_{kj}}}\cdot\phi\left(\boldsymbol{g}_k + \frac{\sqrt{\alpha}\,H_{ki}}{s}\right)\cdot\prod_{j\neq i\in[k-1]}Q\left(\boldsymbol{g}_k + \frac{\sqrt{\alpha}\,H_{kj}}{s}\right),
$$

Next, we use Gaussian integration by parts (GIBP) to further simplify the expression in (I.26). Fix any $i\in[k-1]$. Then, by (GIBP):

$$
\begin{aligned}
A_i &:= \mathbb{E}\left[H_{ki}\cdot\mathcal{A}_i\left(\boldsymbol{g}_k,\{H_{kj}\}_{j\in[k-1]}\right)\right] \qquad\qquad\qquad\qquad\qquad\qquad\qquad\qquad\text{(I.27)} \\
&= \mathbb{E}[H_{ki}^2]\mathbb{E}\left[\frac{\mathrm{d}}{\mathrm{d}H_{ki}}\mathcal{A}_i\left(\boldsymbol{g}_k,\{H_{kj}\}_{j\in[k-1]}\right)\right] + \sum_{\substack{\ell\in[k-1]\\\ell\neq i}}\mathbb{E}[H_{ki}\cdot H_{k\ell}]\mathbb{E}\left[\frac{\mathrm{d}}{\mathrm{d}H_{k\ell}}\mathcal{A}_i\left(\boldsymbol{g}_k,\{H_{kj}\}_{j\in[k-1]}\right)\right] \\
&= \underbrace{2\,\mathbb{E}\left[\frac{\mathrm{d}}{\mathrm{d}H_{ki}}\mathcal{A}_i\left(\boldsymbol{g}_k,\{H_{kj}\}_{j\in[k-1]}\right)\right]}_{\text{TermI}} + \underbrace{\sum_{\substack{\ell\in[k-1]\\\ell\neq i}}\mathbb{E}\left[\frac{\mathrm{d}}{\mathrm{d}H_{k\ell}}\mathcal{A}_i\left(\boldsymbol{g}_k,\{H_{kj}\}_{j\in[k-1]}\right)\right]}_{\text{TermII}},\qquad\text{(I.28)}
\end{aligned}
$$

where in the second line, we used the fact that $\boldsymbol{h}\sim\mathcal{N}(\boldsymbol{0},\boldsymbol{I}_k)$ to compute

$$
\mathbb{E}[H_{ki}^2] = 2,\quad\text{and}\quad\mathbb{E}[H_{ki}H_{k\ell}] = 1,\;\ell\neq i,\;\ell\in[k-1].
$$

We now compute the derivatives in (I.28). First, for any $\ell \in [k-1]$, $\ell \neq i$,

$$\frac{d\mathcal{A}_i \left( g_k, \{H_{kj}\}_{j \in [k-1]} \right)}{dH_{k\ell}} = -\frac{e^{H_{k\ell}}}{\left( 1 + \sum_{j \in [k-1]} e^{H_{kj}} \right)^2} \cdot \phi \left( g_k + \frac{\sqrt{\alpha} \, H_{ki}}{s} \right) \cdot \prod_{j \neq i \in [k-1]} Q \left( g_k + \frac{\sqrt{\alpha} \, H_{kj}}{s} \right) \quad =: \mathrm{TermII(a)}_\ell$$

$$- \frac{\sqrt{\alpha}}{s} \cdot \frac{1}{1 + \sum_{j \in [k-1]} e^{H_{kj}}} \cdot \phi \left( g_k + \frac{\sqrt{\alpha} \, H_{ki}}{s} \right) \cdot \phi \left( g_k + \frac{\sqrt{\alpha} \, H_{k\ell}}{s} \right) \cdot \prod_{j \neq (i,\ell) \in [k-1]} Q \left( g_k + \frac{\sqrt{\alpha} \, H_{kj}}{s} \right) \quad =: \mathrm{TermII(b)}_\ell$$

$$= \mathrm{TermII(a)}_\ell + \mathrm{TermII(b)}_\ell \tag{I.29}$$

Thus,

$$\mathrm{TermII} = \sum_{\ell \neq i \in [k-1]} \mathbb{E} \left[ \mathrm{TermII(a)}_\ell \right] + \mathbb{E} \left[ \mathrm{TermII(b)}_\ell \right] =: \mathbb{E} \left[ \mathrm{TermII(a)} \right] + N_i, \tag{I.30}$$

where we defined

$$N_i = -\frac{\sqrt{\alpha}}{s} \cdot \mathbb{E} \left[ \frac{\phi \left( g_k + \frac{\sqrt{\alpha} \, H_{ki}}{s} \right)}{1 + \sum_{j \in [k-1]} e^{H_{kj}}} \cdot \sum_{\ell \neq i \in [k-1]} \left\{ \phi \left( g_k + \frac{\sqrt{\alpha} \, H_{k\ell}}{s} \right) \cdot \prod_{j \neq (i,\ell) \in [k-1]} Q \left( g_k + \frac{\sqrt{\alpha} \, H_{kj}}{s} \right) \right\} \right] < 0. \tag{I.31}$$

and we remark for later use that

$$\mathbb{E} \left[ \mathrm{TermII(a)} \right] = \sum_{\ell \neq i \in [k-1]} \mathbb{E} \left[ \mathrm{TermII(a)}_\ell \right] < 0. \tag{I.32}$$

Second, it holds that

$$\frac{d\mathcal{A}_i \left( g_k, \{H_{kj}\}_{j \in [k-1]} \right)}{dH_{ki}} = -\frac{e^{H_{ki}}}{\left( 1 + \sum_{j \in [k-1]} e^{H_{kj}} \right)^2} \cdot \phi \left( g_k + \frac{\sqrt{\alpha} \, H_{ki}}{s} \right) \cdot \prod_{j \neq i \in [k-1]} Q \left( g_k + \frac{\sqrt{\alpha} \, H_{kj}}{s} \right)$$

$$+ \frac{d\phi \left( g_k + \frac{\sqrt{\alpha} \, H_{ki}}{s} \right)}{dH_{ki}} \cdot \frac{1}{1 + \sum_{j \in [k-1]} e^{H_{kj}}} \cdot \prod_{j \neq i \in [k-1]} Q \left( g_k + \frac{\sqrt{\alpha} \, H_{kj}}{s} \right)$$

$$= -\frac{e^{H_{ki}}}{\left( 1 + \sum_{j \in [k-1]} e^{H_{kj}} \right)^2} \cdot \phi \left( g_k + \frac{\sqrt{\alpha} \, H_{ki}}{s} \right) \cdot \prod_{j \neq i \in [k-1]} Q \left( g_k + \frac{\sqrt{\alpha} \, H_{kj}}{s} \right) \quad =: \mathrm{TermI(a)}$$

$$- \frac{\sqrt{\alpha}}{s} \left( g_k + \frac{\sqrt{\alpha} \, H_{ki}}{s} \right) \phi \left( g_k + \frac{\sqrt{\alpha} \, H_{ki}}{s} \right) \cdot \frac{1}{1 + \sum_{j \in [k-1]} e^{H_{kj}}} \cdot \prod_{j \neq i \in [k-1]} Q \left( g_k + \frac{\sqrt{\alpha} \, H_{kj}}{s} \right) \quad =: \mathrm{TermI(b)},$$

$$\tag{I.33}$$

$$= \mathrm{TermI(a)} + \mathrm{TermI(b)},$$

where in the penultimate line we used the fact that $\phi'(x) = -x\phi(x)$. Consider the two terms in (I.33). Clearly,

$$\mathbb{E}[\mathrm{TermI(a)}] < 0. \tag{I.34}$$

For the second term we observe that:

$$\mathbb{E} \left[ \mathrm{TermI(b)} \right] = -\frac{\sqrt{\alpha}}{s} \mathbb{E} \left[ \left( g_k + \frac{\sqrt{\alpha} \, H_{ki}}{s} \right) \cdot \mathcal{A}_i \left( g_k, \{H_{kj}\}_{j \in [k-1]} \right) \right] \tag{I.35}$$

$$= -\frac{\alpha}{s^2} \cdot \mathbb{E} \left[ H_{ki} \cdot \mathcal{A}_i \left( g_k, \{H_{kj}\}_{j \in [k-1]} \right) \right] - \frac{\sqrt{\alpha}}{s} \mathbb{E} \left[ g_k \cdot \mathcal{A}_i \left( g_k, \{H_{kj}\}_{j \in [k-1]} \right) \right]$$

$$= -\frac{\alpha}{s^2} \cdot A_i - \frac{\sqrt{\alpha}}{s} \mathbb{E} \left[ g_k \cdot \mathcal{A}_i \left( g_k, \{H_{kj}\}_{j \in [k-1]} \right) \right]. \tag{I.36}$$

Moreover, using again GIBP, $\mathbb{E}[g_k^2] = 1$, $\mathbb{E}[g_k H_{kj}] = 0$, $j \in [k]$ and the fact that $\phi'(x) = -x\phi(x)$,

$$\mathbb{E}\left[g_k \cdot \mathcal{A}_i\left(g_k, \{H_{kj}\}_{j\in[k-1]}\right)\right] = \mathbb{E}\left[\frac{\mathrm{d}\mathcal{A}_i\left(g_k, \{H_{kj}\}_{j\in[k]}\right)}{\mathrm{d}g_k}\right]$$

$$= -\mathbb{E}\left[\left(g_k + \frac{\sqrt{\alpha}\,H_{ki}}{s}\right) \cdot \mathcal{A}_i\left(g_k, \{H_{kj}\}_{j\in[k-1]}\right)\right]$$

$$- \sum_{\ell \neq i \in [k-1]} \mathbb{E}\left[\phi\left(g_k + \frac{\sqrt{\alpha}\,H_{k\ell}}{s}\right) \cdot \frac{1}{1 + \sum_{j\in[k-1]} e^{H_{k\ell}}} \cdot \phi\left(g_k + \frac{\sqrt{\alpha}\,H_{ki}}{s}\right) \cdot \prod_{j\neq(i,\ell)\in[k-1]} Q\left(g_k + \frac{\sqrt{\alpha}\,H_{kj}}{s}\right)\right]$$

$$= \frac{s}{\sqrt{\alpha}}\,\mathbb{E}\left[\mathrm{TermI(b)}\right] + \frac{s}{\sqrt{\alpha}}\,N_i, \tag{I.37}$$

where, we have recalled (I.35) and (I.31). Using (I.37) in (I.36), we find that

$$\mathbb{E}\left[\mathrm{TermI(b)}\right] = -\frac{\alpha}{s^2}\cdot A_i - \mathbb{E}\left[\mathrm{TermI(b)}\right] - N_i \implies \mathbb{E}\left[\mathrm{TermI(b)}\right] = -\frac{\alpha}{2s^2}\cdot A_i - \frac{N_i}{2}. \tag{I.38}$$

We are now ready to put things together:

$$\begin{aligned} A_i &= 2 \cdot \mathrm{TermI} + \mathrm{TermII} && \text{by (I.28)} \\ &= 2\,\mathbb{E}[\mathrm{TermI(a)}] + 2\,\mathbb{E}[\mathrm{TermI(b)}] + \mathbb{E}[\mathrm{TermII(a)}] + N_i && \text{by (I.30)} \\ &= 2\,\mathbb{E}[\mathrm{TermI(a)}] - \frac{\alpha}{s^2}\cdot A_i - N_i + \mathbb{E}[\mathrm{TermII(a)}] + N_i && \text{by (I.38)} \\ \implies A_i &= \frac{s^2}{s^2 + \alpha}\left(2\,\mathbb{E}[\mathrm{TermI(a)}] + \mathbb{E}[\mathrm{TermII(a)}]\right) \\ &< 0 && \text{by (I.34) and (I.32).} \end{aligned}$$

From this, (I.26) and (I.27), we have shown that $\mathcal{G}$ is strictly decreasing in $(0, \infty)$. Recalling the definition of $\mathcal{G}$ in (I.25), this implies that $F(p,q)$ is strictly increasing in $p > 0$, as desired to complete the proof. ∎

## J   Weighted LS for GMM (Proof of Theorem B.1)

### J.1   Computing $\Sigma_{w,\mu}$

The WLS estimator solves:

$$\min_{\boldsymbol{W}\in\mathbb{R}^{k\times d},\, \boldsymbol{b}\in\mathbb{R}^k} \frac{1}{2n}\left\|\left(\boldsymbol{W}\boldsymbol{X} + \boldsymbol{b}\boldsymbol{1}_n^T - \boldsymbol{Y}\right)\boldsymbol{D}\right\|_F^2 = \sum_{\ell=1}^{k} \min_{\boldsymbol{w}_\ell, b_\ell} \frac{1}{2n}\left\|\boldsymbol{D}\left(\boldsymbol{X}^T\boldsymbol{w}_\ell + b_\ell \boldsymbol{1}_n - \boldsymbol{Y}_\ell\right)\right\|_{\ell_2}^2$$

$$= \sum_{\ell=1}^{k} \min_{\boldsymbol{w}_\ell, b_\ell} \frac{1}{2n}\left\|\boldsymbol{D}\left(\boldsymbol{Y}^T\boldsymbol{M}^T\boldsymbol{w}_\ell + \boldsymbol{Z}^T\boldsymbol{w}_\ell + b_\ell \boldsymbol{1}_n - \boldsymbol{Y}_\ell\right)\right\|_{\ell_2}^2.$$

Define

$$\mathcal{L}_{PO}\left(\boldsymbol{w}_\ell, b_\ell\right) := \frac{1}{2n}\left\|\boldsymbol{D}\left(\boldsymbol{Y}^T\boldsymbol{M}^T\boldsymbol{w}_\ell + \boldsymbol{Z}^T\boldsymbol{w}_\ell + b_\ell \boldsymbol{1}_n - \boldsymbol{Y}_\ell\right)\right\|_{\ell_2}^2. \tag{J.1}$$

**Identifying the AO.** By duality we have

$$\min_{\boldsymbol{w}_\ell, b_\ell} \mathcal{L}_{PO}\left(\boldsymbol{w}_\ell, b_\ell\right) = \min_{\boldsymbol{u}, \boldsymbol{w}_\ell, b_\ell} \max_{\boldsymbol{s}} \frac{1}{n}\left(\boldsymbol{s}^T\boldsymbol{D}\boldsymbol{Y}^T\boldsymbol{M}^T\boldsymbol{w}_\ell + \boldsymbol{s}^T\boldsymbol{D}\boldsymbol{Z}^T\boldsymbol{w}_\ell + b_\ell \boldsymbol{s}^T\boldsymbol{D}\boldsymbol{1}_n - \boldsymbol{s}^T\boldsymbol{D}\boldsymbol{Y}_\ell - \boldsymbol{s}^T\boldsymbol{u} + \frac{\|\boldsymbol{u}\|_{\ell_2}^2}{2}\right)$$

Note that the above is jointly convex in $(\boldsymbol{u}, \boldsymbol{w}_\ell, b_\ell)$ and concave in $\boldsymbol{s}$. Thus, we consider the Auxiliary Optimization (AO) problem

$$\min_{\boldsymbol{u}, \boldsymbol{w}_\ell, b_\ell} \max_{\boldsymbol{s}} \frac{1}{n}\left(\boldsymbol{s}^T\boldsymbol{D}\boldsymbol{Y}^T\boldsymbol{M}^T\boldsymbol{w}_\ell + \sigma\|\boldsymbol{w}_\ell\|_{\ell_2}\boldsymbol{g}^T\boldsymbol{D}\boldsymbol{s} + \sigma\|\boldsymbol{D}\boldsymbol{s}\|_{\ell_2}\boldsymbol{h}^T\boldsymbol{w}_\ell + b_\ell \boldsymbol{s}^T\boldsymbol{D}\boldsymbol{1}_n - \boldsymbol{s}^T\boldsymbol{D}\boldsymbol{Y}_\ell - \boldsymbol{s}^T\boldsymbol{u} + \frac{\|\boldsymbol{u}\|_{\ell_2}^2}{2}\right),$$

where $g \in \mathbb{R}^n$ and $h \in \mathbb{R}^d$ are independent Gaussian random vectors with i.i.d. $\mathcal{N}(0,1)$ entries. Moreover, we carry out a change of variable $s \to Ds$ to arrive at

$$\min_{u,w_\ell,b_\ell} \max_{s} \frac{1}{n}\left(s^T Y^T M^T w_\ell + \sigma \|w_\ell\|_{\ell_2} g^T s + \sigma \|s\|_{\ell_2} h^T w_\ell + b_\ell s^T \mathbf{1}_n - s^T Y_\ell - s^T D^{-1} u + \frac{\|u\|_{\ell_2}^2}{2}\right).$$

**Simplification of the AO.** Maximizing over the direction of $s$ and setting its norm $\beta = \|s\|_{\ell_2}$ above we arrive at

$$\min_{u,w_\ell,b_\ell} \max_{\beta \geq 0} \max_{s:\|s\|_{\ell_2}=1} \frac{1}{n}\left(\beta s^T Y^T M^T w_\ell + \sigma\beta \|w_\ell\|_{\ell_2} g^T s + \sigma\beta h^T w_\ell + b_\ell \beta s^T \mathbf{1}_n - \beta s^T Y_\ell - \beta s^T D^{-1} u + \frac{\|u\|_{\ell_2}^2}{2}\right)$$

$$= \min_{u,w_\ell,b_\ell} \max_{\beta \geq 0} \frac{1}{n}\left(\beta \left\|Y^T M^T w_\ell + \sigma \|w_\ell\|_{\ell_2} g + b_\ell \mathbf{1}_n - Y_\ell - D^{-1} u\right\|_{\ell_2} + \sigma\beta h^T w_\ell + \frac{\|u\|_{\ell_2}^2}{2}\right)$$

$$= \min_{u,b_\ell} \max_{\beta \geq 0} \min_{w_\ell} \frac{1}{n}\left(\beta \left\|Y^T M^T w_\ell + \sigma \|w_\ell\|_{\ell_2} g + b_\ell \mathbf{1}_n - Y_\ell - D^{-1} u\right\|_{\ell_2} + \sigma\beta h^T w_\ell + \frac{\|u\|_{\ell_2}^2}{2}\right).$$

To continue, consider the singular value decomposition

$$M = U\Sigma V^T = \begin{bmatrix} u_1 & u_2 & \ldots & u_r \end{bmatrix} \mathrm{diag}(\sigma_1, \sigma_2, \ldots, \sigma_r) \begin{bmatrix} v_1^T \\ v_2^T \\ \ldots \\ v_r^T \end{bmatrix}, \tag{J.2}$$

with $r := rank(M) \leq k$ and define the variable $\alpha = U^T w_\ell$ and $\alpha_\perp = U_\perp^T w_\ell$ where $U_\perp$ is the orthogonal complement of the columns of $U$. With these definitions the above optimization problem reduces to

$$\min_{u,b_\ell} \max_{\beta \geq 0} \min_{\alpha} \min_{\alpha_\perp} \frac{1}{n}\left(\beta \left\|Y^T V\Sigma\alpha + \sigma\sqrt{\|\alpha\|_{\ell_2}^2 + \|\alpha_\perp\|_{\ell_2}^2} g + b_\ell \mathbf{1}_n - Y_\ell - D^{-1} u\right\|_{\ell_2} + \sigma\beta h^T U\alpha + \sigma\beta h^T U_\perp \alpha_\perp + \frac{\|u\|_{\ell_2}^2}{2}\right)$$

Decomposing the optimization over $\alpha_\perp$ in terms of its direction and norm $\alpha_0 = \|\alpha_\perp\|_{\ell_2}$ we arrive at

$$\min_{u,b_\ell} \max_{\beta \geq 0} \min_{\alpha} \min_{\alpha_0 \geq 0} \frac{1}{n}\left(\beta \left\|Y^T V\Sigma\alpha + \sigma\sqrt{\|\alpha\|_{\ell_2}^2 + \alpha_0^2} g + b_\ell \mathbf{1}_n - Y_\ell - D^{-1} u\right\|_{\ell_2} + \sigma\beta h^T U\alpha - \sigma\alpha_0\beta \left\|U_\perp^T h\right\|_{\ell_2} + \frac{\|u\|_{\ell_2}^2}{2}\right).$$

Since $U^T h$ is $r \leq k$ dimensional in our asymptotic regime the term $\frac{h^T U\alpha}{n}$ can be ignored. Also replacing $\beta$ with $\beta/\sqrt{n}$ we thus arrive at

$$\min_{u,b_\ell} \max_{\beta \geq 0} \min_{\alpha} \min_{\alpha_0 \geq 0} \frac{\beta}{\sqrt{n}} \left\|Y^T V\Sigma\alpha + \sigma\sqrt{\|\alpha\|_{\ell_2}^2 + \alpha_0^2} g + b_\ell \mathbf{1}_n - Y_\ell - D^{-1} u\right\|_{\ell_2} - \frac{1}{\sqrt{n}}\sigma\alpha_0\beta \left\|U_\perp^T h\right\|_{\ell_2} + \frac{\|u\|_{\ell_2}^2}{2n}$$

$$= \min_{u,b_\ell} \max_{\beta \geq 0} \min_{\alpha} \min_{\alpha_0 \geq 0} \min_{\tau \geq 0} \frac{\beta}{2n\tau} \left\|Y^T V\Sigma\alpha + \sigma\sqrt{\|\alpha\|_{\ell_2}^2 + \alpha_0^2} g + b_\ell \mathbf{1}_n - Y_\ell - D^{-1} u\right\|_{\ell_2}^2 + \frac{\beta\tau}{2}$$

$$- \frac{1}{\sqrt{n}}\sigma\alpha_0\beta \left\|U_\perp^T h\right\|_{\ell_2} + \frac{\|u\|_{\ell_2}^2}{2n}$$

$$= \min_{b_\ell} \max_{\beta \geq 0} \min_{\alpha} \min_{\alpha_0 \geq 0} \min_{\tau \geq 0} \min_{u} \frac{\beta}{2n\tau} \left\|Y^T V\Sigma\alpha + \sigma\sqrt{\|\alpha\|_{\ell_2}^2 + \alpha_0^2} g + b_\ell \mathbf{1}_n - Y_\ell - D^{-1} u\right\|_{\ell_2}^2 + \frac{\beta\tau}{2}$$

$$- \frac{1}{\sqrt{n}}\sigma\alpha_0\beta \left\|U_\perp^T h\right\|_{\ell_2} + \frac{\|u\|_{\ell_2}^2}{2n}.$$

Setting the derivative with respect to $u$ to zero we arrive at

$$u = \frac{\beta}{\tau} D^{-1} \left(I + \frac{\beta}{\tau} D^{-2}\right)^{-1} \left(Y^T V\Sigma\alpha + \sigma\sqrt{\|\alpha\|_{\ell_2}^2 + \alpha_0^2} g + b_\ell \mathbf{1}_n - Y_\ell\right).$$

Plugging the latter into the above the AO simplifies to

$$\min_{b_\ell} \max_{\beta \geq 0} \min_{\boldsymbol{\alpha}} \min_{\alpha_0 \geq 0} \min_{\tau \geq 0} \frac{\beta}{2\tau n} \mathrm{trace}\left(\boldsymbol{t}^T \left(\boldsymbol{I} + \frac{\beta}{\tau} \boldsymbol{D}^{-2}\right)^{-1} \boldsymbol{t}\right) - \frac{1}{\sqrt{n}} \sigma \alpha_0 \beta \left\|\boldsymbol{U}_\perp^T \boldsymbol{h}\right\|_{\ell_2} + \frac{\beta\tau}{2}$$

where

$$\boldsymbol{t} := \boldsymbol{Y}^T \boldsymbol{V}\boldsymbol{\Sigma}\boldsymbol{\alpha} + \sigma\sqrt{\|\boldsymbol{\alpha}\|_{\ell_2}^2 + \alpha_0^2}\boldsymbol{g} + b_\ell \boldsymbol{1}_n - \boldsymbol{Y}_\ell = \boldsymbol{Y}^T \left(\boldsymbol{V}\boldsymbol{\Sigma}\boldsymbol{\alpha} - \boldsymbol{e}_\ell\right) + \sigma\sqrt{\|\boldsymbol{\alpha}\|_{\ell_2}^2 + \alpha_0^2}\boldsymbol{g} + b_\ell \boldsymbol{1}_n$$

To continue note that in our asymptotic regime we have

$$\frac{1}{\sqrt{n}} \left\|\boldsymbol{U}_\perp^T \boldsymbol{h}\right\|_{\ell_2} \xrightarrow{P} \sqrt{\gamma}$$

and the cross terms can be ignored so that in an asymptotic sense

$$\frac{1}{n}\mathrm{trace}\left(\boldsymbol{t}^T \left(\boldsymbol{I} + \frac{\beta}{\tau} \boldsymbol{D}^{-2}\right)^{-1} \boldsymbol{t}\right)$$

$$= \frac{1}{n}\sigma^2 \left(\|\boldsymbol{\alpha}\|_{\ell_2}^2 + \alpha_0^2\right) \mathrm{trace}\left(\left(\boldsymbol{I} + \frac{\beta}{\tau} \boldsymbol{D}^{-2}\right)^{-1}\right)$$

$$+ \frac{1}{n}\left(\boldsymbol{Y}^T \left(\boldsymbol{V}\boldsymbol{\Sigma}\boldsymbol{\alpha} - \boldsymbol{e}_\ell\right) + b_\ell \boldsymbol{1}_n\right)^T \left(\boldsymbol{I} + \frac{\beta}{\tau} \boldsymbol{D}^{-2}\right)^{-1} \left(\boldsymbol{Y}^T \left(\boldsymbol{V}\boldsymbol{\Sigma}\boldsymbol{\alpha} - \boldsymbol{e}_\ell\right) + b_\ell \boldsymbol{1}_n\right)$$

Therefore we arrive at

$$\min_{b_\ell} \max_{\beta \geq 0} \min_{\boldsymbol{\alpha}} \min_{\tau \geq 0} \min_{\alpha_0 \geq 0} \frac{\beta}{2\tau n}\sigma^2 \left(\|\boldsymbol{\alpha}\|_{\ell_2}^2 + \alpha_0^2\right) \mathrm{trace}\left(\left(\boldsymbol{I} + \frac{\beta}{\tau} \boldsymbol{D}^{-2}\right)^{-1}\right) - \sigma\alpha_0\beta\sqrt{\gamma} + \frac{\beta\tau}{2}$$

$$+ \frac{\beta}{2\tau n}\left(\boldsymbol{Y}^T \left(\boldsymbol{V}\boldsymbol{\Sigma}\boldsymbol{\alpha} - \boldsymbol{e}_\ell\right) + b_\ell \boldsymbol{1}_n\right)^T \left(\boldsymbol{I} + \frac{\beta}{\tau} \boldsymbol{D}^{-2}\right)^{-1} \left(\boldsymbol{Y}^T \left(\boldsymbol{V}\boldsymbol{\Sigma}\boldsymbol{\alpha} - \boldsymbol{e}_\ell\right) + b_\ell \boldsymbol{1}_n\right),$$

which can be rewritten in the form

$$\min_{b_\ell} \max_{\beta \geq 0} \min_{\boldsymbol{\alpha}} \min_{\tau \geq 0} \frac{\beta}{2\tau n}\sigma^2 \|\boldsymbol{\alpha}\|_{\ell_2}^2 \mathrm{trace}\left(\left(\boldsymbol{I} + \frac{\beta}{\tau} \boldsymbol{D}^{-2}\right)^{-1}\right) + \frac{\beta\tau}{2}$$

$$+ \frac{\beta}{2\tau n}\left(\boldsymbol{Y}^T \left(\boldsymbol{V}\boldsymbol{\Sigma}\boldsymbol{\alpha} - \boldsymbol{e}_\ell\right) + b_\ell \boldsymbol{1}_n\right)^T \left(\boldsymbol{I} + \frac{\beta}{\tau} \boldsymbol{D}^{-2}\right)^{-1} \left(\boldsymbol{Y}^T \left(\boldsymbol{V}\boldsymbol{\Sigma}\boldsymbol{\alpha} - \boldsymbol{e}_\ell\right) + b_\ell \boldsymbol{1}_n\right)$$

$$+ \frac{\beta}{2\tau n}\sigma^2\alpha_0^2\mathrm{trace}\left(\left(\boldsymbol{I} + \frac{\beta}{\tau} \boldsymbol{D}^{-2}\right)^{-1}\right) - \alpha_0\sigma\beta\sqrt{\gamma}.$$

To continue further we shall assume $\boldsymbol{D} = \mathrm{diag}\left(\boldsymbol{Y}^T\boldsymbol{\omega}\right)$. Note that in this case

$$\frac{1}{n}\mathrm{trace}\left(\left(\boldsymbol{I} + \frac{\beta}{\tau} \boldsymbol{D}^{-2}\right)^{-1}\right) = \frac{1}{n}\sum_{i=1}^n \frac{(\boldsymbol{y}_i^T\boldsymbol{\omega})^2}{(\boldsymbol{y}_i^T\boldsymbol{\omega})^2 + \frac{\beta}{\tau}} = \sum_{\ell=1}^k \frac{n_\ell}{n}\frac{\omega_\ell^2}{\omega_\ell^2 + \frac{\beta}{\tau}} \xrightarrow{P} \sum_{\ell=1}^k \frac{\pi_\ell\omega_\ell^2}{\omega_\ell^2 + \frac{\beta}{\tau}}.$$

Also,

$$\frac{1}{n}\left(\boldsymbol{Y}^T \left(\boldsymbol{V}\boldsymbol{\Sigma}\boldsymbol{\alpha} - \boldsymbol{e}_\ell\right) + b_\ell \boldsymbol{1}_n\right)^T \left(\boldsymbol{I} + \frac{\beta}{\tau} \boldsymbol{D}^{-2}\right)^{-1} \left(\boldsymbol{Y}^T \left(\boldsymbol{V}\boldsymbol{\Sigma}\boldsymbol{\alpha} - \boldsymbol{e}_\ell\right) + b_\ell \boldsymbol{1}_n\right)$$

$$= \frac{1}{n}\left(\boldsymbol{V}\boldsymbol{\Sigma}\boldsymbol{\alpha} - \boldsymbol{e}_\ell\right)^T \boldsymbol{Y}\left(\boldsymbol{I} + \frac{\beta}{\tau} \boldsymbol{D}^{-2}\right)^{-1} \boldsymbol{Y}^T\left(\boldsymbol{V}\boldsymbol{\Sigma}\boldsymbol{\alpha} - \boldsymbol{e}_\ell\right)$$

$$+ \frac{2}{n}b_\ell\boldsymbol{1}_n^T\left(\boldsymbol{I} + \frac{\beta}{\tau} \boldsymbol{D}^{-2}\right)^{-1} \boldsymbol{Y}^T\left(\boldsymbol{V}\boldsymbol{\Sigma}\boldsymbol{\alpha} - \boldsymbol{e}_\ell\right) + \frac{b_\ell^2}{n}\mathrm{trace}\left(\left(\boldsymbol{I} + \frac{\beta}{\tau} \boldsymbol{D}^{-2}\right)^{-1}\right)$$

$$\xrightarrow{P} \left(\boldsymbol{V}\boldsymbol{\Sigma}\boldsymbol{\alpha} - \boldsymbol{e}_\ell\right)^T \mathrm{diag}\left(\frac{\pi_1\omega_1^2}{\omega_1^2 + \frac{\beta}{\tau}}, \frac{\pi_2\omega_2^2}{\omega_2^2 + \frac{\beta}{\tau}}, \ldots, \frac{\pi_k\omega_k^2}{\omega_k^2 + \frac{\beta}{\tau}}\right)\left(\boldsymbol{V}\boldsymbol{\Sigma}\boldsymbol{\alpha} - \boldsymbol{e}_\ell\right)$$

$$+ 2b_\ell\left[\begin{array}{cccc}\frac{\pi_1\omega_1^2}{\omega_1^2 + \frac{\beta}{\tau}} & \frac{\pi_2\omega_2^2}{\omega_2^2 + \frac{\beta}{\tau}} & \cdots & \frac{\pi_k\omega_k^2}{\omega_k^2 + \frac{\beta}{\tau}}\end{array}\right]\left(\boldsymbol{V}\boldsymbol{\Sigma}\boldsymbol{\alpha} - \boldsymbol{e}_\ell\right) + b_\ell^2\left(\sum_{\ell=1}^k \frac{\pi_\ell\omega_\ell^2}{\omega_\ell^2 + \frac{\beta}{\tau}}\right).$$

Next define

$$A(\eta) := \begin{bmatrix} \sigma^2 \left(\pi^T \nu(\eta)\right) I + \Sigma V^T \mathrm{diag}\left(\pi\right) \mathrm{diag}\left(\nu(\eta)\right) V \Sigma & \Sigma V^T \mathrm{diag}\left(\nu(\eta)\right) \pi \\ \pi^T \mathrm{diag}\left(\nu(\eta)\right) V \Sigma & \pi^T \nu(\eta) \end{bmatrix}$$

$$c_\ell := \begin{bmatrix} \Sigma V^T e_\ell \\ 1 \end{bmatrix}, \tag{J.3}$$

where

$$\nu(\eta) = \frac{1}{\gamma} \begin{bmatrix} \frac{\omega_1^2}{\omega_1^2 + \eta} \\ \frac{\omega_2^2}{\omega_2^2 + \eta} \\ \dots \\ \frac{\omega_k^2}{\omega_k^2 + \eta} \end{bmatrix}. \tag{J.4}$$

We thus arrive at

$$\min_{\alpha} \min_{b_\ell} \max_{\alpha_0 \geq 0} \max_{\beta \geq 0} \min_{\tau \geq 0} \frac{\gamma \beta}{2\tau} \left( \pi_\ell \nu_\ell \left(\frac{\beta}{\tau}\right) + \begin{bmatrix} \alpha^T & b_\ell \end{bmatrix} A\left(\frac{\beta}{\tau}\right) \begin{bmatrix} \alpha \\ b_\ell \end{bmatrix} - 2\pi_\ell \nu_\ell \left(\frac{\beta}{\tau}\right) c_\ell^T \begin{bmatrix} \alpha \\ b_\ell \end{bmatrix} \right)$$

$$+ \frac{\gamma \beta}{2\tau} \sigma^2 \left( \pi^T \nu \left(\frac{\beta}{\tau}\right) \right) \alpha_0^2 - \alpha_0 \sigma \beta \sqrt{\gamma} + \frac{\beta \tau}{2}.$$

**Deterministic Analysis of the AO.** Setting the derivative of the above with respect to $\alpha_0$ to zero we arrive at

$$\frac{\gamma \beta}{\tau} \sigma^2 \left( \pi^T \nu \left(\frac{\beta}{\tau}\right) \right) \alpha_0 - \sigma \beta \sqrt{\gamma} = 0 \quad \Rightarrow \quad \alpha_0 = \frac{\tau}{\sigma \sqrt{\gamma} \left( \pi^T \nu \left(\frac{\beta}{\tau}\right) \right)}.$$

Note that the above objective has the form

$$f\left(\frac{\beta}{\tau}\right) - \alpha_0 \sigma \beta \sqrt{\gamma} + \frac{\beta \tau}{2}$$

with

$$f(\eta) := \frac{\eta \gamma}{2} \left( \pi_\ell \nu_\ell(\eta) + \begin{bmatrix} \alpha^T & b_\ell \end{bmatrix} A(\eta) \begin{bmatrix} \alpha \\ b_\ell \end{bmatrix} - 2\pi_\ell \nu_\ell(\eta) c_\ell^T \begin{bmatrix} \alpha \\ b_\ell \end{bmatrix} \right) + \frac{\gamma \eta}{2} \sigma^2 \left( \pi^T \nu(\eta) \right) \alpha_0^2.$$

Thus setting the derivatives with respect to $\beta$ and $\tau$ to zero, we have

$$\frac{1}{\tau} f'\left(\frac{\beta}{\tau}\right) - \alpha_0 \sigma \sqrt{\gamma} + \frac{\tau}{2} = 0 \quad \Rightarrow \quad f'\left(\frac{\beta}{\tau}\right) - \alpha_0 \sigma \sqrt{\gamma} \tau + \frac{\tau^2}{2} = 0 \quad \Rightarrow \quad f'\left(\frac{\beta}{\tau}\right) = \tau^2 \left( \frac{1}{\pi^T \nu \left(\frac{\beta}{\tau}\right)} - \frac{1}{2} \right)$$

and

$$-\frac{\beta}{\tau^2} f'\left(\frac{\beta}{\tau}\right) + \frac{\beta}{2} = 0 \quad \Rightarrow \quad \tau^2 = 2 f'\left(\frac{\beta}{\tau}\right).$$

Combining the latter two we conclude that $\pi^T \nu \left(\frac{\beta}{\tau}\right) = 1$. Thus, $\eta = \frac{\beta}{\tau}$ is the solution to $\pi^T \nu(\eta) = 1$.
To calculate $\tau$ and hence $\alpha_0$ we calculate $f'$ which is equal to

$$f'(\eta) = \frac{\gamma}{2} \left( \pi_\ell \nu_\ell(\eta) + \begin{bmatrix} \alpha^T & b_\ell \end{bmatrix} A(\eta) \begin{bmatrix} \alpha \\ b_\ell \end{bmatrix} - 2\pi_\ell \nu_\ell(\eta) c_\ell^T \begin{bmatrix} \alpha \\ b_\ell \end{bmatrix} \right) + \frac{\gamma}{2} \sigma^2 \left( \pi^T \nu(\eta) \right) \alpha_0^2 + \frac{\gamma \eta}{2} \sigma^2 \alpha_0^2 (\pi^T \nu'(\eta))$$

$$+ \frac{\gamma \eta}{2} \left( \pi_\ell \nu_\ell'(\eta) + \begin{bmatrix} \alpha^T & b_\ell \end{bmatrix} A'(\eta) \begin{bmatrix} \alpha \\ b_\ell \end{bmatrix} - 2\pi_\ell \nu_\ell'(\eta) c_\ell^T \begin{bmatrix} \alpha \\ b_\ell \end{bmatrix} \right),$$

where

$$\nu'(\eta) = -\frac{1}{\gamma} \begin{bmatrix} \frac{\omega_1^2}{(\omega_1^2 + \eta)^2} \\ \frac{\omega_2^2}{(\omega_2^2 + \eta)^2} \\ \dots \\ \frac{\omega_k^2}{(\omega_k^2 + \eta)^2} \end{bmatrix}$$

$$A'(\eta) := \begin{bmatrix} \sigma^2 \left(\pi^T \nu'(\eta)\right) I + \Sigma V^T \mathrm{diag}\left(\pi\right) \mathrm{diag}\left(\nu'(\eta)\right) V \Sigma & \Sigma V^T \mathrm{diag}\left(\nu'(\eta)\right) \pi \\ \pi^T \mathrm{diag}\left(\nu'(\eta)\right) V \Sigma & \pi^T \nu'(\eta) \end{bmatrix}$$

$$c_\ell := \begin{bmatrix} \Sigma V^T e_\ell \\ 1 \end{bmatrix}.$$

Now note that at the optimal point we have

$$f'(\eta) = \frac{\tau^2}{2} = \frac{\gamma}{2}\sigma^2\left(\boldsymbol{\pi}^T\boldsymbol{\nu}(\eta)\right)^2\alpha_0^2\,.$$

Thus from the above we can conclude that

$$\alpha_0^2 = -\frac{1}{\eta\sigma^2(\boldsymbol{\pi}^T\boldsymbol{\nu}'(\eta))}\left(\pi_\ell\boldsymbol{\nu}_\ell(\eta) + \begin{bmatrix}\boldsymbol{\alpha}^T & b_\ell\end{bmatrix}\boldsymbol{A}(\eta)\begin{bmatrix}\boldsymbol{\alpha}\\b_\ell\end{bmatrix} - 2\pi_\ell\boldsymbol{\nu}_\ell(\eta)\boldsymbol{c}_\ell^T\begin{bmatrix}\boldsymbol{\alpha}\\b_\ell\end{bmatrix}\right)$$

$$-\frac{1}{\sigma^2(\boldsymbol{\pi}^T\boldsymbol{\nu}'(\eta))}\left(\pi_\ell\boldsymbol{\nu}_\ell'(\eta) + \begin{bmatrix}\boldsymbol{\alpha}^T & b_\ell\end{bmatrix}\boldsymbol{A}'(\eta)\begin{bmatrix}\boldsymbol{\alpha}\\b_\ell\end{bmatrix} - 2\pi_\ell\boldsymbol{\nu}_\ell'(\eta)\boldsymbol{c}_\ell^T\begin{bmatrix}\boldsymbol{\alpha}\\b_\ell\end{bmatrix}\right)\,.$$

Thus the AO optimization problem reduces to

$$\min_{b_\ell}\ \min_{\boldsymbol{\alpha}}\ \frac{\eta\gamma}{2}\left(\pi_\ell\boldsymbol{\nu}_\ell + \begin{bmatrix}\boldsymbol{\alpha}^T & b_\ell\end{bmatrix}\boldsymbol{A}\begin{bmatrix}\boldsymbol{\alpha}\\b_\ell\end{bmatrix} - 2\pi_\ell\boldsymbol{\nu}_\ell\boldsymbol{c}_\ell^T\begin{bmatrix}\boldsymbol{\alpha}\\b_\ell\end{bmatrix}\right)\,,$$

where $\eta$ is the solution to

$$\sum_{\ell=1}^{k}\frac{\pi_\ell\omega_\ell^2}{\omega_\ell^2 + \eta} = \gamma\,,$$

and

$$\boldsymbol{A}(\eta) := \begin{bmatrix}\sigma^2\boldsymbol{I}_r + \boldsymbol{\Sigma}\boldsymbol{V}^T\mathrm{diag}\left(\boldsymbol{\pi}\odot\boldsymbol{\nu}\right)\boldsymbol{V}\boldsymbol{\Sigma} & \boldsymbol{\Sigma}\boldsymbol{V}^T\left(\boldsymbol{\pi}\odot\boldsymbol{\nu}\right)\\ \left(\boldsymbol{\pi}\odot\boldsymbol{\nu}\right)^T\boldsymbol{V}\boldsymbol{\Sigma} & 1\end{bmatrix}$$

$$\boldsymbol{c}_\ell := \begin{bmatrix}\boldsymbol{\Sigma}\boldsymbol{V}^T\boldsymbol{e}_\ell\\1\end{bmatrix}\,, \tag{J.5}$$

with

$$\boldsymbol{\nu} := \frac{1}{\gamma}\begin{bmatrix}\frac{\omega_1^2}{\omega_1^2+\eta}\\\frac{\omega_2^2}{\omega_2^2+\eta}\\\dots\\\frac{\omega_k^2}{\omega_k^2+\eta}\end{bmatrix}\,.$$

First, note that the matrix $\boldsymbol{A}$ is positive definite. This can be checked by computing the Schur complement of $\boldsymbol{A}$:

$$\boldsymbol{\Delta} := \sigma^2\boldsymbol{I}_r + \boldsymbol{\Sigma}\boldsymbol{V}^T\boldsymbol{P}\boldsymbol{V}\boldsymbol{\Sigma} := \sigma^2\boldsymbol{I}_r + \boldsymbol{\Sigma}\boldsymbol{V}^T\left(\mathrm{diag}(\boldsymbol{\pi}\odot\boldsymbol{\nu}) - (\boldsymbol{\pi}\odot\boldsymbol{\nu})(\boldsymbol{\pi}\odot\boldsymbol{\nu})^T\right)\boldsymbol{V}\boldsymbol{\Sigma} > \boldsymbol{0}_{r\times r}. \tag{J.6}$$

Positive definiteness above holds because $\boldsymbol{P} := \left(\mathrm{diag}(\boldsymbol{\pi}\odot\boldsymbol{\nu}) - (\boldsymbol{\pi}\odot\boldsymbol{\nu})(\boldsymbol{\pi}\odot\boldsymbol{\nu})^T\right) \succeq \boldsymbol{0}_{k\times k}$. Thus the objective is a strictly convex quadratic and is jointly convex in its arguments. We proceed by minimizing the objective over $(\boldsymbol{\alpha}, b_\ell)$ which is equal to

$$\begin{bmatrix}\widehat{\boldsymbol{\alpha}}\\\widehat{b_\ell}\end{bmatrix} = \pi_\ell\boldsymbol{\nu}_\ell\boldsymbol{A}^{-1}\boldsymbol{c}_\ell = \pi_\ell\boldsymbol{\nu}_\ell\begin{bmatrix}-\boldsymbol{\Delta}^{-1}\boldsymbol{\Sigma}\boldsymbol{V}^T(\boldsymbol{\pi}\odot\boldsymbol{\nu} - \boldsymbol{e}_\ell)\\1 + (\boldsymbol{\pi}\odot\boldsymbol{\nu})^T\boldsymbol{V}\boldsymbol{\Sigma}\boldsymbol{\Delta}^{-1}\boldsymbol{\Sigma}\boldsymbol{V}^T(\boldsymbol{\pi}\odot\boldsymbol{\nu} - \boldsymbol{e}_\ell)\end{bmatrix}\,. \tag{J.7}$$

Thus, the minimum value attained is

$$-\pi_\ell^2\boldsymbol{\nu}_\ell^2\begin{bmatrix}-(\boldsymbol{\pi}\odot\boldsymbol{\nu} - \boldsymbol{e}_\ell)^T\boldsymbol{V}\boldsymbol{\Sigma} & 1\end{bmatrix}\begin{bmatrix}\boldsymbol{\Delta}^{-1} & \boldsymbol{0}\\\boldsymbol{0}^T & 1\end{bmatrix}\begin{bmatrix}-\boldsymbol{\Sigma}\boldsymbol{V}^T(\boldsymbol{\pi}\odot\boldsymbol{\nu} - \boldsymbol{e}_\ell)\\1\end{bmatrix} \tag{J.8}$$

$$= -\pi_\ell^2\boldsymbol{\nu}_\ell^2\left(1 + (\boldsymbol{\pi}\odot\boldsymbol{\nu} - \boldsymbol{e}_\ell)^T\boldsymbol{V}\boldsymbol{\Sigma}\boldsymbol{\Delta}^{-1}\boldsymbol{\Sigma}\boldsymbol{V}^T(\boldsymbol{\pi}\odot\boldsymbol{\nu} - \boldsymbol{e}_\ell)\right)\,.$$

Thus the objective reduces to

$$\frac{\eta\gamma}{2}\left(\pi_\ell\boldsymbol{\nu}_\ell(1 - \pi_\ell\boldsymbol{\nu}_\ell) - \pi_\ell^2\boldsymbol{\nu}_\ell^2(\boldsymbol{\pi}\odot\boldsymbol{\nu} - \boldsymbol{e}_\ell)^T\boldsymbol{V}\boldsymbol{\Sigma}\boldsymbol{\Delta}^{-1}\boldsymbol{\Sigma}\boldsymbol{V}^T(\boldsymbol{\pi}\odot\boldsymbol{\nu} - \boldsymbol{e}_\ell)\right)\,. \tag{J.9}$$

Therefore,

$$\alpha_0^2 = -\frac{1}{\eta\sigma^2(\boldsymbol{\pi}^T\boldsymbol{\nu}'(\eta))}\left(\pi_\ell\boldsymbol{\nu}_\ell(1 - \pi_\ell\boldsymbol{\nu}_\ell) - \pi_\ell^2\boldsymbol{\nu}_\ell^2(\boldsymbol{\pi}\odot\boldsymbol{\nu} - \boldsymbol{e}_\ell)^T\boldsymbol{V}\boldsymbol{\Sigma}\boldsymbol{\Delta}^{-1}\boldsymbol{\Sigma}\boldsymbol{V}^T(\boldsymbol{\pi}\odot\boldsymbol{\nu} - \boldsymbol{e}_\ell)\right)$$

$$-\frac{1}{\sigma^2(\boldsymbol{\pi}^T\boldsymbol{\nu}'(\eta))}\left(\pi_\ell\boldsymbol{\nu}_\ell'(\eta) + \begin{bmatrix}\boldsymbol{\alpha}^T & b_\ell\end{bmatrix}\boldsymbol{A}'(\eta)\begin{bmatrix}\boldsymbol{\alpha}\\b_\ell\end{bmatrix} - 2\pi_\ell\boldsymbol{\nu}_\ell'(\eta)\boldsymbol{c}_\ell^T\begin{bmatrix}\boldsymbol{\alpha}\\b_\ell\end{bmatrix}\right)\,,$$

where

$$\begin{bmatrix} \widehat{\boldsymbol{\alpha}} \\ \widehat{b}_\ell \end{bmatrix} = \pi_\ell \nu_\ell \begin{bmatrix} -\boldsymbol{\Delta}^{-1}\boldsymbol{\Sigma}\boldsymbol{V}^T(\boldsymbol{\pi}\odot\boldsymbol{\nu}-\boldsymbol{e}_\ell) \\ 1+(\boldsymbol{\pi}\odot\boldsymbol{\nu})^T\boldsymbol{V}\boldsymbol{\Sigma}\boldsymbol{\Delta}^{-1}\boldsymbol{\Sigma}\boldsymbol{V}^T(\boldsymbol{\pi}\odot\boldsymbol{\nu}-\boldsymbol{e}_\ell) \end{bmatrix} \tag{J.10}$$

and

$$\boldsymbol{\nu}'(\eta) = -\frac{1}{\gamma}\begin{bmatrix} \frac{\omega_1^2}{(\omega_1^2+\eta)^2} \\ \frac{\omega_2^2}{(\omega_2^2+\eta)^2} \\ \cdots \\ \frac{\omega_k^2}{(\omega_k^2+\eta)^2} \end{bmatrix}$$

$$\boldsymbol{A}'(\eta) := \begin{bmatrix} \sigma^2\left(\boldsymbol{\pi}^T\boldsymbol{\nu}'(\eta)\right)\boldsymbol{I} + \boldsymbol{\Sigma}\boldsymbol{V}^T\mathrm{diag}\left(\boldsymbol{\pi}\right)\mathrm{diag}\left(\boldsymbol{\nu}'(\eta)\right)\boldsymbol{V}\boldsymbol{\Sigma} & \boldsymbol{\Sigma}\boldsymbol{V}^T\mathrm{diag}\left(\boldsymbol{\nu}'(\eta)\right)\boldsymbol{\pi} \\ \boldsymbol{\pi}^T\mathrm{diag}\left(\boldsymbol{\nu}'(\eta)\right)\boldsymbol{V}\boldsymbol{\Sigma} & \boldsymbol{\pi}^T\boldsymbol{\nu}'(\eta) \end{bmatrix}.$$

To continue note that

$$\pi_\ell\boldsymbol{\nu}'_\ell(\eta) + \begin{bmatrix} \boldsymbol{\alpha}^T & b_\ell \end{bmatrix}\boldsymbol{A}'(\eta)\begin{bmatrix} \boldsymbol{\alpha} \\ b_\ell \end{bmatrix} - 2\pi_\ell\boldsymbol{\nu}'_\ell(\eta)\boldsymbol{c}_\ell^T\begin{bmatrix} \boldsymbol{\alpha} \\ b_\ell \end{bmatrix}$$
$$= \pi_\ell\boldsymbol{\nu}'_\ell(\eta) + \pi_\ell^2\nu_\ell^2\boldsymbol{c}_\ell^T\boldsymbol{A}^{-1}\boldsymbol{A}'(\eta)\boldsymbol{A}^{-1}\boldsymbol{c}_\ell - 2\pi_\ell^2\nu'_\ell\nu_\ell\boldsymbol{c}_\ell^T\boldsymbol{A}^{-1}\boldsymbol{c}_\ell.$$

Thus,

$$\begin{aligned}
\alpha_0^2 &= -\frac{1}{\eta\sigma^2(\boldsymbol{\pi}^T\boldsymbol{\nu}'(\eta))}\left(\pi_\ell\nu_\ell\left(1-\pi_\ell\nu_\ell\right) - \pi_\ell^2\nu_\ell^2\left(\boldsymbol{\pi}\odot\boldsymbol{\nu}-\boldsymbol{e}_\ell\right)^T\boldsymbol{V}\boldsymbol{\Sigma}\boldsymbol{\Delta}^{-1}\boldsymbol{\Sigma}\boldsymbol{V}^T\left(\boldsymbol{\pi}\odot\boldsymbol{\nu}-\boldsymbol{e}_\ell\right)\right) \\
&\quad -\frac{1}{\sigma^2(\boldsymbol{\pi}^T\boldsymbol{\nu}'(\eta))}\left(\pi_\ell\boldsymbol{\nu}'_\ell(\eta) + \pi_\ell^2\nu_\ell^2\boldsymbol{c}_\ell^T\boldsymbol{A}^{-1}\boldsymbol{A}'(\eta)\boldsymbol{A}^{-1}\boldsymbol{c}_\ell - 2\pi_\ell^2\nu'_\ell\nu_\ell\boldsymbol{c}_\ell^T\boldsymbol{A}^{-1}\boldsymbol{c}_\ell\right) \\
&= \frac{\gamma}{\eta\sigma^2\left(\sum_{\ell=1}^k\frac{\pi_\ell\omega_\ell^2}{(\omega_\ell^2+\eta)^2}\right)}\left(\pi_\ell\nu_\ell\left(1-\pi_\ell\nu_\ell\right) - \pi_\ell^2\nu_\ell^2\left(\boldsymbol{\pi}\odot\boldsymbol{\nu}-\boldsymbol{e}_\ell\right)^T\boldsymbol{V}\boldsymbol{\Sigma}\boldsymbol{\Delta}^{-1}\boldsymbol{\Sigma}\boldsymbol{V}^T\left(\boldsymbol{\pi}\odot\boldsymbol{\nu}-\boldsymbol{e}_\ell\right)\right) \\
&\quad +\frac{\gamma}{\sigma^2\left(\sum_{\ell=1}^k\frac{\pi_\ell\omega_\ell^2}{(\omega_\ell^2+\eta)^2}\right)}\left(\pi_\ell\boldsymbol{\nu}'_\ell(\eta) + \pi_\ell^2\nu_\ell^2\boldsymbol{c}_\ell^T\boldsymbol{A}^{-1}\boldsymbol{A}'(\eta)\boldsymbol{A}^{-1}\boldsymbol{c}_\ell - 2\pi_\ell^2\nu'_\ell\nu_\ell\boldsymbol{c}_\ell^T\boldsymbol{A}^{-1}\boldsymbol{c}_\ell\right) \\
&:= \frac{\zeta}{\sigma^2}\left(\pi_\ell\nu_\ell\left(1-\pi_\ell\nu_\ell\right) - \pi_\ell^2\nu_\ell^2\left(\boldsymbol{\pi}\odot\boldsymbol{\nu}-\boldsymbol{e}_\ell\right)^T\boldsymbol{V}\boldsymbol{\Sigma}\boldsymbol{\Delta}^{-1}\boldsymbol{\Sigma}\boldsymbol{V}^T\left(\boldsymbol{\pi}\odot\boldsymbol{\nu}-\boldsymbol{e}_\ell\right)\right) \\
&\quad +\frac{\zeta\eta}{\sigma^2}\left(\pi_\ell\boldsymbol{\nu}'_\ell(\eta) + \pi_\ell^2\nu_\ell^2\boldsymbol{c}_\ell^T\boldsymbol{A}^{-1}\boldsymbol{A}'(\eta)\boldsymbol{A}^{-1}\boldsymbol{c}_\ell - 2\pi_\ell^2\nu'_\ell\nu_\ell\boldsymbol{c}_\ell^T\boldsymbol{A}^{-1}\boldsymbol{c}_\ell\right)
\end{aligned}$$

where $\zeta := \frac{\gamma}{\eta\left(\sum_{\ell=1}^k\frac{\pi_\ell\omega_\ell^2}{(\omega_\ell^2+\eta)^2}\right)}$.

**Asymptotic predictions.** First, from (J.7) the bias term converges as follows:

$$\widehat{b}_\ell \xrightarrow{P} \pi_\ell\nu_\ell\left(1+(\boldsymbol{\pi}\odot\boldsymbol{\nu})^T\boldsymbol{V}\boldsymbol{\Sigma}\boldsymbol{\Delta}^{-1}\boldsymbol{\Sigma}\boldsymbol{V}^T(\boldsymbol{\pi}\odot\boldsymbol{\nu}-\boldsymbol{e}_\ell)\right).$$

Thus,

$$\widehat{\boldsymbol{b}} \xrightarrow{P} \left(\boldsymbol{I}_k - \boldsymbol{P}\boldsymbol{V}\boldsymbol{\Sigma}\boldsymbol{\Delta}^{-1}\boldsymbol{\Sigma}\boldsymbol{V}^T\right)(\boldsymbol{\pi}\odot\boldsymbol{\nu}).$$

Recall that $\boldsymbol{\alpha} = \boldsymbol{U}^T\boldsymbol{w}_\ell$. Thus, the correlations $\langle\boldsymbol{\mu}_i,\boldsymbol{w}_\ell\rangle$, $i\in[k]$ converge as follows:

$$\boldsymbol{M}^T\boldsymbol{w}_\ell = \boldsymbol{V}\boldsymbol{\Sigma}\boldsymbol{U}^T\boldsymbol{w}_\ell \xrightarrow{P} \boldsymbol{V}\boldsymbol{\Sigma}\widehat{\boldsymbol{\alpha}} = -\pi_\ell\nu_\ell\boldsymbol{V}\boldsymbol{\Sigma}\boldsymbol{\Delta}^{-1}\boldsymbol{\Sigma}\boldsymbol{V}^T\left(\boldsymbol{\pi}\odot\boldsymbol{\nu}-\boldsymbol{e}_\ell\right). \tag{J.11}$$

Here, convergence applies element-wise to the entries of the involved random vectors. Moreover, from the analysis above we can predict the limit of the norm $\|\boldsymbol{w}_\ell\|_{\ell_2}$. For this, note that $\|\boldsymbol{w}_\ell\|_{\ell_2}^2 = \widehat{\alpha}_0^2 + \widehat{\boldsymbol{\alpha}}^T\widehat{\boldsymbol{\alpha}}$. Thus,

$$\begin{aligned}
\|\boldsymbol{w}_\ell\|_{\ell_2}^2 \xrightarrow{P} &\frac{\zeta}{\sigma^2}\pi_\ell\nu_\ell(1-\pi_\ell\nu_\ell) + \pi_\ell^2\nu_\ell^2\left(\boldsymbol{\pi}\odot\boldsymbol{\nu}-\boldsymbol{e}_\ell\right)^T\boldsymbol{V}\boldsymbol{\Sigma}\boldsymbol{\Delta}^{-1}\left(\boldsymbol{\Delta}^{-1}-\frac{\zeta}{\sigma^2}\boldsymbol{I}_r\right)\boldsymbol{\Sigma}\boldsymbol{V}^T\left(\boldsymbol{\pi}\odot\boldsymbol{\nu}-\boldsymbol{e}_\ell\right) \\
&+\frac{\eta\zeta}{\sigma^2}\left(\pi_\ell\boldsymbol{\nu}'_\ell(\eta) + \pi_\ell^2\nu_\ell^2\boldsymbol{c}_\ell^T\boldsymbol{A}^{-1}\boldsymbol{A}'\boldsymbol{A}^{-1}\boldsymbol{c}_\ell - 2\pi_\ell^2\nu'_\ell\nu_\ell\boldsymbol{c}_\ell^T\boldsymbol{A}^{-1}\boldsymbol{c}_\ell\right).
\end{aligned} \tag{J.12}$$

## J.2 Computing $\Sigma_{w,w}$

In the previous section we used the CGMT to predict the bias $\widehat{b}_\ell$, the correlations $\langle \boldsymbol{\mu}_i, \widehat{\boldsymbol{w}}_\ell \rangle$, $i[k]$ and the norm $\|\widehat{\boldsymbol{w}}_\ell\|_{\ell_2}$ for all $\ell \in [k]$ members of the multi-output classifier. Here, we show how to compute the limits of the cross-correlations $\langle \widehat{\boldsymbol{w}}_\ell, \widehat{\boldsymbol{w}}_j \rangle, \ell \neq j \in [k]$.

**Lemma J.1** *For $\ell \neq j \in [k]$ let $\widehat{\boldsymbol{w}}_\ell$ $\widehat{\boldsymbol{w}}_j$ be solutions to the least-squares minimization (**??**), i.e.,*

$$(\widehat{\boldsymbol{w}}_\ell, \widehat{\boldsymbol{w}}_j, \widehat{b}_\ell, \widehat{b}_j)$$
$$= \arg \min_{\boldsymbol{w}_\ell, \boldsymbol{w}_j, b_\ell, b_j} \left\{ \frac{1}{2n} \left\| \boldsymbol{D} \left( \boldsymbol{Y}_\ell - \boldsymbol{X}^T \boldsymbol{w}_\ell - b_\ell \boldsymbol{1}_n \right) \right\|_{\ell_2}^2 + \frac{1}{2n} \left\| \boldsymbol{D} \left( \boldsymbol{Y}_j - \boldsymbol{X}^T \boldsymbol{w}_j - b_j \boldsymbol{1}_n \right) \right\|_{\ell_2}^2 \right\}.$$

*Denote $\widehat{\boldsymbol{w}}_{\ell,j} := \widehat{\boldsymbol{w}}_\ell + \widehat{\boldsymbol{w}}_j$ and $\widehat{b}_{\ell,j} := \widehat{b}_\ell + \widehat{b}_j$. Then, $(\widehat{\boldsymbol{w}}_{\ell,j}, \widehat{b}_{\ell,j})$ is a minimizer in the following least-squares problem:*

$$(\widehat{\boldsymbol{w}}_{\ell,j}, \widehat{b}_{\ell,j}) = \arg \min_{\boldsymbol{w},b} \frac{1}{2n} \left\| \boldsymbol{D} \left( \boldsymbol{Y}_\ell + \boldsymbol{Y}_j - \boldsymbol{X}^T \boldsymbol{w} - b \boldsymbol{1}_n \right) \right\|_{\ell_2}^2 \tag{J.13}$$

**Proof** Clearly the minimization in (J.13) is convex. Thus, it suffices to prove that $\widehat{\boldsymbol{w}}_\ell + \widehat{\boldsymbol{w}}_j$ satisfies the KKT conditions. First, by optimality of $\widehat{\boldsymbol{w}}_\ell$, we have that

$$\boldsymbol{X} \boldsymbol{D}^2 \left( \boldsymbol{Y}_\ell - \boldsymbol{X}^T \widehat{\boldsymbol{w}}_\ell - \widehat{b}_\ell \boldsymbol{1}_n \right) = 0$$

Similarly, for $\widehat{\boldsymbol{w}}_j$:

$$\boldsymbol{X} \boldsymbol{D}^2 \left( \boldsymbol{Y}_j - \boldsymbol{X}^T \widehat{\boldsymbol{w}}_j - \widehat{b}_j \boldsymbol{1}_n \right) = 0.$$

Adding the equations on the above displays we find that

$$\boldsymbol{X} \boldsymbol{D}^2 \left( \boldsymbol{Y}_\ell + \boldsymbol{Y}_j - \boldsymbol{X}^T (\widehat{\boldsymbol{w}}_j + \widehat{\boldsymbol{w}}_\ell) - (\widehat{b}_j + \widehat{b}_\ell) \boldsymbol{1}_n \right) = 0.$$

Recognize that this coincides with the optimality condition for (J.13). Thus, the proof is complete. ∎

Thanks to Lemma J.1, we can use the CGMT to characterize the limiting behavior of $\|\widehat{\boldsymbol{w}}_\ell + \widehat{\boldsymbol{w}}_j\|_{\ell_2}$. Observe that this immediately gives the limit of $\langle \widehat{\boldsymbol{w}}_\ell, \widehat{\boldsymbol{w}}_j \rangle$ since

$$\langle \widehat{\boldsymbol{w}}_\ell, \widehat{\boldsymbol{w}}_j \rangle = \frac{\|\widehat{\boldsymbol{w}}_\ell + \widehat{\boldsymbol{w}}_j\|_{\ell_2}^2 - \|\widehat{\boldsymbol{w}}_\ell\|_{\ell_2}^2 - \|\widehat{\boldsymbol{w}}_j\|_{\ell_2}^2}{2}. \tag{J.14}$$

The analysis of (J.13) is very similar to that of (H.1). In particular we use the following decomposition

$$\boldsymbol{w}_{\ell,j} = \sum_{i=1}^{r} \beta_i \boldsymbol{u}_i + \beta_0 \boldsymbol{w}_{\ell,j}^\perp,$$

with $\|\boldsymbol{w}_{\ell,j}^\perp\|_{\ell_2} = 1$ and $\boldsymbol{U}^T \boldsymbol{w}_{\ell,j}^\perp = \boldsymbol{0}_r$. This allows us to arrive at

$$\min_{\boldsymbol{\beta}} \min_{\boldsymbol{b}_{\ell,j}} \max_{\alpha_0 \geq 0} \max_{\beta \geq 0} \min_{\tau \geq 0}$$

$$\frac{\gamma \beta}{2\tau} \left( \pi_\ell \boldsymbol{\nu}_\ell \left( \frac{\beta}{\tau} \right) + \pi_j \boldsymbol{\nu}_j \left( \frac{\beta}{\tau} \right) + \begin{bmatrix} \boldsymbol{\beta}^T & b_{\ell,j} \end{bmatrix} \boldsymbol{A} \left( \frac{\beta}{\tau} \right) \begin{bmatrix} \boldsymbol{\beta} \\ b_{\ell,j} \end{bmatrix} - 2 \left( \pi_\ell \boldsymbol{\nu}_\ell \left( \frac{\beta}{\tau} \right) \boldsymbol{c}_\ell + \pi_j \boldsymbol{\nu}_j \left( \frac{\beta}{\tau} \right) \boldsymbol{c}_j \right)^T \begin{bmatrix} \boldsymbol{\beta} \\ b_{\ell,j} \end{bmatrix} \right)$$

$$+ \frac{\gamma \beta}{2\tau} \sigma^2 \left( \boldsymbol{\pi}^T \boldsymbol{\nu} \left( \frac{\beta}{\tau} \right) \right) \beta_0^2 - \beta_0 \sigma \beta \sqrt{\gamma} + \frac{\beta \tau}{2},$$

where $\boldsymbol{A}(\eta)$ and $\boldsymbol{c}_\ell$ are as in (J.3) and $\boldsymbol{\nu}(\eta)$ is as in (J.4).

Setting the derivative of the above with respect to $\alpha_0$ to zero we arrive at

$$\frac{\gamma \beta}{\tau} \sigma^2 \left( \boldsymbol{\pi}^T \boldsymbol{\nu} \left( \frac{\beta}{\tau} \right) \right) \beta_0 - \sigma \beta \sqrt{\gamma} = 0 \quad \Rightarrow \quad \beta_0 = \frac{\tau}{\sigma \sqrt{\gamma} \left( \boldsymbol{\pi}^T \boldsymbol{\nu} \left( \frac{\beta}{\tau} \right) \right)}.$$

Note that the above objective has the form

$$g \left( \frac{\beta}{\tau} \right) - \beta_0 \sigma \beta \sqrt{\gamma} + \frac{\beta \tau}{2},$$

with

$$g(\eta) := \frac{\eta\gamma}{2}\left(\pi_\ell\boldsymbol{\nu}_\ell(\eta) + \pi_j\boldsymbol{\nu}_j(\eta) + \begin{bmatrix}\boldsymbol{\beta}^T & b_{\ell,j}\end{bmatrix}\boldsymbol{A}(\eta)\begin{bmatrix}\boldsymbol{\beta}\\b_{\ell,j}\end{bmatrix} - 2\left(\pi_\ell\boldsymbol{\nu}_\ell(\eta)\,\boldsymbol{c}_\ell + \pi_j\boldsymbol{\nu}_j(\eta)\,\boldsymbol{c}_j\right)^T\begin{bmatrix}\boldsymbol{\beta}\\b_{\ell,j}\end{bmatrix}\right)$$
$$+ \frac{\gamma\eta}{2}\sigma^2\left(\boldsymbol{\pi}^T\boldsymbol{\nu}(\eta)\right)\beta_0^2.$$

Thus, the derivatives with respect to $\beta$ and $\tau$ to zero we have

$$\frac{1}{\tau}g'\left(\frac{\beta}{\tau}\right) - \beta_0\sigma\sqrt{\gamma} + \frac{\tau}{2} = 0 \quad\Rightarrow\quad g'\left(\frac{\beta}{\tau}\right) - \beta_0\sigma\sqrt{\gamma}\tau + \frac{\tau^2}{2} = 0 \quad\Rightarrow\quad g'\left(\frac{\beta}{\tau}\right) = \tau^2\left(\frac{1}{\boldsymbol{\pi}^T\boldsymbol{\nu}\left(\frac{\beta}{\tau}\right)} - \frac{1}{2}\right)$$

and

$$-\frac{\beta}{\tau^2}g'\left(\frac{\beta}{\tau}\right) + \frac{\beta}{2} = 0 \quad\Rightarrow\quad \tau^2 = 2g'\left(\frac{\beta}{\tau}\right).$$

Combining the latter two we conclude that $\boldsymbol{\pi}^T\boldsymbol{\nu}\left(\frac{\beta}{\tau}\right) = 1$. Thus, $\eta = \frac{\beta}{\tau}$ is the solution to $\boldsymbol{\pi}^T\boldsymbol{\nu}(\eta) = 1$. To calculate $\tau$ and hence $\beta_0$ we calculate $g'$ which is equal to

$$g'(\eta) = \frac{\gamma}{2}\left(\pi_\ell\boldsymbol{\nu}_\ell(\eta) + \pi_j\boldsymbol{\nu}_j(\eta) + \begin{bmatrix}\boldsymbol{\beta}^T & b_{\ell,j}\end{bmatrix}\boldsymbol{A}(\eta)\begin{bmatrix}\boldsymbol{\beta}\\b_{\ell,j}\end{bmatrix} - 2\left(\pi_\ell\boldsymbol{\nu}_\ell(\eta)\,\boldsymbol{c}_\ell + \pi_j\boldsymbol{\nu}_j(\eta)\,\boldsymbol{c}_j\right)^T\begin{bmatrix}\boldsymbol{\beta}\\b_{\ell,j}\end{bmatrix}\right)$$
$$+ \frac{\gamma}{2}\sigma^2\left(\boldsymbol{\pi}^T\boldsymbol{\nu}(\eta)\right)\beta_0^2 + \frac{\gamma\eta}{2}\sigma^2\beta_0^2(\boldsymbol{\pi}^T\boldsymbol{\nu}'(\eta))$$
$$+ \frac{\gamma\eta}{2}\left(\pi_\ell\boldsymbol{\nu}_\ell'(\eta) + \pi_j\boldsymbol{\nu}_j'(\eta) + \begin{bmatrix}\boldsymbol{\beta}^T & b_{\ell,j}\end{bmatrix}\boldsymbol{A}'(\eta)\begin{bmatrix}\boldsymbol{\beta}\\b_{\ell,j}\end{bmatrix} - 2\left(\pi_\ell\boldsymbol{\nu}_\ell'(\eta)\,\boldsymbol{c}_\ell + \pi_j\boldsymbol{\nu}_j'(\eta)\,\boldsymbol{c}_j\right)^T\begin{bmatrix}\boldsymbol{\beta}\\b_{\ell,j}\end{bmatrix}\right).$$

Here, we have

$$\boldsymbol{\nu}'(\eta) = -\frac{1}{\gamma}\begin{bmatrix}\frac{\omega_1^2}{(\omega_1^2+\eta)^2}\\\frac{\omega_2^2}{(\omega_2^2+\eta)^2}\\\cdots\\\frac{\omega_k^2}{(\omega_k^2+\eta)^2}\end{bmatrix}$$

$$\boldsymbol{A}'(\eta) := \begin{bmatrix}\sigma^2\left(\boldsymbol{\pi}^T\boldsymbol{\nu}'(\eta)\right)\boldsymbol{I} + \boldsymbol{\Sigma}\boldsymbol{V}^T\mathrm{diag}\left(\boldsymbol{\pi}\right)\mathrm{diag}\left(\boldsymbol{\nu}'(\eta)\right)\boldsymbol{V}\boldsymbol{\Sigma} & \boldsymbol{\Sigma}\boldsymbol{V}^T\mathrm{diag}\left(\boldsymbol{\nu}'(\eta)\right)\boldsymbol{\pi}\\\boldsymbol{\pi}^T\mathrm{diag}\left(\boldsymbol{\nu}'(\eta)\right)\boldsymbol{V}\boldsymbol{\Sigma} & \boldsymbol{\pi}^T\boldsymbol{\nu}'(\eta)\end{bmatrix}$$

$$\boldsymbol{c}_\ell := \begin{bmatrix}\boldsymbol{\Sigma}\boldsymbol{V}^T\boldsymbol{e}_\ell\\1\end{bmatrix}.$$

Now note that at the optimal point we have

$$g'(\eta) = \frac{\tau^2}{2} = \frac{\gamma}{2}\sigma^2\left(\boldsymbol{\pi}^T\boldsymbol{\nu}(\eta)\right)\beta_0^2.$$

Thus from the above we can conclude that

$$\beta_0^2 = -\frac{1}{\eta\sigma^2(\boldsymbol{\pi}^T\boldsymbol{\nu}'(\eta))}\left(\pi_\ell\boldsymbol{\nu}_\ell(\eta) + \pi_j\boldsymbol{\nu}_j(\eta) + \begin{bmatrix}\boldsymbol{\beta}^T & b_{\ell,j}\end{bmatrix}\boldsymbol{A}(\eta)\begin{bmatrix}\boldsymbol{\beta}\\b_{\ell,j}\end{bmatrix} - 2\left(\pi_\ell\boldsymbol{\nu}_\ell(\eta)\,\boldsymbol{c}_\ell + \pi_j\boldsymbol{\nu}_j(\eta)\,\boldsymbol{c}_j\right)^T\begin{bmatrix}\boldsymbol{\beta}\\b_{\ell,j}\end{bmatrix}\right)$$
$$-\frac{1}{\sigma^2(\boldsymbol{\pi}^T\boldsymbol{\nu}'(\eta))}\left(\pi_\ell\boldsymbol{\nu}_\ell'(\eta) + \pi_j\boldsymbol{\nu}_j'(\eta) + \begin{bmatrix}\boldsymbol{\beta}^T & b_{\ell,j}\end{bmatrix}\boldsymbol{A}'(\eta)\begin{bmatrix}\boldsymbol{\beta}\\b_{\ell,j}\end{bmatrix} - 2\left(\pi_\ell\boldsymbol{\nu}_\ell'(\eta)\,\boldsymbol{c}_\ell + \pi_j\boldsymbol{\nu}_j'(\eta)\,\boldsymbol{c}_j\right)^T\begin{bmatrix}\boldsymbol{\beta}\\b_{\ell,j}\end{bmatrix}\right).$$

Thus, the AO problem reduces to

$$\min_{b_{\ell,j}}\ \min_{\boldsymbol{\beta}}\ \frac{\eta\gamma}{2}\left(\pi_\ell\boldsymbol{\nu}_\ell + \pi_j\boldsymbol{\nu}_j + \begin{bmatrix}\boldsymbol{\beta}^T & b_{\ell,j}\end{bmatrix}\boldsymbol{A}\begin{bmatrix}\boldsymbol{\beta}\\b_{\ell,j}\end{bmatrix} - 2\left(\pi_\ell\boldsymbol{\nu}_\ell\boldsymbol{c}_\ell + \pi_j\boldsymbol{\nu}_j\boldsymbol{c}_j\right)^T\begin{bmatrix}\boldsymbol{\beta}\\b_{\ell,j}\end{bmatrix}\right),$$

where $\eta$ is the solution to

$$\sum_{\ell=1}^{k}\frac{\pi_\ell\omega_\ell^2}{\omega_\ell^2+\eta} = \gamma.$$

Thus, similar to (J.10) we can compute the minimizer of the deterministic

$$\begin{bmatrix} \widehat{\boldsymbol{\beta}} \\ \widehat{b}_{\ell,j} \end{bmatrix} = \begin{bmatrix} -\boldsymbol{\Delta}^{-1}\boldsymbol{\Sigma}\boldsymbol{V}^T\left(\pi_\ell\nu_\ell(\boldsymbol{\pi}\odot\boldsymbol{\nu}-\boldsymbol{e}_\ell)+\pi_j\nu_j(\boldsymbol{\pi}\odot\boldsymbol{\nu}-\boldsymbol{e}_j)\right) \\ \pi_\ell\nu_\ell+\pi_j\nu_j+(\boldsymbol{\pi}\odot\boldsymbol{\nu})^T\boldsymbol{V}\boldsymbol{\Sigma}\boldsymbol{\Delta}^{-1}\boldsymbol{\Sigma}\boldsymbol{V}^T\left(\pi_\ell\nu_\ell(\boldsymbol{\pi}\odot\boldsymbol{\nu}-\boldsymbol{e}_\ell)+\pi_j\nu_j(\boldsymbol{\pi}\odot\boldsymbol{\nu}-\boldsymbol{e}_j)\right) \end{bmatrix}$$
$$=\boldsymbol{A}^{-1}\left(\pi_\ell\boldsymbol{\nu}_\ell\boldsymbol{c}_\ell+\pi_j\boldsymbol{\nu}_j\boldsymbol{c}_j\right) \tag{J.15}$$

and

$$\boldsymbol{\nu}'(\eta)=-\frac{1}{\gamma}\begin{bmatrix} \frac{\omega_1^2}{(\omega_1^2+\eta)^2} \\ \frac{\omega_2^2}{(\omega_2^2+\eta)^2} \\ \ldots \\ \frac{\omega_k^2}{(\omega_k^2+\eta)^2} \end{bmatrix}$$

$$\boldsymbol{A}'(\eta):=\begin{bmatrix} \sigma^2\left(\boldsymbol{\pi}^T\boldsymbol{\nu}'(\eta)\right)\boldsymbol{I}+\boldsymbol{\Sigma}\boldsymbol{V}^T\operatorname{diag}(\boldsymbol{\pi})\operatorname{diag}(\boldsymbol{\nu}'(\eta))\boldsymbol{V}\boldsymbol{\Sigma} & \boldsymbol{\Sigma}\boldsymbol{V}^T\operatorname{diag}(\boldsymbol{\nu}'(\eta))\boldsymbol{\pi} \\ \boldsymbol{\pi}^T\operatorname{diag}(\boldsymbol{\nu}'(\eta))\boldsymbol{V}\boldsymbol{\Sigma} & \boldsymbol{\pi}^T\boldsymbol{\nu}'(\eta) \end{bmatrix}.$$

To continue note that

$$\pi_\ell\boldsymbol{\nu}'_\ell(\eta)+\pi_j\boldsymbol{\nu}'_j(\eta)+\begin{bmatrix}\boldsymbol{\beta}^T & b_{\ell,j}\end{bmatrix}\boldsymbol{A}'(\eta)\begin{bmatrix}\boldsymbol{\beta} \\ b_{\ell,j}\end{bmatrix}-2\left(\pi_\ell\boldsymbol{\nu}'_\ell(\eta)\boldsymbol{c}_\ell+\pi_j\boldsymbol{\nu}'_j(\eta)\boldsymbol{c}_j\right)^T\begin{bmatrix}\boldsymbol{\beta} \\ b_{\ell,j}\end{bmatrix}$$

$$=\pi_\ell\boldsymbol{\nu}'_\ell(\eta)+\pi_j\boldsymbol{\nu}'_j(\eta)+\left(\pi_\ell\boldsymbol{\nu}'_\ell(\eta)\boldsymbol{c}_\ell+\pi_j\boldsymbol{\nu}'_j(\eta)\boldsymbol{c}_j\right)^T\boldsymbol{A}^{-1}\boldsymbol{A}'(\eta)\boldsymbol{A}^{-1}\left(\pi_\ell\boldsymbol{\nu}'_\ell(\eta)\boldsymbol{c}_\ell+\pi_j\boldsymbol{\nu}'_j(\eta)\boldsymbol{c}_j\right)$$

$$-2\left(\pi_\ell\boldsymbol{\nu}'_\ell(\eta)\boldsymbol{c}_\ell+\pi_j\boldsymbol{\nu}'_j(\eta)\boldsymbol{c}_j\right)^T\boldsymbol{A}^{-1}\left(\pi_\ell\boldsymbol{\nu}'_\ell(\eta)\boldsymbol{c}_\ell+\pi_j\boldsymbol{\nu}'_j(\eta)\boldsymbol{c}_j\right).$$

Thus,

$$\beta_0^2=-\frac{1}{\eta\sigma^2(\boldsymbol{\pi}^T\boldsymbol{\nu}'(\eta))}\left(\pi_\ell\boldsymbol{\nu}_\ell(\eta)+\pi_j\boldsymbol{\nu}_j(\eta)+\begin{bmatrix}\boldsymbol{\beta}^T & b_{\ell,j}\end{bmatrix}\boldsymbol{A}(\eta)\begin{bmatrix}\boldsymbol{\beta} \\ b_{\ell,j}\end{bmatrix}-2\left(\pi_\ell\boldsymbol{\nu}_\ell(\eta)\boldsymbol{c}_\ell+\pi_j\boldsymbol{\nu}_j(\eta)\boldsymbol{c}_j\right)^T\begin{bmatrix}\boldsymbol{\beta} \\ b_{\ell,j}\end{bmatrix}\right)$$

$$-\frac{1}{\sigma^2(\boldsymbol{\pi}^T\boldsymbol{\nu}'(\eta))}\left(\pi_\ell\boldsymbol{\nu}'_\ell(\eta)+\pi_j\boldsymbol{\nu}'_j(\eta)+\begin{bmatrix}\boldsymbol{\beta}^T & b_{\ell,j}\end{bmatrix}\boldsymbol{A}'(\eta)\begin{bmatrix}\boldsymbol{\beta} \\ b_{\ell,j}\end{bmatrix}-2\left(\pi_\ell\boldsymbol{\nu}'_\ell(\eta)\boldsymbol{c}_\ell+\pi_j\boldsymbol{\nu}'_j(\eta)\boldsymbol{c}_j\right)^T\begin{bmatrix}\boldsymbol{\beta} \\ b_{\ell,j}\end{bmatrix}\right)$$

$$=\frac{\zeta}{\sigma^2}\Bigg(\pi_\ell\nu_\ell+\pi_j\nu_j-\left(\pi_\ell\nu_\ell+\pi_j\nu_j\right)^2$$

$$-\left(\pi_\ell\nu_\ell\left(\boldsymbol{\pi}\odot\boldsymbol{\nu}-\boldsymbol{e}_\ell\right)+\pi_j\nu_j\left(\boldsymbol{\pi}\odot\boldsymbol{\nu}-\boldsymbol{e}_j\right)\right)^T\boldsymbol{V}\boldsymbol{\Sigma}\boldsymbol{\Delta}^{-1}\boldsymbol{\Sigma}\boldsymbol{V}^T\left(\pi_\ell\nu_\ell\left(\boldsymbol{\pi}\odot\boldsymbol{\nu}-\boldsymbol{e}_\ell\right)+\pi_j\nu_j\left(\boldsymbol{\pi}\odot\boldsymbol{\nu}-\boldsymbol{e}_j\right)\right)\Bigg)$$

$$+\frac{\zeta\eta}{\sigma^2}\Bigg(\pi_\ell\boldsymbol{\nu}'_\ell(\eta)+\pi_j\boldsymbol{\nu}'_j(\eta)+\left(\pi_\ell\boldsymbol{\nu}_\ell(\eta)\boldsymbol{c}_\ell+\pi_j\boldsymbol{\nu}_j(\eta)\boldsymbol{c}_j\right)^T\boldsymbol{A}^{-1}\boldsymbol{A}'(\eta)\boldsymbol{A}^{-1}\left(\pi_\ell\boldsymbol{\nu}_\ell(\eta)\boldsymbol{c}_\ell+\pi_j\boldsymbol{\nu}_j(\eta)\boldsymbol{c}_j\right)$$

$$-2\left(\pi_\ell\boldsymbol{\nu}'_\ell(\eta)\boldsymbol{c}_\ell+\pi_j\boldsymbol{\nu}'_j(\eta)\boldsymbol{c}_j\right)^T\boldsymbol{A}^{-1}\left(\pi_\ell\boldsymbol{\nu}_\ell(\eta)\boldsymbol{c}_\ell+\pi_j\boldsymbol{\nu}_j(\eta)\boldsymbol{c}_j\right)\Bigg),$$

where $\zeta:=\frac{\gamma}{\eta\left(\Sigma_{\ell=1}^k\frac{\pi_\ell\omega_\ell^2}{(\omega_\ell^2+\eta)^2}\right)}$. From the CGMT, we have that $\|\widehat{\boldsymbol{w}}_\ell+\widehat{\boldsymbol{w}}_j\|_{\ell_2}^2\xrightarrow{P}\widehat{\beta}_0^2+\|\boldsymbol{\beta}\|_{\ell_2}^2$. Combining this with the calculations above, we conclude that

$$\|\widehat{\boldsymbol{w}}_\ell+\widehat{\boldsymbol{w}}_j\|_{\ell_2}^2\xrightarrow{P}\frac{\zeta}{\sigma^2}\left(\pi_\ell\nu_\ell+\pi_j\nu_j\right)\left(1-\pi_\ell\nu_\ell-\pi_j\nu_j\right)$$

$$+\left(\pi_\ell\nu_\ell\left(\boldsymbol{\pi}\odot\boldsymbol{\nu}-\boldsymbol{e}_\ell\right)+\pi_j\nu_j\left(\boldsymbol{\pi}\odot\boldsymbol{\nu}-\boldsymbol{e}_j\right)\right)^T\boldsymbol{V}\boldsymbol{\Sigma}\boldsymbol{\Delta}^{-1}\left(\boldsymbol{\Delta}^{-1}-\frac{\eta}{\sigma^2}\boldsymbol{I}_r\right)\boldsymbol{\Sigma}\boldsymbol{V}^T\left(\pi_\ell\nu_\ell\left(\boldsymbol{\pi}\odot\boldsymbol{\nu}-\boldsymbol{e}_\ell\right)+\pi_j\nu_j\left(\boldsymbol{\pi}\odot\boldsymbol{\nu}-\boldsymbol{e}_j\right)\right)$$

$$+\frac{\zeta\eta}{\sigma^2}\Bigg(\pi_\ell\boldsymbol{\nu}'_\ell(\eta)+\pi_j\boldsymbol{\nu}'_j(\eta)+\left(\pi_\ell\boldsymbol{\nu}_\ell(\eta)\boldsymbol{c}_\ell+\pi_j\boldsymbol{\nu}_j(\eta)\boldsymbol{c}_j\right)^T\boldsymbol{A}^{-1}\boldsymbol{A}'(\eta)\boldsymbol{A}^{-1}\left(\pi_\ell\boldsymbol{\nu}_\ell(\eta)\boldsymbol{c}_\ell+\pi_j\boldsymbol{\nu}_j(\eta)\boldsymbol{c}_j\right)$$

$$-2\left(\pi_\ell\boldsymbol{\nu}'_\ell(\eta)\boldsymbol{c}_\ell+\pi_j\boldsymbol{\nu}'_j(\eta)\boldsymbol{c}_j\right)^T\boldsymbol{A}^{-1}\left(\pi_\ell\boldsymbol{\nu}_\ell(\eta)\boldsymbol{c}_\ell+\pi_j\boldsymbol{\nu}_j(\eta)\boldsymbol{c}_j\right)\Bigg). \tag{J.16}$$

Finally, using (J.16) and (J.12) in (J.14) it follows that

$$\langle \boldsymbol{w}_\ell, \boldsymbol{w}_j \rangle \xrightarrow{P}$$

$$\pi_\ell \nu_\ell \pi_j \nu_j \left( -\frac{\zeta}{\sigma^2} + (\boldsymbol{\pi} \odot \boldsymbol{\nu} - \boldsymbol{e}_\ell)^T \boldsymbol{V} \boldsymbol{\Sigma} \boldsymbol{\Delta}^{-1} \left( \boldsymbol{\Delta}^{-1} - \frac{\zeta}{\sigma^2} \boldsymbol{I}_r \right) \boldsymbol{\Sigma} \boldsymbol{V}^T (\boldsymbol{\pi} \odot \boldsymbol{\nu} - \boldsymbol{e}_j) \right)$$

$$+ \frac{\zeta \eta}{\sigma^2} \pi_\ell \nu_\ell \pi_j \nu_j \boldsymbol{c}_j^T \boldsymbol{A}^{-1} \boldsymbol{A}' \boldsymbol{A}^{-1} \boldsymbol{c}_\ell - \frac{\zeta \eta}{\sigma^2} \pi_\ell \pi_j (\nu_\ell \nu_j' + \nu_\ell' \nu_j) \boldsymbol{c}_j^T \boldsymbol{A}^{-1} \boldsymbol{c}_\ell$$

$$= \pi_\ell \nu_\ell \pi_j \nu_j \left( -\frac{\zeta}{\sigma^2} + (\boldsymbol{\pi} \odot \boldsymbol{\nu} - \boldsymbol{e}_\ell)^T \boldsymbol{V} \boldsymbol{\Sigma} \boldsymbol{\Delta}^{-1} \left( \boldsymbol{\Delta}^{-1} - \frac{\zeta}{\sigma^2} \boldsymbol{I}_r \right) \boldsymbol{\Sigma} \boldsymbol{V}^T (\boldsymbol{\pi} \odot \boldsymbol{\nu} - \boldsymbol{e}_j) \right)$$

$$+ \frac{\zeta \eta}{\sigma^2} \pi_\ell \nu_\ell \pi_j \nu_j \left( \boldsymbol{e}_j^T \begin{bmatrix} \boldsymbol{\Sigma} \boldsymbol{V}^T \\ \boldsymbol{1}^T \end{bmatrix}^T \boldsymbol{A}^{-1} \boldsymbol{A}' \boldsymbol{A}^{-1} \begin{bmatrix} \boldsymbol{\Sigma} \boldsymbol{V}^T \\ \boldsymbol{1}^T \end{bmatrix} \boldsymbol{e}_\ell \right)$$

$$- \frac{\eta \zeta}{\sigma^2} \pi_\ell \pi_j (\nu_\ell \nu_j' + \nu_\ell' \nu_j) \left( \boldsymbol{e}_j^T \begin{bmatrix} \boldsymbol{\Sigma} \boldsymbol{V}^T \\ \boldsymbol{1}^T \end{bmatrix}^T \boldsymbol{A}^{-1} \begin{bmatrix} \boldsymbol{\Sigma} \boldsymbol{V}^T \\ \boldsymbol{1}^T \end{bmatrix} \boldsymbol{e}_\ell \right. . \tag{J.17}$$

Putting everything together we arrive at

$$\boldsymbol{\Sigma}_{\boldsymbol{w},\boldsymbol{w}} \xrightarrow{P} \frac{\zeta}{\sigma^2} \boldsymbol{P} + \boldsymbol{P} \boldsymbol{V} \boldsymbol{\Sigma} \boldsymbol{\Delta}^{-1} \left( \boldsymbol{\Delta}^{-1} - \frac{\zeta}{\sigma^2} \boldsymbol{I}_r \right) \boldsymbol{\Sigma} \boldsymbol{V}^T \boldsymbol{P} + \frac{\zeta \eta}{\sigma^2} \boldsymbol{Q},$$

where

$$\boldsymbol{Q} := \text{diag}(\boldsymbol{\pi} \odot \boldsymbol{\nu}') + \text{diag}(\boldsymbol{\pi} \odot \boldsymbol{\nu}) \begin{bmatrix} \boldsymbol{\Sigma} \boldsymbol{V}^T \\ \boldsymbol{1}^T \end{bmatrix}^T \left( \boldsymbol{A}^{-1} \boldsymbol{A}' \boldsymbol{A}^{-1} \right) \begin{bmatrix} \boldsymbol{\Sigma} \boldsymbol{V}^T \\ \boldsymbol{1}^T \end{bmatrix} \text{diag}(\boldsymbol{\pi} \odot \boldsymbol{\nu})$$

$$- \text{diag}(\boldsymbol{\pi} \odot \boldsymbol{\nu}') \begin{bmatrix} \boldsymbol{\Sigma} \boldsymbol{V}^T \\ \boldsymbol{1}^T \end{bmatrix}^T \boldsymbol{A}^{-1} \begin{bmatrix} \boldsymbol{\Sigma} \boldsymbol{V}^T \\ \boldsymbol{1}^T \end{bmatrix} \text{diag}(\boldsymbol{\pi} \odot \boldsymbol{\nu}) - \text{diag}(\boldsymbol{\pi} \odot \boldsymbol{\nu}) \begin{bmatrix} \boldsymbol{\Sigma} \boldsymbol{V}^T \\ \boldsymbol{1}^T \end{bmatrix}^T \boldsymbol{A}^{-1} \begin{bmatrix} \boldsymbol{\Sigma} \boldsymbol{V}^T \\ \boldsymbol{1}^T \end{bmatrix} \text{diag}(\boldsymbol{\pi} \odot \boldsymbol{\nu}')$$

and as mentioned earlier

$$\boldsymbol{A}' := \boldsymbol{A}'(\eta) := \begin{bmatrix} \sigma^2 \left( \boldsymbol{\pi}^T \boldsymbol{\nu}'(\eta) \right) \boldsymbol{I} + \boldsymbol{\Sigma} \boldsymbol{V}^T \text{diag}(\boldsymbol{\pi}) \text{diag}(\boldsymbol{\nu}'(\eta)) \boldsymbol{V} \boldsymbol{\Sigma} & \boldsymbol{\Sigma} \boldsymbol{V}^T \text{diag}(\boldsymbol{\nu}'(\eta)) \boldsymbol{\pi} \\ \boldsymbol{\pi}^T \text{diag}(\boldsymbol{\nu}'(\eta)) \boldsymbol{V} \boldsymbol{\Sigma} & \boldsymbol{\pi}^T \boldsymbol{\nu}'(\eta) \end{bmatrix}$$

$$\boldsymbol{A} := \boldsymbol{A}(\eta) := \begin{bmatrix} \sigma^2 \left( \boldsymbol{\pi}^T \boldsymbol{\nu}(\eta) \right) \boldsymbol{I} + \boldsymbol{\Sigma} \boldsymbol{V}^T \text{diag}(\boldsymbol{\pi}) \text{diag}(\boldsymbol{\nu}(\eta)) \boldsymbol{V} \boldsymbol{\Sigma} & \boldsymbol{\Sigma} \boldsymbol{V}^T \text{diag}(\boldsymbol{\nu}(\eta)) \boldsymbol{\pi} \\ \boldsymbol{\pi}^T \text{diag}(\boldsymbol{\nu}(\eta)) \boldsymbol{V} \boldsymbol{\Sigma} & \boldsymbol{\pi}^T \boldsymbol{\nu}(\eta) \end{bmatrix} .$$

Let us end by simplifying $\boldsymbol{Q}$ to this aim

$$\boldsymbol{A}^{-1} \begin{bmatrix} \boldsymbol{\Sigma} \boldsymbol{V}^T \\ \boldsymbol{1}^T \end{bmatrix} \text{diag}(\boldsymbol{\pi} \odot \boldsymbol{\nu}) = \begin{bmatrix} \boldsymbol{I} & \boldsymbol{0} \\ -\widetilde{\boldsymbol{\pi}}^T \boldsymbol{V} \boldsymbol{\Sigma} & 1 \end{bmatrix} \begin{bmatrix} \boldsymbol{\Delta}^{-1} & \boldsymbol{0} \\ \boldsymbol{0}^T & 1 \end{bmatrix} \begin{bmatrix} \boldsymbol{I} & -\boldsymbol{\Sigma} \boldsymbol{V}^T \widetilde{\boldsymbol{\pi}} \\ \boldsymbol{0}^T & 1 \end{bmatrix} \begin{bmatrix} \boldsymbol{\Sigma} \boldsymbol{V}^T \\ \boldsymbol{1}^T \end{bmatrix} \text{diag}(\boldsymbol{\pi} \odot \boldsymbol{\nu})$$

$$= \begin{bmatrix} \boldsymbol{I} & \boldsymbol{0} \\ -\widetilde{\boldsymbol{\pi}}^T \boldsymbol{V} \boldsymbol{\Sigma} & 1 \end{bmatrix} \begin{bmatrix} \boldsymbol{\Delta}^{-1} & \boldsymbol{0} \\ \boldsymbol{0}^T & 1 \end{bmatrix} \begin{bmatrix} \boldsymbol{\Sigma} \boldsymbol{V}^T \left( \boldsymbol{I} - \widetilde{\boldsymbol{\pi}} \boldsymbol{1}^T \right) \\ \boldsymbol{1}^T \end{bmatrix} \text{diag}(\boldsymbol{\pi} \odot \boldsymbol{\nu})$$

$$= \begin{bmatrix} \boldsymbol{I} & \boldsymbol{0} \\ -\widetilde{\boldsymbol{\pi}}^T \boldsymbol{V} \boldsymbol{\Sigma} & 1 \end{bmatrix} \begin{bmatrix} \boldsymbol{\Delta}^{-1} \boldsymbol{\Sigma} \boldsymbol{V}^T \left( \boldsymbol{I} - \widetilde{\boldsymbol{\pi}} \boldsymbol{1}^T \right) \\ \boldsymbol{1}^T \end{bmatrix} \text{diag}(\boldsymbol{\pi} \odot \boldsymbol{\nu})$$

$$= \begin{bmatrix} \boldsymbol{\Delta}^{-1} \boldsymbol{\Sigma} \boldsymbol{V}^T \left( \boldsymbol{I} - \widetilde{\boldsymbol{\pi}} \boldsymbol{1}^T \right) \\ -\widetilde{\boldsymbol{\pi}}^T \boldsymbol{V} \boldsymbol{\Sigma} \boldsymbol{\Delta}^{-1} \boldsymbol{\Sigma} \boldsymbol{V}^T \left( \boldsymbol{I} - \widetilde{\boldsymbol{\pi}} \boldsymbol{1}^T \right) + \boldsymbol{1}^T \end{bmatrix} \text{diag}(\boldsymbol{\pi} \odot \boldsymbol{\nu})$$

$$= \begin{bmatrix} \boldsymbol{\Delta}^{-1} \boldsymbol{\Sigma} \boldsymbol{V}^T \left( \text{diag}(\widetilde{\boldsymbol{\pi}}) - \widetilde{\boldsymbol{\pi}} \widetilde{\boldsymbol{\pi}}^T \right) \\ -\widetilde{\boldsymbol{\pi}}^T \boldsymbol{V} \boldsymbol{\Sigma} \boldsymbol{\Delta}^{-1} \boldsymbol{\Sigma} \boldsymbol{V}^T \left( \text{diag}(\widetilde{\boldsymbol{\pi}}) - \widetilde{\boldsymbol{\pi}} \widetilde{\boldsymbol{\pi}}'^T \right) + \widetilde{\boldsymbol{\pi}}^T \end{bmatrix}$$

Thus, defining $\widetilde{\boldsymbol{\pi}}' = \boldsymbol{\pi} \odot \boldsymbol{\nu}'$ we have

$$\text{diag}(\boldsymbol{\pi} \odot \boldsymbol{\nu}') \begin{bmatrix} \boldsymbol{\Sigma} \boldsymbol{V}^T \\ \boldsymbol{1}^T \end{bmatrix}^T \boldsymbol{A}^{-1} \begin{bmatrix} \boldsymbol{\Sigma} \boldsymbol{V}^T \\ \boldsymbol{1}^T \end{bmatrix} \text{diag}(\boldsymbol{\pi} \odot \boldsymbol{\nu}) = \text{diag}(\boldsymbol{\pi} \odot \boldsymbol{\nu}') \begin{bmatrix} \boldsymbol{\Sigma} \boldsymbol{V}^T \\ \boldsymbol{1}^T \end{bmatrix}^T \begin{bmatrix} \boldsymbol{\Delta}^{-1} \boldsymbol{\Sigma} \boldsymbol{V}^T \left( \text{diag}(\widetilde{\boldsymbol{\pi}}) - \widetilde{\boldsymbol{\pi}} \widetilde{\boldsymbol{\pi}}^T \right) \\ -\widetilde{\boldsymbol{\pi}}^T \boldsymbol{V} \boldsymbol{\Sigma} \boldsymbol{\Delta}^{-1} \boldsymbol{\Sigma} \boldsymbol{V}^T \left( \text{diag}(\widetilde{\boldsymbol{\pi}}) - \widetilde{\boldsymbol{\pi}} \widetilde{\boldsymbol{\pi}}'^T \right) + \widetilde{\boldsymbol{\pi}}^T \end{bmatrix}$$

$$= \left( \text{diag}(\widetilde{\boldsymbol{\pi}}') - \widetilde{\boldsymbol{\pi}}' \widetilde{\boldsymbol{\pi}}^T \right) \boldsymbol{V} \boldsymbol{\Sigma} \boldsymbol{\Delta}^{-1} \boldsymbol{\Sigma} \boldsymbol{V}^T \left( \text{diag}(\widetilde{\boldsymbol{\pi}}) - \widetilde{\boldsymbol{\pi}} \widetilde{\boldsymbol{\pi}}^T \right) + \widetilde{\boldsymbol{\pi}}' \widetilde{\boldsymbol{\pi}}^T$$

Using the above and recalling $\widetilde{\boldsymbol{\pi}}' = \boldsymbol{\pi} \odot \boldsymbol{\nu}'$ we arrive at

$$\boldsymbol{Q} = \mathrm{diag}(\widetilde{\boldsymbol{\pi}}')$$

$$+ \begin{bmatrix} \boldsymbol{\Delta}^{-1}\boldsymbol{\Sigma}\boldsymbol{V}^T\left(\mathrm{diag}(\widetilde{\boldsymbol{\pi}}) - \widetilde{\boldsymbol{\pi}}\widetilde{\boldsymbol{\pi}}^T\right) \\ -\widetilde{\boldsymbol{\pi}}^T\boldsymbol{V}\boldsymbol{\Sigma}\boldsymbol{\Delta}^{-1}\boldsymbol{\Sigma}\boldsymbol{V}^T\left(\mathrm{diag}(\widetilde{\boldsymbol{\pi}}) - \widetilde{\boldsymbol{\pi}}\widetilde{\boldsymbol{\pi}}^T\right) + \widetilde{\boldsymbol{\pi}}^T \end{bmatrix}^T \boldsymbol{A}' \begin{bmatrix} \boldsymbol{\Delta}^{-1}\boldsymbol{\Sigma}\boldsymbol{V}^T\left(\mathrm{diag}(\widetilde{\boldsymbol{\pi}}) - \widetilde{\boldsymbol{\pi}}\widetilde{\boldsymbol{\pi}}^T\right) \\ -\widetilde{\boldsymbol{\pi}}^T\boldsymbol{V}\boldsymbol{\Sigma}\boldsymbol{\Delta}^{-1}\boldsymbol{\Sigma}\boldsymbol{V}^T\left(\mathrm{diag}(\widetilde{\boldsymbol{\pi}}') - \widetilde{\boldsymbol{\pi}}\widetilde{\boldsymbol{\pi}}^T\right) + \widetilde{\boldsymbol{\pi}}^T \end{bmatrix}$$

$$- \left(\mathrm{diag}\left(\widetilde{\boldsymbol{\pi}}'\right) - \widetilde{\boldsymbol{\pi}}'\widetilde{\boldsymbol{\pi}}^T\right)\boldsymbol{V}\boldsymbol{\Sigma}\boldsymbol{\Delta}^{-1}\boldsymbol{\Sigma}\boldsymbol{V}^T\left(\mathrm{diag}(\widetilde{\boldsymbol{\pi}}) - \widetilde{\boldsymbol{\pi}}\widetilde{\boldsymbol{\pi}}^T\right) - \widetilde{\boldsymbol{\pi}}'\widetilde{\boldsymbol{\pi}}^T$$

$$- \left(\mathrm{diag}\left(\widetilde{\boldsymbol{\pi}}\right) - \widetilde{\boldsymbol{\pi}}\widetilde{\boldsymbol{\pi}}^T\right)\boldsymbol{V}\boldsymbol{\Sigma}\boldsymbol{\Delta}^{-1}\boldsymbol{\Sigma}\boldsymbol{V}^T\left(\mathrm{diag}(\widetilde{\boldsymbol{\pi}}') - \widetilde{\boldsymbol{\pi}}\widetilde{\boldsymbol{\pi}}'^T\right) - \widetilde{\boldsymbol{\pi}}\widetilde{\boldsymbol{\pi}}'^T \tag{J.18}$$

where

$$\boldsymbol{A}' = \begin{bmatrix} \sigma^2\left(\widetilde{\boldsymbol{\pi}}'^T\boldsymbol{1}\right)\boldsymbol{I} + \boldsymbol{\Sigma}\boldsymbol{V}^T\mathrm{diag}\left(\widetilde{\boldsymbol{\pi}}'\right)\boldsymbol{V}\boldsymbol{\Sigma} & \boldsymbol{\Sigma}\boldsymbol{V}^T\widetilde{\boldsymbol{\pi}}' \\ \widetilde{\boldsymbol{\pi}}'^T\boldsymbol{V}\boldsymbol{\Sigma} & \widetilde{\boldsymbol{\pi}}'^T\boldsymbol{1} \end{bmatrix}$$

$$\boldsymbol{A} = \begin{bmatrix} \sigma^2\boldsymbol{I} + \boldsymbol{\Sigma}\boldsymbol{V}^T\mathrm{diag}\left(\widetilde{\boldsymbol{\pi}}\right)\boldsymbol{V}\boldsymbol{\Sigma} & \boldsymbol{\Sigma}\boldsymbol{V}^T\widetilde{\boldsymbol{\pi}} \\ \widetilde{\boldsymbol{\pi}}^T\boldsymbol{V}\boldsymbol{\Sigma} & 1 \end{bmatrix}.$$

Using the above the cross-correlation matrix $\boldsymbol{\Sigma}_{\boldsymbol{w},\boldsymbol{w}}$ is given by

$$\boldsymbol{\Sigma}_{\boldsymbol{w},\boldsymbol{w}} \xrightarrow{P} \frac{\zeta}{\sigma^2}\boldsymbol{P} + \boldsymbol{P}\boldsymbol{V}\boldsymbol{\Sigma}\boldsymbol{\Delta}^{-1}\left(\boldsymbol{\Delta}^{-1} - \frac{\zeta}{\sigma^2}\boldsymbol{I}_r\right)\boldsymbol{\Sigma}\boldsymbol{V}^T\boldsymbol{P} + \frac{\zeta\eta}{\sigma^2}\boldsymbol{Q}.$$

# K  Weighted LS for MLM (Proof of Theorem B.2)

Let $\boldsymbol{D} := \boldsymbol{D}^{(n)} := \mathrm{diag}(D_1,\ldots,D_n)$ be a diagonal matrix with non-zero diagonal entries. In particular, assume that the diagonal entries of $\boldsymbol{D}$ are distributed $D_i \overset{iid}{\sim} D$ where the random variable $D$ may depend on the entries of the matrix of response variables $\boldsymbol{Y}$. Here, we focus on the following setting:

$$\boldsymbol{D} = \sum_{j\in[k]} \mathrm{diag}(\omega_j \boldsymbol{Y}_j), \quad \omega_j \geq 0,\ j \in [k]. \tag{K.1}$$

Specifically, for (K.1), we have $D_i \overset{iid}{\sim} D$ with $D = \omega_\ell Y_\ell + \sum_{i\neq\ell\in[k]}\omega_i Y_i$, where for all $c \in [k]$:

$$\mathbb{P}\left([Y_1, Y_2, \ldots, Y_k]^T = \boldsymbol{e}_c\right) = V_c = \frac{e^{\boldsymbol{e}_c^T\boldsymbol{V}\boldsymbol{\Sigma}\boldsymbol{g}}}{\sum_{\ell'=1}^k e^{\boldsymbol{e}_{\ell'}^T\boldsymbol{V}\boldsymbol{\Sigma}\boldsymbol{g}}}, \tag{K.2}$$

$\boldsymbol{M}\boldsymbol{M}^T = \boldsymbol{V}\boldsymbol{\Sigma}^2\boldsymbol{V}^T$, and $\boldsymbol{g} \sim \mathcal{N}(\boldsymbol{0}, \boldsymbol{I}_r)$.

With these, we consider the weighted least-squares (WLS) solution for $\ell \in [k]$:

$$(\widehat{\boldsymbol{w}}_\ell, \widehat{b}) = \arg\min_{\boldsymbol{w},b} \mathcal{L}_{PO}(\boldsymbol{w}, b) := \frac{1}{2n}\left\|\boldsymbol{D}\left(\boldsymbol{X}^T\boldsymbol{w} + b\boldsymbol{1}_n - \boldsymbol{Y}_\ell\right)\right\|_{\ell_2}^2,$$

where $\boldsymbol{D}$ is as in (K.1). In fact, it is convenient to rewrite the above as follows:

$$(\widehat{\boldsymbol{w}}_\ell, \widehat{b}) = \arg\min_{\boldsymbol{w},b,\boldsymbol{u}} \max_{\boldsymbol{s}} \frac{1}{n}\left(\boldsymbol{s}^T\boldsymbol{D}\boldsymbol{X}^T\boldsymbol{w} + b\boldsymbol{s}^T\boldsymbol{D}\boldsymbol{1}_n\boldsymbol{s}^T\boldsymbol{D}\boldsymbol{Y}_\ell - \boldsymbol{s}^T\boldsymbol{u} + \frac{\|\boldsymbol{u}\|_{\ell_2}^2}{2}\right). \tag{K.3}$$

**Identifying the AO.** The PO in (K.3) is very similar to (I.1). In particular, following step by step the same decomposition trick as in Section H.1.1, it can be shown that the AO corresponding to (K.3) becomes (cf. (I.6))

$$\min_{\boldsymbol{w}_\ell, b_\ell, \boldsymbol{u}} \max_{\boldsymbol{s}} \frac{1}{n}\left(\|\boldsymbol{P}^\perp\boldsymbol{w}_\ell\|_{\ell_2}\boldsymbol{g}^T\boldsymbol{D}\boldsymbol{s} + \|\boldsymbol{D}\boldsymbol{s}\|_{\ell_2}\boldsymbol{h}^T\boldsymbol{P}^\perp\boldsymbol{w}_\ell + \boldsymbol{s}^T\boldsymbol{D}\widetilde{\boldsymbol{G}}^T\boldsymbol{U}^T\boldsymbol{w}_\ell + b_\ell\boldsymbol{s}^T\boldsymbol{D}\boldsymbol{1}_n - \boldsymbol{s}^T\boldsymbol{D}\boldsymbol{Y}_\ell - \boldsymbol{u}^T\boldsymbol{s} + \frac{\|\boldsymbol{u}\|_{\ell_2}^2}{2}\right),$$

where we use the same notation as in Section H.1.1 for $\boldsymbol{P}^\perp, \boldsymbol{U}, \widetilde{\boldsymbol{G}}, \boldsymbol{g}$ and $\boldsymbol{h}$. Recall also the relation of $\boldsymbol{Y}_\ell$ to $\widetilde{\boldsymbol{G}}$ in (I.5).

**Scalarization of the AO.** We start the process of simplifying the AO by setting $\beta := \|\boldsymbol{Ds}\|_{\ell_2} / \sqrt{n}$ and optimizing over the direction of $\boldsymbol{Ds}$ to equivalently write the AO as

$$\min_{\boldsymbol{w}_\ell, \boldsymbol{b}_\ell, \boldsymbol{u}} \max_{\beta \geq 0} \frac{1}{\sqrt{n}} \left( \beta \left\| \|\boldsymbol{P}^\perp \boldsymbol{w}_\ell\|_{\ell_2} \boldsymbol{g} + \widetilde{\boldsymbol{G}}^T \boldsymbol{U}^T \boldsymbol{w}_\ell + b_\ell \boldsymbol{1}_n - \boldsymbol{Y}_\ell - \boldsymbol{D}^{-1} \boldsymbol{u} \right\|_{\ell_2} + \beta \boldsymbol{h}^T \boldsymbol{P}^\perp \boldsymbol{w}_\ell \right) + \frac{\|\boldsymbol{u}\|_{\ell_2}^2}{2n},$$
(K.4)

Next, focus on the minimization over $\boldsymbol{w}_\ell$. Let us denote

$$\boldsymbol{a} := \boldsymbol{U}^T \boldsymbol{w}_\ell \quad \text{and} \quad \alpha_0 = \|\boldsymbol{P}^\perp \boldsymbol{w}_\ell\|_{\ell_2}.$$

Notice that $\boldsymbol{a} \perp \boldsymbol{P}^\perp \boldsymbol{w}_\ell$ and thus the orthogonal decomposition $\boldsymbol{w}_\ell = \boldsymbol{U}\boldsymbol{a} + \boldsymbol{P}^\perp \boldsymbol{w}_\ell$. With this observation, note that the optimal direction of $\boldsymbol{P}^T \boldsymbol{w}_\ell$ in (K.4) aligns with $\boldsymbol{P}^T \boldsymbol{h}$ for all values of $\beta$. Therefore, (K.4) reduces to

$$\min_{\boldsymbol{a}, \alpha_0 \geq 0, \boldsymbol{b}_\ell, \boldsymbol{u}} \max_{\beta \geq 0} \frac{1}{\sqrt{n}} \left( \beta \left\| \alpha_0 \boldsymbol{g} + \widetilde{\boldsymbol{G}}^T \boldsymbol{a} + b_\ell \boldsymbol{1}_n - \boldsymbol{Y}_\ell - \boldsymbol{D}^{-1} \boldsymbol{u} \right\|_{\ell_2} - \beta \alpha_0 \|\boldsymbol{P}^\perp \boldsymbol{h}\|_{\ell_2} \right) + \frac{\|\boldsymbol{u}\|_{\ell_2}^2}{2n}, \quad \text{(K.5)}$$

Continuing let us denote $\boldsymbol{t} := \alpha_0 \boldsymbol{g} + \widetilde{\boldsymbol{G}}^T \boldsymbol{a} + b_\ell \boldsymbol{1}_n - \boldsymbol{Y}_\ell$ for convenience and rewrite $\|\boldsymbol{t} - \boldsymbol{D}^{-1}\boldsymbol{u}\|_{\ell_2}$ as follows

$$\frac{\|\boldsymbol{t} - \boldsymbol{D}^{-1}\boldsymbol{u}\|_{\ell_2}}{\sqrt{n}} = \min_{\tau > 0} \frac{\tau}{2} + \frac{\|\boldsymbol{t} - \boldsymbol{D}^{-1}\boldsymbol{u}\|_{\ell_2}^2}{2\tau n}.$$

Note that the resulting minimization is convex in $\boldsymbol{u}$ and concave in $\beta$. Also, by considering the bounded AO (such that $\beta$ is bounded; see [DKT19, Sec. A]), we can flip the order of min-max and optimize over $\boldsymbol{u}$ first. In particular, $\boldsymbol{u}$ minimizes the following strictly convex quadratic

$$\min_{\boldsymbol{u}} \left\{ \frac{1}{n} \left( \frac{\beta}{2\tau} \|\boldsymbol{D}^{-1}\boldsymbol{u}\|_{\ell_2} + \frac{1}{2} \|\boldsymbol{u}\|_{\ell_2}^2 - \frac{\beta}{\tau} \boldsymbol{t}^T \boldsymbol{D}^{-1} \boldsymbol{u} \right) = \frac{1}{2n} \boldsymbol{u}^T \left( \frac{\beta}{\tau} \boldsymbol{D}^{-2} + \boldsymbol{I}_n \right) \boldsymbol{u} - \frac{\beta}{\tau n} \boldsymbol{t}^T \boldsymbol{D}^{-1} \boldsymbol{u} \right\}.$$

In particular,

$$\boldsymbol{u} = \frac{\beta}{\tau} \left( \frac{\beta}{\tau} \boldsymbol{D}^{-2} + \boldsymbol{I} \right)^{-1} \boldsymbol{D}^{-1} \boldsymbol{t} = \left( \boldsymbol{D}^{-1} + \frac{\tau}{\beta} \boldsymbol{D} \right)^{-1} \left( \alpha_0 \boldsymbol{g} + \widetilde{\boldsymbol{G}}^T \boldsymbol{a} + b_\ell \boldsymbol{1}_n - \boldsymbol{Y}_\ell \right)$$

Putting things together, the new objective function of (K.5) becomes

$$\min_{\boldsymbol{a}, \alpha_0 \geq 0, \boldsymbol{b}_\ell, \tau > 0} \max_{\beta \geq 0} \mathcal{R}(\boldsymbol{a}, \alpha_0, b_\ell, \tau, \beta) \tag{K.6}$$

$$\text{where} \quad \mathcal{R}(\boldsymbol{a}, \alpha_0, b_\ell, \tau, \beta) := \frac{\beta\tau}{2n} + \frac{\beta}{2\tau n} \|\boldsymbol{t}\|_{\ell_2}^2 - \frac{\beta}{2\tau n} \boldsymbol{t}^T \left( \boldsymbol{I} + \frac{\tau}{\beta} \boldsymbol{D}^2 \right)^{-1} \boldsymbol{t} - \frac{\beta\alpha_0}{\sqrt{n}} \|\boldsymbol{P}^\perp \boldsymbol{h}\|_{\ell_2}.$$

**Convergence of the AO** After having simplified the AO into an optimization problem over $r + 4$ variables, we are ready to study its asymptotic behavior. First, we argue on point-wise convergence of $\mathcal{R}$ in (K.6). Fix $\boldsymbol{a}, \alpha_0, \boldsymbol{b}_\ell, \tau$ and $\beta$. From the WLLN, $\frac{1}{\sqrt{n}} \|\boldsymbol{P}^\perp \boldsymbol{h}\|_{\ell_2} \xrightarrow{P} \sqrt{\gamma}$ and as in (I.9)

$$\frac{1}{n} \|\boldsymbol{t}\|_{\ell_2}^2 = \frac{1}{n} \sum_{i=1}^n \left( \alpha_0 \boldsymbol{g}_i + \boldsymbol{a}^T \tilde{\boldsymbol{g}}_i + b_\ell - [\boldsymbol{Y}_\ell]_i \right)^2 \xrightarrow{P} \mathbb{E} \left[ \left( \alpha_0 G_0 + \boldsymbol{a}^T \boldsymbol{g} + b_\ell - Y_\ell \right)^2 \right],$$

where the expectation is over $\boldsymbol{g} \sim \mathcal{N}(\boldsymbol{0}_r, \boldsymbol{I}_r)$ (with some abuse of notation) and

$$Y_\ell \sim \mathrm{Bern}(V_\ell) \quad \text{and} \quad V_\ell = \frac{e^{\boldsymbol{e}_\ell^T \boldsymbol{V} \boldsymbol{\Sigma} \boldsymbol{g}}}{\sum_{\ell'=1}^r e^{\boldsymbol{e}_{\ell'}^T \boldsymbol{V} \boldsymbol{\Sigma} \boldsymbol{g}}}. \tag{K.7}$$

Furthermore,

$$\frac{1}{n} \boldsymbol{t}^T \left( \boldsymbol{I} + \frac{\tau}{\beta} \boldsymbol{D}^2 \right)^{-1} \boldsymbol{t} = \frac{1}{n} \sum_{i=1}^n \frac{\left( \alpha_0 \boldsymbol{g}_i + \boldsymbol{a}^T \tilde{\boldsymbol{g}}_i + b_\ell - [\boldsymbol{Y}_\ell]_i \right)^2}{1 + \frac{\tau}{\beta} d_i^2} \xrightarrow{P} \mathbb{E} \left[ \frac{\left( \alpha_0 G_0 + \boldsymbol{a}^T \boldsymbol{g} + b_\ell - Y_\ell \right)^2}{1 + \frac{\tau}{\beta} D^2} \right]$$

Therefore, point-wise on $\boldsymbol{a}, \alpha_0, \boldsymbol{b}_\ell, \tau$ and $\beta$, the objective $\mathcal{R}$ of the AO converges to

$$
\begin{aligned}
\mathcal{D}_\ell(\alpha_0, \boldsymbol{\alpha}, b_\ell, \tau, \beta) &:= \frac{\beta\tau}{2} + \frac{\beta}{2\tau} \mathbb{E}\left[\left(\alpha_0 G_0 + \boldsymbol{a}^T \boldsymbol{g} + \boldsymbol{b}_\ell - Y_\ell\right)^2\right] - \frac{\beta}{2\tau} \mathbb{E}\left[\frac{\left(\alpha_0 G_0 + \boldsymbol{a}^T \boldsymbol{g} + \boldsymbol{b}_\ell - Y_\ell\right)^2}{1 + \frac{\tau}{\beta} D^2}\right] - \beta\alpha_0\sqrt{\gamma} \\
&= \frac{\beta\tau}{2} + \frac{1}{2} \mathbb{E}\left[\frac{D^2\left(\alpha_0 G_0 + \boldsymbol{a}^T \boldsymbol{g} + \boldsymbol{b}_\ell - Y_\ell\right)^2}{1 + \frac{\tau}{\beta} D^2}\right] - \beta\alpha_0\sqrt{\gamma} \\
&= \frac{\beta\tau}{2} + \frac{1}{2} \mathbb{E}\left[\frac{\left(\alpha_0 G_0 + \boldsymbol{a}^T \boldsymbol{g} + \boldsymbol{b}_\ell - Y_\ell\right)^2}{D^{-2} + (\tau/\beta)}\right] - \beta\alpha_0\sqrt{\gamma}.
\end{aligned}
\tag{K.8}
$$

We note that the function above is jointly convex in $(\alpha_0, \boldsymbol{\alpha}, b_\ell, \tau)$ and concave in $\beta$.

## K.1 Computing $\Sigma_{w,\mu}$

It can be checked that the first order optimality conditions of $\mathcal{D}_\ell(\alpha_0, \boldsymbol{\alpha}, b_\ell, \tau, \beta)$ with respect to $\beta$ and $\tau > 0$ are given as follows:

$$
\beta^2 = \mathbb{E}\left[\frac{\left(\alpha_0 G_0 + \boldsymbol{a}^T \boldsymbol{g} + \boldsymbol{b}_\ell - Y_\ell\right)^2}{\left(D^{-2} + (\tau/\beta)\right)^2}\right] \quad \text{or} \quad \beta = 0,
\tag{K.9}
$$

$$
\alpha_0\sqrt{\gamma} = \frac{\tau}{2} + \frac{\tau}{2\beta^2} \cdot \mathbb{E}\left[\frac{\left(\alpha_0 G_0 + \boldsymbol{a}^T \boldsymbol{g} + \boldsymbol{b}_\ell - Y_\ell\right)^2}{\left(D^{-2} + (\tau/\beta)\right)^2}\right].
\tag{K.10}
$$

Thus, at optimality either $\beta = 0$ or $\tau = \alpha_0\sqrt{\gamma}$. In what follows, consider the solution $\tau = \alpha_0\sqrt{\gamma}$. We will show that this leads to the true saddle point of $\mathcal{D}$.

Moreover, by denoting $\eta := \frac{\beta}{\tau}$ and recalling from (K.7) that $Y_\ell = \text{Bern}(V_\ell)$, we can express $\mathcal{D}_\ell(\alpha_0, \boldsymbol{\alpha}, b_\ell, \tau, \beta)$ as follows

$$
\frac{\beta\tau}{2} + \frac{\alpha_0^2}{2} \mathbb{E}\left[\frac{1}{D^{-2} + 1/\eta}\right] - \beta\alpha_0\sqrt{\gamma} + \frac{1}{2}\begin{bmatrix}\boldsymbol{a}^T & \boldsymbol{b}_\ell\end{bmatrix} \cdot \boldsymbol{A}(\eta) \cdot \begin{bmatrix}\boldsymbol{a} \\ \boldsymbol{b}_\ell\end{bmatrix} - \boldsymbol{c}_\ell^T(\eta)\begin{bmatrix}\boldsymbol{a} \\ \boldsymbol{b}_\ell\end{bmatrix} + \frac{1}{2}\mathbb{E}\left[\frac{Y_\ell^2}{D^{-2} + 1/\eta}\right],
$$

where

$$
\boldsymbol{A}(\eta) := \begin{bmatrix} \mathbb{E}\left[\frac{\boldsymbol{g}\boldsymbol{g}^T}{D^{-2}+1/\eta}\right] & \mathbb{E}\left[\frac{\boldsymbol{g}}{D^{-2}+1/\eta}\right] \\ \mathbb{E}\left[\frac{\boldsymbol{g}^T}{D^{-2}+1/\eta}\right] & \mathbb{E}\left[\frac{1}{D^{-2}+1/\eta}\right] \end{bmatrix}
\tag{K.11a}
$$

$$
\boldsymbol{c}_\ell(\eta) := \begin{bmatrix} \mathbb{E}\left[\frac{\boldsymbol{g}Y_\ell}{D^{-2}+1/\eta}\right] \end{bmatrix}
\tag{K.11b}
$$

we have the following first-order optimality conditions for $\alpha_0, \boldsymbol{a}$ and $\boldsymbol{b}_\ell$:

$$
\begin{bmatrix}\boldsymbol{a} \\ \boldsymbol{b}_\ell\end{bmatrix} = \boldsymbol{A}^{-1}(\eta) \cdot \boldsymbol{c}_\ell(\eta)
\tag{K.12}
$$

$$
\alpha_0 = \beta\sqrt{\gamma} \Big/ \mathbb{E}\left[\frac{1}{D^{-2} + 1/\eta}\right].
\tag{K.13}
$$

Rearranging (K.13) and using $\tau = \alpha_0\sqrt{\gamma}$ gives the following equation for $\eta$:

$$
\frac{\alpha_0\sqrt{\gamma}}{\beta} \mathbb{E}\left[\frac{1}{D^{-2} + 1/\eta}\right] = \gamma \overset{\tau=\alpha_0\sqrt{\gamma}}{\Longrightarrow} \mathbb{E}\left[\frac{1/\eta}{D^{-2} + 1/\eta}\right] = \gamma.
\tag{K.14}
$$

Thus, the optimal values of $\boldsymbol{a}$ and $\boldsymbol{b}_\ell$ are found by (K.12) for $\eta$ the positive solution of the equation in (K.14). To solve for $\alpha_0$, we combine (K.13) and (K.9) which leads to

$$
\alpha_0^2\left(\gamma\eta^2 - \mathbb{E}\left[\left(\frac{1}{D^{-2} + 1/\eta}\right)^2\right]\right) = \mathbb{E}\left[\frac{\left(\boldsymbol{a}^T \boldsymbol{g} + \boldsymbol{b}_\ell - Y_\ell\right)^2}{\left(D^{-2} + 1/\eta\right)^2}\right],
\tag{K.15}
$$

where we have also used the RHS of (K.14). Next, we specialize these findings to the special structure of the weighting matrix $D$ in (K.1).

**Applying weighting** (K.1). Assume (K.1) holds. In this case, Equation (K.14) that determines the value of $\eta > 0$ becomes

$$F(\eta) := \sum_{i \in [k]} \frac{\pi_i \omega_i^2}{\omega_i^2 + \eta} = \gamma, \tag{K.16}$$

where we have recalled the notation in (4.1) $\pi_i := \mathbb{E}[V_i] > 0$, $i \in [k]$. It can be easily checked by direct differentiation that $\eta \mapsto F$ is strictly decreasing in $(0, \infty)$. Also, using $\sum_{i \in [k]} \pi_i = 1$ the range of $F$ in $(0, \infty)$ is $(0, 1)$. Thus, it follows that (K.16) has a unique solution for all $\gamma \in (0, 1)$.

Also, in this case we can write (K.11) in the following more convenient form:

$$A(\eta) := \sum_{i \in [k]} \left( \frac{\omega_i^2 \eta}{\omega_i^2 + \eta} \right) \underbrace{\mathbb{E}\left[ \begin{bmatrix} g \\ 1 \end{bmatrix} \begin{bmatrix} g^T & 1 \end{bmatrix} V_i \right]}_{=: \widetilde{A}_i} \tag{K.17}$$

$$c_\ell(\eta) := \left( \frac{\omega_\ell^2 \eta}{\omega_\ell^2 + \eta} \right) \underbrace{\mathbb{E}\left[ \begin{bmatrix} g \\ 1 \end{bmatrix} V_\ell \right]}_{=: \widetilde{c}_\ell}. \tag{K.18}$$

For convenience let us define vectors $\nu := \nu(\eta), \widetilde{\pi} = \widetilde{\pi}(\eta) \in \mathbb{R}^k$ with entries:

$$\widetilde{\pi}_i := \pi_i \left( \frac{1}{\gamma} \cdot \frac{\omega_i^2}{\omega_i^2 + \eta} \right) =: \pi_i \cdot \nu_i \tag{K.19}$$

Because of (K.16), notice that $\widetilde{\pi}$ is a probability vector, i.e.

$$\widetilde{\pi}^T \mathbf{1}_k = \pi^T \nu = 1.$$

With the notation above, it holds

$$A(\eta) = \gamma \cdot \eta \cdot \begin{bmatrix} \sum_{i \in [k]} \nu_i \cdot \mathbb{E}[V_i g g^T] & \sum_{i \in [k]} \nu_i \cdot \mathbb{E}[V_i g] \\ \sum_{i \in [k]} \nu_i \cdot \mathbb{E}[V_i g^T] & 1 \end{bmatrix}$$

$$= \gamma \cdot \eta \cdot \begin{bmatrix} \sum_{i \in [k]} \nu_i \cdot \mathbb{E}[V_i g g^T] & \Sigma V^T (\operatorname{diag}(\pi) - \Pi) \nu \\ \nu^T (\operatorname{diag}(\pi) - \Pi) V \Sigma & 1 \end{bmatrix} \tag{K.20}$$

$$= \gamma \cdot \eta \cdot \begin{bmatrix} \mathbb{E}[(\nu^T v) g g^T] & \Sigma V^T (\operatorname{diag}(\pi) - \Pi) \nu \\ \nu^T (\operatorname{diag}(\pi) - \Pi) V \Sigma & 1 \end{bmatrix} \tag{K.21}$$

$$c_\ell(\eta) = \gamma \cdot \eta \cdot \begin{bmatrix} \nu_\ell \, \mathbb{E}[V_\ell g] \\ \widetilde{\pi}_\ell \end{bmatrix}$$

$$= \gamma \cdot \eta \cdot \begin{bmatrix} \Sigma V^T (\operatorname{diag}(\pi) - \Pi) \nu_\ell e_\ell \\ \widetilde{\pi}_\ell \end{bmatrix} \tag{K.22}$$

where we have also used the fact that $\mathbb{E}[V_i g] = \Sigma V^T (\operatorname{diag}(\pi) - \Pi) e_i$, $i \in [k]$ and recalled the notation

$$v = [V_1, \dots, V_k]^T.$$

Using (K.30) and (K.31), we conclude from (K.12) the following expressions for $a$ and $b$:

$$a = \Delta^{-1} \Sigma V^T (\operatorname{diag}(\pi) - \Pi) \cdot \nu_\ell \cdot (e_\ell - \pi_\ell \nu), \tag{K.23}$$

$$b_\ell = \widetilde{\pi}_\ell - \nu^T (\operatorname{diag}(\pi) - \Pi) V \Sigma \Delta^{-1} \Sigma V^T (\operatorname{diag}(\pi) - \Pi) \cdot \nu_\ell \cdot (e_\ell - \pi_\ell \nu), \tag{K.24}$$

where we defined

$$\Delta = \mathbb{E}[(\nu^T v) g g^T] - \Sigma V^T (\operatorname{diag}(\pi) - \Pi) \nu \nu^T (\operatorname{diag}(\pi) - \Pi) V \Sigma > 0_{r \times r}. \tag{K.25}$$

Finally, we show how to compute $\alpha_0$ using (K.15). The RHS in (K.15) can be computed as

$$\sum_{i\neq\ell\in[k]} \frac{\begin{bmatrix}\boldsymbol{a}^T & \boldsymbol{b}_\ell\end{bmatrix}\widetilde{\boldsymbol{A}}_i\begin{bmatrix}\boldsymbol{a}\\\boldsymbol{b}_\ell\end{bmatrix}}{\left(\omega_i^{-2}+1/\eta\right)^2} + \frac{\begin{bmatrix}\boldsymbol{a}^T & \boldsymbol{b}_\ell\end{bmatrix}\widetilde{\boldsymbol{A}}_\ell\begin{bmatrix}\boldsymbol{a}\\\boldsymbol{b}_\ell\end{bmatrix}-2\begin{bmatrix}\boldsymbol{a}^T & \boldsymbol{b}_\ell\end{bmatrix}\widetilde{\boldsymbol{c}}_\ell+\pi_\ell}{\left(\omega_\ell^{-2}+1/\eta\right)^2}$$

$$= \eta^2\cdot\gamma^2\cdot\left\{\begin{bmatrix}\boldsymbol{a}^T & \boldsymbol{b}_\ell\end{bmatrix}\left(\sum_{i\in[k]}\nu_i^2\widetilde{\boldsymbol{A}}_i\right)\begin{bmatrix}\boldsymbol{a}\\\boldsymbol{b}_\ell\end{bmatrix}-2\begin{bmatrix}\boldsymbol{a}^T & \boldsymbol{b}_\ell\end{bmatrix}\nu_\ell^2\widetilde{\boldsymbol{c}}_\ell+\pi_\ell\nu_\ell^2\right\},$$

where $\boldsymbol{a},\boldsymbol{b}_\ell$ are as in (K.23) and (K.24). Also, note that

$$\mathbb{E}\left[\left(\frac{1}{D^{-2}+1/\eta}\right)^2\right]=\eta^2\sum_{i\in[k]}\frac{\pi_i\omega_i^4}{\left(\omega_i^2+\eta\right)^2}=\eta^2\cdot\gamma^2\cdot\boldsymbol{\pi}^T\mathrm{diag}(\boldsymbol{\nu})\boldsymbol{\nu}=\eta^2\cdot\gamma^2\cdot\widetilde{\boldsymbol{\pi}}^T\boldsymbol{\nu}.$$

Put together, we have the following expression for $\alpha_0$:

$$\alpha_0^2 = \frac{1}{(1/\gamma-\widetilde{\boldsymbol{\pi}}^T\boldsymbol{\nu})}\cdot\left\{\begin{bmatrix}\boldsymbol{a}^T & \boldsymbol{b}_\ell\end{bmatrix}\left(\sum_{i\in[k]}\nu_i^2\widetilde{\boldsymbol{A}}_i\right)\begin{bmatrix}\boldsymbol{a}\\\boldsymbol{b}_\ell\end{bmatrix}-2\begin{bmatrix}\boldsymbol{a}^T & \boldsymbol{b}_\ell\end{bmatrix}\nu_\ell^2\widetilde{\boldsymbol{c}}_\ell+\pi_\ell\nu_\ell^2\right\}$$

$$= \frac{1}{(1/\gamma-\widetilde{\boldsymbol{\pi}}^T\boldsymbol{\nu})}\cdot\left\{\begin{bmatrix}\boldsymbol{a}^T & \boldsymbol{b}_\ell\end{bmatrix}\boldsymbol{A}'\begin{bmatrix}\boldsymbol{a}\\\boldsymbol{b}_\ell\end{bmatrix}-2\begin{bmatrix}\boldsymbol{a}^T & \boldsymbol{b}_\ell\end{bmatrix}\begin{bmatrix}\boldsymbol{\Sigma}\boldsymbol{V}^T\left(\mathrm{diag}(\boldsymbol{\pi})-\boldsymbol{\Pi}\right)\nu_\ell^2\,\boldsymbol{e}_\ell\\\widetilde{\boldsymbol{\pi}}_\ell\cdot\boldsymbol{\nu}_\ell\end{bmatrix}+\widetilde{\boldsymbol{\pi}}_\ell\cdot\boldsymbol{\nu}_\ell\right\},$$
(K.26)

where $\boldsymbol{a},\boldsymbol{b}_\ell$ are as in (K.23), (K.24) and we have also defined

$$\boldsymbol{A}' = \begin{bmatrix}\mathbb{E}\left[\left(\boldsymbol{\nu}^T\mathrm{diag}(\boldsymbol{\nu})\boldsymbol{v}\right)\boldsymbol{g}\boldsymbol{g}^T\right] & \boldsymbol{\Sigma}\boldsymbol{V}^T\left(\mathrm{diag}(\boldsymbol{\pi})-\boldsymbol{\Pi}\right)\mathrm{diag}(\boldsymbol{\nu})\boldsymbol{\nu}\\\boldsymbol{\nu}^T\mathrm{diag}(\boldsymbol{\nu})\left(\mathrm{diag}(\boldsymbol{\pi})-\boldsymbol{\Pi}\right)\boldsymbol{V}\boldsymbol{\Sigma} & \boldsymbol{\nu}^T\mathrm{diag}(\boldsymbol{\nu})\boldsymbol{\pi}\end{bmatrix}.$$
(K.27)

**Asymptotic Predictions.** Writing (K.24) in vector form we find that

$$\widehat{\boldsymbol{b}}\xrightarrow{P}\widetilde{\boldsymbol{\pi}}-\mathrm{diag}(\boldsymbol{\nu})\left(\boldsymbol{I}_k-\boldsymbol{\pi}\boldsymbol{\nu}^T\right)\left(\mathrm{diag}(\boldsymbol{\pi})-\boldsymbol{\Pi}\right)\boldsymbol{V}\boldsymbol{\Sigma}\boldsymbol{\Delta}^{-1}\boldsymbol{\Sigma}\boldsymbol{V}^T\left(\mathrm{diag}(\boldsymbol{\pi})-\boldsymbol{\Pi}\right)\boldsymbol{\nu}.$$
(K.28)

Also, recalling that $\boldsymbol{e}_\ell^T\boldsymbol{\Sigma}_{\boldsymbol{w},\boldsymbol{\mu}}=\widehat{\boldsymbol{w}}_\ell^T\boldsymbol{U}\boldsymbol{\Sigma}\boldsymbol{V}^T\xrightarrow{P}\boldsymbol{a}^T\boldsymbol{\Sigma}\boldsymbol{V}^T$ and using (K.23):

$$\boldsymbol{\Sigma}_{\boldsymbol{w},\boldsymbol{\mu}}\xrightarrow{P}\mathrm{diag}(\boldsymbol{\nu})\left(\boldsymbol{I}_k-\boldsymbol{\pi}\boldsymbol{\nu}^T\right)\left(\mathrm{diag}(\boldsymbol{\pi})-\boldsymbol{\Pi}\right)\boldsymbol{V}\boldsymbol{\Sigma}\boldsymbol{\Delta}^{-1}\boldsymbol{\Sigma}\boldsymbol{V}^T.$$
(K.29)

Finally, for the magnitudes of the weight vectors, recall that $\|\widehat{\boldsymbol{w}}_\ell\|_{\ell_2}^2\xrightarrow{P}\|\boldsymbol{a}\|_{\ell_2}^2+\alpha_0^2$. Thus, to find the limiting values of the norms, we can combine (K.26) and (K.23)-(K.24). For convenience, we summarize the final expression here. Define the following[1]

$$\boldsymbol{A}:=\begin{bmatrix}\mathbb{E}\left[\left(\boldsymbol{\nu}^T\boldsymbol{v}\right)\boldsymbol{g}\boldsymbol{g}^T\right] & \boldsymbol{\Sigma}\boldsymbol{V}^T\left(\mathrm{diag}(\boldsymbol{\pi})-\boldsymbol{\Pi}\right)\boldsymbol{\nu}\\\boldsymbol{\nu}^T\left(\mathrm{diag}(\boldsymbol{\pi})-\boldsymbol{\Pi}\right)\boldsymbol{V}\boldsymbol{\Sigma} & 1\end{bmatrix}$$
(K.30)

$$\boldsymbol{c}_\ell:=\begin{bmatrix}\boldsymbol{\Sigma}\boldsymbol{V}^T\left(\mathrm{diag}(\boldsymbol{\pi})-\boldsymbol{\Pi}\right)\boldsymbol{\nu}_\ell\,\boldsymbol{e}_\ell\\\widetilde{\boldsymbol{\pi}}_\ell\end{bmatrix}.$$
(K.31)

Further recall the matrix $\boldsymbol{A}'$ in (K.27).

$$\|\widehat{\boldsymbol{w}}_\ell\|_{\ell_2}^2\xrightarrow{P}\left\|\boldsymbol{\Delta}^{-1}\boldsymbol{\Sigma}\boldsymbol{V}^T\left(\mathrm{diag}(\boldsymbol{\pi})-\boldsymbol{\Pi}\right)\cdot\boldsymbol{\nu}_\ell\cdot\left(\boldsymbol{e}_\ell-\pi_\ell\boldsymbol{\nu}\right)\right\|_{\ell_2}^2$$

$$+\frac{1}{(1/\gamma-\widetilde{\boldsymbol{\pi}}^T\boldsymbol{\nu})}\cdot\left\{\boldsymbol{c}_\ell^T\boldsymbol{A}^{-1}\boldsymbol{A}'\boldsymbol{A}^{-1}\boldsymbol{c}_\ell-2\boldsymbol{\nu}_\ell\boldsymbol{c}_\ell^T\boldsymbol{A}^{-1}\boldsymbol{c}_\ell+\widetilde{\boldsymbol{\pi}}_\ell\cdot\boldsymbol{\nu}_\ell\right\}.$$
(K.32)

**Remark K.2** *Consider the special case $\omega_i=1$, $i\in[k]$. We show how the above recovers the solution for (un-weighted) LS. First, note that in this case (K.16) simply gives $\eta=\frac{1}{\gamma}-1$. Thus, $\boldsymbol{\nu}=\boldsymbol{1}_k$ and $\widetilde{\boldsymbol{\pi}}=\boldsymbol{\pi}$. Also, recall that $(\mathrm{diag}(\boldsymbol{\pi})-\boldsymbol{\Pi})\boldsymbol{1}_k=\boldsymbol{0}$ and $\boldsymbol{1}^T\boldsymbol{v}=1$. Thus, (K.25) simply gives $\boldsymbol{\Delta}=\mathbb{E}[\boldsymbol{g}\boldsymbol{g}^T]=\boldsymbol{I}_r$. With these, it can be readily checked that (K.28) and (K.23) simplify to the expressions in (4.4a). Similarly, $\boldsymbol{A}=\boldsymbol{A}'=\boldsymbol{I}_{r+1}$ and (K.26) reduces in this case to (I.1.1). For general weight coefficients, such simplifications do not seem possible and one needs to compute the matrix $\mathbb{E}\left[\left(\boldsymbol{\nu}^T\boldsymbol{v}\right)\boldsymbol{g}\boldsymbol{g}^T\right]$ that appears in the definitions of $\boldsymbol{\Delta}$, $\boldsymbol{A}$ and $\boldsymbol{A}'$. We note that this calculation can be somewhat simplified by applying Gaussian integration by parts similar to lemma C.3.*

## K.3 Computing $\Sigma_{w,w}$

In this section, we use Lemma H.1 to compute the cross-correlations $\langle \widehat{\boldsymbol{w}}_\ell, \widehat{\boldsymbol{w}}_j \rangle$, $j \neq \ell \in [k]$. Specifically, the analysis of (J.13) is almost identical to the analysis of (K.3) in the previous section. Specifically, without repeating all the details for brevity, it can be shown that the AO of (J.13) converges to $\min_{\boldsymbol{a}, \alpha_0 \geq 0, b_\ell, \tau > 0} \ \max_{\beta \geq 0} \ \mathcal{D}(\boldsymbol{a}, \alpha_0, b_\ell, \tau, \beta)$ where $\mathcal{D}(\boldsymbol{a}, \alpha_0, b_\ell, \tau, \beta)$ is as in (K.8) only with $Y_\ell$ substituted by $Y_{\ell,c}$:

$$Y_{\ell,c} \sim \mathrm{Bern}(V_c + V_\ell) \quad \text{and as before:} \quad V_i = \frac{e^{\boldsymbol{e}_i^T \boldsymbol{V} \boldsymbol{\Sigma} \boldsymbol{g}}}{\sum_{\ell'=1}^r e^{\boldsymbol{e}_{\ell'} \boldsymbol{V} \boldsymbol{\Sigma} \boldsymbol{g}}}, \quad i = \ell, c. \tag{K.33}$$

Thus, what changes in the calculations above is in (K.18) and (K.35), where we now have instead

$$\boldsymbol{c}(\eta) := \left( \frac{1}{\frac{1}{\omega_\ell^2} + 1/\eta} \right) \underbrace{\mathbb{E}\left[ \begin{bmatrix} \boldsymbol{g} \\ 1 \end{bmatrix} V_\ell \right]}_{=: \widetilde{\boldsymbol{c}}_\ell} + \left( \frac{1}{\frac{1}{\omega_c^2} + 1/\eta} \right) \underbrace{\mathbb{E}\left[ \begin{bmatrix} \boldsymbol{g} \\ 1 \end{bmatrix} V_c \right]}_{=: \widetilde{\boldsymbol{c}}_c} \tag{K.34}$$

and

$$\sum_{i \neq \{\ell, c\} \in [k]} \frac{\begin{bmatrix} \boldsymbol{a}^T & \boldsymbol{b}_\ell \end{bmatrix} \widetilde{\boldsymbol{A}}_i \begin{bmatrix} \boldsymbol{a} \\ \boldsymbol{b}_\ell \end{bmatrix}}{\left( \omega_i^{-2} + 1/\eta \right)^2} + \frac{\begin{bmatrix} \boldsymbol{a}^T & \boldsymbol{b}_\ell \end{bmatrix} \widetilde{\boldsymbol{A}}_\ell \begin{bmatrix} \boldsymbol{a} \\ \boldsymbol{b}_\ell \end{bmatrix} - 2 \begin{bmatrix} \boldsymbol{a}^T & \boldsymbol{b}_\ell \end{bmatrix} \widetilde{\boldsymbol{c}}_\ell + \boldsymbol{\pi}_\ell}{\left( \omega_\ell^{-2} + \eta \right)^2} \tag{K.35}$$

$$+ \frac{\begin{bmatrix} \boldsymbol{a}^T & \boldsymbol{b}_\ell \end{bmatrix} \widetilde{\boldsymbol{A}}_c \begin{bmatrix} \boldsymbol{a} \\ \boldsymbol{b}_\ell \end{bmatrix} - 2 \begin{bmatrix} \boldsymbol{a}^T & \boldsymbol{b}_\ell \end{bmatrix} \widetilde{\boldsymbol{c}}_c + \boldsymbol{\pi}_c}{\left( \omega_c^{-2} + 1/\eta \right)^2},$$

respectively. With these and following mutatis-mutandis the steps and the notation in the previous section, we find the following asymptotic expression for the magnitude of $\widehat{\boldsymbol{w}}_\ell + \widehat{\boldsymbol{w}}_c$:

$$\| \widehat{\boldsymbol{w}}_\ell + \widehat{\boldsymbol{w}}_c \|_{\ell_2}^2 \xrightarrow{P} \left\| \boldsymbol{\Delta}^{-1} \boldsymbol{\Sigma} \boldsymbol{V}^T \left( \mathrm{diag}(\boldsymbol{\pi}) - \boldsymbol{\Pi} \right) \cdot \left( \boldsymbol{\nu}_\ell \cdot (\boldsymbol{e}_\ell - \boldsymbol{\pi}_\ell \boldsymbol{\nu}) + \boldsymbol{\nu}_c \cdot (\boldsymbol{e}_c - \boldsymbol{\pi}_c \boldsymbol{\nu}) \right) \right\|_{\ell_2}^2$$

$$+ \frac{1}{(1/\gamma - \widetilde{\boldsymbol{\pi}}^T \boldsymbol{\nu})} \cdot \left\{ (\boldsymbol{c}_\ell + \boldsymbol{c}_c)^T \boldsymbol{A}^{-1} \boldsymbol{A}' \boldsymbol{A}^{-1} (\boldsymbol{c}_\ell + \boldsymbol{c}_c) - 2 (\boldsymbol{\nu}_\ell \boldsymbol{c}_\ell + \boldsymbol{\nu}_c \boldsymbol{c}_c)^T \boldsymbol{A}^{-1} (\boldsymbol{\nu}_\ell \boldsymbol{c}_\ell + \boldsymbol{\nu}_c \boldsymbol{c}_c) + \widetilde{\boldsymbol{\pi}}_\ell \cdot \boldsymbol{\nu}_\ell + \widetilde{\boldsymbol{\pi}}_c \cdot \boldsymbol{\nu}_c \right\}.$$

We may now combine this with (K.32) to conclude with the following asymptotic limits for the cross-correlations for all $\ell \neq c \in [k]$:

$$\langle \widehat{\boldsymbol{w}}_\ell, \widehat{\boldsymbol{w}}_c \rangle \xrightarrow{P} \boldsymbol{\nu}_c (\boldsymbol{e}_c - \boldsymbol{\pi}_c \boldsymbol{\nu})^T (\mathrm{diag}(\boldsymbol{\pi}) - \boldsymbol{\Pi}) \boldsymbol{V} \boldsymbol{\Sigma} \boldsymbol{\Delta}^{-2} \boldsymbol{\Sigma} \boldsymbol{V}^T (\mathrm{diag}(\boldsymbol{\pi}) - \boldsymbol{\Pi}) \boldsymbol{\nu}_\ell (\boldsymbol{e}_\ell - \boldsymbol{\pi}_\ell \boldsymbol{\nu})$$

$$+ \frac{1}{(1/\gamma - \widetilde{\boldsymbol{\pi}}^T \boldsymbol{\nu})} \cdot \left\{ \boldsymbol{c}_c^T \boldsymbol{A}^{-1} \boldsymbol{A}' \boldsymbol{A}^{-1} \boldsymbol{c}_\ell - 2 \boldsymbol{\nu}_c \boldsymbol{\nu}_\ell \boldsymbol{c}_c^T \boldsymbol{A}^{-1} \boldsymbol{c}_\ell \right\}. \tag{K.36}$$

$$= \boldsymbol{\nu}_c (\boldsymbol{e}_c - \boldsymbol{\pi}_c \boldsymbol{\nu})^T (\mathrm{diag}(\boldsymbol{\pi}) - \boldsymbol{\Pi}) \boldsymbol{V} \boldsymbol{\Sigma} \boldsymbol{\Delta}^{-2} \boldsymbol{\Sigma} \boldsymbol{V}^T (\mathrm{diag}(\boldsymbol{\pi}) - \boldsymbol{\Pi}) \boldsymbol{\nu}_\ell (\boldsymbol{e}_\ell - \boldsymbol{\pi}_\ell \boldsymbol{\nu})$$

$$+ \frac{1}{(1/\gamma - \widetilde{\boldsymbol{\pi}}^T \boldsymbol{\nu})} \cdot \left\{ \boldsymbol{c}_c^T \left( \boldsymbol{A}^{-1} \boldsymbol{A}' \boldsymbol{A}^{-1} - 2 \boldsymbol{\nu}_c \boldsymbol{\nu}_\ell \boldsymbol{A}^{-1} \right) \boldsymbol{c}_\ell \right\}. \tag{K.37}$$

In matrix form, we have

$$\Sigma_{\boldsymbol{w},\boldsymbol{w}} \xrightarrow{P} \mathrm{diag}(\boldsymbol{\nu}) \left( \boldsymbol{I}_k - \boldsymbol{\pi} \boldsymbol{\nu}^T \right) (\mathrm{diag}(\boldsymbol{\pi}) - \boldsymbol{\Pi}) \boldsymbol{V} \boldsymbol{\Sigma} \boldsymbol{\Delta}^{-2} \boldsymbol{\Sigma} \boldsymbol{V}^T (\mathrm{diag}(\boldsymbol{\pi}) - \boldsymbol{\Pi}) \left( \boldsymbol{I}_k - \boldsymbol{\nu} \boldsymbol{\pi}^T \right) \mathrm{diag}(\boldsymbol{\nu})$$

$$+ \frac{1}{(1/\gamma - \widetilde{\boldsymbol{\pi}}^T \boldsymbol{\nu})} \left\{ \begin{bmatrix} \mathrm{diag}(\boldsymbol{\nu}) (\mathrm{diag}(\boldsymbol{\pi}) - \boldsymbol{\Pi}) \boldsymbol{V} \boldsymbol{\Sigma} & \widetilde{\boldsymbol{\pi}} \end{bmatrix} \boldsymbol{A}^{-1} \boldsymbol{A}' \boldsymbol{A}^{-1} \begin{bmatrix} \boldsymbol{\Sigma} \boldsymbol{V}^T (\mathrm{diag}(\boldsymbol{\pi}) - \boldsymbol{\Pi}) \mathrm{diag}(\boldsymbol{\nu}) \\ \widetilde{\boldsymbol{\pi}}^T \end{bmatrix} \right\}$$

$$- 2 \frac{1}{(1/\gamma - \widetilde{\boldsymbol{\pi}}^T \boldsymbol{\nu})} \left\{ \mathrm{diag}(\boldsymbol{\nu}) \begin{bmatrix} \mathrm{diag}(\boldsymbol{\nu}) (\mathrm{diag}(\boldsymbol{\pi}) - \boldsymbol{\Pi}) \boldsymbol{V} \boldsymbol{\Sigma} & \widetilde{\boldsymbol{\pi}} \end{bmatrix} \boldsymbol{A}^{-1} \begin{bmatrix} \boldsymbol{\Sigma} \boldsymbol{V}^T (\mathrm{diag}(\boldsymbol{\pi}) - \boldsymbol{\Pi}) \mathrm{diag}(\boldsymbol{\nu}) \\ \widetilde{\boldsymbol{\pi}}^T \end{bmatrix} \mathrm{diag}(\boldsymbol{\nu}) \right\}$$

$$+ \frac{1}{(1/\gamma - \widetilde{\boldsymbol{\pi}}^T \boldsymbol{\nu})} \left\{ \mathrm{diag}(\boldsymbol{\nu}) \, \mathrm{diag}(\widetilde{\boldsymbol{\pi}}) \right\}. \tag{K.38}$$