[Reviews · NeurIPS 2020]

Review 1

Summary and Contributions: The paper studies the test accuracy in linear multiclass classification in the high-dimensional regime. Specifically, they studied two models -- Gaussian mixture model and multinomial logit model -- and three algorithms -- least-squares, weighted LS, and per class averaging -- under the setting where the sample size and the number of features tend to infinity proportionally. Their main contribution is the formulae of the limits of certain quantities (intercepts and correlation matrices in the linear classifier) which can be used to characterize the limiting classification error. The theory is validated by simulation experiments.

Strengths: The paper precisely characterizes the asymptotic accuracy in multiclass classification under Gaussian mixture model and multinomial logit model trained by least squares and per class averaging. The theoretical results agree with the numerical experiments quite well.

Weaknesses: The algorithms do not seem natural. This paper studies two simple parametric models for classification: Gaussian mixture model (with identity covariance) and multinomial logit model, which bring with them some natural algorithms, for instance, maximum likelihood and Bayesian methods. However, the algorithms analyzed in the paper, (weighted) least-squares and per class averaging, do not seem a good fit for the models. What is the rational for doing linear regression on a categorical response, or setting each coefficient as the sample mean of the corresponding class? I could not find any justification for this. The results are only asymptotic and only for linear classifier.

Correctness: The theorem and proofs look correct to me.

Clarity: Generally the paper is well written. Though the main body is mostly a summary of results without much intuition. For instance, why is the averaging estimator better for GMM while the LS is better for MLM? Moreover, no intuition is given regarding why the result should hold or how the proof goes. The proof techniques in the supplementary seem interesting and I think if they were sketched in the main paper, the readers could have a better understanding of the theory.

Relation to Prior Work: The paper has a good literature review and their difference from prior works is clear.

Reproducibility: Yes

Additional Feedback:


Review 2

Summary and Contributions: The paper considers the multi-class classification problem using the linear model, in a case of high dimension and large sample size. In the double asymptotic regime where n, d \to \infty, basical statistics of linear classifiers are asymptotically calculated and performance of the proposed schime is numerically verified.

Strengths: In the asymptotic assumption w.r.t. dimension and sample size, explicit forms of statistics of linear classifiers are shown.

Weaknesses: Calculations in the double asymptotic regime are not quite new. Settings of numerical experiments seems to be arbitrary.

Correctness: Yes

Clarity: Yes

Relation to Prior Work: I donot think so. There are few mentions about asymptotic theory of the case (n,d\to \infty).

Reproducibility: Yes

Additional Feedback: Calculations of limits of statistics themselves do not bring any insights for us and theoretical evaluations of generalization performance of the linear classifiers are required. How do the linear classifiers perform on the double asymptotic situation, in the sense of generalization error?


Review 3

Summary and Contributions: This article deals with multi-category pattern recognition. The authors characterize the asymptotic behaviour of four linear classifiers applied to data generated according to two models. The four classifiers differ according to their loss function: least-squares, class averaging, weighted least-squares and cross-entropy. The data are obtained through a Gaussian mixture or a multinomial logit model. The main results are convergences in probability of the parameters (intercepts and "correlation" matrices). The total and class-wise accuracies are also characterized. Experimental results (obtained on artificial data following the aforementioned models) are also provided in Section 5.

Strengths: This contribution fully characterizes the asymptotic behaviour of popular multivariate linear classifiers.

Weaknesses: The limitations directly spring from the very restrictive theoretical framework : linear classifiers and data generated according to two very simple models. It is unlikely that the results obtained extend nicely to more general settings.

Correctness: I was unable to check the technical sanity. On the other hand, I could not find any flaw either.

Clarity: The paper is clearly written. There are only a few typos. Three examples: -line 35: "the number of classes are large?" -> "the number of classes is large?" -line 82: "none of these prior works have" -> "none of these prior works has" -line 89 "correleations" -> "correlations".

Relation to Prior Work: This seems to be the case, although I am not familiar with the state of the art.

Reproducibility: Yes

Additional Feedback:


Review 4

Summary and Contributions: The paper studies the asymptotic probabilities of error of simple linear classifiers under certain model assumptions: Gaussian mixture model and a multiclass logit model. This refers to providing precise asymptotic distributions of the classification error and not just order-wise bounds. As a key step the paper points out that for both models, the error depends on two correlation matrices, the one of the class-dependent weight vectors with each other and the weight vectors with the class mean feature vectors.

Strengths: The theoretical analysis of multiclass classification is an open problem at the core of machine learning / statistical modelling. While the the specific setting considered seem limited, they are insightful and likely an important stepping stone in the full analysis of multi-class classification. For instance, it is very interesting that the simple approach of findings weight vectors (per class) through per class averaging is Bayes optimal for the Gaussian mixture model (under balanced classes). Such results are a very good lead for future investigations of more general settings. The theory accurately predicts the outcomes of numerical experiments.

Weaknesses: I don’t see any major weaknesses in the work.

Correctness: While the basic mathematical setup including the importance of the correlation matrices are straightforward to follow, the main results are technically involved and I did not check the proofs in the appendix. However, the paper makes a 100 percent rigorous impression and the theoretical results are recovered in the numerical experiments. Therefore I have little concerns about correctness.

Clarity: The paper is written extremely densely and challenging to follow. However, I believe this is due to the intrinsic complexity of the problem, and the authors did a great job in providing an accessible introduction and formal setup in Section 2. Given the space limits there is not much that can be done to improve the presentation. Potentially, a little more room could be found for the interpretation of some results.

Relation to Prior Work: The authors give thorough pointers to related work. On question is whether existing results for binary classification could have been lifted in a straightforward way to compare their predictions with the novel more specific results in the numerical experiments. This might give a quantitative sense for the improvement in understanding of the multilclass problem through this work.

Reproducibility: Yes

Additional Feedback: UPDATE: I thank the authors for the useful and specific comments regarding my inquiry about the differences/relation to binary classification results. In general, the author response was very convincing and further corroborated my high opinion of the paper.


Review 5

Summary and Contributions: The submission discusses a high-dimensional asymptotic analysis of the linear/weighted-linear and plug-in estimators for the multiclass classification problem in the setting where the number of training samples and the number of input features grows to infinity at a proportional rate, while the number of classes is O(1). While a lot of works based on CGMT have recently looked at the binary classification problem, this paper looks at the important extension to multi-class classification. Such theory is welcome at a venue like NeurIPS. The experiments corroborate the theoretical predictions made by the authors.

Strengths: The paper is very clearly written. The problem is well motivated and is of significance to the broad ML community at large. Important aspects of the problem are discussed and the main ingredients to look out for in multi-class classification problem are brought out. Overall, the results are very informative and very interesting to the NeurIPS as well as the broad scientific community at large.

Weaknesses: The paper only considers simple estimators for the problem (for which closed form solutions of the estimator are known in terms of the training data), in the absence of any regularization for the weights. Hence it could be that the results could perhaps have been derived from I think the regularized estimators could have been considered by the authors but has been ignored (perhaps for the sake of brevity).

Correctness: The claims and methods seem correct

Clarity: Yes

Relation to Prior Work: Yes

Reproducibility: Yes

Additional Feedback:

[Author Response · NeurIPS 2020]

We thank the reviewers for their feedback and the time spent on our submission.

**First,** let us elaborate on the concerns by **Reviewers 1 & 3** regarding restrictions to specific linear classifiers. To quote
**Reviewer 4** – who we thank for the encouraging feedback – : "*The theoretical analysis of multiclass classification is an*
*open problem at the core of machine learning/statistical modelling. While the specific setting considered seem limited,*
*they are insightful and likely an important stepping stone in the full analysis of multi-class classification...Such results*
*are a very good lead for future investigations of more general settings*". Below, we remark on the following regarding
the motivation behind the studied classifiers and the impact of our results. We will expand upon these in the revision.

**(I)** On the averaging estimator: The averaging estimator is Bayes optimal for balanced GMM with Gaussian means
(Prop. 3.4). As such, it serves well as a baseline and a "natural algorithm" for this data model. One then might wonder
how the performance of this algorithm depends on the data model. Thus, we further analyze its performance for
the second basic model considered here: the MLM. **(II)** On the LS classifier: There have been numerous empirical
works that investigate the role of the loss function in classification tasks for various data models. Several of these
find empirically that simple LS can have comparable performance compared to the (perhaps most commonly used)
hinge/logistic losses, e.g. [53,74,75]. Quoting [p.105, 53]: "*Intuitively, it seems that the square loss may be less*
*well suited to classification than the hinge loss (...) However, in practice, we have found that the accuracy of RLSC*
*(regularized LS classification) is essentially equivalent to that of SVMs*." One of the long-term goals of our project is to
provide theoretical evidence against/in-favor of such empirical findings and to characterize what loss is suitable for
each setting. As a first step, we naturally ask whether these claims are already justified (or not) in simple linear models,
and if so, under what conditions. Along these lines, there are several recent works that theoretically study the role of LS
in high-dimensional *binary* linear classification. For example, under the same asymptotic regime as in our paper, [44]
proves that LS is optimal for GMM within the family of convex un-regularized empirical-risk minimization, and, [60]
proves that LS is approximately optimal (thus, comparable to the ML solution: logistic loss) for logistic data. We take
the first steps towards extending these to the more challenging, but more versatile, *multiclass* setting. **(III)** On WLS: (a)
We are motivated by recent findings [13] that "weighted" variations of LS can significantly boost the performance over
simple LS. (b) Compared to LS, WLS offers the flexibility to adjust the algorithm to balance performance between
majority vs minority classes (together with our ability to accurately predict class-wise errors).

While there is a lot to do further down the road, our results (model setup, analysis, sharp asymptotics) are the first step
towards this direction and facing some of the new challenges in multiclass settings (see lines 45-52). Certain important
additions such as *regularized* (W)LS and correlated Gaussian features – while requiring extra work – are almost direct
extensions of our current framework. Others, such as the study of cross-entropy minimization or extreme classification
will likely require combining elements of our work with new ideas. We believe that our paper sets the fundamentals and
will inspire further investigations in this direction. Of course, extensions to non-asymptotic regimes and non-linear
models (e.g., RFF, NTK) are highly desirable. Such results, only recently obtained for regression settings, are typically
founded on long prior work on simpler regression models – linear, (isotropic) Gaussian, etc.. Our work, together with
refs. in (lines 81-82) for binary settings, resemble these essential precursor works for the setting of classification.

**Second,** on **Reviewer's 1** question on the relative performance of the averaging and LS estimators for the two data
models: This is discussed in Sec. 3.2 and 4.2. In Prop. 3.3, we prove that averaging outperforms LS for balanced GMM
with orthogonal means. Intuitively, this is because "compared to the weight vectors $\mathbf{w}_i$ of the class averaging classifier
that are also (asymptotically) orthogonal when means are orthogonal, this is not the case for LS" (line 222, pg.6). In
fact in Sec. 3.3 we formally study the optimality of the averaging estimator in a Bayesian GMM setting. Similarly, in
Prop. 4.3 and Sec. 4.2, we show that LS outperforms the averaging estimator in MLM for large data samples.

**Third,** we agree with **Reviewers 1 and 4** that sketching key proof ideas in the main body of the paper will benefit the
reader. If accepted, we will use the extra space to move the corresponding discussions from App. F to the main body.

**Fourth,** in response to **Reviewer's 4** suggestion. Indeed, results for binary classification can be lifted to characterize
the limit of $\mathbf{\Sigma}_{w,\mu}$ and diagonal entries of $\mathbf{\Sigma}_{w,w}$ for one-vs-all classifiers (including LS) (with some additional technical
work to capture correlations $\mathbf{\Sigma}_{\mu,\mu}$ for k>2). As mentioned in the paper, this alone does not give any information
on the off-diagonals of $\mathbf{\Sigma}_{w,w}$, needed in the exact test-error formulas (2.3)/(2.4). It is possible to derive heuristic
approximations and union bound arguments leading to error expressions that depend only on the diagonals of $\mathbf{\Sigma}_{w,w}$.
Indeed, Fig. 5 in App. A provides a result of this flavor and gives a sense of how our exact results improve upon such
approximations. We will expand upon this comparison in Appendix A in the revision.

**Reviewer 2:** With respect to test error, our paper is precisely about characterizing the performance of the studied linear
classifiers *in the sense of test error.* The formulas of Thms. 3.1,3.2,4.1,4.2 can be directly plugged in (2.3) and (2.4)
to obtain test error. Regarding "calculations in the double asymptotic regime are not quite new": Of course, there are
numerous works in this regime under numerous settings over the last decade (lines 77-79). However, ours is *the first*
*such work in multiclass classification* This point is well-articulated in the introduction (lines 73-76, 82-91).

[Meta-Review · NeurIPS 2020]

The paper studies the statistical behaviour of certain multiclass classification algorithms in the doubly-asymptotic limit of n, d -> ∞. The results elucidate certain differences compared to the analysis of binary classifiers, such as dependence on class-correlation matrices. One reviewer raised concerns about the results not providing insight into generalisation performance. The response indicates this is not the case, and this was corroborated by other reviews and my own reading. One critique raised by a couple of reviewers was regarding the specialised nature of the results, which are for linear classifiers and specific data models. These critiques are valid, but given that the paper is the first extension of existing work to the multiclass setting, it seems permissible to make tractable simplifications. Beyond these, three reviewers found the results to be interesting and well-presented. This was corroborated by an additional review sought after the author feedback. From my own reading of the paper, I concur that it is well written and the theoretical results seem of interest to the community. We thus recommend acceptance.